# Understanding the Forgetting of (Replay-based) Continual Learning via Feature Learning: Angle Matters

Hongyi Wan [* 1]  Shiyuan Ren [* 1]  Wei Huang [2]  Miao Zhang [1]  Xiang Deng [1]  Yixin Bao [1]  Liqiang Nie [1]

## Abstract

Continual learning (CL) is crucial for advancing human-level intelligence, but its theoretical understanding, especially regarding factors influencing forgetting, is still relatively limited. This work aims to build a unified theoretical framework for understanding CL using feature learning theory. Different from most existing studies that analyze forgetting under linear regression model or lazy training, we focus on a more practical two-layer convolutional neural network (CNN) with polynomial ReLU activation for sequential tasks within a signal-noise data model. Specifically, we theoretically reveal how the angle between task signal vectors influences forgetting that: *acute or small obtuse angles lead to benign forgetting, whereas larger obtuse angles result in harmful forgetting*. Furthermore, we demonstrate that the replay method alleviates forgetting by expanding the range of angles corresponding to benign forgetting. Our theoretical results suggest that mid-angle sampling, which selects examples with moderate angles to the prototype, can enhance the replay method's ability to mitigate forgetting. Experiments on synthetic and real-world datasets confirm our theoretical results and highlight the effectiveness of our mid-angle sampling strategy.

## 1. Introduction

Continual learning (CL) involves learning a series of tasks in sequence (Wang et al., 2024), where the model adapts as if all tasks are learned simultaneously. Unlike traditional machine learning models which deal with static data, CL focuses on dynamic and evolving distributions. A major

*Equal contribution [1]Harbin Institute of Technology (Shenzhen), Shenzhen, China [2]Independent Researcher. Correspondence to: Miao Zhang <zhangmiao@hit.edu.cn>, Wei Huang <weihuang.uts@gmail.com>.

*Proceedings of the 42^nd International Conference on Machine Learning*, Vancouver, Canada. PMLR 267, 2025. Copyright 2025 by the author(s).

challenge in CL is the *catastrophic forgetting* phenomenon (McCloskey & Cohen, 1989), where learning from new distributions often causes a significant decline in the model's ability to retain knowledge from previous tasks.

Catastrophic forgetting in CL has prompted various empirical methods, which can be broadly categorized into three types: regularization-based methods, replay-based methods, and architecture-based methods. Regularization-based methods (Kirkpatrick et al., 2017; Chaudhry et al., 2018a; Benzing, 2022) selectively regularize parameter changes to balance the learning of new and old tasks. Replay-based methods (Rebuffi et al., 2017; Lopez-Paz & Ranzato, 2017; Chaudhry et al., 2018b; Rolnick et al., 2019; Buzzega et al., 2020; Boschini et al., 2022) approximate and restore old data distributions by storing training samples or gradient information from previous tasks. Architecture-based methods (Mallya et al., 2018; Ostapenko et al., 2019; Qin et al., 2021; Kumar et al., 2021) allocate specific network parameters to each task to counteract forgetting.

Despite the success of experimental studies in addressing forgetting, theoretical understanding of forgetting in CL remains limited. Most of the recent works concentrate on linear models (Evron et al., 2022; 2023; Lin et al., 2023; Ding et al., 2024; Zhao et al., 2024). Analyses beyond linear models are primarily confined to the teacher-student setup (Lee et al., 2021) and the neural tangent kernel (Bennani et al., 2020; Doan et al., 2021). However, these theories often rely on strong assumptions, such as infinitely wide networks or linearizing assumptions in the weight space, which may not fully capture the generalization and forgetting dynamics in CL. As a result, a unified theoretical framework for CL with practical neural networks remains absent.

In this paper, we fill the theoretical gap in CL with neural networks through the application of feature learning theory (Allen-Zhu & Li, 2020; Cao et al., 2022; Huang et al., 2023), establishing a unified theoretical framework to analyze the convergence, generalization and forgetting in CL. We provide a precise characterization to how the angle between task signal vectors influences forgetting in polynomial ReLU neural networks during continual learning. Moreover, we reveal the mechanisms by which replay-based methods help alleviate forgetting. Our theoretical findings inspire mid-

angle sampling, which further enhances the effectiveness of replay-based methods in mitigating forgetting.

## 1.1. Problem Setup

Our focus is primarily on two binary classification tasks in CL, using a data distribution similar to that in Cao et al. (2022). The input data consists of two components: signals which correspond to image labels, and noises which represent background information unrelated to the labels.

**Definition 1.1.** *Let $\boldsymbol{\mu}_k \in \mathbb{R}^d$ be a fixed vector representing the signal contained in each data point of the $k$-th task $T_k$, and let $\mathbf{U} = \mathrm{orth}(\{\boldsymbol{\mu}_k\}) \in \mathbb{R}^{d \times k}$ represent the collection of signal vectors, where $\mathrm{orth}(\cdot)$ denotes Gram-Schmidt orthogonalization and normalization. Then each data point $(\mathbf{x}_k, y_k)$ of task $T_k$ with $\mathbf{x}_k = [\mathbf{x}_k^{(1)\top}, \mathbf{x}_k^{(2)\top}]^\top \in \mathbb{R}^{2d}$ and $y_k \in \{-1, 1\}$ is generated from the distribution $\mathcal{D}_k$:*

1. *Label $y_k$ is generated as a Rademacher random variable.*

2. *A noise vector $\boldsymbol{\xi}_k$ is generated from the Gaussian distribution $N(\mathbf{0}, \sigma_{p_k}^2 \cdot (\mathbf{I} - \mathbf{U}(\mathbf{U}^\top \mathbf{U})^{-1} \mathbf{U}^\top))$, where $\mathbf{U} \in \mathbb{R}^{d \times k}$.*

3. *One of $\mathbf{x}_k^{(1)}, \mathbf{x}_k^{(2)}$ is given as $y_k \cdot \boldsymbol{\mu}_k$, which represents the signal, the other is given by $\boldsymbol{\xi}_k$, which represents noises.*

Definition 1.1 adapts the data distribution from Cao et al. (2022) for the CL task setting, with the noise patch following $N(\mathbf{0}, \sigma_{p_k}^2 \cdot (\mathbf{I} - \mathbf{U}(\mathbf{U}^\top \mathbf{U})^{-1} \mathbf{U}^\top))$, ensuring orthogonality to the signal vector $\boldsymbol{\mu}_k$ of each task $T_k$. We train a two-layer CNN with polynomial ReLU activation (ReLU$^q$, $q > 2$) by minimizing empirical cross-entropy loss:

$$L_{S_k}(\mathbf{W}^{(T_k)}) = \frac{1}{n_k} \sum_{i=1}^{n_k} \ell[y_{k,i} \cdot f(\mathbf{W}^{(T_k)}, \mathbf{x}_{k,i})], \quad (1.1)$$

where $\ell(z) = \log(1 + \exp(-z))$, $S_k = \{(\mathbf{x}_{k,i}, y_{k,i})\}_{i=1}^{n_k}$ is the training data set for task $T_k$, and $f(\mathbf{W}^{(T_k)}, \mathbf{x}_{k,i})$ is the two-layer CNN as defined in Section 2. We use gradient descent to minimize (1.1). Additionally, we define the true loss (test loss) $L_{\mathcal{D}_k}(\mathbf{W}^{(T_k)}) := \mathbb{E}_{(\mathbf{x}_k, y_k) \sim \mathcal{D}_k} \ell[y_k \cdot f(\mathbf{W}^{(T_k)}, \mathbf{x}_k)]$ and the true error (test error):

$$L_{\mathcal{D}_k}^{0-1}(\mathbf{W}^{(T_k)}) := \mathbb{P}_{(\mathbf{x}_k, y_k) \sim \mathcal{D}_k}\big[y_k \neq \mathrm{sign}\big(f(\mathbf{W}^{(T_k)}, \mathbf{x}_k)\big)\big].$$

Then the forgetting is naturally measured by the true error on the first task:

$$L_{\mathcal{D}_1}^{0-1}(\mathbf{W}^{(T_2)}) := \mathbb{P}_{(\mathbf{x}_1, y_1) \sim \mathcal{D}_1}\big[y_1 \neq \mathrm{sign}\big(f(\mathbf{W}^{(T_2)}, \mathbf{x}_1)\big)\big].$$

## 1.2. Main Contributions

For ease of discussion, we define the signal-to-noise ratio for task $T_k$ as $\mathrm{SNR}_k = \|\boldsymbol{\mu}_k\|_2 / (\sigma_{p_k} \sqrt{d})$, and the cosine similarity between the signal vectors of task $T_1$ and $T_2$

as $\cos \theta_{1,2} = \langle \boldsymbol{\mu}_1, \boldsymbol{\mu}_2 \rangle / (\|\boldsymbol{\mu}_1\|_2 \cdot \|\boldsymbol{\mu}_2\|_2)$. We prove the following Theorem 1.2, which characterizes the training loss and test loss on the current task $T_2$, as well as the true error on the previous task $T_1$ of the two-layer polynomial ReLU CNN. Figure 1 visually illustrates the relationship between forgetting and $\cos \theta_{1,2}$ from Theorem 1.2.

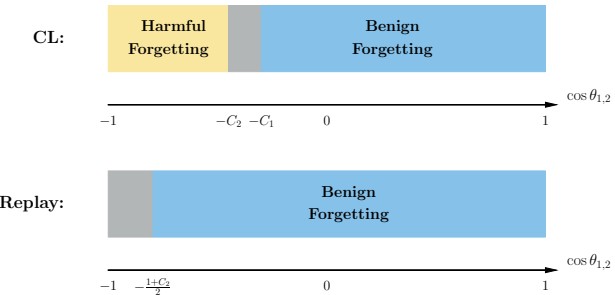

*Figure 1.* Illustration of the relationship between forgetting and cosine similarity in CL wo/w replay. The blue region represents the cosine setting with benign forgetting, while the yellow region corresponds to the setting with harmful forgetting. The gray region indicates the setting where forgetting is not well characterized.

**Theorem 1.2** (Informal). *For any $\epsilon > 0$, under certain regularity condition, if $n_k \cdot \mathrm{SNR}_k^q = \widetilde{\Omega}(1)$ for $k \in [2]$, with probability at least $1 - \delta$, there exists $t_{end}$ such that the training loss of task $T_2$ converges to $\epsilon$, i.e., $L_S(\mathbf{W}^{(T_2, t_{end})}) \leq \epsilon$. Furthermore, the model obtained through CL achieves a small test loss on the current task: $L_{\mathcal{D}_2}(\mathbf{W}^{(T_2, t_{end})}) \leq 6\epsilon + \exp(-n_2^2)$. Regarding forgetting on the previous task in CL, we present the following statements to illustrate its relationship with cosine similarity.*

1. *When $-C_1 \leq \cos \theta_{1,2} \leq 1$, the forgetting on task $T_1$: $L_{\mathcal{D}_1}^{0-1}(\mathbf{W}^{(T_2, t_{end})}) \leq \exp(-C \cdot m^{2q-2} n^{2q} / q^2)$.*

2. *When $-1 \leq \cos \theta_{1,2} < -C_2$, the forgetting on task $T_1$: $L_{\mathcal{D}_1}^{0-1}(\mathbf{W}^{(T_2, t_{end})}) \geq 1 - \exp(-C \cdot m^{2q-2} n^{2q} / q^2)$.*

3. *With replay, when $-\frac{C_2 + 1}{2} \leq \cos \theta_{1,2} < 0$, the forgetting on task $T_1$ is bounded by: $L_{\mathcal{D}_1}^{0-1}(\mathbf{W}^{(T_2, t_{end})}) \leq \exp(-C \cdot m^{2q-2} n^{2q} / q^2)$.*

*Here, $C_1$ and $C_2$ are positive constants with $C_1 < C_2 < 1$.*

The significance of Theorem 1.2 is summarized as follows:

- Under the condition on the signal-to-noise ratio that $n_k \cdot \mathrm{SNR}_k^q = \widetilde{\Omega}(1)$, the learned CNN can achieve both small training and test losses on the current task.

- When the angle $\theta_{1,2}$ between two tasks satisfies $-C_1 \leq \cos \theta_{1,2} \leq 1$, the CNN can achieve benign forgetting,

where performance on the previous task slightly deteriorates to accommodate the new task, but test error remains small. This contrasts with harmful forgetting, which occurs when $-1 \leq \cos\theta_{1,2} < -C_2$ leading to significant test error on the old task. Notably, Jeon et al. (2022) also introduce the benign forgetting, but they refer to the forgetting of misguidance learned from biased datasets rather than the balance between old and new tasks.

- With replay, benign forgetting occurs when the angle between two tasks satisfies $-\frac{C_2+1}{2} \leq \cos\theta_{1,2} \leq 1$, indicating an expanded angular range compared to the case without replay ($-C_1 \leq \cos\theta_{1,2} \leq 1$).

To the best of our knowledge, we are the first to apply feature learning theory to analyze the convergence, generalization, and forgetting of the polynomial ReLU CNN in CL. This analysis extends beyond the previous works that rely on the linearization assumption in lazy training (Bennani et al., 2020; Doan et al., 2021), where network parameters remain close to their initial values throughout the training process. Our theoretical results demonstrate that replay-based methods mitigate forgetting by expanding the angular range of benign forgetting and inspire mid-angle sampling to enhance replay strategies. Furthermore, we conduct comprehensive experiments on both synthetic and real-world datasets to validate our theoretical findings and the effectiveness of mid-angle sampling.

## 2. Preliminaries

In this section, we present the notation, the neural network model, and the training algorithm based on gradient descent.

**Notation.** We represent scalars with lowercase letters, vectors with lowercase boldface letters, and matrices with uppercase boldface letters. The $l_2$ norm of a vector $v$ is denoted as $\|v_2\|_2$. For a matrix $\mathbf{A}$, we denote its spectral norm by $\|\mathbf{A}\|_2$ and its Frobenius norm by $\|\mathbf{A}\|_F$. To compare two sequences, we use standard asymptotic notations such as $o(\cdot)$, $O(\cdot)$, $\Omega(\cdot)$ and $\Theta(\cdot)$ to characterize their limiting behavior. Additionally, we utilize $\widetilde{O}(\cdot)$, $\widetilde{\Omega}(\cdot)$, and $\widetilde{\Theta}(\cdot)$ to suppress logarithmic factors in these notations. Moreover, we write $x_n = \text{poly}(y_n)$ if $x_n = O(y_n^D)$ for some positive constant $D$. Lastly, sequences of integers are represented as $[n] = \{1, 2, \ldots, n\}$, and $a \vee b$ denotes $\max\{a, b\}$.

**Neural Network Model.** We consider a two-layer CNN model with a ReLU$^q$ activation function, defined as $\sigma(z) = (\max\{0, z\})^q$. When learning task $T_k$, the neural network's output for the input data $\mathbf{x}_k$ is expressed as $f(\mathbf{W}^{(T_k)}, \mathbf{x}_k) = F_{+1}(\mathbf{W}_{+1}^{(T_k)}, \mathbf{x}_k) - F_{-1}(\mathbf{W}_{-1}^{(T_k)}, \mathbf{x}_k)$, where the sign corresponds to the neuron fixed as either $+1/m$ or $-1/m$ in the second layer. The terms $F_{+1}(\mathbf{W}_{+1}^{(T_k)}, \mathbf{x}_k)$ and

$F_{-1}(\mathbf{W}_{-1}^{(T_k)}, \mathbf{x}_k)$ are defined as:

$$F_j(\mathbf{W}_j^{(T_k)}, \mathbf{x}_k) = \frac{1}{m}\sum_{r=1}^{m}\left[\sigma(\langle\mathbf{w}_{j,r}^{(T_k)}, \mathbf{x}_k^{(1)}\rangle) + \sigma(\langle\mathbf{w}_{j,r}^{(T_k)}, \mathbf{x}_k^{(2)}\rangle)\right]$$

where $m$ is the network width, and $\mathbf{w}_{j,r}^{(T_k)}$ represents the $r$-th neuron in the first layer during task $T_k$. The entire parameter set of the model is denoted by $\mathbf{W}^{(T_k)}$.

**Training Algorithm.** We use gradient descent to optimize the objective defined in (1.1). For the first task, the weights are initialized using Gaussian initialization, where each entry of $\mathbf{W}_{+1}^{(T_1,0)}$ and $\mathbf{W}_{-1}^{(T_1,0)}$ is independently drawn from a Gaussian distribution $N(0, \sigma_0^2)$. In the context of CL, the initial weight of the current task naturally equal the final weight of the previous task, i.e., $\mathbf{w}_{j,r}^{(T_k,0)} = \mathbf{w}_{j,r}^{(T_{k-1},t_{end})}(k \geq 2)$. Note that $(T_k, t_{end})$ represents the final iteration of task $T_k$, where the corresponding $t_{end}$ can vary for different tasks. Then the update of the filters in the CNN during learning task $T_k$ can be written as

$$\mathbf{w}_{j,r}^{(T_k,t+1)} = \mathbf{w}_{j,r}^{(T_k,t)} - \eta_k \cdot \nabla_{\mathbf{w}_{j,r}} L_{S_k}(\mathbf{W}^{(T_k,t)})$$

$$= \mathbf{w}_{j,r}^{(T_k,t)} - \frac{\eta_k}{n_k m}\sum_{i=1}^{n_k}\ell_{k,i}^{\prime(T_k,t)}j[\sigma'(\langle\mathbf{w}_{j,r}^{(T_k,t)}, \boldsymbol{\xi}_{k,i}\rangle)$$

$$\cdot y_{k,i}\boldsymbol{\xi}_{k,i} + \sigma'(\langle\mathbf{w}_{j,r}^{(T_k,t)}, y_{k,i}\boldsymbol{\mu}_k\rangle)\boldsymbol{\mu}_k]$$

$$(2.1)$$

for $j \in \{\pm 1\}$ and $r \in [m]$, where the derivative of the loss is defined as $\ell_{k,i}^{\prime(T_k,t)} = \ell'[y_{k,i} \cdot f(\mathbf{W}^{(T_k,t)}, \mathbf{x}_{k,i})]$.

## 3. Main Results

In this section, we present our main theoretical results, which are established based on the following conditions.

**Condition 3.1.** *Suppose that*

1. *The dimension $d$ is sufficiently large: $d = \widetilde{\Omega}(m^{2\vee[4/(q-2)]} \cdot \max\{n_k\}^{4\vee[(2q-2)/(q-2)]})$.*

2. *The neural network width $m$ and training sample size $n_k$ of task $T_k$ satisfy $m, n_k = \Omega(\text{polylog}(d))$ for $k \in [2]$.*

3. *The learning rate $\eta_k$ of task $T_k$ satisfies $\eta_k \leq \widetilde{O}(\min\{\|\boldsymbol{\mu}_k\|_2^{-2}, \sigma_{p_k}^{-2}d^{-1}\})$ for $k \in [2]$. The standard deviation of Gaussian initialization $\sigma_0$ is chosen such that $\sigma_0 \leq \widetilde{O}(m^{-[2/(q-2)]\vee 1} \cdot \max\{n_k\}^{-[1/(q-2)]\vee 1}) \cdot \min\{(\sigma_{p_k}\sqrt{d})^{-1}, \|\boldsymbol{\mu}_k\|_2^{-1}\}$.*

The first two conditions regarding $d, m, n_k$ are designed to ensure a sufficiently over-parameterized learning setup, as well as certain statistical properties of the training data and weight initialization based on concentration inequalities. Similar conditions have also been employed in Chatterji & Long (2021); Frei et al. (2022); Cao et al. (2022); Kou et al.

(2023). Condition on $\eta_k$ and $\sigma_0$ is imposed to guarantee that gradient descent can effectively minimize the training loss.

The definition of $\cos\theta_{1,2}$ follows Subsection 1.2, representing the cosine similarity between task $T_1$ and $T_2$. Based on the aforementioned conditions and definitions, we present the main result in the following theorems.

**Theorem 3.2** (standard CL)**.** *For any $\epsilon > 0$, under Condition 3.1, if $n_k \cdot \mathrm{SNR}_k^q = \widetilde{\Omega}(1)$, then with probability at least $1 - \delta$ there exists $t_{end} = \widetilde{O}(m\eta_2^{-1}\sigma_0^{-(q-2)}\|\boldsymbol{\mu}_2\|_2^{-q} + m^3\eta_2^{-1}\epsilon^{-1}\|\boldsymbol{\mu}_2\|_2^{-2})$ such that the training loss on task $T_2$ converges to $\epsilon$, i.e., $L_S(\mathbf{W}^{(T_2,t_{end})}) \leq \epsilon$. Moreover, the trained CNN will generalize with a small test loss on task $T_2$: $L_{\mathcal{D}_2}(\mathbf{W}^{(T_2,t_{end})}) \leq 6\epsilon + \exp(-n_2^2)$. The connection between forgetting and cosine similarity can be summarized in the following three points:*

1. *When $\cos\theta_{1,2} \geq 0$, the trained CNN achieves a small test loss on task $T_1$: $L_{\mathcal{D}_1}(\mathbf{W}^{(T_2,t_{end})}) \leq 18\epsilon + \exp(-n_1^2)$.*

2. *When $-C_1 \leq \cos\theta_{1,2} < 0$, the trained CNN achieves a small test error on task $T_1$: $L_{\mathcal{D}_1}^{0-1}(\mathbf{W}^{(T_2,t_{end})}) \leq \exp(-C \cdot m^{2q-2}n^{2q}/q^2)$.*

3. *When $-1 \leq \cos\theta_{1,2} < -C_2$, the test error on task $T_1$: $L_{\mathcal{D}_1}^{0-1}(\mathbf{W}^{(T_2,t_{end})}) \geq 1 - \exp(-C \cdot m^{2q-2}n^{2q}/q^2)$.*

*Here, $C_1$ and $C_2$ are positive constants, with $C_1 < C_2 < 1$.*

Theorem 3.2 shows that the learned CNN can achieve small training and test losses on the current task if the signal-to-noise ratio is large. As the training loss converges to $\epsilon$, the angle $\theta_{1,2}$ plays a crucial role in determining the nature of forgetting: acute or small obtuse angles lead to benign forgetting (small test error on the previous task), while larger obtuse angles cause harmful forgetting (large test error on the previous task). Note that a small test loss implies a small test error, as the loss function in (1.1) penalizes incorrect classifications more heavily. Based on Theorem 3.2, we further examine the relationship between forgetting and cosine similarity in CL with replay.

**Theorem 3.3** (Replay-based CL)**.** *For any $\epsilon > 0$, under the same conditions as Theorem 3.2, when $-\frac{1+C_2}{2} \leq \cos\theta_{1,2} < 0$ and replay buffer size $n_1^* \geq n_2 \cdot \widetilde{\Theta}\{(-\cos\theta_{1,2})^q\eta_2 m^{-2/q}\}$, then with probability at least $1 - \delta$, there exists $t_{end} = \widetilde{\Theta}(\eta_2^{-1}m\sigma_0^{2-q}\|\boldsymbol{\mu}_2\|_2^{-q} + m^{\frac{2}{q}}\eta_2^{-1}\epsilon^{-k^q})$ such that:*

1. *The training loss on task $T_2$: $L_{S_2}(\mathbf{W}^{(T_2,t_{end})}) \leq \epsilon^C$, where $C = O(1)$ is a positive constant.*

2. *The trained CNN achieves a small test loss on task $T_2$: $L_{D_2}(\mathbf{W}^{(T_2,t_{end})}) \leq 6\epsilon^C + \exp(-n_2^2)$.*

3. *The CNN achieves a small test error on task $T_1$: $L_{\mathcal{D}_1}^{0-1}(\mathbf{W}^{(T_2,t_{end})}) \leq \exp(-C \cdot m^{2q-2}n^{2q}/q^2)$.*

The first two points of Theorem 3.3 demonstrate that the CNN trained with replay in CL can achieve small training and test losses on the current task, similar to Theorem 3.2. The final point states that if the replay buffer size for task $T_1$ exceeds a certain threshold, the replay method can mitigate forgetting by expanding the range of angles corresponding to benign forgetting.

# 4. Proof Roadmap

In this section, we provide a proof roadmap for Theorem 3.2 and Theorem 3.3. Based on the gradient descent update rule in (2.1), it can be observed that the weights $\mathbf{w}_{j,r}^{(T_k,t)}$ for task $T_k$ are a linear combination of the initialization $\mathbf{w}_{j,r}^{(T_k,0)}$, the signal vectors $\boldsymbol{\mu}_k$, and the noise vectors $\boldsymbol{\xi}_{k,i}$ for $i \in [n_k]$ and $k \in [2]$. Therefore, for $r \in [m]$, the weight vector can be decomposed as such:

$$\mathbf{w}_{j,r}^{(T_k,t)} = \mathbf{w}_{j,r}^{(T_k,0)} + j\gamma(\boldsymbol{\mu}_k)_{j,r}^{(T_k,t)}\frac{\boldsymbol{\mu}_k}{\|\boldsymbol{\mu}_k\|_2^2} + \sum_{i=1}^{n_k}\rho(\boldsymbol{\xi}_k)_{j,r,i}^{(T_k,t)}\frac{\boldsymbol{\xi}_{k,i}}{\|\boldsymbol{\xi}_{k,i}\|_2^2}.$$

Further denote $\overline{\rho}(\boldsymbol{\xi}_k)_{j,r,i}^{(T_k,t)} := \rho(\boldsymbol{\xi}_k)_{j,r,i}^{(T_k,t)}\mathbb{1}(\rho(\boldsymbol{\xi}_k)_{j,r,i}^{(T_k,t)} \geq 0)$, $\underline{\rho}(\boldsymbol{\xi}_k)_{j,r,i}^{(T_k,t)} := \rho(\boldsymbol{\xi}_k)_{j,r,i}^{(T_k,t)}\mathbb{1}(\rho(\boldsymbol{\xi}_k)_{j,r,i}^{(T_k,t)} \leq 0)$. Then

$$\begin{aligned}
\mathbf{w}_{j,r}^{(T_k,t)} &= \mathbf{w}_{j,r}^{(T_k,0)} + j \cdot \gamma(\boldsymbol{\mu}_k)_{j,r}^{(T_k,t)} \cdot \|\boldsymbol{\mu}_k\|_2^{-2} \cdot \boldsymbol{\mu}_k \\
&\quad + \sum_{i=1}^{n_k}[\overline{\rho}(\boldsymbol{\xi}_k)_{j,r,i}^{(T_k,t)} + \underline{\rho}(\boldsymbol{\xi}_k)_{j,r,i}^{(T_k,t)}] \cdot \|\boldsymbol{\xi}_{k,i}\|_2^{-2} \cdot \boldsymbol{\xi}_{k,i},
\end{aligned} \tag{4.1}$$

where $\gamma(\boldsymbol{\mu}_k)_{j,r}^{(T_k,t)}, \overline{\rho}(\boldsymbol{\xi}_k)_{j,r,i}^{(T_k,t)}, \underline{\rho}(\boldsymbol{\xi}_k)_{j,r,i}^{(T_k,t)}$ represent the relevant coefficients. We refer to Equation (4.1) as the signal-noise decomposition of $\mathbf{w}_{j,r}^{(T_k,t)}$, where $\|\boldsymbol{\mu}_k\|_2^{-2}$ and $\|\boldsymbol{\xi}_{k,i}\|_2^{-2}$ serve as normalization factors. The key aspect of the signal-noise decomposition model is to simplify the dynamic process of CNN learning into the dynamic update of the coefficients, which forms the basis of our analysis.

## 4.1. Proof Sketch for Continual Learning

Based on the decomposition in (4.1) and gradient descent update (2.1), the iteration of coefficients are given:

**Lemma 4.1.** *The coefficients $\gamma(\boldsymbol{\mu}_k)_{j,r}^{(T_k,t)}, \overline{\rho}_{j,r,i}^{(T_k,t)}, \underline{\rho}_{j,r,i}^{(T_k,t)}$ in decomposition (4.1) satisfy the following equations:*

$$\gamma(\boldsymbol{\mu}_k)_{j,r}^{(T_k,0)}, \overline{\rho}(\boldsymbol{\xi}_k)_{j,r,i}^{(T_k,0)}, \underline{\rho}(\boldsymbol{\xi}_k)_{j,r,i}^{(T_k,0)} = 0, \tag{4.2}$$

$$\begin{aligned}
\gamma(\boldsymbol{\mu}_k)_{j,r}^{(T_k,t+1)} &= \gamma(\boldsymbol{\mu}_k)_{j,r}^{(T_k,t)} - \frac{\eta_k}{n_k m} \cdot \sum_{i=1}^{n_k}\ell_{k,i}'^{(T_k,t)} \\
&\quad \cdot \sigma'(\langle\mathbf{w}_{j,r}^{(T_k,t)}, y_{k,i} \cdot \boldsymbol{\mu}_k\rangle) \cdot \|\boldsymbol{\mu}_k\|_2^2,
\end{aligned} \tag{4.3}$$

$$\overline{\rho}(\boldsymbol{\xi}_k)_{j,r,i}^{(T_k,t+1)} = \overline{\rho}(\boldsymbol{\xi}_k)_{j,r,i}^{(T_k,t)} - \frac{\eta_k}{n_k m} \cdot \ell_{k,i}'^{(T_k,t)}$$

$$\cdot \sigma'(\langle \mathbf{w}_{j,r}^{(T_k,t)}, \boldsymbol{\xi}_{k,i}\rangle) \cdot \|\boldsymbol{\xi}_{k,i}\|_2^2 \cdot \mathbb{1}(y_{k,i}=j), \quad (4.4)$$

$$\underline{\rho}(\boldsymbol{\xi}_k)_{j,r,i}^{(T_k,t+1)} = \underline{\rho}(\boldsymbol{\xi}_k)_{j,r,i}^{(T_k,t)} + \frac{\eta_k}{n_k m} \cdot \ell_{k,i}'^{(T_k,t)}$$

$$\cdot \sigma'(\langle \mathbf{w}_{j,r}^{(T_k,t)}, \boldsymbol{\xi}_{k,i}\rangle) \cdot \|\boldsymbol{\xi}_{k,i}\|_2^2 \cdot \mathbb{1}(y_{k,i}=-j) \quad (4.5)$$

Lemma 4.1 describes the iteration rule for the coefficients under gradient descent update. When $\cos\theta_{1,2} = 0$ (right angle), the absence of interaction between signal vectors ensures small test loss on both tasks following the approach of Cao et al. (2022). We then analyze the convergence, generalization, and forgetting in CL under two cases: $\cos\theta_{1,2} > 0$ (acute angle) and $\cos\theta_{1,2} < 0$ (obtuse angle).

**Acute angle.** When $\cos\theta_{1,2} > 0$, we apply a two-stage technique to analyze the convergence and generalization on the current task $T_2$ similarly to Cao et al. (2022). The key difference is that since task $T_1$ has already been learned, a component $\gamma(\boldsymbol{\mu}_1)_{j,r}^{(T_1,t_{end})}\|\boldsymbol{\mu}_2\|_2 \cos\theta_{1,2}/\|\boldsymbol{\mu}_1\|_2$ is present in the signal vector $\boldsymbol{\mu}_2$ during the initial phase of training task $T_2$. If this component is large, the training process can skip the first stage and proceed directly to the second stage. As a result, small training and test losses on task $T_2$ in Theorem 3.2 naturally follow. The complete proof can be found in Appendix E.2.

After learning task $T_2$, forgetting can be analyzed by examining the true loss on task $T_1$, i.e., $L_{\mathcal{D}_1}(\mathbf{W}^{(T_2,t_{end})})$. Under the same conditions as Theorem 3.2, Cao et al. (2022) show that learning a single task can result in a small true loss, i.e., $L_{\mathcal{D}_1}(\mathbf{W}^{(T_1,t_{end})}) \le 6\epsilon + \exp(-n_1^2)$. Consequently, we can bound $L_{\mathcal{D}_1}(\mathbf{W}^{(T_2,t_{end})})$ by analyzing its relationship with $L_{\mathcal{D}_1}(\mathbf{W}^{(T_1,t_{end})})$. When $\cos\theta_{1,2} > 0$, due to the positive interaction between the signal vectors of the two tasks, the inequalities $\langle \mathbf{w}_{y_1,r}^{(T_2,t_{end})}, y_1\boldsymbol{\mu}_1\rangle \ge \langle \mathbf{w}_{y_1,r}^{(T_1,t_{end})}, y_1\boldsymbol{\mu}_1\rangle$ and $\langle \mathbf{w}_{-y_1,r}^{(T_2,t_{end})}, y_1\boldsymbol{\mu}_1\rangle \le \langle \mathbf{w}_{-y_1,r}^{(T_1,t_{end})}, y_1\boldsymbol{\mu}_1\rangle$ can be established, which naturally lead to the following lemma.

**Lemma 4.2.** *Under the same conditions as Theorem 3.2, when $\cos\theta_{1,2} > 0$, the following bounds hold for $t_{end}$:*

$$y_1 f(\mathbf{W}^{(T_1,t_{end})}, \mathbf{x}_1) - y_1 f(\mathbf{W}^{(T_2,t_{end})}, \mathbf{x}_1) \le \kappa, \quad (4.6)$$

*where $\kappa = \Theta(1)$ is a positive constant.*

Note that Eq.(4.6) essentially characterizes the connection between $L_{\mathcal{D}_1}(\mathbf{W}^{(T_2,t_{end})})$ and $L_{\mathcal{D}_1}(\mathbf{W}^{(T_1,t_{end})})$ by the definition $L_{\mathcal{D}_1}(\mathbf{W}^{(T_k,t_{end})}) = \mathbb{E}_{(\mathbf{x}_1,y_1)\sim\mathcal{D}_1}\ell[y_1 \cdot f(\mathbf{W}^{(T_k,t_{end})}, \mathbf{x}_1)]$ in Subsection 1.1. Then with the property of cross-entropy loss that $\ell(z) \le \exp(-z)$ for all $z$, the test loss on the previous task $L_{\mathcal{D}_1}(\mathbf{W}^{(T_2,t_{end})}) \le 18\epsilon + \exp(-n_1^2)$ in Theorem 3.2 naturally follows. The complete proof is provided in Appendix E.3.

**Obtuse angle.** Due to the complex interaction between the signal vectors when $\cos\theta_{1,2} < 0$, a comprehensive analysis of neuron behavior is required to characterize the convergence and generalization of the current task $T_2$, rather than

studying a subset of neurons for a single task in Cao et al. (2022). Specifically, Cao et al. (2022) focus on neurons $\mathbf{w}_{j,r}$ for $r \in I_{j,1} = \{r \in [m] : \langle \mathbf{w}_{j,r}^{(T_1,0)}, j\boldsymbol{\mu}_1\rangle > 0\}$. These neurons induce a negative effect $\langle \mathbf{w}_{j,r}^{(T_2,0)}, -j\boldsymbol{\mu}_2\rangle$ during the initial phase of training task $T_2$. As training progresses, $\langle \mathbf{w}_{j,r}^{(T_2,t)}, -j\boldsymbol{\mu}_2\rangle$ gradually decays to 0, and the corresponding $\gamma(\boldsymbol{\mu}_2)_{j,r}^{(T_2,t_{end})}$ that characterizes signal learning for task $T_2$ is bounded by $-\gamma(\boldsymbol{\mu}_1)_{j,r}^{(T_1,t_{end})}\|\boldsymbol{\mu}_2\|_2 \cos\theta_{1,2}/\|\boldsymbol{\mu}_1\|_2$. In contrast, the neurons $\mathbf{w}_{j,r}$ for $r \in I_{j,2} = \{r \in [m] : \langle \mathbf{w}_{j,r}^{(T_1,0)}, j\boldsymbol{\mu}_1\rangle \le 0\} \cap \{r \in [m] : \langle \mathbf{w}_{j,r}^{(T_1,0)}, j\boldsymbol{\mu}_1^{\perp}\rangle > 0\}$, where $\boldsymbol{\mu}_1^{\perp} = \boldsymbol{\mu}_2 - \|\boldsymbol{\mu}_2\|_2 \cos\theta_{1,2} \cdot \boldsymbol{\mu}_1/\|\boldsymbol{\mu}_1\|_2$, actually learn the feature $\boldsymbol{\mu}_2$ and exhibit increasing inner products $\langle \mathbf{w}_{j,r}^{(T_2,t)}, j\boldsymbol{\mu}_2\rangle$. The learning of neurons $\mathbf{w}_{j,r}$ for $r \in I_{j,2}$ is key to achieving small training and test losses on task $T_2$ in Theorem 3.2. Note that while the two-stage signal learning for the tasks is similar, the neurons responsible for learning signals $\boldsymbol{\mu}_1$ and $\boldsymbol{\mu}_2$ are distinct. The complete proof and detailed explanation can be found in Appendix F.

By the definition of forgetting, $L_{\mathcal{D}_1}^{0-1}(\mathbf{W}^{(T_2,t_{end})}) = \mathbb{P}_{(\mathbf{x}_1,y_1)\sim\mathcal{D}_1}[y_1 \ne \text{sign}(f(\mathbf{W}^{(T_2,t_{end})}, \mathbf{x}_1))]$ in Subsection 1.1, we know that forgetting is closely related to the sign of $y_1 f(\mathbf{W}^{(T_2,t_{end})}, \mathbf{x}_1)$. Under the condition that the signal-to-noise ratio $n_k \cdot \text{SNR}_k^q = \widetilde{\Omega}(1)$, the noise memory $|\rho(\boldsymbol{\xi}_k)_{j,r,i}^{(T_k,t_{end})}|$ will be bounded by $\sigma_0\sigma_{p_k}\sqrt{d}$, which allows us to approximate $y_1 f(\mathbf{W}^{(T_2,t_{end})}, \mathbf{x}_1)$ as $\sum_{r=1}^{m}[\sigma(\langle \mathbf{w}_{y_1,r}^{(T_2,t_{end})}, y_1\boldsymbol{\mu}_1\rangle) - \sigma(\langle \mathbf{w}_{-y_1,r}^{(T_2,t_{end})}, y_1\boldsymbol{\mu}_1\rangle)]$. Through a comprehensive analysis of neuron learning, $\sum_{r=1}^{m}\sigma(\langle \mathbf{w}_{y_1,r}^{(T_2,t_{end})}, y_1\boldsymbol{\mu}_1\rangle)$ can be characterized by $\Theta(m\overline{\gamma(\boldsymbol{\mu}_1)_{y_1}}(1 - \cos^2\theta_{1,2})^q)$, where $\overline{\gamma(\boldsymbol{\mu}_1)_{y_1}} = \frac{1}{m}\sum_{r\in I_{j,1}}[\gamma(\boldsymbol{\mu}_1)_{j,r}^{(T_1,t_{end})}]^q$ for $j = y_1$ represents the average value of $\gamma(\boldsymbol{\mu}_1)_{y_1,r}^{(T_1,t_{end})}$ corresponding to the neurons that learn the signal $\boldsymbol{\mu}_1$. Similarly, $\sum_{r=1}^{m}\sigma(\langle \mathbf{w}_{-y_1,r}^{(T_2,t_{end})}, y_1\boldsymbol{\mu}_1\rangle)$ can be characterized by $\Theta(m\overline{\gamma(\boldsymbol{\mu}_2)_{-y_1}}\|\boldsymbol{\mu}_1\|_2^q(-\cos\theta_{1,2})^q/\|\boldsymbol{\mu}_2\|_2^q)$, where $\overline{\gamma(\boldsymbol{\mu}_2)_{-y_1}} = \frac{1}{m}\sum_{r\in I_{j,2}}[\gamma(\boldsymbol{\mu}_2)_{j,r}^{(T_2,t_{end})}]^q$ for $j = -y_1$ represents the average value of $\gamma(\boldsymbol{\mu}_2)_{-y_1,r}^{(T_2,t_{end})}$ corresponding to the neurons that learn the signal $\boldsymbol{\mu}_2$. Based on the above analysis, the study of forgetting can be transformed into the study of the antagonism between $\sum_{r=1}^{m}\sigma(\langle \mathbf{w}_{y_1,r}^{(T_2,t_{end})}, y_1\boldsymbol{\mu}_1\rangle)$ and $\sum_{r=1}^{m}\sigma(\langle \mathbf{w}_{-y_1,r}^{(T_2,t_{end})}, y_1\boldsymbol{\mu}_1\rangle)$, which is directly related to $\cos\theta_{1,2}$. Specifically, the relationship between $\sum_{r=1}^{m}[\sigma(\langle \mathbf{w}_{y_1,r}^{(T_2,t_{end})}, y_1\boldsymbol{\mu}_1\rangle) - \sigma(\langle \mathbf{w}_{-y_1,r}^{(T_2,t_{end})}, y_1\boldsymbol{\mu}_1\rangle)]$ and $\cos\theta_{1,2}$ can be characterized by the following lemma.

**Lemma 4.3.** *Under the same conditions as Theorem 3.2, when $-C_1 \le \cos\theta_{1,2} < 0$, it holds that*

$$\sum_{r=1}^{m}\big[\sigma(\langle \mathbf{w}_{y_1,r}^{(T_2,t_{end})}, y_1\boldsymbol{\mu}_1\rangle) - \sigma(\langle \mathbf{w}_{-y_1,r}^{(T_2,t_{end})}, y_1\boldsymbol{\mu}_1\rangle)\big] \ge C_3,$$

*where $C_1 < 1$ and $C_3$ are positive constants.*

In contrast, under the condition that the cosine similarity $\cos\theta_{1,2}$ is relatively small, the relationship between $\sum_{r=1}^{m}[\sigma(\langle \mathbf{w}_{-y_1,r}^{(T_2,t_{end})}, y_1\boldsymbol{\mu}_1\rangle) - \sigma(\langle \mathbf{w}_{y_1,r}^{(T_2,t_{end})}, y_1\boldsymbol{\mu}_1\rangle)]$ and $\cos\theta_{1,2}$ is described by the following lemma.

**Lemma 4.4.** *Under the same conditions as Theorem 3.2, when $-1 < \cos\theta_{1,2} \le -C_2$, it holds that*

$$\sum_{r=1}^{m}\big[\sigma(\langle \mathbf{w}_{-y_1,r}^{(T_2,t_{end})}, y_1\boldsymbol{\mu}_1\rangle) - \sigma(\langle \mathbf{w}_{y_1,r}^{(T_2,t_{end})}, y_1\boldsymbol{\mu}_1\rangle)\big] \ge C_3,$$

*where $C_1 < C_2 < 1$ and $C_3$ are positive constants.*

Using the antagonistic relationship in Lemma 4.3, along with the Gaussian concentration of Lipschitz function, we can directly derive an upper bound for benign forgetting (the second part of Theorem 3.2), following a similar idea to the proof of Theorem E.1 in Kou et al. (2023). Similarly, by Lemma 4.4, the lower bound for harmful forgetting (the third part of Theorem 3.2) naturally follows. The complete proof can be found in Appendix F.3.

### 4.2. Proof Sketch for Replay Methods

We also analyze the replay method using signal-noise decomposition and a two-stage approach. After applying signal-to-noise decomposition, we derive the inner product of network weights $\mathbf{w}_{j,r}^{(T_2,t)}$ and signal vector $\boldsymbol{\mu}_1$:

$$\langle \mathbf{w}_{j,r}^{(T_2,t)}, j\boldsymbol{\mu}_1\rangle = \langle \mathbf{w}_{j,r}^{(T_2,0)}, j\boldsymbol{\mu}_1\rangle + \gamma(\boldsymbol{\mu}_1)_{j,r}^{(T_2,t)}$$
$$+ \cos\theta_{1,2}\frac{\|\boldsymbol{\mu}_1\|_2}{\|\boldsymbol{\mu}_2\|_2}\gamma(\boldsymbol{\mu}_2)_{j,r}^{(T_2,t)}.$$

From this formula, it is evident that the inner product $\langle \mathbf{w}_{j,r}^{(T_2,t)}, j\boldsymbol{\mu}_1\rangle$ is influenced by two dynamic coefficients, $\gamma(\boldsymbol{\mu}_1)_{j,r}^{(T_2,t)}$ and $\gamma(\boldsymbol{\mu}_2)_{j,r}^{(T_2,t)}$. These coefficients evolve during neural network training, and analyzing the changes in the cross-entropy loss derivatives of task $T_1$ and task $T_2$ is complex. To simplify the derivative form, we first introduce the following lemma.

**Lemma 4.5.** *Under Condition 3.1, during task $T_1$ training, when $t = \tilde{\Theta}(1/\epsilon)$, we have*

$$\langle \mathbf{w}_{j,r}^{(T_1,t)}, j\boldsymbol{\mu}_1\rangle = o(\langle \mathbf{w}_{j,r^*}^{(T_1,t)}, j\boldsymbol{\mu}_1\rangle)$$

*for $j \in \{\pm1\}$ and $r \neq r^*$.*

$r^*$ represents the maximum value of all $\langle \mathbf{w}_{j,r}^{(T_1,t)}, j\boldsymbol{\mu}_1\rangle$, where $r \in [m]$. The proof of this lemma relies on the relationship between infinite series and infinite products, which is detailed in the Appendix G. By this lemma, we can show that: $\sum_{r=1}^{m}\sigma(\langle \mathbf{w}_{j,r}^{(T_1,t)}, j\boldsymbol{\mu}_1\rangle) = \sigma(\langle \mathbf{w}_{j,r^*}^{(T_1,t)}, j\boldsymbol{\mu}_1\rangle)[1 + o(1)]$, indicating that the sum of

inner products is dominated by the largest inner product. This allows us to simplify the derivative as follows: $-\ell_{1,i}^{\prime(T_1,t)} = 1/\exp\{\frac{1}{m}\sigma(\langle \mathbf{w}_{j,r^*}^{(T_1,t)}, j\boldsymbol{\mu}_1\rangle)\}[1 + o(1)]$. When task $T_1$ is trained until the training loss reaches $\epsilon$, we can estimate $\langle \mathbf{w}_{j,r^*}^{(T_1,t_{end})}, j\boldsymbol{\mu}_1\rangle$ by $(m\log(1/\epsilon))^{1/q}$. Consequently, the derivative of task $T_1$ becomes very small. Thus, when switching to task training $T_2$, neither the coefficients $\gamma(\boldsymbol{\mu}_1)_{j,r}^{(T_2,t)}$ nor $\rho(\boldsymbol{\mu}_1)_{j,r,i}^{(T_2,t)}$ of task $T_1$ will change significantly. During the initial phase of task $T_2$ training, the maximum inner product $\langle \mathbf{w}_{j,r^*}^{T_2,t}, j\boldsymbol{\mu}_1\rangle$ will initially decrease and then increase again, a phenomenon we prove in the Appendix F. Additionally, it is shown that the changes in $\langle \mathbf{w}_{j,r}^{(T_2,t)}, j\boldsymbol{\mu}_2\rangle$ are essentially similar to those in $\langle \mathbf{w}_{j,r}^{(T_1,t)}, j\boldsymbol{\mu}_1\rangle$, meaning that the largest $\langle \mathbf{w}_{j,r}^{(T_2,t)}, j\boldsymbol{\mu}_2\rangle$ is the higher order infinity of the other inner products. The overall analysis of $\langle \mathbf{w}_{j,r}^{(T_2,t)}, j\boldsymbol{\mu}_2\rangle$ is also based on the two-stage approach.

***Stage 1.*** When $\langle \mathbf{w}_{j,r^*}^{(T_2,t)}, j\boldsymbol{\mu}_2\rangle$ is large enough, similar to $-\ell_{1,i}^{\prime(T_1,t)}$, we can give a rough estimate of $-\ell_{2,i}^{\prime(T_2,t)}$: $-\ell_{2,i}^{\prime(T_2,t)} \approx 1/\exp\{\frac{1}{m}\sigma(\langle \mathbf{w}_{j,r^*}^{(T_2,t)}, j\boldsymbol{\mu}_2\rangle)\}$. Thus, when $\langle \mathbf{w}_{j,r^*}^{(T_2,t)}, j\boldsymbol{\mu}_2\rangle \le m^{\frac{1}{q}}$, we can assert that $-\ell_{2,i}^{\prime(T_2,t)} = \Theta(1)$. The following lemma gives our specification of stage 1:

**Lemma 4.6.** *Under the same condition as Theorem 3.3, there exists time*

$$T_4^+ = \frac{C(n_1^* + n_2)2^4 m \log\big(\frac{2m^{\frac{1}{q}}}{\sqrt{1-(\cos\theta_{1,2})^2}\sigma_0\|\boldsymbol{\mu}_2\|_2}\big)}{\eta_2 q n_2 (1-(\cos\theta_{1,2})^2)^{q/2}\sigma_0^{q-2}\|\boldsymbol{\mu}_2\|_2^q}$$

*such that*

- $\max_r \gamma(\boldsymbol{\mu}_2)_{j,r}^{(T_2,t)} \ge \frac{m^{\frac{1}{q}}}{1-(cos\theta_{1,2})^2}$ *for $j \in \{\pm1\}$.*

- $\max_{j,r,i}|\rho(\boldsymbol{\mu}_k)_{j,r,i_k}^{(T_2,t)}| \le \sigma_0\sigma_{p_k}\sqrt{d}$ *for all $j \in \{\pm1\}$, $r \in [m]$, $i_k \in [n_k]$ and $0 \le t \le T_4^+$.*

Lemma 4.6 gives a significant difference between $\gamma(\boldsymbol{\mu}_2)_{j,r^*}^{(T_2,t)}$ and $\rho(\boldsymbol{\mu}_k)_{j,r,i_k}^{(T_2,t)}$ within time $T_4^+$, during which $\gamma(\boldsymbol{\mu}_2)_{j,r^*}^{(T_2,t)}$ reaches $\frac{m^{\frac{1}{q}}}{1-(cos\theta_{1,2})^2}$, while $|\rho(\boldsymbol{\mu}_k)_{j,r,i_k}^{(T_2,t)}|$ is upper bounded by $\sigma_0\sigma_{p_k}\sqrt{d}$. After this, as $\gamma(\boldsymbol{\mu}_2)_{j,r^*}^{(T_2,t)}$ grows, the derivative $-\ell_{2,i}^{\prime(T_2,t)}$ cannot be maintained at a constant level so that the training for task $T_2$ enters stage 2.

***Stage 2.*** At this stage, $\gamma(\boldsymbol{\mu}_2)_{j,r^*}^{(T_2,t)}$ reaches a higher value, but $\rho(\boldsymbol{\mu}_k)_{j,r,i_k}^{(t)}$ is still very different from $\gamma(\boldsymbol{\mu}_2)_{j,r^*}^{(T_2,t)}$. Specifically, $\gamma(\boldsymbol{\mu}_2)_{j,r^*}^{(T_2,t^*)}$ reaches a large and reasonable value $\frac{k(m\log(\frac{1}{\epsilon}))^{\frac{1}{q}}}{1-(cos\theta_{1,2})^2}$, where $k$ is a positive constant, and $\rho(\boldsymbol{\mu}_k)_{j,r,i_k}^{(t)}$ can be bounded by $2\sigma_0\sigma_{p_k}\sqrt{d}$. The following lemma gives some specifications.

**Lemma 4.7.** *Under Condition 3.1, let* $T_5^+ = T_4^+ + \widetilde{\Theta}\big(m^{\frac{2}{q}}\eta_2^{-1}\epsilon^{-k^q}\big)$. *Then we have* $\max_{j,r,i} |\rho(\boldsymbol{\mu}_k)_{j,r,i_k}^{(t)}| \leq 2\sigma_0\sigma_{p_k}\sqrt{d}$ *for all* $T_4^+ \leq t \leq T_5^+$. *And we can also find a time* $t^*$ *that* $\gamma(\boldsymbol{\mu}_2)_{j,r^*}^{(T_2,t^*)} \geq \frac{k(m\log(\frac{1}{\epsilon}))^{\frac{1}{q}}}{1-(\cos\theta_{1,2})^2}$, *where* $T_4^+ \leq t^* \leq T_5^+$.

In Lemma 4.7, $k$ is a parameter that measures feature learning on task $T_2$. $|\rho(\boldsymbol{\mu}_k)_{j,r,i_k}^{(t)}|$ can be bounded by $2\sigma_0\sigma_{p_k}\sqrt{d}$. This suggests that the model is far inferior to feature learning in terms of noise memorization. The time to stop training for task $T_2$ is the moment when $\gamma(\boldsymbol{\mu}_2)_{j,r^*}^{(T_2,t^*)}$ reaches $\frac{k(m\log(\frac{1}{\epsilon}))^{\frac{1}{q}}}{1-(\cos\theta_{1,2})^2}$ for the first time, which we denote by $t_{end}$. At the moment $t_{end}$, loss of training for task $T_2$ will be reduced to a minimal level: $L_{S_2}(\mathbf{W}^{(T_2,t_{end})}) \leq \epsilon^{k^q/2}$. Further, we will prove in the Appendix H.5 that when $-\frac{1+C_2}{2} \leq \cos\theta_{1,2} \leq 0$, $\sum_{r=1}^{m}\big[\sigma(\langle\mathbf{w}_{y_1,r}^{(T_2,t_{end})}, y_1\boldsymbol{\mu}_1\rangle) - \sigma(\langle\mathbf{w}_{-y_1,r}^{(T_2,t_{end})}, y_1\boldsymbol{\mu}_1\rangle)\big] \geq C_3$.

It should be clear that our proof for the replay method is based on the fact that $\langle\mathbf{w}_{j,r}^{(T_2,t)}, \boldsymbol{\xi}_{1,i}\rangle$ and $\langle\mathbf{w}_{j,r}^{(T_2,t)}, \boldsymbol{\xi}_{2,i}\rangle$ can be bounded. The following lemma will give this fact.

**Lemma 4.8.** *Under Condition 3.1, when* $0 \leq t \leq T_5^+ = \widetilde{\Theta}(\eta_2^{-1}m\sigma_0^{2-q}\|\boldsymbol{\mu}_2\|_2^{-q} + m^{\frac{2}{q}}\eta_2^{-1}\epsilon^{-k^q})$, *we have*

$$\langle\mathbf{w}_{j,r}^{(T_2,t)}, \boldsymbol{\xi}_{1,i}\rangle, \langle\mathbf{w}_{j,r}^{(T_2,t)}, \boldsymbol{\xi}_{2,i}\rangle \leq C_{\boldsymbol{\xi}},$$

*where* $C_{\boldsymbol{\xi}} = \widetilde{O}(m^{-[2/(q-2)]\vee 1} \cdot \max\{n_k\}^{-[1/(q-2)]\vee 1})$ *for all* $j \in \{\pm 1\}$ *and* $r \in [m]$.

In this case, we complete the proof of the replay method.

## 5. Experiments

**Synthetic experiments**  We validate theoretical findings with synthetic data as per Definition 1.1, setting training data size to $n_k = 100$ for $k \in [2]$ and dimension $d = 100$. Signal vector $\boldsymbol{\mu}_1$ for task $T_1$ is $\|\boldsymbol{\mu}_1\|_2 \cdot [1,0,\cdots,0]^\top$ with $\|\boldsymbol{\mu}_1\|_2 = 8$, and $\boldsymbol{\mu}_2$ for $T_2$ is obtained by rotating $\boldsymbol{\mu}_1$ by angle $\theta_{1,2}$. Noise variance is $\sigma_{p_k} = 1$ for $k \in [2]$.

We sequentially train a two-layer ReLU$^q$ CNN defined in Section 2 on the datasets of the two tasks described above, where width $m = 10$ and $q = 3$. We initialize the model with a Gaussian distribution with $\sigma_0 = 0.02$ and train it using full-batch gradient descent with a learning rate of 0.001 for 3000 iterations on each task.

After training on two tasks, we estimate forgetting (test error on the previous task $T_1$) using 1000 test data points. In Figure 2, we see that acute or small obtuse angles are associated with benign forgetting, while large obtuse angles lead to harmful forgetting. Moreover, the replay method alleviates forgetting by expanding the range of angles corresponding

to benign forgetting. These results are attributed to a comprehensive analysis of neuron behavior. The second plot of Figure 2 verifies that $\langle\mathbf{w}_{j,r}^{(T_1,t)}, j\boldsymbol{\mu}_1\rangle = o(\langle\mathbf{w}_{j,r^*}^{(T_1,t)}, j\boldsymbol{\mu}_1\rangle)$ holds for $j \in \{\pm 1\}$ and $r \neq r^*$ as $t$ approaches $\widetilde{\Theta}(1/\epsilon)$ in Lemma 4.5 during the training of task $T_1$, where $r^* = \arg\max_{r\in[m]}\langle\mathbf{w}_{j,r}^{(T_1,t)}, j\boldsymbol{\mu}_1\rangle$. The neuron behavior during task $T_2$ in the case of obtuse angle can be verified by the third plot of Figure 2. The descending curve corresponds to neurons $\mathbf{w}_{j,r}$ for $r \in I_{j,1}$, which learn the signal $\boldsymbol{\mu}_1$ of task $T_1$. These neurons initially induce a negative effect $\langle\mathbf{w}_{j,r}^{(T_2,0)}, -j\boldsymbol{\mu}_2\rangle$, which gradually decays to 0. The ascending curve corresponds to neurons $\mathbf{w}_{j,r}$ for $r \in I_{j,2}$, which learn the signal $\boldsymbol{\mu}_2$ of task $T_2$ and exhibit an increasing $\langle\mathbf{w}_{j,r}^{(T_2,t)}, j\boldsymbol{\mu}_2\rangle$. The sets $I_{j,1}$ and $I_{j,2}$ are defined as in Subsection 4.1. The relationship between forgetting and the angle $\theta_{1,2}$ arises from the antagonism between these two types of neurons.

**Real-world experiments**  We validate our conclusions on the MNIST dataset (LeCun, 1998) using a two-layer ReLU$^q$ CNN for binary classification with labels $+1$ and $-1$ (See Section 1.1). We denote the average flattened image of all positive labeled examples (label $+1$) in task $T_1$ as the signal vector $\boldsymbol{\mu}_1$. Similarly, we obtain $\boldsymbol{\mu}_2$ for task $T_2$. Then the angle $\theta_{1,2}$ between the two signal vectors naturally follows. We conduct the following four sets of experiments:

1. **Experiment 1**: Task $T_1$: classification of the original digit 5 (label $+1$) and its inverted version (label $-1$). Task $T_2$: classification of the original digit 8 (label $+1$) and its inverted version (label $-1$). Then the angle between two tasks $\theta_{1,2} = 30.19°$.

2. **Experiment 2**: Task $T_1$: classification of the original digit 0 (label $+1$) and its inverted version (label $-1$). Task $T_2$: classification of the original digit 1 (label $-1$) and its inverted version (label $+1$). Then the angle between two tasks $\theta_{1,2} = 102.97°$.

3. **Experiment 3**: Task $T_1$: classification of the original digit 2 (label $+1$) and its inverted version (label $-1$). Task $T_2$: classification of the original digit 6 (label $-1$) and its inverted version (label $+1$). Then the angle between two tasks $\theta_{1,2} = 140.26°$.

4. **Experiment 4**: Task $T_1$: classification of the original digit 4 (label $+1$) and its inverted version (label $-1$). Task $T_2$: classification of the original digit 9 (label $-1$) and its inverted version (label $+1$). Then the angle between two tasks $\theta_{1,2} = 153.90°$.

Figure 3 illustrates the relationship between test accuracy on task $T_1$ and the replay buffer size across four settings. For tasks with acute angles (30.19°) or smaller obtuse angles

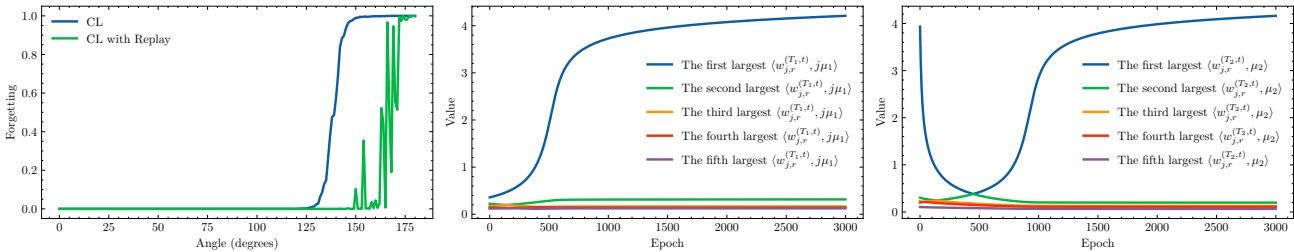

*Figure 2.* validation of the relationship between forgetting and the angle $\theta_{1,2}$ between the signal vectors of two tasks under both CL and CL with replay in Section 3, and we provide experimental support for the key analysis of neuron behavior discussed in Section 4.

(102.97°), the CNN exhibits benign forgetting even without replay, supporting Theorem 3.2. In contrast, for larger obtuse angles (140.26° and 153.90°), harmful forgetting occurs unless sufficient replay data is provided. When the replay buffer size reaches a threshold, forgetting is significantly alleviated, aligning with Theorem 3.3. These results highlight the angle-dependent effectiveness of replay in mitigating forgetting.

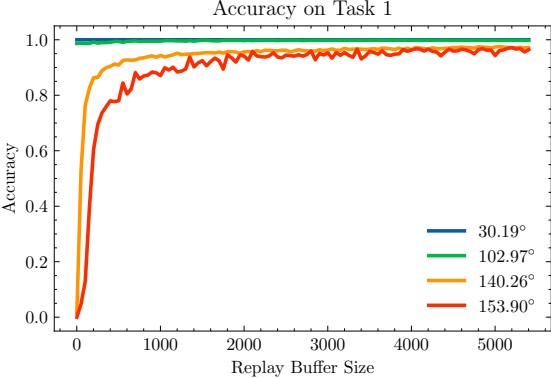

*Figure 3.* Experimental results of CL on the MNIST dataset with varying angles between task signal vectors: reported are accuracies on task $T_1$ for different replay buffer sizes.

**Mid-angle sampling:** Inspired by our theoretical results, we can leverage the angle $\theta_{1,2}$ to balance the stability and plasticity. Instead of random sampling old task examples, we select examples that have moderate cosine similarity (mid-angle) with the prototype for each class, which we refer as mid-angle sampling. Examples with high or low cosine similarity typically correspond to higher or lower levels of forgetting during new task training respectively. In contrast, mid-angle sampling selects examples with a moderate degree of forgetting, which proves to be a more cost-effective approach.

We conduct an initial validation of the effectiveness of mid-angle sampling using a two-layer CNN on the MNIST dataset, which can be found in Appendix B.1. To further

*Table 1.* Accuracy comparison on previously trained classes using different replay buffer sampling methods within the iCaRL framework on the CIFAR100 dataset (percentage omitted).

| Sampling | CIFAR100-10 | CIFAR100-5 |
|---|---|---|
| Random | $47.17 \pm 0.45$ | $56.08 \pm 0.12$ |
| Small-angle | $45.63 \pm 0.12$ | $54.36 \pm 0.35$ |
| **Mid-angle** | $\mathbf{48.02 \pm 0.27}$ | $\mathbf{56.51 \pm 0.06}$ |
| Big-angle | $45.34 \pm 0.76$ | $54.77 \pm 0.29$ |
| Herding | $47.40 \pm 0.17$ | $56.12 \pm 0.20$ |

validate the universality of mid-angle sampling, we conduct experiments on the CIFAR100 dataset (Krizhevsky et al., 2009) following the iCaRL replay framework from Rebuffi et al. (2017). The data is split into $T$ tasks, denoted as dataset-T. For example, CIFAR100-10 represents 10 tasks, each containing 10 distinct classes. Specifically, we train a model $f$ consisting of a feature extractor $\varphi$ and a classification layer $e$. The prototype of the $j$-th class is defined as $\boldsymbol{\mu}_j = \frac{1}{|S_j|} \Sigma_{x_{j,i} \in S_j} \varphi(x_{j,i})$, where $S_j$ represents the dataset of class $j$. Based on the prototype, mid-angle sampling, big-angle sampling and small-angle sampling for class $j$ can be performed according to $\text{sim}(\varphi(x_{j,i}), \boldsymbol{\mu}_j)$, where $\text{sim}(\cdot)$ denotes the cosine similarity function. Further details can be found in Appendix B.2. Table 1 shows the test accuracy on previously trained classes for the three above sampling methods, random sampling, and herding. In addition, we include in Table 2 the average forgetting metric used by Wang et al. (2024) on previously learned tasks to further validate the effectiveness of mid-angle sampling. As shown, mid-angle sampling outperforms the other sampling methods, aligning with our theoretical predictions.

## 6. Conclusion and Future Work

This paper establishes a unified theoretical framework for understanding CL through feature learning theory. By analyzing neuron behavior, we reveal the relationship between forgetting and the angle between task signal vectors. Furthermore, we demonstrate that the replay method alleviates

*Table 2.* Average forgetting comparison on previously learned tasks using different replay buffer sampling methods within the iCaRL framework on the CIFAR100 dataset (percentage omitted).

| Sampling | CIFAR100-10 | CIFAR100-5 |
|---|---|---|
| Random | $15.72 \pm 0.31$ | $11.15 \pm 0.46$ |
| Small-angle | $19.47 \pm 0.39$ | $14.10 \pm 0.26$ |
| **Mid-angle** | $\mathbf{14.84 \pm 0.26}$ | $\mathbf{10.15 \pm 0.28}$ |
| Big-angle | $18.04 \pm 0.49$ | $12.50 \pm 0.72$ |
| Herding | $15.51 \pm 0.08$ | $10.64 \pm 0.13$ |

forgetting by expanding the range of angles associated with benign forgetting. Our theoretical results also inspire a mid-angle sampling strategy, which can help the replay method better mitigate forgetting. As a pioneering study in feature learning within the context of CL, our theoretical framework is limited to binary classification under a single-head setting. Moreover, the contribution of mid-angle sampling is relatively modest. Future work can extend our framework to more complex task settings and explore more effective replay methods guided by theoretical insights into how the angle between task signal vectors influences forgetting.

## Acknowledgements

Miao Zhang was partially sponsored by the National Natural Science Foundation of China under Grant 62306084 and U23B2051, and Shenzhen College Stability Support Plan under Grant GXWD20231128102243003 and Grant ZDSYS20230626091203008.

## Impact Statement

This paper presents work whose goal is to advance the field of Machine Learning. There are many potential societal consequences of our work, none which we feel must be specifically highlighted here.

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

# A. Related Work

## A.1. Feature Learning Theory

Unlike some of the more traditional feature extraction methods, such as recursive feature elimination (Darst et al., 2018), which typically require the knowledge and experience of domain experts to guide the selection and construction of features, feature learning theory (Cao et al., 2022) reveals an efficient method to automatically extract useful features from raw data. It aims to find feature representations that better represent the data by mining the intrinsic structure of the data. Feature learning theory has a wide range of applications. Cao et al. (2022) and Kou et al. (2023) address the phenomenon of benign overfitting that may occur in modern deep learning models, and study benign overfitting in two-layer ReLU convolutional neural networks through a feature learning theoretical approach to reveal changes under different conditions; Huang et al. (2023) and Pan et al. (2024) establish a unified theoretical foundation for understanding federated learning through feature learning theory, revealing that federated averaging improves the signal-to-noise ratio to achieve near-zero test error and weighted federated averaging to solve the problem of data heterogeneity in feature learning; Yang & Hu (2021) point out that the standard and NTK parameterisations of neural networks cannot reach the infinite-width limit at which feature learning can take place, and propose a simple modification of the standard parameterisation to achieve feature learning at the limit, which is found to outperform the neural tangent kernel baseline and finite-width networks, and the finite-width networks converge to the infinite-width feature learning performance as the width increases; Zou et al. (2023) analyse Mixup theoretically from the perspective of feature learning, and construct a feature-noise data model to carry out the study, which reveals that Mixup training can effectively learn rare features from a mixture of rare and common features, while standard training cannot do so, which in turn leads to the problem of poor generalisation performance; Jelassi et al. (2022); Jiang et al. (2024) conduct in-depth research on Vision Transformers (ViTs). Given that ViTs can achieve comparable or even better performance than Convolutional Neural Networks (CNNs) in computer vision without incorporating the visual inductive bias of spatial locality and can learn spatially localized patterns, the authors aim to provide theoretical explanations for this phenomenon. In contrast, our research employs feature learning theory to investigate how the angle between task signal vectors relates to benign and harmful forgetting in continual learning (CL), introducing additional techniques designed for CL. Furthermore, it reveals the theoretical mechanism underlying replay-based methods, which mitigate forgetting by expanding the angular range associated with benign forgetting.

## A.2. Theoretical Analysis of CL

Since McCloskey & Cohen (1989) first put forward the phenomenon of catastrophic forgetting in continual learning in 1989, researchers have been committed to finding effective methods to balance the learning plasticity of models on new tasks and the catastrophic forgetting of old tasks. In recent years, researchers have started from the theory of continual learning to conduct theoretical interpretations of continual learning and explore new methods. Bennani et al. (2020) proposed that in over-parameterised neural networks, the OGD algorithm can effectively solve the catastrophic forgetting problem in continuous learning, and also revealed the need to take into account the effect of NTK variations on the algorithm's performance in practical applications. Doan et al. (2021) proposes a task similarity metric based on the neural tangent kernel (NTK) overlap matrix, analyses how common projected gradient algorithms can mitigate catastrophic forgetting, and proposes a variant of Orthogonal Gradient Descent (OGD) using Principal Component Analysis (PCA) structured data, with experimental results supporting the theoretical findings and demonstrating the approach's forgetting potential. Evron et al. (2022) studies the problem of catastrophic forgetting when fitting over-parameterised linear models to cope with sequences of tasks with different input distributions, analysing the extent of forgetting, establishing relevant domain links and demonstrating a situation-specific upper bound on forgetting and pointing out how it varies when the tasks are differently ordered. Lin et al. (2023) conducts a theoretical analysis based on an over-parameterised linear model, give explicit forms of the expected forgetting and generalisation errors under a general CL setting, analyse the effects of relevant factors on CL forgetting and generalisation errors, and show that some of the insights can be used in practice and can contribute to the design of CL algorithms through experiments with deep neural networks on real datasets. Ding et al. (2024) provides a generalised theoretical analysis of forgetting in linear regression models via stochastic gradient descent (SGD), revealing interesting insights into the relationship between task sequences and algorithm parameters, showing the impact of task alignment and the choice of an appropriate step size on forgetting in large data volumes. Zhao et al. (2024) presents a statistical analysis of continuous learning in a sequence of regularisation-based linear regression tasks, compares the minimum-paradigm estimator with the lower bound of continuous ridge regression to reflect its non-optimality, and concludes with an experimental validation of its theory.

# B. Supplements for Experiments

## B.1. Mid-angle Sampling with a Two-layer CNN on MNIST

We conduct experiments to validate the effectiveness of mid-angle sampling using a two-layer ReLU$^q$ CNN with width $m = 256$ and $q = 3$ on the MNIST dataset (LeCun, 1998). We define two tasks, where task $T_1$ involves classifying digits 4 and 5, and task $T_2$ involves classifying digits 8 and 9. We denote the flattened image of the $i$-th example in the $j$-th class as $\boldsymbol{\mu}_{j,i}$, and the prototype of the $j$-th class is represented as $\boldsymbol{\mu}_j = \frac{1}{|S_j|}\Sigma_{x_{j,i} \in S_j}\boldsymbol{\mu}_{j,i}$, where $S_j$ is the dataset of class $j$. We perform sampling based on the cosine similarity $\text{sim}(\boldsymbol{\mu}_{j,i}, \boldsymbol{\mu}_j)$, where $\text{sim}(\cdot)$ denotes the cosine similarity function. Examples with moderate similarity $\text{sim}(\boldsymbol{\mu}_{j,i}, \boldsymbol{\mu}_j)$ are selected for mid-angle sampling, while examples with higher or lower similarity correspond to small-angle sampling and big-angle sampling respectively. We compare the three sampling methods above with random sampling in terms of the test accuracy on the previous task $T_1$ in Figure 4. As shown, mid-angle sampling outperforms the other three methods, which is consistent with our analysis.

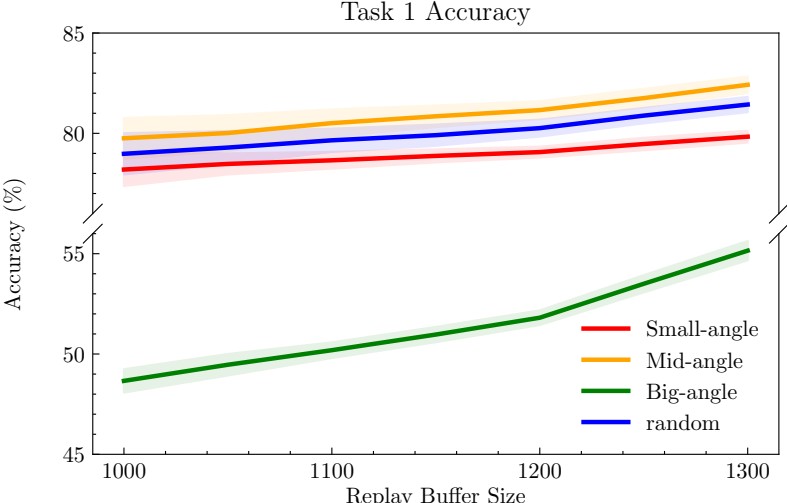

*Figure 4.* Experimental results on the MNIST dataset using different replay buffer sampling methods: reported are accuracies on task $T_1$ for various replay buffer sizes.

## B.2. Experimental Details of Mid-angle Sampling on CIFAR-100

We evaluate five replay buffer sampling methods within the iCaRL framework (Rebuffi et al., 2017): small-angle, mid-angle, big-angle, random sampling and herding. We rely on PyTorch (Paszke et al., 2019) and train an 18-layers CBAM-ResNet (He et al., 2016; Woo et al., 2018) as the feature extractor $\varphi$, allowing iCaRL to store up to $K = 5000$ exemplars. Each task is trained for 100 epochs, with the learning rate starting at 2.0 and divided by 5 after 48, 62 and 80 epochs. The network is trained via standard backpropagation with minibatches of size 128 and a weight decay of 0.00001.

# C. Preliminary Lemmas

**Lemma C.1** (Cao et al. (2022)). *Suppose that $\delta > 0$ and $d = \Omega(\log(4n_k/\delta))$. Then with probability at least $1 - \delta$,*

$$\sigma_{p_k}^2 d/2 \le \|\boldsymbol{\xi}_{k,i}\|_2^2 \le 3\sigma_{p_k}^2 d/2,$$

$$|\langle \boldsymbol{\xi}_{k,i}, \boldsymbol{\xi}_{k',i'} \rangle| \le 2\sigma_{p_k}\sigma_{p_{k'}} \cdot \sqrt{d\log(4n_k^2/\delta)}$$

*for $i \in [n_k], i' \in [n_{k'}]$ and $k, k' \in [2]$.*

**Lemma C.2** (Cao et al. (2022)). *Suppose that $d \ge \Omega(\log(mn_k/\delta))$, $m = \Omega(\log(1/\delta))$. Then with probability at least $1 - \delta$,*

$$|\langle \mathbf{w}_{j,r}^{(T_1,0)}, \boldsymbol{\mu}_k \rangle| \le \sqrt{2\log(8m/\delta)} \cdot \sigma_0\|\boldsymbol{\mu}_k\|_2,$$

$$|\langle \mathbf{w}_{j,r}^{(T_1,0)}, \boldsymbol{\xi}_{k,i} \rangle| \leq 2\sqrt{\log(8mn_k/\delta)} \cdot \sigma_0 \sigma_{p_k} \sqrt{d}$$

*for all $r \in [m]$, $j \in \{\pm 1\}$ and $i \in [n_k]$. Moreover,*

$$\sigma_0 \|\boldsymbol{\mu}_k\|_2/2 \leq \max_{r \in [m]} j \cdot \langle \mathbf{w}_{j,r}^{(T_1,0)}, \boldsymbol{\mu}_k \rangle \leq \sqrt{2\log(8m/\delta)} \cdot \sigma_0 \|\boldsymbol{\mu}_k\|_2,$$

$$\sigma_0 \sigma_{p_k} \sqrt{d}/4 \leq \max_{r \in [m]} j \cdot \langle \mathbf{w}_{j,r}^{(T_1,0)}, \boldsymbol{\xi}_{k,i} \rangle \leq 2\sqrt{\log(8mn_k/\delta)} \cdot \sigma_0 \sigma_{p_k} \sqrt{d}$$

*for all $j \in \{\pm 1\}$ and $i \in [n_k]$.*

## D. Signal-noise Decomposition for Task T1

Based on the analysis of training task $T_1$ by Cao et al. (2022) we know that

$$\mathbf{w}_{j,r}^{(T_2,0)} = \mathbf{w}_{j,r}^{(T_1,t_{end})} = \mathbf{w}_{j,r}^{(T_1,0)} + j \cdot \gamma(\boldsymbol{\mu}_1)_{j,r}^{(T_1,t_{end})} \cdot \frac{\boldsymbol{\mu}_1}{\|\boldsymbol{\mu}_1\|_2^2} + \sum_{i=1}^{n_1} \overline{\rho}(\boldsymbol{\xi}_1)_{j,r,i}^{(T_1,t_{end})} \cdot \frac{\boldsymbol{\xi}_{1,i}}{\|\boldsymbol{\xi}_{1,i}\|_2^2} + \sum_{i=1}^{n_1} \underline{\rho}(\boldsymbol{\xi}_1)_{j,r,i}^{(T_1,t_{end})} \cdot \frac{\boldsymbol{\xi}_{1,i}}{\|\boldsymbol{\xi}_{1,i}\|_2^2}. \tag{D.1}$$

And if $n_1 \cdot \mathrm{SNR}_1^q = \widetilde{\Omega}(1)$, the learned neural network can achieve small training and test losses. So at the beginning of training task $T_2$, we have following property holds:

- $\max_r \gamma(\boldsymbol{\mu}_1)_{j,r}^{(T_1,t_{end})} = O([m\log(1/\epsilon)]^{1/q}), \forall j \in \{\pm 1\}$.

- $\max_{j,r,i} |\rho(\boldsymbol{\xi}_1)_{j,r,i}^{(T_1,t_{end})}| \leq \sigma_0 \sigma_{p_1} \sqrt{d}$.

We also know that

$$\langle \mathbf{w}_{j,r}^{(T_2,0)}, \boldsymbol{\mu}_2 \rangle = \langle \mathbf{w}_{j,r}^{(T_1,0)}, \boldsymbol{\mu}_2 \rangle + j \cdot \gamma(\boldsymbol{\mu}_1)_{j,r}^{(T_1,t_{end})} \cdot \frac{\|\boldsymbol{\mu}_2\|_2 \cos\theta_{1,2}}{\|\boldsymbol{\mu}_1\|_2}$$

$$\langle \mathbf{w}_{j,r}^{(T_2,0)}, \boldsymbol{\xi}_{2,i} \rangle = \langle \mathbf{w}_{j,r}^{(T_1,0)}, \boldsymbol{\xi}_{2,i} \rangle + \sum_{i'=1}^{n_1} \rho(\boldsymbol{\xi}_1)_{j,r,i}^{(T_1,t_{end})} \cdot \frac{\langle \boldsymbol{\xi}_{1,i'}, \boldsymbol{\xi}_{2,i} \rangle}{\|\boldsymbol{\xi}_{1,i'}\|_2^2}$$

By Condition 3.1, Lemma C.1 and C.2, we further have

$$\max_r |\langle \mathbf{w}_{j,r}^{(T_2,0)}, \boldsymbol{\mu}_2 \rangle| = O([m\log(1/\epsilon)]^{1/q}), \forall j \in \{\pm 1\} \tag{D.2}$$

$$\max_{j,r,i} |\langle \mathbf{w}_{j,r}^{(T_2,0)}, \boldsymbol{\xi}_{2,i} \rangle| \leq 4\sqrt{\log(8mn_2/\delta)} \cdot \sigma_0 \sigma_{p_2} \sqrt{d} \tag{D.3}$$

## E. Learning of Task T2 (Standard CL with Acute Angle)

In this section, we consider continual learning where the signal vectors between task $T_1$ and task $T_2$ form an acute angle, i.e., $\langle \boldsymbol{\mu}_1, \boldsymbol{\mu}_2 \rangle > 0$. Denote $\cos\theta_{1,2} = \langle \boldsymbol{\mu}_1, \boldsymbol{\mu}_2 \rangle/(\|\boldsymbol{\mu}_1\|_2 \cdot \|\boldsymbol{\mu}_2\|_2)$. Then we have that $\cos\theta_{1,2} > 0$. We remind the readers that the proofs in this section are based on the results in Section D, which can achieve small training loss and test loss during learning task $T_1$.

### E.1. Signal-noise Decomposition Analysis

**Lemma E.1** (Restatement of Lemma 4.1). *The coefficients $\gamma(\boldsymbol{\mu}_2)_{j,r}^{(T_2,t)}, \overline{\rho}(\boldsymbol{\xi}_2)_{j,r,i}^{(T_2,t)}, \underline{\rho}(\boldsymbol{\xi}_2)_{j,r,i}^{(T_2,t)}$ in decomposition (4.1) satisfy the following iterative equations:*

$$\gamma(\boldsymbol{\mu}_2)_{j,r}^{(T_2,0)}, \overline{\rho}(\boldsymbol{\xi}_2)_{j,r,i}^{(T_2,0)}, \underline{\rho}(\boldsymbol{\xi}_2)_{j,r,i}^{(T_2,0)} = 0,$$

$$\gamma(\boldsymbol{\mu}_2)_{j,r}^{(T_2,t+1)} = \gamma(\boldsymbol{\mu}_2)_{j,r}^{(T_2,t)} - \frac{\eta_2}{n_2 m} \cdot \sum_{i=1}^{n_2} \ell_{2,i}'^{(T_2,t)} \cdot \sigma'(\langle \mathbf{w}_{j,r}^{(T_2,t)}, y_{2,i} \cdot \boldsymbol{\mu}_2 \rangle) \cdot \|\boldsymbol{\mu}_2\|_2^2,$$

$$\overline{\rho}(\boldsymbol{\xi}_2)_{j,r,i}^{(T_2,t+1)} = \overline{\rho}(\boldsymbol{\xi}_2)_{j,r,i}^{(T_2,t)} - \frac{\eta_2}{n_2 m} \cdot \ell_{2,i}'^{(T_2,t)} \cdot \sigma'(\langle \mathbf{w}_{j,r}^{(T_2,t)}, \boldsymbol{\xi}_{2,i}\rangle) \cdot \|\boldsymbol{\xi}_{2,i}\|_2^2 \cdot \mathbb{1}(y_{2,i} = j),$$

$$\underline{\rho}(\boldsymbol{\xi}_2)_{j,r,i}^{(T_2,t+1)} = \underline{\rho}(\boldsymbol{\xi}_2)_{j,r,i}^{(T_2,t)} + \frac{\eta_2}{n_2 m} \cdot \ell_{2,i}'^{(T_2,t)} \cdot \sigma'(\langle \mathbf{w}_{j,r}^{(T_2,t)}, \boldsymbol{\xi}_{2,i}\rangle) \cdot \|\boldsymbol{\xi}_{2,i}\|_2^2 \cdot \mathbb{1}(y_{2,i} = -j)$$

*for all $r \in [m]$, $j \in \{\pm 1\}$ and $i \in [n]$.*

*Proof of Lemma E.1.* First, we iterate the gradient descent update rule (2.1) $t$ times and get

$$\mathbf{w}_{j,r}^{(T_2,t)} = \mathbf{w}_{j,r}^{(T_2,0)} - \frac{\eta_2}{n_2 m} \sum_{s=0}^{t-1} \sum_{i=1}^{n_2} \ell_{2,i}'^{(T_2,s)} \cdot \sigma'(\langle \mathbf{w}_{j,r}^{(T_2,s)}, \boldsymbol{\xi}_{2,i}\rangle) \cdot j y_{2,i} \boldsymbol{\xi}_{2,i}$$

$$- \frac{\eta_2}{n_2 m} \sum_{s=0}^{t-1} \sum_{i=1}^{n_2} \ell_{2,i}'^{(T_2,s)} \cdot \sigma'(\langle \mathbf{w}_{j,r}^{(T_2,s)}, y_{2,i} \boldsymbol{\mu}_2\rangle) \cdot j \boldsymbol{\mu}_2.$$

According to the decomposition (4.1),

$$\mathbf{w}_{j,r}^{(T_2,t)} = \mathbf{w}_{j,r}^{(T_2,0)} + j \cdot \gamma(\boldsymbol{\mu}_2)_{j,r}^{(T_2,t)} \cdot \|\boldsymbol{\mu}_2\|_2^{-2} \cdot \boldsymbol{\mu}_2 + \sum_{i=1}^{n_2} \rho(\boldsymbol{\xi}_2)_{j,r,i}^{(T_2,t)} \cdot \|\boldsymbol{\xi}_{2,i}\|_2^{-2} \cdot \boldsymbol{\xi}_{2,i}.$$

Note that the vectors are linearly independent with probability 1, under which condition we have the unique representation

$$\gamma(\boldsymbol{\mu}_2)_{j,r}^{(T_2,t)} = -\frac{\eta_2}{n_2 m} \sum_{s=0}^{t-1} \sum_{i=1}^{n_2} \ell_{2,i}'^{(T_2,s)} \cdot \sigma'(\langle \mathbf{w}_{j,r}^{(T_2,s)}, y_{2,i} \boldsymbol{\mu}_2\rangle) \cdot \|\boldsymbol{\mu}_2\|_2^2, \tag{E.1}$$

$$\rho(\boldsymbol{\xi}_2)_{j,r,i}^{(T_2,t)} = -\frac{\eta_2}{n_2 m} \sum_{s=0}^{t-1} \ell_{2,i}'^{(T_2,s)} \cdot \sigma'(\langle \mathbf{w}_{j,r}^{(T_2,s)}, \boldsymbol{\xi}_{2,i}\rangle) \cdot \|\boldsymbol{\xi}_{2,i}\|_2^2 \cdot j y_{2,i}.$$

Now with the notation $\overline{\rho}(\boldsymbol{\xi}_2)_{j,r,i}^{(T_2,t)} := \rho(\boldsymbol{\xi}_2)_{j,r,i}^{(T_2,t)} \mathbb{1}(\rho(\boldsymbol{\xi}_2)_{j,r,i}^{(T_2,t)} \geq 0)$, $\underline{\rho}(\boldsymbol{\xi}_2)_{j,r,i}^{(T_2,t)} := \rho(\boldsymbol{\xi}_2)_{j,r,i}^{(T_2,t)} \mathbb{1}(\rho(\boldsymbol{\xi}_2)_{j,r,i}^{(T_2,t)} \leq 0)$ and the fact $\ell_{2,i}'^{(T_2,s)} < 0$, we get

$$\overline{\rho}(\boldsymbol{\xi}_2)_{j,r,i}^{(T_2,t)} = -\frac{\eta_2}{n_2 m} \sum_{s=0}^{t-1} \ell_{2,i}'^{(T_2,s)} \cdot \sigma'(\langle \mathbf{w}_{j,r}^{(T_2,s)}, \boldsymbol{\xi}_{2,i}\rangle) \cdot \|\boldsymbol{\xi}_{2,i}\|_2^2 \cdot \mathbb{1}(y_{2,i} = j), \tag{E.2}$$

$$\underline{\rho}(\boldsymbol{\xi}_2)_{j,r,i}^{(T_2,t)} = \frac{\eta_2}{n_2 m} \sum_{s=0}^{t-1} \ell_{2,i}'^{(T_2,s)} \cdot \sigma'(\langle \mathbf{w}_{j,r}^{(T_2,s)}, \boldsymbol{\xi}_{2,i}\rangle) \cdot \|\boldsymbol{\xi}_{2,i}\|_2^2 \cdot \mathbb{1}(y_{2,i} = -j). \tag{E.3}$$

Writing out the iterative versions of (E.1), (E.2) and (E.3) completes the proof. $\square$

We will prove the Proposition E.2 following Cao et al. (2022), which shows that the coefficients in the signal-noise decomposition (4.1) stay a reasonable scale for a long time of training. Consider the learning period $0 \leq t \leq T^*$, where $T^* = \text{poly}(\eta_1^{-1}, \eta_2^{-1}, \epsilon^{-1}, \|\boldsymbol{\mu}_1\|_2^{-1}, \|\boldsymbol{\mu}_2\|_2^{-1}, d^{-1}\sigma_{p_1}^{-2}, d^{-1}\sigma_{p_1}^{-2}, \sigma_0^{-1}, n_1, n_2, m, d)$ is the maximum admissible iterations. Note that we can consider any polynomial training time $T^*$. Denote $\alpha = 4\log(T^*)$. Here we list the exact conditions on $\eta_2, \sigma_0, d$ for the proofs, which are part of Condition 3.1:

$$\eta_2 = O\left(\min\{n_2 m/(q\sigma_{p_2}^2 d), n_2 m/(q 2^{q+2}\alpha^{q-2}\sigma_{p_2}^2 d), n_2 m/(q 2^{q+2}\alpha^{q-2}\|\boldsymbol{\mu}_2\|_2^2)\}\right), \tag{E.4}$$

$$\sigma_0 \leq [32\sqrt{\log(8mn_2/\delta)}]^{-1} \min\{\|\boldsymbol{\mu}_1\|_2^{-1}, \|\boldsymbol{\mu}_2\|_2^{-1}, (\sigma_{p_2}\sqrt{d})^{-1}\}, \tag{E.5}$$

$$d \geq 1024\log(4n_2^2/\delta)\alpha^2 n_2^2. \tag{E.6}$$

Denote $\beta_0 = 2\max_{i,j,r}\{|\langle \mathbf{w}_{j,r}^{(T_1,0)}, \boldsymbol{\mu}_2\rangle|, |\langle \mathbf{w}_{j,r}^{(T_1,0)}, \boldsymbol{\xi}_{2,i}\rangle|\}$. By Lemma C.2, we can upper bound $\beta_0$ by $4\sqrt{\log(8mn_2/\delta)} \cdot \sigma_0 \cdot \max\{\|\boldsymbol{\mu}_2\|_2, \sigma_{p_2}\sqrt{d}\}$. Then, by (E.5) and (E.6), it is straightforward to verify the following inequality:

$$4\max\left\{\beta_0, 8n_2\sqrt{\frac{\log(4n_2^2/\delta)}{d}}\alpha\right\} \leq 1. \tag{E.7}$$

Denote $\beta_1 = 2\max_{j,r}\{|\langle \mathbf{w}_{j,r}^{(T_2,0)}, \boldsymbol{\mu}_2\rangle|\}$, $\beta_2 = 2\max_{i,j,r}\{|\langle \mathbf{w}_{j,r}^{(T_2,0)}, \boldsymbol{\xi}_{2,i}\rangle|\}$.Then, by (D.2), (D.3) and conditions listed in (E.5), we have that

$$\beta_1 = O([m\log(1/\epsilon)]^{1/q}) \tag{E.8}$$

$$\beta_2 \leq \frac{1}{4} \tag{E.9}$$

**Proposition E.2** (Acute-angle Case). *Under Condition 3.1, and when $\cos\theta_{1,2} > 0$, for $0 \leq t \leq T^*$, we have that*

$$0 \leq \gamma(\boldsymbol{\mu}_2)_{j,r}^{(T_2,t)}, \overline{\rho}(\boldsymbol{\xi}_2)_{j,r,i}^{(T_2,t)} \leq \alpha, \tag{E.10}$$

$$0 \geq \underline{\rho}(\boldsymbol{\xi}_2)_{j,r,i}^{(T_2,t)} \geq -\beta_2 - 16n_2\sqrt{\frac{\log(4n_2^2/\delta)}{d}}\alpha \geq -\alpha. \tag{E.11}$$

*for all $r \in [m]$, $j \in \{\pm 1\}$ and $i \in [n_2]$.*

We prove Proposition E.2 by induction, first introducing several technical lemmas for the proof.

**Lemma E.3** (Cao et al. (2022)). *For $t \geq 0$, it holds that $\langle \mathbf{w}_{j,r}^{(T_2,t)} - \mathbf{w}_{j,r}^{(T_2,0)}, \boldsymbol{\mu}_2\rangle = j \cdot \gamma(\boldsymbol{\mu}_2)_{j,r}^{(T_2,t)}$ for $r \in [m]$, $j \in \{\pm 1\}$.*

**Lemma E.4** (Cao et al. (2022)). *Under Condition 3.1, suppose (E.10) and (E.11) hold at iteration $t$. Then*

$$\underline{\rho}(\boldsymbol{\xi}_2)_{j,r,i}^{(T_2,t)} - 8n_2\sqrt{\frac{\log(4n_2^2/\delta)}{d}}\alpha \leq \langle \mathbf{w}_{j,r}^{(T_2,t)} - \mathbf{w}_{j,r}^{(T_2,0)}, \boldsymbol{\xi}_{2,i}\rangle \leq \underline{\rho}(\boldsymbol{\xi}_2)_{j,r,i}^{(T_2,t)} + 8n_2\sqrt{\frac{\log(4n_2^2/\delta)}{d}}\alpha, \ j \neq y_{2,i},$$

$$\overline{\rho}(\boldsymbol{\xi}_2)_{j,r,i}^{(T_2,t)} - 8n_2\sqrt{\frac{\log(4n_2^2/\delta)}{d}}\alpha \leq \langle \mathbf{w}_{j,r}^{(T_2,t)} - \mathbf{w}_{j,r}^{(T_2,0)}, \boldsymbol{\xi}_{2,i}\rangle \leq \overline{\rho}(\boldsymbol{\xi}_2)_{j,r,i}^{(T_2,t)} + 8n_2\sqrt{\frac{\log(4n_2^2/\delta)}{d}}\alpha, \ j = y_{2,i}$$

*for all $r \in [m]$, $j \in \{\pm 1\}$ and $i \in [n_2]$.*

**Lemma E.5.** *Under Condition 3.1, suppose (E.10) and (E.11) hold at iteration $t$. Then*

$$\langle \mathbf{w}_{j,r}^{(T_2,t)}, y_{2,i}\boldsymbol{\mu}_2\rangle \leq \langle \mathbf{w}_{j,r}^{(T_1,0)}, y_{2,i}\boldsymbol{\mu}_2\rangle,$$

$$\langle \mathbf{w}_{j,r}^{(T_2,t)}, \boldsymbol{\xi}_{2,i}\rangle \leq \langle \mathbf{w}_{j,r}^{(T_2,0)}, \boldsymbol{\xi}_{2,i}\rangle + 8n_2\sqrt{\frac{\log(4n_2^2/\delta)}{d}}\alpha,$$

$$F_j(\mathbf{W}_j^{(T_2,t)}, \mathbf{x}_{2,i}) \leq 1$$

*for all $r \in [m]$ and $j \neq y_{2,i}$.*

*Proof of Lemma E.5.* For $j \neq y_{2,i}$, we have that

$$\langle \mathbf{w}_{j,r}^{(T_2,t)}, y_{2,i}\boldsymbol{\mu}_2\rangle = \langle \mathbf{w}_{j,r}^{(T_1,0)}, y_{2,i}\boldsymbol{\mu}_2\rangle + y_{2,i} \cdot j \cdot \gamma(\boldsymbol{\mu}_1)_{j,r}^{(T_1,t_{end})} \cdot \frac{\|\boldsymbol{\mu}_2\|_2 \cos\theta_{1,2}}{\|\boldsymbol{\mu}_1\|_2} + y_{2,i} \cdot j \cdot \gamma(\boldsymbol{\mu}_2)_{j,r}^{(T_2,t)}$$

$$\leq \langle \mathbf{w}_{j,r}^{(T_1,0)}, y_{2,i}\boldsymbol{\mu}_2\rangle, \tag{E.12}$$

where the inequality is by $\gamma(\boldsymbol{\mu}_1)_{j,r}^{(T_1,t_{end})} \geq 0$ and $\gamma(\boldsymbol{\mu}_2)_{j,r}^{(T_2,t)} \geq 0$. In addition, we have

$$\langle \mathbf{w}_{j,r}^{(T_2,t)}, \boldsymbol{\xi}_{2,i}\rangle \leq \langle \mathbf{w}_{j,r}^{(T_2,0)}, \boldsymbol{\xi}_{2,i}\rangle + \underline{\rho}(\boldsymbol{\xi}_2)_{j,r,i}^{(T_2,t)} + 8n_2\sqrt{\frac{\log(4n_2^2/\delta)}{d}}\alpha \leq \langle \mathbf{w}_{j,r}^{(T_2,0)}, \boldsymbol{\xi}_{2,i}\rangle + 8n_2\sqrt{\frac{\log(4n_2^2/\delta)}{d}}\alpha, \tag{E.13}$$

where the first inequality is by Lemma E.4 and the second inequality is due to $\underline{\rho}(\boldsymbol{\xi}_2)_{j,r,i}^{(T_2,t)} \leq 0$. Then we can get that

$$F_j(\mathbf{W}_j^{(T_2,t)}, \mathbf{x}_{2,i}) = \frac{1}{m}\sum_{r=1}^m [\sigma(\langle \mathbf{w}_{j,r}^{(T_2,t)}, -j \cdot \boldsymbol{\mu}_2\rangle) + \sigma(\langle \mathbf{w}_{j,r}^{(T_2,t)}, \boldsymbol{\xi}_{2,i}\rangle)]$$

$$\leq 2^{q+1}\max_{j,r,i}\left\{|\langle \mathbf{w}_{j,r}^{(T_1,0)}, \boldsymbol{\mu}_2\rangle|, |\langle \mathbf{w}_{j,r}^{(T_2,0)}, \boldsymbol{\xi}_{2,i}\rangle|, 8n_2\sqrt{\frac{\log(4n_2^2/\delta)}{d}}\alpha\right\}^q$$

$$\leq 1,$$

where the first inequality is by (E.12), (E.13) and the second inequality is by (E.7), (E.9). $\square$

Now we are ready to prove Proposition E.2.

*Proof of Proposition E.2.* Our proof proceeds by induction. The claim is trivially true at $t = 0$ since all coefficients are initially zero. Suppose that there exists $\widetilde{T} \leq T^*$ such that the results in Proposition E.2 hold for all time $0 \leq t \leq \widetilde{T} - 1$. We aim to prove that they also hold for $t = \widetilde{T}$.

We first prove that (E.11) holds for $t = \widetilde{T}$, i.e., $\underline{\rho}(\boldsymbol{\xi}_2)_{j,r,i}^{(T_2,t)} \geq -\beta_2 - 16n_2\sqrt{\frac{\log(4n_2^2/\delta)}{d}}\alpha$ for $t = \widetilde{T}$, $r \in [m]$, $j \in \{\pm 1\}$ and $i \in [n_2]$. Since $\underline{\rho}(\boldsymbol{\xi}_2)_{j,r,i}^{(T_2,t)} = 0$ for $j = y_{2,i}$, it suffices to consider the case where $j \neq y_{2,i}$. When $\underline{\rho}(\boldsymbol{\xi}_2)_{j,r,i}^{(T_2,\widetilde{T}-1)} \leq -0.5\beta_2 - 8n_2\sqrt{\frac{\log(4n_2^2/\delta)}{d}}\alpha$, by Lemma E.4 we have that

$$\langle \mathbf{w}_{j,r}^{(T_2,\widetilde{T}-1)}, \boldsymbol{\xi}_{2,i} \rangle \leq \underline{\rho}(\boldsymbol{\xi}_2)_{j,r,i}^{(T_2,\widetilde{T}-1)} + \langle \mathbf{w}_{j,r}^{(T_2,0)}, \boldsymbol{\xi}_{2,i} \rangle + 8n_2\sqrt{\frac{\log(4n_2^2/\delta)}{d}}\alpha \leq 0,$$

and thus $\underline{\rho}(\boldsymbol{\xi}_2)_{j,r,i}^{(T_2,\widetilde{T})} = \underline{\rho}(\boldsymbol{\xi}_2)_{j,r,i}^{(T_2,\widetilde{T}-1)} \geq -\beta_2 - 16n_2\sqrt{\frac{\log(4n_2^2/\delta)}{d}}\alpha$ by induction hypothesis. When $\underline{\rho}(\boldsymbol{\xi}_2)_{j,r,i}^{(T_2,\widetilde{T}-1)} \geq -0.5\beta_2 - 8n_2\sqrt{\frac{\log(4n_2^2/\delta)}{d}}\alpha$, we have that

$$
\begin{aligned}
\underline{\rho}(\boldsymbol{\xi}_2)_{j,r,i}^{(T_2,\widetilde{T})} &= \underline{\rho}(\boldsymbol{\xi}_2)_{j,r,i}^{(T_2,\widetilde{T}-1)} + \frac{\eta_2}{n_2 m} \cdot \ell_{2,i}'^{(T_2,\widetilde{T}-1)} \cdot \sigma'(\langle \mathbf{w}_{j,r}^{(T_2,\widetilde{T}-1)}, \boldsymbol{\xi}_{2,i} \rangle) \cdot \mathbb{1}(y_{2,i} = -j)\|\boldsymbol{\xi}_{2,i}\|_2^2 \\
&\geq -0.5\beta_2 - 8n_2\sqrt{\frac{\log(4n_2^2/\delta)}{d}}\alpha - O\left(\frac{\eta_2\sigma_{p_2}^2 d}{n_2 m}\right)\sigma'\left(0.5\beta_2 + 8n_2\sqrt{\frac{\log(4n_2^2/\delta)}{d}}\alpha\right) \\
&\geq -0.5\beta - 8n_2\sqrt{\frac{\log(4n_2^2/\delta)}{d}}\alpha - O\left(\frac{\eta_2 q\sigma_{p_2}^2 d}{n_2 m}\right)\left(0.5\beta_2 + 8n_2\sqrt{\frac{\log(4n_2^2/\delta)}{d}}\alpha\right) \\
&\geq -\beta_2 - 16n_2\sqrt{\frac{\log(4n_2^2/\delta)}{d}}\alpha,
\end{aligned}
$$

where we use $|\ell_{2,i}'^{(T_2,\widetilde{T}-1)}| \leq 1$ and $\|\boldsymbol{\xi}_{2,i}\|_2^2 = O(\sigma_{p_2}^2 d)$ in the first inequality, the second inequality is by $0.5\beta_2 + 8n_2\sqrt{\frac{\log(4n_2^2/\delta)}{d}}\alpha \leq 1$, and the last inequality is by $\eta_2 = O(n_2 m/(q\sigma_{p_2}^2 d))$ in (E.4).

Next we prove (E.10) holds for $t = \widetilde{T}$. By Lemma E.5, we have

$$
\begin{aligned}
|\ell_{2,i}'^{(T_2,t)}| &= \frac{1}{1 + \exp\{y_{2,i} \cdot [F_{+1}(\mathbf{W}_{+1}^{(T_2,t)}, \mathbf{x}_{2,i}) - F_{-1}(\mathbf{W}_{-1}^{(T_2,t)}, \mathbf{x}_{2,i})]\}} \\
&\leq \exp\{-y_{2,i} \cdot [F_{+1}(\mathbf{W}_{+1}^{(T_2,t)}, \mathbf{x}_{2,i}) - F_{-1}(\mathbf{W}_{-1}^{(T_2,t)}, \mathbf{x}_{2,i})]\} \\
&\leq \exp\{-F_{y_{2,i}}(\mathbf{W}_{y_{2,i}}^{(T_2,t)}, \mathbf{x}_{2,i}) + 1\}.
\end{aligned}
\tag{E.14}
$$

Following Cao et al. (2022), let $t_{j,r,i}$ denote the last time $t < T^*$ that $\overline{\rho}(\boldsymbol{\xi}_2)_{j,r,i}^{(T_2,t)} \leq 0.5\alpha$. By the update rule of $\overline{\rho}(\boldsymbol{\xi}_2)_{j,r,i}^{(T_2,t)}$, we have that

$$
\begin{aligned}
\overline{\rho}_{j,r,i}^{(T_2,\widetilde{T})} = \overline{\rho}_{j,r,i}^{(T_2,t_{j,r,i})} &\underbrace{- \frac{\eta_2}{n_2 m} \cdot \ell_{2,i}'^{(T_2,t_{j,r,i})} \cdot \sigma'(\langle \mathbf{w}_{j,r}^{(T_2,t_{j,r,i})}, \boldsymbol{\xi}_{2,i} \rangle) \cdot \mathbb{1}(y_{2,i} = j)\|\boldsymbol{\xi}_{2,i}\|_2^2}_{I_1} \\
&\underbrace{- \sum_{t_{j,r,i} < t < \widetilde{T}} \frac{\eta_2}{n_2 m} \cdot \ell_{2,i}'^{(T_2,t)} \cdot \sigma'(\langle \mathbf{w}_{j,r}^{(T_2,t)}, \boldsymbol{\xi}_{2,i} \rangle) \cdot \mathbb{1}(y_{2,i} = j)\|\boldsymbol{\xi}_{2,i}\|_2^2}_{I_2}.
\end{aligned}
\tag{E.15}
$$

We first bound $I_1$ as follows,

$$|I_1| \leq 2qn_2^{-1}m^{-1}\eta_2\left(\overline{\rho}(\boldsymbol{\xi}_2)_{j,r,i}^{(T_2,t_{j,r,i})} + 0.5\beta_2 + 8n_2\sqrt{\frac{\log(4n_2^2/\delta)}{d}}\alpha\right)^{q-1}\sigma_{p_2}^2 d \leq q2^q n_2^{-1}m^{-1}\eta_2\alpha^{q-1}\sigma_{p_2}^2 d \leq 0.25\alpha,$$

where the first inequality is by Lemmas E.4 and C.1, the second inequality is by $\beta_2 \leq 0.1\alpha$ and $8n_2\sqrt{\frac{\log(4n_2^2/\delta)}{d}}\alpha \leq 0.1\alpha$, the last inequality is by $\eta_2 \leq n_2 m/(q2^{q+2}\alpha^{q-2}\sigma_{p_2}^2 d)$.

For $t_{j,r,i} < t < \widetilde{T}$ and $y_{2,i} = j$, applying Lemma E.4 and a technique from Cao et al. (2022), $\langle \mathbf{w}_{j,r}^{(T_2,t)}, \boldsymbol{\xi}_{2,i}\rangle$ is bounded by $0.25\alpha \leq \langle \mathbf{w}_{j,r}^{(T_2,t)}, \boldsymbol{\xi}_{2,i}\rangle \leq 2\alpha$. Plugging these bounds into $I_2$ yields

$$
\begin{aligned}
|I_2| &\leq \sum_{t_{j,r,i} < t < \widetilde{T}} \frac{\eta_2}{n_2 m} \cdot \exp(-\sigma(\langle \mathbf{w}_{j,r}^{(T_2,t)}, \boldsymbol{\xi}_{2,i}\rangle) + 1) \cdot \sigma'(\langle \mathbf{w}_{j,r}^{(T_2,t)}, \boldsymbol{\xi}_{2,i}\rangle) \cdot \mathbb{1}(y_{2,i} = j)\|\boldsymbol{\xi}_{2,i}\|_2^2 \\
&\leq \frac{eq2^q \eta_2 T^*}{n_2 m} \exp(-\alpha^q/4^q)\alpha^{q-1}\sigma_{p_2}^2 d \\
&\leq 0.25 T^* \exp(-\alpha^q/4^q)\alpha \\
&\leq 0.25 T^* \exp(-\log(T^*)^q)\alpha \\
&\leq 0.25\alpha,
\end{aligned}
$$

where the first inequality is by (E.14), the second inequality is by Lemma C.1, the third inequality is by $\eta_2 = O(n_2 m/(q2^{q+2}\alpha^{q-2}\sigma_{p_2}^2 d))$ in (E.4), the fourth inequality is by $\alpha = 4\log(T^*)$ and the last inequality is by the fact that $\log(T^*)^q \geq \log(T^*)$. Plugging the bound of $I_1, I_2$ into (E.15) completes the proof for $\overline{\rho}$. Similarly, using $\beta_1 \leq 0.1\alpha$ in (E.8) and $\eta_2 = O(n_2 m/(q2^{q+2}\alpha^{q-2}\|\boldsymbol{\mu}_2\|_2^2))$ in (E.4), we obtain $\gamma(\boldsymbol{\mu}_2)_{j,r}^{(T_2,\widetilde{T})} \leq \alpha$. Therefore Proposition E.2 holds for $t = \widetilde{T}$, which completes the induction. $\qquad\square$

## E.2. Signal Learning

In this section, we consider the signal learning case under the condition that $n_2\|\boldsymbol{\mu}_2\|_2^q \geq \widetilde{\Omega}(\sigma_{p_2}^q(\sqrt{d})^q)$. We remind the readers that the proofs in this section are based on the results in Section C and D, which hold with high probability. For the ease of discussion, we decompose $\boldsymbol{\mu}_1$ into components along $\boldsymbol{\mu}_2$ and the direction orthogonal to $\boldsymbol{\mu}_2$. Then by (2.1) and (4.1), we have that

$$
\begin{aligned}
\mathbf{w}_{j,r}^{(T_2,t)} &= \mathbf{w}_{j,r}^{(T_1,0)} + j \cdot [\gamma(\boldsymbol{\mu}_1)_{j,r}^{(T_1,t_{end})} \cdot \frac{\|\boldsymbol{\mu}_2\|_2 \cos\theta_{1,2}}{\|\boldsymbol{\mu}_1\|_2} + \gamma(\boldsymbol{\mu}_2)_{j,r}^{(T_2,t)}] \cdot \frac{\boldsymbol{\mu}_2}{\|\boldsymbol{\mu}_2\|_2^2} + j \cdot [\gamma(\boldsymbol{\mu}_1)_{j,r}^{(T_1,t_{end})} \cdot \frac{\|\boldsymbol{\mu}_2\|_2 \sin\theta_{1,2}}{\|\boldsymbol{\mu}_1\|_2}] \cdot \frac{\boldsymbol{\mu}_2^\perp}{\|\boldsymbol{\mu}_2^\perp\|_2^2} \\
&\quad + \sum_{i=1}^{n_1} \overline{\rho}(\boldsymbol{\xi}_1)_{j,r,i}^{(T_1,t_{end})} \cdot \frac{\boldsymbol{\xi}_{1,i}}{\|\boldsymbol{\xi}_{1,i}\|_2^2} + \sum_{i=1}^{n_1} \underline{\rho}(\boldsymbol{\xi}_1)_{j,r,i}^{(T_1,t_{end})} \cdot \frac{\boldsymbol{\xi}_{1,i}}{\|\boldsymbol{\xi}_{1,i}\|_2^2} + \sum_{i=1}^{n_2} \overline{\rho}(\boldsymbol{\xi}_2)_{j,r,i}^{(T_2,t)} \cdot \frac{\boldsymbol{\xi}_{2,i}}{\|\boldsymbol{\xi}_{2,i}\|_2^2} + \sum_{i=1}^{n_2} \underline{\rho}(\boldsymbol{\xi}_2)_{j,r,i}^{(T_2,t)} \cdot \frac{\boldsymbol{\xi}_{2,i}}{\|\boldsymbol{\xi}_{2,i}\|_2^2},
\end{aligned}
$$

where $\boldsymbol{\mu}_2^\perp \in \text{span}\{\boldsymbol{\mu}_1, \boldsymbol{\mu}_2\}$ and $\boldsymbol{\mu}_2^\perp$ is orthogonal to $\boldsymbol{\mu}_2$, with $\|\boldsymbol{\mu}_2^\perp\|_2 = \|\boldsymbol{\mu}_2\|_2$. Denote $\widetilde{\gamma}(\boldsymbol{\mu}_2)_{j,r}^{(T_2,t)} = \gamma(\boldsymbol{\mu}_1)_{j,r}^{(T_1,t_{end})} \cdot \|\boldsymbol{\mu}_2\|_2 \cos\theta_{1,2}/\|\boldsymbol{\mu}_1\|_2 + \gamma(\boldsymbol{\mu}_2)_{j,r}^{(T_2,t)}$. By (4.3), then

$$
\widetilde{\gamma}(\boldsymbol{\mu}_2)_{j,r}^{(T_2,t+1)} = \widetilde{\gamma}(\boldsymbol{\mu}_2)_{j,r}^{(T_2,t)} - \frac{\eta_2}{n_2 m} \cdot \sum_{i=1}^{n_2} \ell_{2,i}'^{(T_2,t)} \cdot \sigma'(y_{2,i} \cdot \langle \mathbf{w}_{j,r}^{(T_1,0)}, \boldsymbol{\mu}_2\rangle + y_{2,i} \cdot j \cdot \widetilde{\gamma}(\boldsymbol{\mu}_2)_{j,r}^{(T_2,t)}) \cdot \|\boldsymbol{\mu}_2\|_2^2.
$$

We know that if $\widetilde{\gamma}(\boldsymbol{\mu}_2)_{j,r}^{(T_2,0)} = \gamma(\boldsymbol{\mu}_1)_{j,r}^{(T_1,t_{end})} \cdot \|\boldsymbol{\mu}_2\|_2 \cos\theta_{1,2}/\|\boldsymbol{\mu}_1\|_2 < 2$, the learning process enters the first stage; otherwise, it directly proceeds to the second stage.

### E.2.1. FIRST STAGE

**Lemma E.6** (Cao et al. (2022)). *Under the same conditions as Theorem 3.2, in particular if we choose*

$$
n_2 \cdot \text{SNR}_2^q \geq C \log(6/\sigma_0\|\boldsymbol{\mu}_2\|_2)2^{2q+6}[6\log(8mn_2/\delta)]^{(q-1)/2}, \tag{E.16}
$$

*where $C = O(1)$ is a positive constant, there exists time*

$$
t_1 = \frac{C \log(6/\sigma_0\|\boldsymbol{\mu}_2\|_2)2^{q+1}m}{\eta_2\sigma_0^{q-2}\|\boldsymbol{\mu}_2\|_2^q}
$$

*such that*

- $\max_r \widetilde{\gamma}(\boldsymbol{\mu}_2)_{j,r}^{(T_2,t_1)} \geq 2$ *for $j \in \{\pm 1\}$.*

- $|\rho_{j,r,i}^{(T_2,t)}| \leq \sigma_0\sigma_{p_2}\sqrt{d}/2$ *for all $j \in \{\pm 1\}, r \in [m], i \in [n_2]$ and $0 \leq t \leq t_1$.*

E.2.2. SECOND STAGE

By the results in the first stage, the following property holds at the beginning of the second stage:

- $\max_r \widetilde{\gamma}(\boldsymbol{\mu}_2)_{j,r}^{(T_2,t_1)} \geq 2, \forall j \in \{\pm 1\}$.

- $\max_{j,r,i} |\rho_{j,r,i}^{(T_2,t_1)}| \leq \widehat{\beta}$ where $\widehat{\beta} = \sigma_0 \sigma_{p_2} \sqrt{d}/2$.

Lemma 4.1 ensures that the learned feature $\widetilde{\gamma}(\boldsymbol{\mu}_2)_{j,r}^{(T_2,t)}$ will not deteriorate, i.e., $\widetilde{\gamma}(\boldsymbol{\mu}_2)_{j,r}^{(T_2,t+1)} \geq \widetilde{\gamma}(\boldsymbol{\mu}_2)_{j,r}^{(T_2,t)}$ for $t \geq t_1$. Thus, $\max_r \widetilde{\gamma}(\boldsymbol{\mu}_2)_{j,r}^{(T_2,t)} \geq 2$. Now we choose $\mathbf{W}^*$ as follows:

$$\mathbf{w}_{j,r}^* = \mathbf{w}_{j,r}^{(T_1,0)} + 2qm \log(2q/\epsilon) \cdot j \cdot \frac{\boldsymbol{\mu}_2}{\|\boldsymbol{\mu}_2\|_2^2} + j \cdot \left[\gamma(\boldsymbol{\mu}_1)_{j,r}^{(T_1,t_{end})} \cdot \frac{\|\boldsymbol{\mu}_2\|_2 \sin\theta_{1,2}}{\|\boldsymbol{\mu}_1\|_2}\right] \cdot \frac{\boldsymbol{\mu}_2^\perp}{\|\boldsymbol{\mu}_2^\perp\|_2^2}$$
$$+ \sum_{i=1}^{n_1} \overline{\rho}(\boldsymbol{\xi}_1)_{j,r,i}^{(T_1,t_{end})} \cdot \frac{\boldsymbol{\xi}_{1,i}}{\|\boldsymbol{\xi}_{1,i}\|_2^2} + \sum_{i=1}^{n_1} \underline{\rho}(\boldsymbol{\xi}_1)_{j,r,i}^{(T_1,t_{end})} \cdot \frac{\boldsymbol{\xi}_{1,i}}{\|\boldsymbol{\xi}_{1,i}\|_2^2}.$$

Based on the above definition of $\mathbf{W}^*$, we have the following lemmas.

**Lemma E.7** (Cao et al. (2022)). *Under the same conditions as Theorem 3.2, let $t^+ = t_1 + \left\lfloor \frac{\|\mathbf{W}^{(T_2,t_1)} - \mathbf{W}^*\|_F^2}{2\eta_2 \epsilon} \right\rfloor = t_1 + \widetilde{O}(m^3 \eta_2^{-1} \epsilon^{-1} \|\boldsymbol{\mu}_2\|_2^{-2})$. Then we have $\max_{j,r,i} |\rho_{j,r,i}^{(T_2,t)}| \leq 2\widehat{\beta} = \sigma_0 \sigma_{p_2} \sqrt{d}$ for all $t_1 \leq t \leq t^+$. Besides,*

$$\frac{1}{t - t_1 + 1} \sum_{s=t_1}^t L_{S_2}(\mathbf{W}^{(T_2,s)}) \leq \frac{\|\mathbf{W}^{(T_2,t_1)} - \mathbf{W}^*\|_F^2}{(2q-1)\eta_2(t - t_1 + 1)} + \frac{\epsilon}{2q - 1}$$

*for all $t_1 \leq t \leq t^+$, and we can find an iterate $t_{end}$ with training loss smaller than $\epsilon$ within $t^+$ iterations.*

**Lemma E.8** (Cao et al. (2022)). *Let $t^+$ and $t_{end}$ be defined in Lemma E.7 respectively. Under the same conditions as Theorem 3.2, for any $0 \leq t \leq t^+$ with $L_{S_2}(\mathbf{W}^{(T_2,t)}) \leq 1$, it holds that $L_{\mathcal{D}_2}(\mathbf{W}^{(T_2,t)}) \leq 6 \cdot L_{S_2}(\mathbf{W}^{(T_2,t)}) + \exp(-n_2^2)$. Furthermore, since $L_{S_2}(\mathbf{W}^{(T_2,t_{end})}) \leq \epsilon$, it follows that $L_{\mathcal{D}_2}(\mathbf{W}^{(T_2,t_{end})}) \leq 6\epsilon + \exp(-n_2^2)$.*

### E.3. Forgetting

Consider a new data point $(\mathbf{x}_1, y_1)$ drawn from the distribution of task $T_1$ defined in Definition 1.1. Without loss of generality, we suppose that the first patch is the signal patch and the second patch is the noise patch, i.e., $\mathbf{x}_1 = [y_1 \boldsymbol{\mu}_1, \boldsymbol{\xi}_1]$. Moreover, based on the analysis of signal learning we know that

$$\mathbf{w}_{j,r}^{(T_2,t_{end})} = \mathbf{w}_{j,r}^{(T_1,0)} + j \cdot \gamma(\boldsymbol{\mu}_1)_{j,r}^{(T_1,t_{end})} \cdot \frac{\boldsymbol{\mu}_1}{\|\boldsymbol{\mu}_1\|_2^2} + j \cdot \gamma(\boldsymbol{\mu}_2)_{j,r}^{(T_2,t_{end})} \cdot \frac{\boldsymbol{\mu}_2}{\|\boldsymbol{\mu}_2\|_2^2}$$
$$+ \sum_{i=1}^{n_1} \rho(\boldsymbol{\xi}_1)_{j,r,i}^{(T_1,t_{end})} \cdot \frac{\boldsymbol{\xi}_{1,i}}{\|\boldsymbol{\xi}_{1,i}\|_2^2} + \sum_{i=1}^{n_2} \rho(\boldsymbol{\xi}_2)_{j,r,i}^{(T_2,t_{end})} \cdot \frac{\boldsymbol{\xi}_{2,i}}{\|\boldsymbol{\xi}_{2,i}\|_2^2}. \qquad (E.17)$$

And at the end of training task $T_2$, we have following property holds:

- $\max_{j,r,i} |\rho(\boldsymbol{\xi}_1)_{j,r,i}^{(T_1,t_{end})}| \leq \sigma_0 \sigma_{p_1} \sqrt{d}$.

- $\max_{j,r,i} |\rho(\boldsymbol{\xi}_2)_{j,r,i}^{(T_2,t_{end})}| \leq \sigma_0 \sigma_{p_2} \sqrt{d}$.

**Lemma E.9.** *Under Condition 3.1, we have*

$$\frac{1}{m} \sum_{r=1}^m \sigma(\langle \mathbf{w}_{-y_1,r}^{(T_2,t_{end})}, y_1 \cdot \boldsymbol{\mu}_1 \rangle) \leq 1$$

*Proof of Lemma E.9.* We have that

$$\langle \mathbf{w}_{-y_1,r}^{(T_2,t_{end})}, y_1 \cdot \boldsymbol{\mu}_1 \rangle = \langle \mathbf{w}_{-y_1,r}^{(T_1,0)}, y_1 \cdot \boldsymbol{\mu}_1 \rangle - \gamma(\boldsymbol{\mu}_1)_{-y_1,r}^{(T_1,t_{end})} - \gamma(\boldsymbol{\mu}_2)_{-y_1,r}^{(T_2,t_{end})} \cdot \frac{\|\boldsymbol{\mu}_1\|_2 \cos\theta_{1,2}}{\|\boldsymbol{\mu}_2\|_2}$$

$$\leq \langle \mathbf{w}_{-y_1,r}^{(T_1,0)}, y_1 \cdot \boldsymbol{\mu}_1 \rangle, \tag{E.18}$$

where the inequality is by $\gamma(\boldsymbol{\mu}_1)_{-y_1,r}^{(T_1,t_{end})} \geq 0$ and $\gamma(\boldsymbol{\mu}_2)_{-y_1,r}^{(T_2,t_{end})} \geq 0$. Then we can get that

$$\frac{1}{m}\sum_{r=1}^{m}\sigma(\langle \mathbf{w}_{-y_1,r}^{(T_2,t_{end})}, y_1 \cdot \boldsymbol{\mu}_1 \rangle) \leq \max_r |\langle \mathbf{w}_{-y_1,r}^{(T_1,0)}, y_1 \cdot \boldsymbol{\mu}_1 \rangle|^q$$

$$\leq 1,$$

where the first inequality is by (E.18) and the second inequality is by (E.5) and Lemma C.2. $\qquad\square$

**Lemma E.10** (Cao et al. (2022))**.** *Under the same conditions as Theorem 3.2, with probability at least $1 - 4m \cdot \exp(-C_2^{-1}\sigma_0^{-2}\sigma_{p_1}^{-2}d^{-1})$, we have that $\max_{j,r}|\langle \mathbf{w}_{j,r}^{(T_1,t_{end})}, \boldsymbol{\xi}_1 \rangle| \leq 1/2$, where $C_2 = \widetilde{O}(1)$.*

**Lemma E.11.** *Under the same conditions as Theorem 3.2, with probability at least $1 - 4m \cdot \exp(-C_2^{-1}\sigma_0^{-2}\sigma_{p_1}^{-2}d^{-1})$, we have that $\max_{j,r}|\langle \mathbf{w}_{j,r}^{(T_2,t_{end})}, \boldsymbol{\xi}_1 \rangle| \leq 1/2$, where $C_2 = \widetilde{O}(1)$.*

*Proof of Lemma E.11.* Let $\widetilde{\mathbf{w}}_{j,r}^{(T_2,t_{end})} = \mathbf{w}_{j,r}^{(T_2,t_{end})} - j \cdot \gamma(\boldsymbol{\mu}_1)_{j,r}^{(T_1,t_{end})} \cdot \frac{\boldsymbol{\mu}_1}{\|\boldsymbol{\mu}_1\|_2^2} - j \cdot \gamma(\boldsymbol{\mu}_2)_{j,r}^{(T_2,t_{end})} \cdot \frac{\boldsymbol{\mu}_2}{\|\boldsymbol{\mu}_2\|_2^2}$, then we have that $\langle \widetilde{\mathbf{w}}_{j,r}^{(T_2,t_{end})}, \boldsymbol{\xi}_1 \rangle = \langle \mathbf{w}_{j,r}^{(T_2,t_{end})}, \boldsymbol{\xi}_1 \rangle$ and

$$\|\widetilde{\mathbf{w}}_{j,r}^{(T_2,t_{end})}\|_2 \leq \widetilde{O}(\sigma_0\sqrt{d} + n_1\sigma_0 + n_2\sigma_0) = \widetilde{O}(\sigma_0\sqrt{d}), \tag{E.19}$$

where the equality is due to $d \geq \widetilde{\Omega}(m^2 \cdot \max\{n_1^4, n_2^4\})$ by Condition 3.1. Then similar to Cao et al. (2022), applying the properties of Gaussian distribution completes the proof. $\qquad\square$

**Lemma E.12.** *Let $t_{end}$ be defined in Lemma E.7. Under the same conditions as Theorem 3.2, it holds that $L_{\mathcal{D}_1}(\mathbf{W}^{(T_2,t_{end})}) \leq 18\epsilon + \exp(-n_1^2)$.*

*Proof of Lemma E.8.* By Lemma D.8 in Cao et al. (2022), we know that $L_{\mathcal{D}_1}(\mathbf{W}^{(T_1,t_{end})}) \leq 6\epsilon + \exp(-n_1^2)$. We intend to determine an upper bound for $L_{\mathcal{D}_1}(\mathbf{W}^{(T_2,t_{end})})$ by leveraging the relationship between $L_{\mathcal{D}_1}(\mathbf{W}^{(T_2,t_{end})})$ and $L_{\mathcal{D}_1}(\mathbf{W}^{(T_1,t_{end})})$. According to the definition of true loss in Two-layer CNNs, we have

$$L_{\mathcal{D}_1}(\mathbf{W}^{(T_1,t_{end})}) = \mathbb{E}_{(\mathbf{x}_1,y_1)\sim\mathcal{D}_1}\ell[y_1 \cdot f(\mathbf{W}^{(T_1,t_{end})}, \mathbf{x}_1)]$$
$$L_{\mathcal{D}_1}(\mathbf{W}^{(T_2,t_{end})}) = \mathbb{E}_{(\mathbf{x}_1,y_1)\sim\mathcal{D}_1}\ell[y_1 \cdot f(\mathbf{W}^{(T_2,t_{end})}, \mathbf{x}_1)].$$

When $y_1 \cdot f(\mathbf{W}^{(T_2,t_{end})}, \mathbf{x}_1) \geq y_1 \cdot f(\mathbf{W}^{(T_1,t_{end})}, \mathbf{x}_1)$, by $\ell(z) = \log(1 + \exp(-z))$, we have

$$L_{\mathcal{D}_1}(\mathbf{W}^{(T_2,t_{end})}) \leq L_{\mathcal{D}_1}(\mathbf{W}^{(T_1,t_{end})}) \leq 6\epsilon + \exp(-n_1^2).$$

When $y_1 \cdot f(\mathbf{W}^{(T_2,t_{end})}, \mathbf{x}_1) < y_1 \cdot f(\mathbf{W}^{(T_1,t_{end})}, \mathbf{x}_1)$, let event $\mathcal{E}$ to be the event that Lemma E.10 and E.11 holds. Then we can divide $L_{\mathcal{D}_1}(\mathbf{W}^{(T_2,t_{end})})$ into two parts:

$$\mathbb{E}\big[\ell\big(y_1 f(\mathbf{W}^{(T_2,t_{end})}, \mathbf{x}_1)\big)\big] = \underbrace{\mathbb{E}[\mathbb{1}(\mathcal{E})\ell\big(y_1 f(\mathbf{W}^{(T_2,t_{end})}, \mathbf{x}_1)\big)]}_{I_1} + \underbrace{\mathbb{E}[\mathbb{1}(\mathcal{E}^c)\ell\big(y_1 f(\mathbf{W}^{(T_2,t_{end})}, \mathbf{x}_1)\big)]}_{I_2}. \tag{E.20}$$

In the following, we bound $I_1$ and $I_2$ respectively.

**Bounding $I_1$:** By the signal-noise decomposition (D.1) and (E.17), we have

$$\langle \mathbf{w}_{y_1,r}^{(T_1,t_{end})}, y_1 \cdot \boldsymbol{\mu}_1 \rangle = \langle \mathbf{w}_{y_1,r}^{(T_1,0)}, y_1 \cdot \boldsymbol{\mu}_1 \rangle + \gamma(\boldsymbol{\mu}_1)_{y_1,r}^{(T_1,t_{end})}$$
$$\langle \mathbf{w}_{-y_1,r}^{(T_1,t_{end})}, y_1 \cdot \boldsymbol{\mu}_1 \rangle = \langle \mathbf{w}_{-y_1,r}^{(T_1,0)}, y_1 \cdot \boldsymbol{\mu}_1 \rangle - \gamma(\boldsymbol{\mu}_1)_{-y_1,r}^{(T_1,t_{end})}$$
$$\langle \mathbf{w}_{y_1,r}^{(T_2,t_{end})}, y_1 \cdot \boldsymbol{\mu}_1 \rangle = \langle \mathbf{w}_{y_1,r}^{(T_1,0)}, y_1 \cdot \boldsymbol{\mu}_1 \rangle + \gamma(\boldsymbol{\mu}_1)_{y_1,r}^{(T_1,t_{end})} + \gamma(\boldsymbol{\mu}_2)_{y_1,r}^{(T_2,t_{end})} \cdot \frac{\|\boldsymbol{\mu}_1\|_2 \cos\theta_{1,2}}{\|\boldsymbol{\mu}_2\|_2}$$
$$\langle \mathbf{w}_{-y_1,r}^{(T_2,t_{end})}, y_1 \cdot \boldsymbol{\mu}_1 \rangle = \langle \mathbf{w}_{-y_1,r}^{(T_1,0)}, y_1 \cdot \boldsymbol{\mu}_1 \rangle - \gamma(\boldsymbol{\mu}_1)_{-y_1,r}^{(T_1,t_{end})} - \gamma(\boldsymbol{\mu}_2)_{-y_1,r}^{(T_2,t_{end})} \cdot \frac{\|\boldsymbol{\mu}_1\|_2 \cos\theta_{1,2}}{\|\boldsymbol{\mu}_2\|_2}.$$

Since $\gamma(\boldsymbol{\mu}_2)_{j,r}^{(T_2,t_{end})} \geq 0$ and $\cos\theta_{1,2} \geq 0$, we know that $\langle \mathbf{w}_{y_1,r}^{(T_2,t_{end})}, y_1 \cdot \boldsymbol{\mu}_1 \rangle \geq \langle \mathbf{w}_{y_1,r}^{(T_1,t_{end})}, y_1 \cdot \boldsymbol{\mu}_1 \rangle$ and $\langle \mathbf{w}_{-y_1,r}^{(T_2,t_{end})}, y_1 \cdot \boldsymbol{\mu}_1 \rangle \leq \langle \mathbf{w}_{-y_1,r}^{(T_1,t_{end})}, y_1 \cdot \boldsymbol{\mu}_1 \rangle$. Then if event $\mathcal{E}$ holds, we have that

$$
\begin{aligned}
-y_1 f(\mathbf{W}^{(T_2,t_{end})}, \mathbf{x}_1) = & -y_1 f(\mathbf{W}^{(T_1,t_{end})}, \mathbf{x}_1) + \frac{1}{m}\sum_{r=1}^{m}\left[\sigma(\langle \mathbf{w}_{y_1,r}^{(T_1,t_{end})}, y_1 \cdot \boldsymbol{\mu}_1 \rangle) - \sigma(\langle \mathbf{w}_{y_1,r}^{(T_2,t_{end})}, y_1 \cdot \boldsymbol{\mu}_1 \rangle)\right] \\
& + \frac{1}{m}\sum_{r=1}^{m}\left[\sigma(\langle \mathbf{w}_{-y_1,r}^{(T_2,t_{end})}, y_1 \cdot \boldsymbol{\mu}_1 \rangle) - \sigma(\langle \mathbf{w}_{-y_1,r}^{(T_1,t_{end})}, y_1 \cdot \boldsymbol{\mu}_1 \rangle)\right] + \frac{1}{m}\sum_{r=1}^{m}\sigma(\langle \mathbf{w}_{y_1,r}^{(T_1,t_{end})}, \boldsymbol{\xi}_1 \rangle) \\
& + \frac{1}{m}\sum_{r=1}^{m}\sigma(\langle \mathbf{w}_{-y_1,r}^{(T_2,t_{end})}, \boldsymbol{\xi}_1 \rangle) - \frac{1}{m}\sum_{r=1}^{m}\sigma(\langle \mathbf{w}_{-y_1,r}^{(T_1,t_{end})}, \boldsymbol{\xi}_1 \rangle) - \frac{1}{m}\sum_{r=1}^{m}\sigma(\langle \mathbf{w}_{y_1,r}^{(T_2,t_{end})}, \boldsymbol{\xi}_1 \rangle) \\
\leq & -y_1 f(\mathbf{W}^{(T_1,t_{end})}, \mathbf{x}_1) + \frac{1}{m}\sum_{r=1}^{m}\sigma(\langle \mathbf{w}_{y_1,r}^{(T_1,t_{end})}, \boldsymbol{\xi}_1 \rangle) + \frac{1}{m}\sum_{r=1}^{m}\sigma(\langle \mathbf{w}_{-y_1,r}^{(T_2,t_{end})}, \boldsymbol{\xi}_1 \rangle) \\
\leq & -y_1 f(\mathbf{W}^{(T_1,t_{end})}, \mathbf{x}_1) + \frac{1}{m}\sum_{r=1}^{m}\sigma(1/2) + \frac{1}{m}\sum_{r=1}^{m}\sigma(1/2) \\
\leq & -y_1 f(\mathbf{W}^{(T_1,t_{end})}, \mathbf{x}_1) + 1,
\end{aligned}
\tag{E.21}
$$

where the second inequality is by $\max_{j,r}|\langle \mathbf{w}_{j,r}^{(T_1,t_{end})}, \boldsymbol{\xi}_1 \rangle| \leq 1/2$ in Lemma E.10 and $\max_{j,r}|\langle \mathbf{w}_{j,r}^{(T_2,t_{end})}, \boldsymbol{\xi}_1 \rangle| \leq 1/2$ in Lemma E.11. Therefore, we have that

$$
\begin{aligned}
I_1 &\leq \mathbb{E}[\mathbb{1}(\mathcal{E})\exp(-y_1 f(\mathbf{W}^{(T_2,t_{end})}, \mathbf{x}_1))] \\
&\leq e \cdot \mathbb{E}[\mathbb{1}(\mathcal{E})\exp(-y_1 f(\mathbf{W}^{(T_1,t_{end})}, \mathbf{x}_1))] \\
&\leq 18\epsilon,
\end{aligned}
$$

where the first inequality is by the property of cross-entropy loss that $\ell(z) \leq \exp(-z)$ for all $z$, the second inequality is by (E.21), and the third inequality is by $\mathbb{E}[\mathbb{1}(\mathcal{E})\exp(-y_1 f(\mathbf{W}^{(T_1,t_{end})}, \mathbf{x}_1))] \leq 6\epsilon$ in the proof of Lemma D.8 in Cao et al. (2022).

**Bounding $I_2$:** Next we bound the second term $I_2$. We choose an arbitrary training data $(\mathbf{x}_{i'}, y_{i'})$ such that $y_{i'} = y$. Then we have

$$
\begin{aligned}
\ell\left(y_1 f(\mathbf{W}^{(T_2,t_{end})}, \mathbf{x}_1)\right) &\leq \log(1 + \exp(F_{-y_1}(\mathbf{W}^{(T_2,t_{end})}, \mathbf{x}_1))) \\
&\leq 1 + F_{-y_1}(\mathbf{W}^{(T_2,t_{end})}, \mathbf{x}_1)) \\
&= 1 + \frac{1}{m}\sum_{r=1}^{m}\sigma(\langle \mathbf{w}_{-y_1,r}^{(T_2,t_{end})}, y_1\boldsymbol{\mu}_1 \rangle) + \frac{1}{m}\sum_{r=1}^{m}\sigma(\langle \mathbf{w}_{-y_1,r}^{(T_2,t_{end})}, \boldsymbol{\xi}_1 \rangle) \\
&\leq 2 + \frac{1}{m}\sum_{r=1}^{m}\sigma(\langle \mathbf{w}_{-y_1,r}^{(T_2,t_{end})}, \boldsymbol{\xi}_1 \rangle) \\
&\leq 2 + \widetilde{O}((\sigma_0\sqrt{d})^q)\|\boldsymbol{\xi}_1\|^q,
\end{aligned}
\tag{E.22}
$$

where the first inequality is due to $F_{y_1}(\mathbf{W}^{(T_2,t_{end})}, \mathbf{x}_1)) \geq 0$, the second inequality is by the property of cross-entropy loss, i.e., $\log(1 + \exp(z)) \leq 1 + z$ for all $z \geq 0$, the third inequality is by $\frac{1}{m}\sum_{r=1}^{m}\sigma(\langle \mathbf{w}_{-y_1,r}^{(T_2,t_{end})}, y_1\boldsymbol{\mu}_1 \rangle) \leq 1$ in Lemma E.9, and the last inequality is due to $\langle \widetilde{\mathbf{w}}_{j,r}^{(T_2,t_{end})}, \boldsymbol{\xi}_1 \rangle = \langle \mathbf{w}_{j,r}^{(T_2,t_{end})}, \boldsymbol{\xi}_1 \rangle \leq \|\widetilde{\mathbf{w}}_{j,r}^{(T_2,t_{end})}\|_2\|\boldsymbol{\xi}_1\|_2 \leq \widetilde{O}(\sigma_0\sqrt{d})\|\boldsymbol{\xi}_1\|_2$ in (E.19). Then we further have that

$$
\begin{aligned}
I_2 &\leq \sqrt{\mathbb{E}[\mathbb{1}(\mathcal{E}^c)]} \cdot \sqrt{\mathbb{E}\left[\ell\left(y_1 f(\mathbf{W}^{(T_2,t_{end})}, \mathbf{x}_1)\right)^2\right]} \\
&\leq \sqrt{\mathbb{P}(\mathcal{E}^c)} \cdot \sqrt{4 + \widetilde{O}((\sigma_0\sqrt{d})^{2q})\mathbb{E}[\|\boldsymbol{\xi}_1\|_2^{2q}]} \\
&\leq \exp[-\widetilde{\Omega}(\sigma_0^{-2}\sigma_{p_1}^{-2}d^{-1}) + \text{polylog}(d)] \\
&\leq \exp(-n_1^2),
\end{aligned}
$$

where the first inequality is by Cauchy-Schwartz inequality, the second inequality is by (E.22), the third inequality is by the fact that $\sqrt{4 + \widetilde{O}((\sigma_0\sqrt{d})^{2q})\mathbb{E}[\|\boldsymbol{\xi}_1\|_2^{2q}]} = O(\text{poly}(d))$, Lemma E.10 and Lemma E.11, and the last inequality is by our condition $\sigma_0 \leq \widetilde{O}(m^{-2/(q-2)}n_1^{-1}) \cdot (\sigma_{p_1}\sqrt{d})^{-1}$ in Condition 3.1. Plugging the bounds of $I_1$, $I_2$ into (E.20) completes the proof. $\qquad\square$

## F. Learning of Task T2 (Standard CL with Obtuse Angle)

In this section, we consider continual learning where the signal vector of task $T_1$ forms an obtuse angle with the signal vector of task $T_2$. When the angle between the two task signal vectors $0 \geq \cos\theta_{1,2} \geq -\sqrt{2\log(8m/\delta)} \cdot \sigma_0\|\boldsymbol{\mu}_1\|_2/([m\log(1/\epsilon)]^{1/q})$, through Condition 3.1, we know that $-\sqrt{2\log(8m/\delta)} \cdot \sigma_0\|\boldsymbol{\mu}_1\|_2/([m\log(1/\epsilon)]^{1/q}) \to 0$ and $\theta_{1,2} \to \pi/2$. For continual learning where the signal vectors of the two tasks are orthogonal, since the components in the signal directions $\gamma(\boldsymbol{\mu}_1)$ and $\gamma(\boldsymbol{\mu}_2)$ do not interfere with each other, the analysis of training task $T_2$ is similar to the analysis of task $T_1$ by Cao et al. (2022), resulting in a small amount of forgetting. Next, we consider a less extreme scenario where $-1 \leq \cos\theta_{1,2} \leq -\sqrt{2\log(8m/\delta)} \cdot \sigma_0\|\boldsymbol{\mu}_1\|_2/([m\log(1/\epsilon)]^{1/q})$. We intend to investigate the mechanisms of signal learning in task $T_2$ and forgetting on task $T_1$ by conducting a detailed analysis of the behavior of various types of neurons during the training process of task $T_2$.

### F.1. Behavior of Different Types of Neurons

By the decomposition (4.1), we have that

$$\mathbf{w}_{j,r}^{(T_2,t)} = \mathbf{w}_{j,r}^{(T_2,0)} + j \cdot \gamma(\boldsymbol{\mu}_2)_{j,r}^{(T_2,t)} \cdot \|\boldsymbol{\mu}_2\|_2^{-2} \cdot \boldsymbol{\mu}_2 + \sum_{i=1}^{n_2} \overline{\rho}(\boldsymbol{\xi}_2)_{j,r,i}^{(T_2,t)} \cdot \|\boldsymbol{\xi}_{2,i}\|_2^{-2} \cdot \boldsymbol{\xi}_{2,i} + \sum_{i=1}^{n_2} \underline{\rho}(\boldsymbol{\xi}_2)_{j,r,i}^{(T_2,t)} \cdot \|\boldsymbol{\xi}_{2,i}\|_2^{-2} \cdot \boldsymbol{\xi}_{2,i}.$$

Next, we intend to categorize the neurons $\mathbf{w}_{j,r}^{(T_2)}$ into three groups based on the following conditions: $r \in \{r \in [m] : \langle\mathbf{w}_{j,r}^{(T_1,0)}, j\boldsymbol{\mu}_1\rangle > 0\}$, $r \in \{r \in [m] : \langle\mathbf{w}_{j,r}^{(T_1,0)}, j\boldsymbol{\mu}_1\rangle \leq 0\} \cap \{r \in [m] : \langle\mathbf{w}_{j,r}^{(T_1,0)}, j\boldsymbol{\mu}_1^{\perp}\rangle > 0\}$ and $r \in \{r \in [m] : \langle\mathbf{w}_{j,r}^{(T_1,0)}, j\boldsymbol{\mu}_1\rangle \leq 0\} \cap \{r \in [m] : \langle\mathbf{w}_{j,r}^{(T_1,0)}, j\boldsymbol{\mu}_1^{\perp}\rangle \leq 0\}$. Subsequently, we will analyze their behavior during the training process of task $T_2$.

**Lemma F.1.** *Under the same conditions as Theorem 3.2, when* $r \in \{r \in [m] : \langle\mathbf{w}_{j,r}^{(T_1,0)}, j\boldsymbol{\mu}_1\rangle \leq 0\}$, *we have that* $\gamma(\boldsymbol{\mu}_1)_{j,r}^{(T_1,t_{end})} = -\langle\mathbf{w}_{j,r}^{(T_1,0)}, j\boldsymbol{\mu}_1\rangle$.

*Proof of Lemma F.1.* We first prove that $\gamma(\boldsymbol{\mu}_1)_{j,r}^{(T_1,t)} \leq -\langle\mathbf{w}_{j,r}^{(T_1,0)}, j\boldsymbol{\mu}_1\rangle$ for $t \leq t_{end}$ based on induction. The result is obvious at $t = 0$ as all the coefficients are zero. Suppose that there exists $\widetilde{t} \leq t_{end}$ such that the result holds for all time $0 \leq t \leq \widetilde{t} - 1$. We aim to prove that it also holds for $t = \widetilde{t}$. By the update formula for the signal vector of task $T_1$ similar to that in Equation (4.3), we have that

$$\gamma(\boldsymbol{\mu}_1)_{j,r}^{(T_1,\widetilde{t})} = \gamma(\boldsymbol{\mu}_1)_{j,r}^{(T_1,\widetilde{t}-1)} - \frac{\eta_1}{n_1 m} \cdot \sum_{i=1}^{n_1} \ell_{1,i}'^{(T_1,\widetilde{t}-1)} \cdot \sigma'(\langle\mathbf{w}_{j,r}^{(T_1,0)}, y_{1,i} \cdot \boldsymbol{\mu}_1\rangle + y_{1,i} \cdot j \cdot \gamma(\boldsymbol{\mu}_1)_{j,r}^{(T_1,\widetilde{t}-1)}) \cdot \|\boldsymbol{\mu}_1\|_2^2$$

$$= \gamma(\boldsymbol{\mu}_1)_{j,r}^{(T_1,\widetilde{t}-1)} - \frac{\eta_1}{n_1 m} \cdot \sum_{y_{1,i}=-j}^{n_1} \ell_{1,i}'^{(T_1,\widetilde{t}-1)} \cdot \sigma'(-\langle\mathbf{w}_{j,r}^{(T_1,0)}, j \cdot \boldsymbol{\mu}_1\rangle - \gamma(\boldsymbol{\mu}_1)_{j,r}^{(T_1,\widetilde{t}-1)}) \cdot \|\boldsymbol{\mu}_1\|_2^2.$$

Denote $\widehat{\gamma}(\boldsymbol{\mu}_1)_{j,r}^{(T_1,t)} = -\gamma(\boldsymbol{\mu}_1)_{j,r}^{(T_1,t)} - \langle\mathbf{w}_{j,r}^{(T_1,0)}, j\boldsymbol{\mu}_1\rangle$, then we have

$$\widehat{\gamma}(\boldsymbol{\mu}_1)_{j,r}^{(T_1,\widetilde{t})} = \widehat{\gamma}(\boldsymbol{\mu}_1)_{j,r}^{(T_1,\widetilde{t}-1)} + \frac{\eta_1}{n_1 m} \cdot \sum_{y_{1,i}=-j}^{n_1} \ell_{1,i}'^{(T_1,\widetilde{t}-1)} \cdot \sigma'(\widehat{\gamma}(\boldsymbol{\mu}_1)_{j,r}^{(T_1,\widetilde{t}-1)}) \cdot \|\boldsymbol{\mu}_1\|_2^2$$

$$\geq \widehat{\gamma}(\boldsymbol{\mu}_1)_{j,r}^{(T_1,\widetilde{t}-1)} - \frac{q\eta_1\|\boldsymbol{\mu}_1\|_2^2}{m} \cdot (\widehat{\gamma}(\boldsymbol{\mu}_1)_{j,r}^{(T_1,\widetilde{t}-1)})^{q-1}$$

$$= \widehat{\gamma}(\boldsymbol{\mu}_1)_{j,r}^{(T_1,\widetilde{t}-1)} \cdot [1 - \frac{q\eta_1\|\boldsymbol{\mu}_1\|_2^2}{m} \cdot (\widehat{\gamma}(\boldsymbol{\mu}_1)_{j,r}^{(T_1,\widetilde{t}-1)})^{q-2}]$$

$$\geq \widehat{\gamma}(\boldsymbol{\mu}_1)_{j,r}^{(T_1,\widetilde{t}-1)} \cdot [1 - \frac{q\eta_1\|\boldsymbol{\mu}_1\|_2^2}{m} \cdot (-\langle \mathbf{w}_{j,r}^{(T_1,0)}, j\boldsymbol{\mu}_1\rangle)^{q-2}]$$

$$\geq 0,$$

where we use $0 > \ell_{1,i}'^{(T_1,\widetilde{t}-1)} \geq -1$ and $\widehat{\gamma}(\boldsymbol{\mu}_1)_{j,r}^{(T_1,\widetilde{t}-1)} \geq 0$ in the first inequality, the second inequality is by $\widehat{\gamma}(\boldsymbol{\mu}_1)_{j,r}^{(T_1,\widetilde{t}-1)} \leq \widehat{\gamma}(\boldsymbol{\mu}_1)_{j,r}^{(T_1,0)} = -\langle \mathbf{w}_{j,r}^{(T_1,0)}, j\boldsymbol{\mu}_1\rangle$, and the last inequality is by $\eta_1 \leq m/(q\alpha^{q-2}\|\boldsymbol{\mu}_1\|_2^2)$ in Condition 3.1. Now we know that $\widehat{\gamma}(\boldsymbol{\mu}_1)_{j,r}^{(T_1,t)}$ is monotonically decreasing and bounded below, thus $\widehat{\gamma}(\boldsymbol{\mu}_1)_{j,r}^{(T_1,t)}$ must converge to a limit, which we denote by $C$. Then we have that

$$C = C + \frac{\eta_1}{n_1 m} \cdot \sum_{y_{1,i}=-j}^{n_1} \ell_{1,i}'^{(T_1,t_{end})} \cdot \sigma'(C) \cdot \|\boldsymbol{\mu}_1\|_2^2.$$

By $\ell_{1,i}'^{(T_1,t_{end})} < 0$ and $C \geq 0$, we know that $C$ must be equal to 0. It follows that $\widehat{\gamma}(\boldsymbol{\mu}_1)_{j,r}^{(T_1,t)}$ converges to 0, implying that $\gamma(\boldsymbol{\mu}_1)_{j,r}^{(T_1,t_{end})} = -\langle \mathbf{w}_{j,r}^{(T_1,0)}, j\boldsymbol{\mu}_1\rangle$, which completes the proof. $\qquad\square$

**Lemma F.2.** *Under the same conditions as Theorem 3.2, when $r \in \{r \in [m] : \langle \mathbf{w}_{j,r}^{(T_1,0)}, j\boldsymbol{\mu}_1\rangle > 0\}$, we have that* $\langle \mathbf{w}_{j,r}^{(T_2,0)}, j\boldsymbol{\mu}_2\rangle \leq 0$ *and* $\gamma(\boldsymbol{\mu}_2)_{j,r}^{(T_2,t_{end})} = -\langle \mathbf{w}_{j,r}^{(T_2,0)}, j\boldsymbol{\mu}_2\rangle$ *for $j \in \{\pm1\}$.*

*proof of Lemma F.2.* By (D.1), we have that

$$\langle \mathbf{w}_{j,r}^{(T_2,0)}, j\boldsymbol{\mu}_2\rangle = \langle \mathbf{w}_{j,r}^{(T_1,0)}, j\boldsymbol{\mu}_2\rangle + \gamma(\boldsymbol{\mu}_1)_{j,r}^{(T_1,t_{end})} \cdot \frac{\|\boldsymbol{\mu}_2\|_2 \cos\theta_{1,2}}{\|\boldsymbol{\mu}_1\|_2}$$

$$\leq \langle \mathbf{w}_{j,r}^{(T_1,0)}, j\boldsymbol{\mu}_2\rangle - \sqrt{2\log(8m/\delta)} \cdot \sigma_0\|\boldsymbol{\mu}_2\|_2$$

$$\leq 0,$$

where the first inequality is by $\cos\theta_{1,2} \leq -\sqrt{2\log(8m/\delta)} \cdot \sigma_0\|\boldsymbol{\mu}_1\|_2/([m\log(1/\epsilon)]^{1/q})$, and the second inequality is by $|\langle \mathbf{w}_{j,r}^{(T_1,0)}, \boldsymbol{\mu}_2\rangle| \leq \sqrt{2\log(8m/\delta)} \cdot \sigma_0\|\boldsymbol{\mu}_2\|_2$ in Lemma C.2. Then, we can further prove that $\gamma(\boldsymbol{\mu}_2)_{j,r}^{(T_2,t_{end})} = -\langle \mathbf{w}_{j,r}^{(T_2,0)}, j\boldsymbol{\mu}_2\rangle$ by $\eta_2 \leq m/(q\alpha^{q-2}\|\boldsymbol{\mu}_2\|_2^2)$ in Condition 3.1, which is analogous to the proof of Lemma F.1. $\qquad\square$

**Lemma F.3.** *Under the same conditions as Theorem 3.2, when $r \in \{r \in [m] : \langle \mathbf{w}_{j,r}^{(T_1,0)}, j\boldsymbol{\mu}_1\rangle \leq 0\} \cap \{r \in [m] : \langle \mathbf{w}_{j,r}^{(T_1,0)}, j\boldsymbol{\mu}_1^\perp\rangle > 0\}$, we have that $\langle \mathbf{w}_{j,r}^{(T_2,0)}, j\boldsymbol{\mu}_2\rangle = \langle \mathbf{w}_{j,r}^{(T_1,0)}, j\boldsymbol{\mu}_1^\perp\rangle$, $|\langle \mathbf{w}_{j,r}^{(T_2,0)}, \boldsymbol{\mu}_2\rangle| \leq \sqrt{2\log(8m/\delta)} \cdot \sigma_0\|\boldsymbol{\mu}_1^\perp\|_2$ and* $\max_{r\in[m]}\langle \mathbf{w}_{j,r}^{(T_2,0)}, j\boldsymbol{\mu}_2\rangle \geq \sigma_0\|\boldsymbol{\mu}_1^\perp\|_2/2$ *for $j \in \{\pm1\}$, where $\boldsymbol{\mu}_1^\perp = \boldsymbol{\mu}_2 - \|\boldsymbol{\mu}_2\|_2 \cos\theta_{1,2} \cdot \frac{\boldsymbol{\mu}_1}{\|\boldsymbol{\mu}_1\|_2}$ is orthogonal to the signal vector $\boldsymbol{\mu}_1$.*

*proof of Lemma F.3.* By $\langle \mathbf{w}_{j,r}^{(T_1,0)}, j\boldsymbol{\mu}_1\rangle \leq 0$ and Lemma F.1, we know that $\gamma(\boldsymbol{\mu}_1)_{j,r}^{(T_1,t_{end})} = -\langle \mathbf{w}_{j,r}^{(T_1,0)}, j\boldsymbol{\mu}_1\rangle$. Then

$$\langle \mathbf{w}_{j,r}^{(T_2,0)}, j\boldsymbol{\mu}_2\rangle = \langle \mathbf{w}_{j,r}^{(T_1,0)}, j\boldsymbol{\mu}_2\rangle + \gamma(\boldsymbol{\mu}_1)_{j,r}^{(T_1,t_{end})} \cdot \frac{\|\boldsymbol{\mu}_2\|_2 \cos\theta_{1,2}}{\|\boldsymbol{\mu}_1\|_2}$$

$$= \langle \mathbf{w}_{j,r}^{(T_1,0)}, j\boldsymbol{\mu}_2\rangle - \langle \mathbf{w}_{j,r}^{(T_1,0)}, j\boldsymbol{\mu}_1\rangle \cdot \frac{\|\boldsymbol{\mu}_2\|_2 \cos\theta_{1,2}}{\|\boldsymbol{\mu}_1\|_2}$$

$$= j \cdot \langle \mathbf{w}_{j,r}^{(T_1,0)}, \boldsymbol{\mu}_2 - \|\boldsymbol{\mu}_2\|_2 \cos\theta_{1,2} \cdot \frac{\boldsymbol{\mu}_1}{\|\boldsymbol{\mu}_1\|_2}\rangle$$

$$= \langle \mathbf{w}_{j,r}^{(T_1,0)}, j\boldsymbol{\mu}_1^\perp\rangle$$

$$> 0.$$

By Lemma C.2, we further have that $|\langle \mathbf{w}_{j,r}^{(T_2,0)}, \boldsymbol{\mu}_2\rangle| \leq \sqrt{2\log(8m/\delta)} \cdot \sigma_0\|\boldsymbol{\mu}_1^\perp\|_2$ and $\max_{r\in[m]}\langle \mathbf{w}_{j,r}^{(T_2,0)}, j\boldsymbol{\mu}_2\rangle \geq \sigma_0\|\boldsymbol{\mu}_1^\perp\|_2/2$, which completes the proof. $\qquad\square$

**Lemma F.4.** *Under the same conditions as Theorem 3.2, when $r \in \{r \in [m] : \langle \mathbf{w}_{j,r}^{(T_1,0)}, j\boldsymbol{\mu}_1\rangle \leq 0\} \cap \{r \in [m] : \langle \mathbf{w}_{j,r}^{(T_1,0)}, j\boldsymbol{\mu}_1^\perp\rangle \leq 0\}$, we have that $\langle \mathbf{w}_{j,r}^{(T_2,0)}, j\boldsymbol{\mu}_2\rangle = \langle \mathbf{w}_{j,r}^{(T_1,0)}, j\boldsymbol{\mu}_1^\perp\rangle \leq 0$ and $\gamma(\boldsymbol{\mu}_2)_{j,r}^{(T_2,t_{end})} = -\langle \mathbf{w}_{j,r}^{(T_1,0)}, j\boldsymbol{\mu}_1^\perp\rangle$ for $j \in \{\pm1\}$, where $\boldsymbol{\mu}_1^\perp = \boldsymbol{\mu}_2 - \|\boldsymbol{\mu}_2\|_2 \cos\theta_{1,2} \cdot \frac{\boldsymbol{\mu}_1}{\|\boldsymbol{\mu}_1\|_2}$ is orthogonal to the signal vector $\boldsymbol{\mu}_1$.*

*proof of Lemma F.4.* By Lemma F.3, we have that $\langle \mathbf{w}_{j,r}^{(T_2,0)}, j\boldsymbol{\mu}_2 \rangle = \langle \mathbf{w}_{j,r}^{(T_1,0)}, j\boldsymbol{\mu}_1^\perp \rangle \leq 0$. Then, we can further prove that $\gamma(\boldsymbol{\mu}_2)_{j,r}^{(T_2,t_{end})} = -\langle \mathbf{w}_{j,r}^{(T_2,0)}, j\boldsymbol{\mu}_2 \rangle = -\langle \mathbf{w}_{j,r}^{(T_1,0)}, j\boldsymbol{\mu}_1^\perp \rangle$, which is similar to the proof of Lemma F.1. □

**Proposition F.5** (Obtuse-angle Case). *Under Condition 3.1, and when* $\cos\theta_{1,2} < 0$, *for* $0 \leq t \leq T^*$, *we have that*

$$0 \leq \gamma(\boldsymbol{\mu}_2)_{j,r}^{(T_2,t)}, \overline{\rho}(\boldsymbol{\xi}_2)_{j,r,i}^{(T_2,t)} \leq \alpha,$$

$$0 \geq \underline{\rho}(\boldsymbol{\xi}_2)_{j,r,i}^{(T_2,t)} \geq -\beta_2 - 16n_2\sqrt{\frac{\log(4n_2^2/\delta)}{d}}\alpha \geq -\alpha.$$

*for all* $r \in [m]$, $j \in \{\pm 1\}$ *and* $i \in [n_2]$.

*Proof of Proposition F.5.* It is known that the noise vector is orthogonal to the signal vector, which implies that the proof for $0 \leq \overline{\rho}(\boldsymbol{\xi}_2)_{j,r,i}^{(T_2,t)} \leq \alpha$ and $0 \geq \underline{\rho}(\boldsymbol{\xi}_2)_{j,r,i}^{(T_2,t)} \geq -\beta_2 - 16n_2\sqrt{\frac{\log(4n_2^2/\delta)}{d}}\alpha \geq -\alpha$ follows directly from the proof provided for the Acute Angle Case in Proposition E.2. Therefore, it remains to prove $0 \leq \gamma(\boldsymbol{\mu}_2)_{j,r}^{(T_2,t)} \leq \alpha$. When $r \in \{r \in [m] : \langle \mathbf{w}_{j,r}^{(T_1,0)}, j\boldsymbol{\mu}_1 \rangle > 0\}$, by the proof of Lemma F.2, we have that

$$\gamma(\boldsymbol{\mu}_2)_{j,r}^{(T_2,t)} \leq -\langle \mathbf{w}_{j,r}^{(T_1,0)}, j\boldsymbol{\mu}_2 \rangle - \gamma(\boldsymbol{\mu}_1)_{j,r}^{(T_1,t_{end})} \cdot \frac{\|\boldsymbol{\mu}_2\|_2 \cos\theta_{1,2}}{\|\boldsymbol{\mu}_1\|_2}$$

$$\leq \alpha,$$

where the last inequality is by $|\langle \mathbf{w}_{j,r}^{(T_1,0)}, \boldsymbol{\mu}_2 \rangle| \leq \sqrt{2\log(8m/\delta)} \cdot \sigma_0 \|\boldsymbol{\mu}_2\|_2$ in Lemma C.2 and $\gamma(\boldsymbol{\mu}_1)_{j,r}^{(T_1,t_{end})} \leq \alpha$ in Proposition 5.3 by Cao et al. (2022). When $r \in \{r \in [m] : \langle \mathbf{w}_{j,r}^{(T_1,0)}, j\boldsymbol{\mu}_1 \rangle \leq 0\} \cap \{r \in [m] : \langle \mathbf{w}_{j,r}^{(T_1,0)}, j\boldsymbol{\mu}_1^\perp \rangle \leq 0\}$, by the proof of Lemma F.4, we have that

$$\gamma(\boldsymbol{\mu}_2)_{j,r}^{(T_2,t)} \leq -\langle \mathbf{w}_{j,r}^{(T_1,0)}, j\boldsymbol{\mu}_1^\perp \rangle \leq \alpha,$$

where the last inequality is by $|\langle \mathbf{w}_{j,r}^{(T_1,0)}, \boldsymbol{\mu}_1^\perp \rangle| \leq \sqrt{2\log(8m/\delta)} \cdot \sigma_0 \|\boldsymbol{\mu}_1^\perp\|_2$ in Lemma C.2. When $r \in \{r \in [m] : \langle \mathbf{w}_{j,r}^{(T_1,0)}, j\boldsymbol{\mu}_1 \rangle \leq 0\} \cap \{r \in [m] : \langle \mathbf{w}_{j,r}^{(T_1,0)}, j\boldsymbol{\mu}_1^\perp \rangle > 0\}$, by Lemma F.3, we have that

$$|\langle \mathbf{w}_{j,r}^{(T_2,0)}, \boldsymbol{\mu}_2 \rangle| \leq \sqrt{2\log(8m/\delta)} \cdot \sigma_0 \|\boldsymbol{\mu}_1^\perp\|_2$$

$$\max_{r \in [m]} \langle \mathbf{w}_{j,r}^{(T_2,0)}, j\boldsymbol{\mu}_2 \rangle \geq \sigma_0 \|\boldsymbol{\mu}_1^\perp\|_2/2.$$

Then we can use induction analogous to Proposition 5.3 in Cao et al. (2022), to prove $0 \leq \gamma(\boldsymbol{\mu}_2)_{j,r}^{(T_2,t)} \leq \alpha$ for $r \in \{r \in [m] : \langle \mathbf{w}_{j,r}^{(T_1,0)}, j\boldsymbol{\mu}_1 \rangle \leq 0\} \cap \{r \in [m] : \langle \mathbf{w}_{j,r}^{(T_1,0)}, j\boldsymbol{\mu}_1^\perp \rangle > 0\}$, which completes the proof. □

### F.2. Signal Learning

In this section, we consider the signal learning case under the condition that $n\|\boldsymbol{\mu}_1^\perp\|_2^q \geq \widetilde{\Omega}(\sigma_{p_2}^q(\sqrt{d})^q)$. Based on the analysis of neural behavior in Subsection F.1, we have that

$$|\langle \mathbf{w}_{j,r}^{(T_2,0)}, \boldsymbol{\mu}_2 \rangle| \leq \sqrt{2\log(8m/\delta)} \cdot \sigma_0 \|\boldsymbol{\mu}_1^\perp\|_2$$

$$\max_{r \in [m]} \langle \mathbf{w}_{j,r}^{(T_2,0)}, j\boldsymbol{\mu}_2 \rangle \geq \sigma_0 \|\boldsymbol{\mu}_1^\perp\|_2/2,$$

for $r \in \{r \in [m] : \langle \mathbf{w}_{j,r}^{(T_1,0)}, j\boldsymbol{\mu}_1 \rangle \leq 0\} \cap \{r \in [m] : \langle \mathbf{w}_{j,r}^{(T_1,0)}, j\boldsymbol{\mu}_1^\perp \rangle > 0\}$ and $j \in \{\pm 1\}$, where $\boldsymbol{\mu}_1^\perp = \boldsymbol{\mu}_2 - \|\boldsymbol{\mu}_2\|_2 \cos\theta_{1,2} \cdot \frac{\boldsymbol{\mu}_1}{\|\boldsymbol{\mu}_1\|_2}$ is orthogonal to the signal vector $\boldsymbol{\mu}_1$. Note that the component of these neurons along the signal direction exhibits an evolution similar to that of the growing signal component in task $T_1$, as analyzed by Cao et al. (2022) Furthermore, we know that the noise vector is orthogonal to the signal vector. Thus, similarly to the acute angle case, we have $\max_{j,r,i} |\langle \mathbf{w}_{j,r}^{(T_2,0)}, \boldsymbol{\xi}_{2,i} \rangle| \leq 4\sqrt{\log(8mn_2/\delta)} \cdot \sigma_0\sigma_{p_2}\sqrt{d}$ in D.3. Then we utilize a two-stage technique similar to Lemma 5.5 and Lemma 5.6 in Cao et al. (2022) to analyze the learning process.

### F.2.1. FIRST STAGE

**Lemma F.6** (Cao et al. (2022))**.** *Under the same conditions as Theorem 3.2, in particular if we choose*

$$n_2 \cdot \|\boldsymbol{\mu}_1^\perp\|_2^q / (\sigma_{p_2}\sqrt{d})^q \geq C \log(6/\sigma_0\|\boldsymbol{\mu}_1^\perp\|_2) 2^{2q+6} [6\log(8mn_2/\delta)]^{(q-1)/2}, \tag{F.1}$$

*where $C = O(1)$ is a positive constant, there exists time*

$$t_1 = \frac{C \log(6/\sigma_0\|\boldsymbol{\mu}_1^\perp\|_2) 2^{q+1} m}{\eta_2 \sigma_0^{q-2}\|\boldsymbol{\mu}_1^\perp\|_2^q}$$

*such that*

- $\max_r \gamma(\boldsymbol{\mu}_2)_{j,r}^{(T_2,t_1)} \geq 2$ *for* $j \in \{\pm 1\}$.

- $|\rho_{j,r,i}^{(T_2,t_1)}| \leq \sigma_0\sigma_{p_2}\sqrt{d}/2$ *for all* $j \in \{\pm 1\}, r \in [m], i \in [n_2]$ *and* $0 \leq t \leq t_1$.

### F.2.2. SECOND STAGE

By the results we get in the first stage, the following property holds:

- $\max_r \gamma(\boldsymbol{\mu}_2)_{j,r}^{(T_2,t_1)} \geq 2, \forall j \in \{\pm 1\}$.

- $\max_{j,r,i} |\rho_{j,r,i}^{(T_2,t_1)}| \leq \widehat{\beta}$ where $\widehat{\beta} = \sigma_0\sigma_{p_2}\sqrt{d}/2$.

Now we choose $\mathbf{W}^*$ as follows:

$$\mathbf{w}_{j,r}^* = \mathbf{w}_{j,r}^{(T_1,0)} + 2qm \log(2q/\epsilon) \cdot j \cdot \frac{\boldsymbol{\mu}_2}{\|\boldsymbol{\mu}_2\|_2^2}.$$

Based on the above definition of $\mathbf{W}^*$, we have the following lemmas.

**Lemma F.7** (Cao et al. (2022))**.** *Under the same conditions as Theorem 3.2, let* $t^+ = t_1 + \left\lfloor \frac{\|\mathbf{W}^{(T_2,t_1)} - \mathbf{W}^*\|_F^2}{2\eta_2\epsilon} \right\rfloor = t_1 + \widetilde{O}(m^3\eta_2^{-1}\epsilon^{-1}\|\boldsymbol{\mu}_2\|_2^{-2})$. *Then we have* $\max_{j,r,i} |\rho_{j,r,i}^{(T_2,t)}| \leq 2\widehat{\beta} = \sigma_0\sigma_{p_2}\sqrt{d}$ *for all* $t_1 \leq t \leq t^+$. *Besides,*

$$\frac{1}{t - t_1 + 1} \sum_{s=t_1}^{t} L_{S_2}(\mathbf{W}^{(T_2,s)}) \leq \frac{\|\mathbf{W}^{(T_2,t_1)} - \mathbf{W}^*\|_F^2}{(2q-1)\eta_2(t - t_1 + 1)} + \frac{\epsilon}{2q-1}$$

*for all* $t_1 \leq t \leq t^+$, *and we can find an iterate* $t_{end}$ *with training loss smaller than* $\epsilon$ *within* $t^+$ *iterations.*

**Lemma F.8** (Cao et al. (2022))**.** *Let* $t^+$ *and* $t_{end}$ *be defined in Lemma F.7 respectively. Under the same conditions as Theorem 3.2, for any* $0 \leq t \leq t^+$ *with* $L_{S_2}(\mathbf{W}^{(T_2,t)}) \leq 1$, *it holds that* $L_{\mathcal{D}_2}(\mathbf{W}^{(T_2,t)}) \leq 6 \cdot L_{S_2}(\mathbf{W}^{(T_2,t)}) + \exp(-n_2^2)$. *Furthermore, since* $L_{S_2}(\mathbf{W}^{(T_2,t_{end})}) \leq \epsilon$, *it follows that* $L_{\mathcal{D}_2}(\mathbf{W}^{(T_2,t_{end})}) \leq 6\epsilon + \exp(-n_2^2)$.

### F.3. Forgetting

Similar to the acute-angle case, consider a new data point $(\mathbf{x}_1, y_1)$ with $\mathbf{x}_1 = [y_1\boldsymbol{\mu}_1, \boldsymbol{\xi}_1]$ following the distribution of task $T_1$ as defined in Definition 1.1. Furthermore, based on the analysis of signal learning, it follows that

$$\mathbf{w}_{j,r}^{(T_2,t_{end})} = \mathbf{w}_{j,r}^{(T_1,0)} + j \cdot \gamma(\boldsymbol{\mu}_1)_{j,r}^{(T_1,t_{end})} \cdot \frac{\boldsymbol{\mu}_1}{\|\boldsymbol{\mu}_1\|_2^2} + j \cdot \gamma(\boldsymbol{\mu}_2)_{j,r}^{(T_2,t_{end})} \cdot \frac{\boldsymbol{\mu}_2}{\|\boldsymbol{\mu}_2\|_2^2}$$

$$+ \sum_{i=1}^{n_1} \rho(\boldsymbol{\xi}_1)_{j,r,i}^{(T_1,t_{end})} \cdot \frac{\boldsymbol{\xi}_{1,i}}{\|\boldsymbol{\xi}_{1,i}\|_2^2} + \sum_{i=1}^{n_2} \rho(\boldsymbol{\xi}_2)_{j,r,i}^{(T_2,t_{end})} \cdot \frac{\boldsymbol{\xi}_{2,i}}{\|\boldsymbol{\xi}_{2,i}\|_2^2}. \tag{F.2}$$

- $\max_{j,r,i} |\rho(\boldsymbol{\xi}_1)_{j,r,i}^{(T_1,t_{end})}| \leq \sigma_0\sigma_{p_1}\sqrt{d}$.

- $\max_{j,r,i} |\rho(\boldsymbol{\xi}_2)_{j,r,i}^{(T_2,t_{end})}| \leq \sigma_0\sigma_{p_2}\sqrt{d}$.

**Lemma F.9.** *Under the same conditions as Theorem 3.2, when $r \in \{r \in [m] : \langle \mathbf{w}_{j,r}^{(T_1,0)}, j\boldsymbol{\mu}_1 \rangle > 0\}$, we have that*

$$\langle \mathbf{w}_{j,r}^{(T_1,t_{end})}, j\boldsymbol{\mu}_1 \rangle = \langle \mathbf{w}_{j,r}^{(T_1,0)}, j\boldsymbol{\mu}_1 \rangle + \gamma(\boldsymbol{\mu}_1)_{j,r}^{(T_1,t_{end})}$$
$$\langle \mathbf{w}_{j,r}^{(T_1,t_{end})}, -j\boldsymbol{\mu}_1 \rangle \leq 0,$$

*for $j \in \{\pm 1\}$.*

*Proof of Lemma F.9.* By the signal-noise decomposition (D.1), we have that

$$\langle \mathbf{w}_{j,r}^{(T_1,t_{end})}, j\boldsymbol{\mu}_1 \rangle = \langle \mathbf{w}_{j,r}^{(T_1,0)}, j\boldsymbol{\mu}_1 \rangle + \gamma(\boldsymbol{\mu}_1)_{j,r}^{(T_1,t_{end})}.$$

Then we further have that

$$\langle \mathbf{w}_{j,r}^{(T_1,t_{end})}, -j\boldsymbol{\mu}_1 \rangle = \langle \mathbf{w}_{j,r}^{(T_1,0)}, -j\boldsymbol{\mu}_1 \rangle - \gamma(\boldsymbol{\mu}_1)_{j,r}^{(T_1,t_{end})}$$
$$\leq 0,$$

where the inequality is by $\langle \mathbf{w}_{j,r}^{(T_1,0)}, j\boldsymbol{\mu}_1 \rangle \geq 0$ and $\gamma(\boldsymbol{\mu}_1)_{j,r}^{(T_1,t_{end})} \geq 0$, which completes the proof. $\square$

**Lemma F.10.** *Under the same conditions as Theorem 3.2, when $r \in \{r \in [m] : \langle \mathbf{w}_{j,r}^{(T_1,0)}, j\boldsymbol{\mu}_1 \rangle \leq 0\}$, we have that*

$$\langle \mathbf{w}_{j,r}^{(T_1,t_{end})}, j\boldsymbol{\mu}_1 \rangle = 0$$
$$\langle \mathbf{w}_{j,r}^{(T_1,t_{end})}, -j\boldsymbol{\mu}_1 \rangle = 0,$$

*for $j \in \{\pm 1\}$.*

*Proof of Lemma F.10.* By the signal-noise decomposition (D.1) and Lemma F.1, we have that $\langle \mathbf{w}_{j,r}^{(T_1,t_{end})}, j\boldsymbol{\mu}_1 \rangle = \langle \mathbf{w}_{j,r}^{(T_1,0)}, j\boldsymbol{\mu}_1 \rangle + \gamma(\boldsymbol{\mu}_1)_{j,r}^{(T_1,t_{end})} = 0$ and $\langle \mathbf{w}_{j,r}^{(T_1,t_{end})}, -j\boldsymbol{\mu}_1 \rangle = -\langle \mathbf{w}_{j,r}^{(T_1,t_{end})}, j\boldsymbol{\mu}_1 \rangle = 0$, which completes the proof. $\square$

**Lemma F.11.** *Under the same conditions as Theorem 3.2, when $r \in \{r \in [m] : \langle \mathbf{w}_{j,r}^{(T_1,0)}, j\boldsymbol{\mu}_1 \rangle > 0\}$, we have that*

$$\langle \mathbf{w}_{j,r}^{(T_2,t_{end})}, j\boldsymbol{\mu}_1 \rangle \geq \gamma(\boldsymbol{\mu}_1)_{j,r}^{(T_1,t_{end})} \cdot (1 - \cos^2 \theta_{1,2}) - \widetilde{O}(m^{-1}n^{-1})$$
$$\langle \mathbf{w}_{j,r}^{(T_2,t_{end})}, -j\boldsymbol{\mu}_1 \rangle \leq \sqrt{2\log(8m/\delta)} \cdot \sigma_0 \|\boldsymbol{\mu}_1\|_2$$
$$\langle \mathbf{w}_{j,r}^{(T_2,t_{end})}, j\boldsymbol{\mu}_2 \rangle = 0$$
$$\langle \mathbf{w}_{j,r}^{(T_2,t_{end})}, -j\boldsymbol{\mu}_2 \rangle = 0,$$

*for $j \in \{\pm 1\}$, where $n = \max\{n_1, n_2\}$ and $\boldsymbol{\mu}_2^{\perp} = \boldsymbol{\mu}_1 - \|\boldsymbol{\mu}_1\|_2 \cos\theta_{1,2} \cdot \frac{\boldsymbol{\mu}_2}{\|\boldsymbol{\mu}_2\|_2}$ is orthogonal to the signal vector $\boldsymbol{\mu}_2$.*

*Proof of Lemma F.11.* By Equation (F.2) and Lemma F.2, we have that

$$\langle \mathbf{w}_{j,r}^{(T_2,t_{end})}, j\boldsymbol{\mu}_1 \rangle = \langle \mathbf{w}_{j,r}^{(T_1,0)}, j\boldsymbol{\mu}_1 \rangle + \gamma(\boldsymbol{\mu}_1)_{j,r}^{(T_1,t_{end})} + \gamma(\boldsymbol{\mu}_2)_{j,r}^{(T_2,t_{end})} \cdot \frac{\|\boldsymbol{\mu}_1\|_2 \cos\theta_{1,2}}{\|\boldsymbol{\mu}_2\|_2}$$
$$= \langle \mathbf{w}_{j,r}^{(T_1,0)}, j\boldsymbol{\mu}_2^{\perp} \rangle + \gamma(\boldsymbol{\mu}_1)_{j,r}^{(T_1,t_{end})} \cdot (1 - \cos^2 \theta_{1,2})$$
$$\geq \gamma(\boldsymbol{\mu}_1)_{j,r}^{(T_1,t_{end})} \cdot (1 - \cos^2 \theta_{1,2}) - \widetilde{O}(m^{-1}n^{-1}),$$

where the inequality is by $|\langle \mathbf{w}_{j,r}^{(T_1,0)}, \boldsymbol{\mu}_2^{\perp} \rangle| \leq \sqrt{2\log(8m/\delta)} \cdot \sigma_0 \|\boldsymbol{\mu}_1\|_2$ similar to Lemma C.2 and $\sigma_0 \|\boldsymbol{\mu}_1\|_2 \leq \widetilde{O}(m^{-1}n^{-1})$ in Condition 3.1. Similarly,

$$\langle \mathbf{w}_{j,r}^{(T_2,t_{end})}, -j\boldsymbol{\mu}_1 \rangle = \langle \mathbf{w}_{j,r}^{(T_1,0)}, -j\boldsymbol{\mu}_2^{\perp} \rangle - \gamma(\boldsymbol{\mu}_1)_{j,r}^{(T_1,t_{end})} \cdot (1 - \cos^2 \theta_{1,2})$$
$$\leq \sqrt{2\log(8m/\delta)} \cdot \sigma_0 \|\boldsymbol{\mu}_2^{\perp}\|_2$$

$$\leq \sqrt{2 \log(8m/\delta)} \cdot \sigma_0 \|\boldsymbol{\mu}_1\|_2,$$

where the first inequality is by $|\langle \mathbf{w}_{j,r}^{(T_1,0)}, \boldsymbol{\mu}_2^\perp \rangle| \leq \sqrt{2 \log(8m/\delta)} \cdot \sigma_0 \|\boldsymbol{\mu}_2^\perp\|_2$ similar to Lemma C.2, the second inequality is by the definition of $\boldsymbol{\mu}_2^\perp$. Then by Equation (4.1) and Lemma F.2, we have that $\langle \mathbf{w}_{j,r}^{(T_2,t_{end})}, j\boldsymbol{\mu}_2 \rangle = \langle \mathbf{w}_{j,r}^{(T_2,0)}, j\boldsymbol{\mu}_2 \rangle + \gamma(\boldsymbol{\mu}_2)_{j,r}^{(T_2,t_{end})} = 0$ and $\langle \mathbf{w}_{j,r}^{(T_2,t_{end})}, -j\boldsymbol{\mu}_2 \rangle = \langle \mathbf{w}_{j,r}^{(T_2,0)}, -j\boldsymbol{\mu}_2 \rangle - \gamma(\boldsymbol{\mu}_2)_{j,r}^{(T_2,t_{end})} = 0$, which completes the proof. $\square$

**Lemma F.12.** *Under the same conditions as Theorem 3.2, when* $r \in \{r \in [m] : \langle \mathbf{w}_{j,r}^{(T_1,0)}, j\boldsymbol{\mu}_1 \rangle \leq 0\} \cap \{r \in [m] : \langle \mathbf{w}_{j,r}^{(T_1,0)}, j\boldsymbol{\mu}_1^\perp \rangle > 0\}$, *we have that*

$$\langle \mathbf{w}_{j,r}^{(T_2,t_{end})}, j\boldsymbol{\mu}_1 \rangle \leq 0$$

$$\langle \mathbf{w}_{j,r}^{(T_2,t_{end})}, -j\boldsymbol{\mu}_1 \rangle = -\gamma(\boldsymbol{\mu}_2)_{j,r}^{(T_2,t_{end})} \cdot \frac{\|\boldsymbol{\mu}_1\|_2 \cos\theta_{1,2}}{\|\boldsymbol{\mu}_2\|_2}$$

$$\langle \mathbf{w}_{j,r}^{(T_2,t_{end})}, j\boldsymbol{\mu}_2 \rangle \geq \gamma(\boldsymbol{\mu}_2)_{j,r}^{(T_2,t_{end})}$$

$$\langle \mathbf{w}_{j,r}^{(T_2,t_{end})}, -j\boldsymbol{\mu}_2 \rangle \leq 0,$$

*for* $j \in \{\pm 1\}$, *where* $\boldsymbol{\mu}_1^\perp = \boldsymbol{\mu}_2 - \|\boldsymbol{\mu}_2\|_2 \cos\theta_{1,2} \cdot \frac{\boldsymbol{\mu}_1}{\|\boldsymbol{\mu}_1\|_2}$ *is orthogonal to the signal vector* $\boldsymbol{\mu}_1$.

*Proof of Lemma F.12.* By Equation (F.2) and Lemma F.1, we have that

$$\langle \mathbf{w}_{j,r}^{(T_2,t_{end})}, j\boldsymbol{\mu}_1 \rangle = \langle \mathbf{w}_{j,r}^{(T_1,0)}, j\boldsymbol{\mu}_1 \rangle + \gamma(\boldsymbol{\mu}_1)_{j,r}^{(T_1,t_{end})} + \gamma(\boldsymbol{\mu}_2)_{j,r}^{(T_2,t_{end})} \cdot \frac{\|\boldsymbol{\mu}_1\|_2 \cos\theta_{1,2}}{\|\boldsymbol{\mu}_2\|_2}$$

$$= \gamma(\boldsymbol{\mu}_2)_{j,r}^{(T_2,t_{end})} \cdot \frac{\|\boldsymbol{\mu}_1\|_2 \cos\theta_{1,2}}{\|\boldsymbol{\mu}_2\|_2}$$

$$\leq 0,$$

where the inequality is by $\cos\theta_{1,2} \leq 0$. Similarly,

$$\langle \mathbf{w}_{j,r}^{(T_2,t_{end})}, -j\boldsymbol{\mu}_1 \rangle = -\gamma(\boldsymbol{\mu}_2)_{j,r}^{(T_2,t_{end})} \cdot \frac{\|\boldsymbol{\mu}_1\|_2 \cos\theta_{1,2}}{\|\boldsymbol{\mu}_2\|_2}.$$

Then by Equation (F.2) and Lemma F.3, we have that

$$\langle \mathbf{w}_{j,r}^{(T_2,t_{end})}, j\boldsymbol{\mu}_2 \rangle = \langle \mathbf{w}_{j,r}^{(T_1,0)}, j\boldsymbol{\mu}_1^\perp \rangle + \gamma(\boldsymbol{\mu}_2)_{j,r}^{(T_2,t_{end})}$$

$$\geq \gamma(\boldsymbol{\mu}_2)_{j,r}^{(T_2,t_{end})},$$

where the inequality is by $\langle \mathbf{w}_{j,r}^{(T_1,0)}, j\boldsymbol{\mu}_1^\perp \rangle > 0$. Then we further have that $\langle \mathbf{w}_{j,r}^{(T_2,t_{end})}, -j\boldsymbol{\mu}_2 \rangle = -\langle \mathbf{w}_{j,r}^{(T_2,t_{end})}, j\boldsymbol{\mu}_2 \rangle \leq 0$, which completes the proof. $\square$

**Lemma F.13.** *Under the same conditions as Theorem 3.2, when* $r \in \{r \in [m] : \langle \mathbf{w}_{j,r}^{(T_1,0)}, j\boldsymbol{\mu}_1 \rangle \leq 0\} \cap \{r \in [m] : \langle \mathbf{w}_{j,r}^{(T_1,0)}, j\boldsymbol{\mu}_1^\perp \rangle \leq 0\}$, *we have that*

$$\langle \mathbf{w}_{j,r}^{(T_2,t_{end})}, j\boldsymbol{\mu}_1 \rangle \leq 0$$

$$\langle \mathbf{w}_{j,r}^{(T_2,t_{end})}, -j\boldsymbol{\mu}_1 \rangle \leq \sqrt{2 \log(8m/\delta)} \cdot \sigma_0 \|\boldsymbol{\mu}_1\|_2$$

$$\langle \mathbf{w}_{j,r}^{(T_2,t_{end})}, j\boldsymbol{\mu}_2 \rangle = 0$$

$$\langle \mathbf{w}_{j,r}^{(T_2,t_{end})}, -j\boldsymbol{\mu}_2 \rangle = 0,$$

*for* $j \in \{\pm 1\}$, *where* $\boldsymbol{\mu}_1^\perp = \boldsymbol{\mu}_2 - \|\boldsymbol{\mu}_2\|_2 \cos\theta_{1,2} \cdot \frac{\boldsymbol{\mu}_1}{\|\boldsymbol{\mu}_1\|_2}$ *is orthogonal to the signal vector* $\boldsymbol{\mu}_1$.

*Proof of Lemma F.13.* Similar to the proof of Lemma F.12, we have that

$$\langle \mathbf{w}_{j,r}^{(T_2,t_{end})}, j\boldsymbol{\mu}_1 \rangle = \gamma(\boldsymbol{\mu}_2)_{j,r}^{(T_2,t_{end})} \cdot \frac{\|\boldsymbol{\mu}_1\|_2 \cos \theta_{1,2}}{\|\boldsymbol{\mu}_2\|_2} \le 0,$$

$$\langle \mathbf{w}_{j,r}^{(T_2,t_{end})}, -j\boldsymbol{\mu}_1 \rangle = -\gamma(\boldsymbol{\mu}_2)_{j,r}^{(T_2,t_{end})} \cdot \frac{\|\boldsymbol{\mu}_1\|_2 \cos \theta_{1,2}}{\|\boldsymbol{\mu}_2\|_2}$$

$$= \langle \mathbf{w}_{j,r}^{(T_1,0)}, j\boldsymbol{\mu}_1^\perp \rangle \cdot \frac{\|\boldsymbol{\mu}_1\|_2 \cos \theta_{1,2}}{\|\boldsymbol{\mu}_2\|_2}$$

$$\le \sqrt{2\log(8m/\delta)} \cdot \sigma_0 \|\boldsymbol{\mu}_1\|_2,$$

where the third equality is by Lemma F.4, the inequality is by $|\langle \mathbf{w}_{j,r}^{(T_1,0)}, \boldsymbol{\mu}_1^\perp \rangle| \le \sqrt{2\log(8m/\delta)} \cdot \sigma_0 \|\boldsymbol{\mu}_1^\perp\|_2$ similar to Lemma C.2 and the definition of $\boldsymbol{\mu}_1^\perp$. Then by Equation (F.2) and Lemma F.4, we have that $\langle \mathbf{w}_{j,r}^{(T_2,t_{end})}, j\boldsymbol{\mu}_2 \rangle = \langle \mathbf{w}_{j,r}^{(T_1,0)}, j\boldsymbol{\mu}_1^\perp \rangle + \gamma(\boldsymbol{\mu}_2)_{j,r}^{(T_2,t_{end})} = 0$ and $\langle \mathbf{w}_{j,r}^{(T_2,t_{end})}, -j\boldsymbol{\mu}_2 \rangle = -\langle \mathbf{w}_{j,r}^{(T_2,t_{end})}, j\boldsymbol{\mu}_2 \rangle = 0$, which completes the proof. □

**Lemma F.14.** *Under Condition 3.1, we have that* $|\langle \mathbf{w}_{j,r}^{(T_2,t_{end})}, \boldsymbol{\xi}_1 \rangle| \le 6\sqrt{\log(8mn/\delta)}\sigma_0\sigma_{p_1}\sqrt{d}$ *for all* $r \in [m]$ *and* $j \in \{\pm 1\}$, *where* $n = \max\{n_1, n_2\}$.

*Proof of Lemma F.14.* By (F.2), we have that

$$\langle \mathbf{w}_{j,r}^{(T_2,t_{end})}, \boldsymbol{\xi}_1 \rangle = \langle \mathbf{w}_{j,r}^{(T_1,0)}, \boldsymbol{\xi}_1 \rangle + \sum_{i=1}^{n_1} \rho_{j,r,i}^{(T_1,t_{end})} \cdot \|\boldsymbol{\xi}_{1,i}\|_2^{-2} \cdot \langle \boldsymbol{\xi}_{1,i}, \boldsymbol{\xi}_1 \rangle + \sum_{i=1}^{n_2} \rho_{j,r,i}^{(T_2,t_{end})} \cdot \|\boldsymbol{\xi}_{2,i}\|_2^{-2} \cdot \langle \boldsymbol{\xi}_{2,i}, \boldsymbol{\xi}_1 \rangle$$

$$\le 2\sqrt{\log(8mn_1/\delta)}\sigma_0\sigma_{p_1}\sqrt{d} + (4n_1\sqrt{\frac{\log(4n_1^2/\delta)}{d}} + 1)\sigma_0\sigma_{p_1}\sqrt{d} + (4n_2\sqrt{\frac{\log(4n_2^2/\delta)}{d}} + 1)\sigma_0\sigma_{p_1}\sqrt{d}$$

$$\le 2\sqrt{\log(8mn_1/\delta)}\sigma_0\sigma_{p_1}\sqrt{d} + 2\sqrt{\log(8mn_1/\delta)}\sigma_0\sigma_{p_1}\sqrt{d} + 2\sqrt{\log(8mn_2/\delta)}\sigma_0\sigma_{p_1}\sqrt{d}$$

$$\le 6\sqrt{\log(8mn/\delta)}\sigma_0\sigma_{p_1}\sqrt{d},$$

where the first inequality is by $\max_{j,r,i}|\rho(\boldsymbol{\xi}_1)_{j,r,i}^{(T_1,t_{end})}| \le \sigma_0\sigma_{p_1}\sqrt{d}$ and $\max_{j,r,i}|\rho(\boldsymbol{\xi}_2)_{j,r,i}^{(T_2,t_{end})}| \le \sigma_0\sigma_{p_2}\sqrt{d}$, the second inequality is due to $d \ge \widetilde{\Omega}(m^2 \cdot \max\{n_1^4, n_2^4\})$ by Condition 3.1, and the last inequality is by $n = \max\{n_1, n_2\}$. □

Using a proof technique similar to that in Lemma F.14, we can further prove the following lemma.

**Lemma F.15.** *Under Condition 3.1, we have that* $|\langle \mathbf{w}_{j,r}^{(T_1,t_{end})}, \boldsymbol{\xi}_{1,i'} \rangle| \le 6\sqrt{\log(8mn/\delta)}\sigma_0\sigma_{p_1}\sqrt{d}$ *and* $|\langle \mathbf{w}_{j,r}^{(T_2,t_{end})}, \boldsymbol{\xi}_{2,i''} \rangle| \le 6\sqrt{\log(8mn/\delta)}\sigma_0\sigma_{p_1}\sqrt{d}$ *for all* $r \in [m]$, $j \in \{\pm 1\}$, $i' \in [n_1]$ *and* $i'' \in [n_2]$, *where* $n = \max\{n_1, n_2\}$.

For ease of discussion, let $I_{j,1} = \{r \in [m] : \langle \mathbf{w}_{j,r}^{(T_1,0)}, j\boldsymbol{\mu}_1 \rangle > 0\}$, $I_{j,1}^c = \{r \in [m] : \langle \mathbf{w}_{j,r}^{(T_1,0)}, j\boldsymbol{\mu}_1 \rangle \le 0\}$, $I_{j,2} = \{r \in [m] : \langle \mathbf{w}_{j,r}^{(T_1,0)}, j\boldsymbol{\mu}_1 \rangle \le 0\} \cap \{r \in [m] : \langle \mathbf{w}_{j,r}^{(T_1,0)}, j\boldsymbol{\mu}_1^\perp \rangle > 0\}$ and $I_{j,3} = \{r \in [m] : \langle \mathbf{w}_{j,r}^{(T_1,0)}, j\boldsymbol{\mu}_1 \rangle \le 0\} \cap \{r \in [m] : \langle \mathbf{w}_{j,r}^{(T_1,0)}, j\boldsymbol{\mu}_1^\perp \rangle \le 0\}$ for $j \in \{\pm 1\}$, then the following lemma holds.

**Lemma F.16** (Cao et al. (2022)). *Suppose that* $\delta > 0$ *and* $n \ge 32\log(4/\delta)$. *Then with probability at least* $1 - \delta$, $|I_{j,1}|, |I_{j,1}^c| \ge m/4$ *and* $|I_{j,2}|, |I_{j,3}| \ge m/8$ *for* $j \in \{\pm 1\}$.

**Lemma F.17.** *Under the same conditions as Theorem 3.2, denote* $\overline{\gamma(\boldsymbol{\mu}_1)_j} = \frac{1}{m}\sum_{r \in I_{j,1}}[\gamma(\boldsymbol{\mu}_1)_{j,r}^{(T_1,t_{end})}]^q$. *Then we have*

$$\log(\frac{1}{\epsilon + o(\epsilon)}) - \widetilde{O}(m^{-1}n^{-1}) \le \overline{\gamma(\boldsymbol{\mu}_1)_j} \le \log(\frac{C_0}{\epsilon + o(\epsilon)}) + \widetilde{O}(m^{-1}n^{-1}),$$

*for* $j \in \{\pm 1\}$, *where* $C_0 > 1$ *is a constant and* $n = \max\{n_1, n_2\}$.

*Proof of Lemma F.17.* Since $L_{S_1}(\mathbf{W}^{(T_1,t_{end})}) \le \epsilon$, there must exist one $(\mathbf{x}_{1,i}, y_{1,i})$ and a constant $C_0 \ge 0$ such that $\epsilon/C_0 \le \ell(y_{1,i} \cdot f(\mathbf{W}^{(T_1,t_{end})}, \mathbf{x}_{1,i})) \le \epsilon$, which implies that

$$\log(\frac{1}{\epsilon + o(\epsilon)}) \le y_{1,i} \cdot f(\mathbf{W}^{(T_1,t_{end})}, \mathbf{x}_{1,i}) \le \log(\frac{C_0}{\epsilon + o(\epsilon)}).$$

By the definition of $f(\mathbf{W}^{(T_1, t_{end})}, \mathbf{x}_{1,i})$, we have that

$$
\begin{aligned}
y_{1,i} \cdot f(\mathbf{W}^{(T_1, t_{end})}, \mathbf{x}_{1,i}) &= F_{y_{1,i}}(\mathbf{W}_{y_{1,i}}^{(T_1, t_{end})}, \mathbf{x}_{1,i}) - F_{-y_{1,i}}(\mathbf{W}_{-y_{1,i}}^{(T_1, t_{end})}, \mathbf{x}_{1,i}) \\
&= \frac{1}{m} \sum_{r=1}^{m} \left[ \sigma(\langle \mathbf{w}_{y_{1,i},r}^{(T_1, t_{end})}, y_{1,i} \cdot \boldsymbol{\mu}_1 \rangle) + \sigma(\langle \mathbf{w}_{y_{1,i},r}^{(T_1, t_{end})}, \boldsymbol{\xi}_{1,i} \rangle) \right] \\
&\quad - \frac{1}{m} \sum_{r=1}^{m} \left[ \sigma(\langle \mathbf{w}_{-y_{1,i},r}^{(T_1, t_{end})}, y_{1,i} \cdot \boldsymbol{\mu}_1 \rangle) + \sigma(\langle \mathbf{w}_{-y_{1,i},r}^{(T_1, t_{end})}, \boldsymbol{\xi}_{1,i} \rangle) \right] \\
&= \frac{1}{m} \sum_{r=1}^{m} \sigma(\langle \mathbf{w}_{y_{1,i},r}^{(T_1, t_{end})}, y_{1,i} \cdot \boldsymbol{\mu}_1 \rangle) + \frac{1}{m} \sum_{r=1}^{m} \left[ \sigma(\langle \mathbf{w}_{y_{1,i},r}^{(T_1, t_{end})}, \boldsymbol{\xi}_{1,i} \rangle) - \sigma(\langle \mathbf{w}_{-y_{1,i},r}^{(T_1, t_{end})}, \boldsymbol{\xi}_{1,i} \rangle) \right],
\end{aligned}
$$

where the last equality is by $\langle \mathbf{w}_{-y_{1,i},r}^{(T_1, t_{end})}, y_{1,i} \cdot \boldsymbol{\mu}_1 \rangle \leq 0$ in Lemma F.9 and Lemma F.10. Then by $|\langle \mathbf{w}_{j,r}^{(T_1, t_{end})}, \boldsymbol{\xi}_{1,i} \rangle| \leq 6\sqrt{\log(8mn/\delta)} \sigma_0 \sigma_{p_1} \sqrt{d}$ in Lemma F.15 and $\sigma_0 \sigma_{p_1} \sqrt{d} \leq \widetilde{O}(m^{-1}n^{-1})$ in Condition 3.1, we have that

$$
\log\left(\frac{1}{\epsilon + o(\epsilon)}\right) - \widetilde{O}(m^{-1}n^{-1}) \leq \frac{1}{m} \sum_{r=1}^{m} \sigma(\langle \mathbf{w}_{y_{1,i},r}^{(T_1, t_{end})}, y_{1,i} \cdot \boldsymbol{\mu}_1 \rangle) \leq \log\left(\frac{C_0}{\epsilon + o(\epsilon)}\right) + \widetilde{O}(m^{-1}n^{-1}).
$$

By Lemma F.10, we have that

$$
\begin{aligned}
\frac{1}{m} \sum_{r=1}^{m} \sigma(\langle \mathbf{w}_{y_{1,i},r}^{(T_1, t_{end})}, y_{1,i} \cdot \boldsymbol{\mu}_1 \rangle) &= \frac{1}{m} \sum_{r \in I_{y_{1,i},1}} \sigma(\langle \mathbf{w}_{y_{1,i},r}^{(T_1, t_{end})}, y_{1,i} \cdot \boldsymbol{\mu}_1 \rangle) \\
&= \frac{1}{m} \sum_{r \in I_{y_{1,i},1}} \sigma(\langle \mathbf{w}_{y_{1,i},r}^{(T_1, 0)}, y_{1,i} \cdot \boldsymbol{\mu}_1 \rangle + \gamma(\boldsymbol{\mu}_1)_{y_{1,i},r}^{(T_1, t_{end})}).
\end{aligned}
$$

Then by $|\langle \mathbf{w}_{j,r}^{(T_1, 0)}, \boldsymbol{\mu}_1 \rangle| \leq \sqrt{2\log(8m/\delta)} \cdot \sigma_0 \|\boldsymbol{\mu}_1\|_2$ in Lemma C.2 and $\sigma_0 \|\boldsymbol{\mu}_1\|_2 \leq \widetilde{O}(m^{-1}n^{-1})$ in Condition 3.1, we have that

$$
\log\left(\frac{1}{\epsilon + o(\epsilon)}\right) - \widetilde{O}(m^{-1}n^{-1}) \leq \frac{1}{m} \sum_{r \in I_{y_{1,i},1}} [\gamma(\boldsymbol{\mu}_1)_{y_{1,i},r}^{(T_1, t_{end})}]^q \leq \log\left(\frac{C_0}{\epsilon + o(\epsilon)}\right) + \widetilde{O}(m^{-1}n^{-1}),
$$

which completes the proof. $\qquad\square$

Using a method analogous to that in Lemma F.17, we can prove the following lemma.

**Lemma F.18.** *Under the same conditions as Theorem 3.2, denote $\overline{\gamma(\boldsymbol{\mu}_2)_j} = \frac{1}{m} \sum_{r \in I_{j,2}} [\gamma(\boldsymbol{\mu}_2)_{j,r}^{(T_2, t_{end})}]^q$. Then we have*

$$
\log\left(\frac{1}{\epsilon + o(\epsilon)}\right) - \widetilde{O}(m^{-1}n^{-1}) \leq \overline{\gamma(\boldsymbol{\mu}_2)_j} \leq \log\left(\frac{C_0}{\epsilon + o(\epsilon)}\right) + \widetilde{O}(m^{-1}n^{-1}),
$$

*for $j \in \{\pm 1\}$, where $C_0 > 1$ is a constant and $n = \max\{n_1, n_2\}$.*

**Lemma F.19.** *Under the same conditions as Theorem 3.2, when $-C_1 \leq \cos\theta_{1,2} < 0$, we have that*

$$
\sum_{r=1}^{m} \left[ \sigma(\langle \mathbf{w}_{y_1,r}^{(T_2, t_{end})}, y_1 \boldsymbol{\mu}_1 \rangle) - \sigma(\langle \mathbf{w}_{-y_1,r}^{(T_2, t_{end})}, y_1 \boldsymbol{\mu}_1 \rangle) \right] \geq C_3,
$$

*where $0 < C_1 < 1$ and $C_3$ are positive constants.*

*Proof of Lemma F.19.* By Lemma F.12 and Lemma F.13, we have that

$$
\sum_{r=1}^{m} \sigma(\langle \mathbf{w}_{y_1,r}^{(T_2, t_{end})}, y_1 \boldsymbol{\mu}_1 \rangle) = \sum_{r \in I_{y_1,1}} \sigma(\langle \mathbf{w}_{y_1,r}^{(T_2, t_{end})}, y_1 \boldsymbol{\mu}_1 \rangle)
$$

$$\geq \frac{m}{4} \cdot \overline{\gamma(\boldsymbol{\mu}_1)_{y_1}} \cdot (1 - \cos^2 \theta_{1,2})^q - \widetilde{O}(n^{-1}),$$

where the inequality is by Lemma F.11 and $|I_{y_1,1}| \geq m/4$ in Lemma F.16. Then by Lemma F.11 and Lemma F.13, we have that

$$\sum_{r=1}^{m} \sigma(\langle \mathbf{w}_{-y_1,r}^{(T_2,t_{end})}, y_1\boldsymbol{\mu}_1\rangle) \leq \sum_{r \in I_{y_1,2}} \sigma(\langle \mathbf{w}_{-y_1,r}^{(T_2,t_{end})}, y_1\boldsymbol{\mu}_1\rangle) + \widetilde{O}(n^{-1})$$

$$\leq m \cdot \overline{\gamma(\boldsymbol{\mu}_2)_{-y_1}} \cdot \frac{\|\boldsymbol{\mu}_1\|_2^q \cdot (-\cos\theta_{1,2})^q}{\|\boldsymbol{\mu}_2\|_2^q} + \widetilde{O}(n^{-1}),$$

where the first inequality is by $\sigma_0\|\boldsymbol{\mu}_1\|_2 \leq \widetilde{O}(m^{-1}n^{-1})$ in Condition 3.1, the second inequality is by Lemma F.12. Then we have that

$$\sum_{r=1}^{m} \left[ \sigma(\langle \mathbf{w}_{y_1,r}^{(T_2,t_{end})}, y_1\boldsymbol{\mu}_1\rangle) - \sigma(\langle \mathbf{w}_{-y_1,r}^{(T_2,t_{end})}, y_1\boldsymbol{\mu}_1\rangle) \right]$$

$$\geq \frac{m}{4} \cdot \overline{\gamma(\boldsymbol{\mu}_1)_{y_1}} \cdot (1 - \cos^2 \theta_{1,2})^q - m \cdot \overline{\gamma(\boldsymbol{\mu}_2)_{-y_1}} \cdot \frac{\|\boldsymbol{\mu}_1\|_2^q \cdot (-\cos\theta_{1,2})^q}{\|\boldsymbol{\mu}_2\|_2^q} - \widetilde{O}(n^{-1}).$$

By Lemma F.17 and Lemma F.18, we have that

$$\left( \frac{\log(\frac{1}{\epsilon+o(\epsilon)}) - \widetilde{O}(m^{-1}n^{-1})}{\log(C_0) + \log(\frac{1}{\epsilon+o(\epsilon)}) + \widetilde{O}(m^{-1}n^{-1})} \right)^{\frac{1}{q}} \leq \left( \frac{\overline{\gamma(\boldsymbol{\mu}_2)_{-y_1}}}{\overline{\gamma(\boldsymbol{\mu}_1)_{y_1}}} \right)^{\frac{1}{q}} \leq \left( \frac{\log(C_0) + \log(\frac{1}{\epsilon+o(\epsilon)}) + \widetilde{O}(m^{-1}n^{-1})}{\log(\frac{1}{\epsilon+o(\epsilon)}) - \widetilde{O}(m^{-1}n^{-1})} \right)^{\frac{1}{q}}.$$

Then we know that $\left( \overline{\gamma(\boldsymbol{\mu}_2)_{-y_1}} / \overline{\gamma(\boldsymbol{\mu}_1)_{y_1}} \right)^{1/q}$ is a constant, denoted as $C_4$. Now we choose $C_1$ as follows:

$$C_1 = \frac{-C_4 C_5 + \sqrt{C_4^2 C_5^2 + 4 - (16C_3 + \widetilde{O}(n^{-1}))/(mq \cdot \overline{\gamma(\boldsymbol{\mu}_1)_{y_1}}^{1/q})}}{2},$$

where $C_5 = 4^{1/q} \cdot \|\boldsymbol{\mu}_1\|_2/\|\boldsymbol{\mu}_2\|_2$, $\overline{\gamma(\boldsymbol{\mu}_1)_{y_1}} = \frac{1}{m}\sum_{r \in I_{y_1,1}} [\gamma(\boldsymbol{\mu}_1)_{y_1,r}^{(T_1,t_{end})}]^q \geq \log(\frac{1}{\epsilon+o(\epsilon)}) - \widetilde{O}(m^{-1}n^{-1})$ by Lemma F.17. Then when

$$\frac{(1/q)^{\frac{1}{q-1}} \cdot \|\boldsymbol{\mu}_2\|_2}{4^{1/q} \cdot \overline{\gamma(\boldsymbol{\mu}_2)_{-y_1}}^{1/q} \cdot \|\boldsymbol{\mu}_1\|_2} \leq -\cos\theta_{1,2} \leq C_1,$$

it is easy to verify that

$$q[4^{1/q} \cdot \overline{\gamma(\boldsymbol{\mu}_2)_{-y_1}}^{1/q} \cdot \frac{\|\boldsymbol{\mu}_1\|_2}{\|\boldsymbol{\mu}_2\|_2} \cdot (-\cos\theta_{1,2})]^{q-1} \geq 1$$

$$\overline{\gamma(\boldsymbol{\mu}_1)_{y_1}}^{1/q} \cdot (1 - \cos^2 \theta_{1,2}) - 4^{1/q} \cdot \overline{\gamma(\boldsymbol{\mu}_2)_{-y_1}}^{1/q} \cdot \frac{\|\boldsymbol{\mu}_1\|_2}{\|\boldsymbol{\mu}_2\|_2} \cdot (-\cos\theta_{1,2}) \geq \frac{4[C_3 + \widetilde{O}(n^{-1})]}{m},$$

where $\overline{\gamma(\boldsymbol{\mu}_2)_{-y_1}} = \frac{1}{m}\sum_{r \in I_{-y_1,2}} [\gamma(\boldsymbol{\mu}_2)_{-y_1,r}^{(T_2,t_{end})}]^q$. Then we have that

$$\frac{m}{4} \cdot \overline{\gamma(\boldsymbol{\mu}_1)_{y_1}} \cdot (1 - \cos^2 \theta_{1,2})^q - m \cdot \overline{\gamma(\boldsymbol{\mu}_2)_{-y_1}} \cdot \frac{\|\boldsymbol{\mu}_1\|_2^q \cdot (-\cos\theta_{1,2})^q}{\|\boldsymbol{\mu}_2\|_2^q}$$

$$\geq \frac{m}{4} \cdot q[4^{1/q} \cdot \overline{\gamma(\boldsymbol{\mu}_2)_{-y_1}}^{1/q} \cdot \frac{\|\boldsymbol{\mu}_1\|_2}{\|\boldsymbol{\mu}_2\|_2} \cdot (-\cos\theta_{1,2})]^{q-1}$$

$$\cdot \left[ \overline{\gamma(\boldsymbol{\mu}_1)_{y_1}}^{1/q} \cdot (1 - \cos^2 \theta_{1,2}) - 4^{1/q} \cdot \overline{\gamma(\boldsymbol{\mu}_2)_{-y_1}}^{1/q} \cdot \frac{\|\boldsymbol{\mu}_1\|_2}{\|\boldsymbol{\mu}_2\|_2} \cdot (-\cos\theta_{1,2}) \right]$$

$$\geq C_3 + \widetilde{O}(n^{-1}),$$

where the first inequality is by the convexity of $\mathrm{ReLU}^q$. Denote $h(x)$ as

$$h(x) = \frac{m}{4} \cdot \overline{\gamma(\boldsymbol{\mu}_1)_{y_1}} \cdot (1 - x^2)^q - m \cdot \overline{\gamma(\boldsymbol{\mu}_2)_{-y_1}} \cdot \frac{\|\boldsymbol{\mu}_1\|_2^q \cdot x^q}{\|\boldsymbol{\mu}_2\|_2^q} - \widetilde{O}(n^{-1}), \quad x \in (0, 1).$$

We know that $h(x)$ is monotonically decreasing. So when

$$0 < -\cos\theta_{1,2} \le \frac{(1/q)^{\frac{1}{q-1}} \cdot \|\boldsymbol{\mu}_2\|_2}{4^{1/q} \cdot \overline{\gamma(\boldsymbol{\mu}_2)_{-y_1}}^{1/q} \cdot \|\boldsymbol{\mu}_1\|_2},$$

we have that

$$\frac{m}{4} \cdot \overline{\gamma(\boldsymbol{\mu}_1)_{y_1}} \cdot (1 - \cos^2\theta_{1,2})^q - m \cdot \overline{\gamma(\boldsymbol{\mu}_2)_{-y_1}} \cdot \frac{\|\boldsymbol{\mu}_1\|_2^q \cdot (-\cos\theta_{1,2})^q}{\|\boldsymbol{\mu}_2\|_2^q} \ge C_3 + \widetilde{O}(n^{-1}).$$

Then when $0 < -\cos\theta_{1,2} \le C_1$, we have that

$$\sum_{r=1}^{m} \left[ \sigma(\langle \mathbf{w}_{y_1,r}^{(T_2,t_{end})}, y_1\boldsymbol{\mu}_1\rangle) - \sigma(\langle \mathbf{w}_{-y_1,r}^{(T_2,t_{end})}, y_1\boldsymbol{\mu}_1\rangle) \right] \ge C_3,$$

which completes the proof. $\qquad\square$

**Lemma F.20.** *Under the same conditions as Theorem 3.2, when $-1 < \cos\theta_{1,2} \le -C_2$, we have that*

$$\sum_{r=1}^{m} \left[ \sigma(\langle \mathbf{w}_{-y_1,r}^{(T_2,t_{end})}, y_1\boldsymbol{\mu}_1\rangle) - \sigma(\langle \mathbf{w}_{y_1,r}^{(T_2,t_{end})}, y_1\boldsymbol{\mu}_1\rangle) \right] \ge C_3,$$

*where $0 < C_2 < 1$ and $C_3$ are positive constants.*

*Proof of Lemma F.20.* By $\sigma(z) \ge 0$, we have that

$$\sum_{r=1}^{m} \sigma(\langle \mathbf{w}_{-y_1,r}^{(T_2,t_{end})}, y_1\boldsymbol{\mu}_1\rangle) \ge \sum_{r\in I_{-y_1,2}} \sigma(\langle \mathbf{w}_{-y_1,r}^{(T_2,t_{end})}, y_1\boldsymbol{\mu}_1\rangle)$$

$$\ge \frac{m}{8} \cdot \overline{\gamma(\boldsymbol{\mu}_2)_{-y_1}} \cdot \frac{\|\boldsymbol{\mu}_1\|_2^q \cdot (-\cos\theta_{1,2})^q}{\|\boldsymbol{\mu}_2\|_2^q},$$

where the second inequality is by Lemma F.12 and $|I_{-y_1,2}| \ge m/8$ in Lemma F.16. Then by Lemma F.12 and Lemma F.13, we have that

$$\sum_{r=1}^{m} \sigma(\langle \mathbf{w}_{y_1,r}^{(T_2,t_{end})}, y_1\boldsymbol{\mu}_1\rangle) = \sum_{r\in I_{y_1,1}} \sigma(\langle \mathbf{w}_{y_1,r}^{(T_2,t_{end})}, y_1\boldsymbol{\mu}_1\rangle)$$

$$\le m \cdot \overline{\gamma(\boldsymbol{\mu}_1)_{y_1}} \cdot (1 - \cos^2\theta_{1,2})^q + \widetilde{O}(n^{-1}),$$

where the inequality is by Lemma F.11 and $\sigma_0\|\boldsymbol{\mu}_1\|_2 \le \widetilde{O}(m^{-1}n^{-1})$ in Condition 3.1. Then we have that

$$\sum_{r=1}^{m} \left[ \sigma(\langle \mathbf{w}_{-y_1,r}^{(T_2,t_{end})}, y_1\boldsymbol{\mu}_1\rangle) - \sigma(\langle \mathbf{w}_{y_1,r}^{(T_2,t_{end})}, y_1\boldsymbol{\mu}_1\rangle) \right]$$

$$\ge \frac{m}{8} \cdot \overline{\gamma(\boldsymbol{\mu}_2)_{-y_1}} \cdot \frac{\|\boldsymbol{\mu}_1\|_2^q \cdot (-\cos\theta_{1,2})^q}{\|\boldsymbol{\mu}_2\|_2^q} - m \cdot \overline{\gamma(\boldsymbol{\mu}_1)_{y_1}} \cdot (1 - \cos^2\theta_{1,2})^q - \widetilde{O}(n^{-1}).$$

Now we choose $C_2$ as follows:

$$C_2 = \frac{-C_4 C_6 + \sqrt{C_4^2 C_6^2 + 4 + (32C_3 + \widetilde{O}(n^{-1}))/(8^{1/q}mq \cdot \overline{\gamma(\boldsymbol{\mu}_1)_{y_1}}^{1/q})}}{2},$$

where $C_4 = \left(\overline{\gamma(\boldsymbol{\mu}_2)_{-y_1}}/\overline{\gamma(\boldsymbol{\mu}_1)_{y_1}}\right)^{1/q}$, $C_6 = \|\boldsymbol{\mu}_1\|_2/(8^{1/q} \cdot \|\boldsymbol{\mu}_2\|_2)$ and $\overline{\gamma(\boldsymbol{\mu}_1)_{y_1}} = \frac{1}{m}\sum_{r\in I_{y_1,1}}[\gamma(\boldsymbol{\mu}_1)_{y_1,r}^{(T_1,t_{end})}]^q \ge \log(\frac{1}{\epsilon+o(\epsilon)}) - \widetilde{O}(m^{-1}n^{-1})$ by Lemma F.17. Then when $C_2 \le -\cos\theta_{1,2} < 1$, using a proof technique similar to that in Lemma F.19, we have that

$$\frac{m}{8} \cdot \overline{\gamma(\boldsymbol{\mu}_2)_{-y_1}} \cdot \frac{\|\boldsymbol{\mu}_1\|_2^q \cdot (-\cos\theta_{1,2})^q}{\|\boldsymbol{\mu}_2\|_2^q} - m \cdot \overline{\gamma(\boldsymbol{\mu}_1)_{y_1}} \cdot (1 - \cos^2\theta_{1,2})^q \ge C_3 + \widetilde{O}(n^{-1}).$$

Then when $C_2 \leq -\cos\theta_{1,2} < 1$, we have that

$$\sum_{r=1}^{m}\left[\sigma(\langle\mathbf{w}_{-y_1,r}^{(T_2,t_{end})}, y_1\boldsymbol{\mu}_1\rangle) - \sigma(\langle\mathbf{w}_{y_1,r}^{(T_2,t_{end})}, y_1\boldsymbol{\mu}_1\rangle)\right] \geq C_3,$$

which completes the proof. $\qquad\square$

**Lemma F.21.** *Let $C_1$ be defined in Lemma F.19. Under the same conditions as Theorem 3.2, when $-C_1 \leq \cos\theta_{1,2} < 0$, we have that $\mathbb{P}_{(\mathbf{x}_1,y_1)\sim\mathcal{D}_1}\big(y_1 \neq \mathrm{sign}(f(\mathbf{W}^{(T_2,t_{end})}, \mathbf{x}_1))\big) \leq \exp(-C \cdot m^{2q-2}n^{2q}/q^2)$, where $C = O(1)$.*

*Proof of Lemma F.21.* For the sake of convenience, we use $(\mathbf{x}_1, y_1) \sim \mathcal{D}_1$ to denote the following: data point $(\mathbf{x}_1, y_1)$ follows distribution $\mathcal{D}_1$ of Task $T_1$ defined in Definition 1.1. We can write out the test error as

$$\mathbb{P}_{(\mathbf{x}_1,y_1)\sim\mathcal{D}_1}\big(y_1 \neq \mathrm{sign}(f(\mathbf{W}^{(T_2,t_{end})}, \mathbf{x}_1))\big) = \mathbb{P}_{(\mathbf{x}_1,y_1)\sim\mathcal{D}_1}\big(y_1 f(\mathbf{W}^{(T_2,t_{end})}, \mathbf{x}_1) \leq 0\big).$$

It therefore suffices to provide an upper bound for $\mathbb{P}_{(\mathbf{x}_1,y_1)\sim\mathcal{D}_1}\big(y_1 f(\mathbf{W}^{(T_2,t_{end})}, \mathbf{x}_1) \leq 0\big)$. To achieve this, we write $\mathbf{x}_1 = (y_1\boldsymbol{\mu}_1, \boldsymbol{\xi}_1)$, and get

$$y_1 f(\mathbf{W}^{(T_2,t_{end})}, \mathbf{x}_1) = F_{y_1}(\mathbf{W}_{y_1}^{(T_2,t_{end})}, \mathbf{x}_1) - F_{-y_1}(\mathbf{W}_{-y_1}^{(T_2,t_{end})}, \mathbf{x}_1)$$

$$= \frac{1}{m}\sum_{r=1}^{m}\left[\sigma(\langle\mathbf{w}_{y_1,r}^{(T_2,t_{end})}, y_1\boldsymbol{\mu}_1\rangle) + \sigma(\langle\mathbf{w}_{y_1,r}^{(T_2,t_{end})}, \boldsymbol{\xi}_1\rangle)\right]$$

$$- \frac{1}{m}\sum_{r=1}^{m}\left[\sigma(\langle\mathbf{w}_{-y_1,r}^{(T_2,t_{end})}, y_1\boldsymbol{\mu}_1\rangle) + \sigma(\langle\mathbf{w}_{-y_1,r}^{(T_2,t_{end})}, \boldsymbol{\xi}_1\rangle)\right]$$

Then

$$\mathbb{P}_{(\mathbf{x}_1,y_1)\sim\mathcal{D}_1}\big(y_1 f(\mathbf{W}^{(T_2,t_{end})}, \mathbf{x}_1) \leq 0\big)$$

$$\leq \mathbb{P}_{(\mathbf{x}_1,y_1)\sim\mathcal{D}_1}\left(\sum_{r=1}^{m}\left[\sigma(\langle\mathbf{w}_{-y_1,r}^{(T_2,t_{end})}, \boldsymbol{\xi}_1\rangle)\right] \geq \sum_{r=1}^{m}\left[\sigma(\langle\mathbf{w}_{y_1,r}^{(T_2,t_{end})}, y_1\boldsymbol{\mu}_1\rangle) - \sigma(\langle\mathbf{w}_{-y_1,r}^{(T_2,t_{end})}, y_1\boldsymbol{\mu}_1\rangle)\right]\right). \tag{F.3}$$

Let $\widetilde{\mathbf{w}}_{j,r}^{(T_2,t)} = \mathbf{w}_{j,r}^{(T_2,t)} - j\cdot\gamma_{j,r}^{(T_1,t_{end})}\cdot\frac{\boldsymbol{\mu}_1}{\|\boldsymbol{\mu}_1\|_2^2} - j\cdot\gamma_{j,r}^{(T_2,t)}\cdot\frac{\boldsymbol{\mu}_2}{\|\boldsymbol{\mu}_2\|_2^2}$, then we have that $\langle\widetilde{\mathbf{w}}_{j,r}^{(T_2,t)}, \boldsymbol{\xi}_1\rangle = \langle\mathbf{w}_{j,r}^{(T_2,t)}, \boldsymbol{\xi}_1\rangle$ and

$$\|\widetilde{\mathbf{w}}_{j,r}^{(T_2,t)}\|_2 \leq \widetilde{O}(\sigma_0\sqrt{d} + \max\{n_1, n_2\}\cdot\sigma_0) = \widetilde{O}(\sigma_0\sqrt{d}), \tag{F.4}$$

where the equality is due to $d \geq \widetilde{\Omega}(m^2 \cdot \max\{n_1, n_2\}^4)$ by Condition 3.1. By (F.4), let $\max_{j,r}\|\widetilde{\mathbf{w}}_{j,r}^{(T_2,t)}\|_2 \leq C_7\sigma_0\sqrt{d}$, where $C_7 = \widetilde{O}(1)$. Denote $g(\boldsymbol{\xi}_1)$ as $\sum_r \sigma(\langle\widetilde{\mathbf{w}}_{-y_1,r}^{(T_2,t_{end})}, \boldsymbol{\xi}_1\rangle)$. According to Theorem 5.2.2 in, we know that for any $x \geq 0$ it holds that

$$\mathbb{P}(g(\boldsymbol{\xi}_1) - \mathbb{E}g(\boldsymbol{\xi}_1) \geq x) \leq \exp\left(-\frac{cx^2}{\sigma_{p_1}^2\|g\|_{\mathrm{Lip}}^2}\right), \tag{F.5}$$

where $c$ is a constant. To calculate the Lipschitz norm, we have

$$|g(\boldsymbol{\xi}) - g(\boldsymbol{\xi}')| = \left|\sum_{r=1}^{m}\sigma(\langle\widetilde{\mathbf{w}}_{-y_1,r}^{(T_2,t_{end})}, \boldsymbol{\xi}\rangle) - \sum_{r=1}^{m}\sigma(\langle\widetilde{\mathbf{w}}_{-y_1,r}^{(T_2,t_{end})}, \boldsymbol{\xi}'\rangle)\right|$$

$$\leq \sum_{r=1}^{m}\left|\sigma(\langle\widetilde{\mathbf{w}}_{-y_1,r}^{(T_2,t_{end})}, \boldsymbol{\xi}\rangle) - \sigma(\langle\widetilde{\mathbf{w}}_{-y_1,r}^{(T_2,t_{end})}, \boldsymbol{\xi}'\rangle)\right|$$

$$\leq q(6\sqrt{\log(8mn/\delta)}\sigma_0\sigma_{p_1}\sqrt{d})^{q-1}\sum_{r=1}^{m}|\langle\widetilde{\mathbf{w}}_{-y_1,r}^{(T_2,t_{end})}, \boldsymbol{\xi} - \boldsymbol{\xi}'\rangle|$$

$$\leq q(6\sqrt{\log(8mn/\delta)}\sigma_0\sigma_{p_1}\sqrt{d})^{q-1}\sum_{r=1}^{m}\|\widetilde{\mathbf{w}}_{-y_1,r}^{(T_2,t_{end})}\|_2 \cdot \|\boldsymbol{\xi} - \boldsymbol{\xi}'\|_2,$$

where the first inequality is by triangle inequality; the second inequality is by the convexity of $\mathrm{ReLU}^q$; the last inequality is by Cauchy-Schwartz inequality. Therefore, we have

$$\|g\|_{\mathrm{Lip}} \le q(6\sqrt{\log(8mn/\delta)}\sigma_0\sigma_{p_1}\sqrt{d})^{q-1}\sum_{r=1}^{m}\|\widetilde{\mathbf{w}}_{-y_1,r}^{(T_2,t_{end})}\|_2, \tag{F.6}$$

and since $\langle\widetilde{\mathbf{w}}_{-y_1,r}^{(T_2,t_{end})}, \boldsymbol{\xi}_1\rangle \sim \mathcal{N}\big(0, \|\widetilde{\mathbf{w}}_{-y_1,r}^{(T_2,t_{end})}\|_2^2\sigma_{p_1}^2\big)$, we can get

$$\mathbb{E}g(\boldsymbol{\xi}_1) = \sum_{r=1}^{m}\mathbb{E}\sigma(\langle\widetilde{\mathbf{w}}_{-y_1,r}^{(T_2,t_{end})}, \boldsymbol{\xi}_1\rangle) = \sum_{r=1}^{m}\frac{2^{\frac{q-1}{2}}\cdot\Gamma(\frac{q+1}{2})\cdot(\|\widetilde{\mathbf{w}}_{-y_1,r}^{(T_2,t_{end})}\|_2\sigma_{p_1})^q}{\sqrt{2\pi}} = C_8\cdot\frac{\sigma_{p_1}^q}{\sqrt{2\pi}}\sum_{r=1}^{m}\|\widetilde{\mathbf{w}}_{-y_1,r}^{(T_2,t_{end})}\|_2^q,$$

where $C_8 = 2^{\frac{q-1}{2}}\cdot\Gamma(\frac{q+1}{2})$ is a constant, with $\Gamma(z) = \int_0^{+\infty}t^{z-1}e^{-t}\,dt$ denoting the Gamma function. When $-C_1 \le \cos\theta_{1,2} < 0$, by (F.3) and $\langle\widetilde{\mathbf{w}}_{j,r}^{(T_2,t)}, \boldsymbol{\xi}_1\rangle = \langle\mathbf{w}_{j,r}^{(T_2,t)}, \boldsymbol{\xi}_1\rangle$, we have that

$$\mathbb{P}_{(\mathbf{x}_1,y_1)\sim\mathcal{D}_1}\big(y_1 f(\mathbf{W}^{(T_2,t_{end})}, \mathbf{x}_1) \le 0\big)$$

$$\le \mathbb{P}_{(\mathbf{x}_1,y_1)\sim\mathcal{D}_1}\left(\sum_{r=1}^{m}\big[\sigma(\langle\widetilde{\mathbf{w}}_{-y_1,r}^{(T_2,t_{end})}, \boldsymbol{\xi}_1\rangle)\big] \ge \sum_{r=1}^{m}\big[\sigma(\langle\mathbf{w}_{y_1,r}^{(T_2,t_{end})}, y_1\boldsymbol{\mu}_1\rangle) - \sigma(\langle\mathbf{w}_{-y_1,r}^{(T_2,t_{end})}, y_1\boldsymbol{\mu}_1\rangle)\big]\right)$$

$$= \mathbb{P}_{(\mathbf{x}_1,y_1)\sim\mathcal{D}_1}\left(g(\boldsymbol{\xi}_1) - \mathbb{E}g(\boldsymbol{\xi}_1) \ge \sum_{r=1}^{m}\big[\sigma(\langle\mathbf{w}_{y_1,r}^{(T_2,t_{end})}, y_1\boldsymbol{\mu}_1\rangle) - \sigma(\langle\mathbf{w}_{-y_1,r}^{(T_2,t_{end})}, y_1\boldsymbol{\mu}_1\rangle)\big] - C_8\cdot\frac{\sigma_{p_1}^q}{\sqrt{2\pi}}\sum_{r=1}^{m}\|\widetilde{\mathbf{w}}_{-y_1,r}^{(T_2,t_{end})}\|_2^q\right)$$

$$\le \exp\left[-\frac{c\left(\sum_{r=1}^{m}\big[\sigma(\langle\mathbf{w}_{y_1,r}^{(T_2,t_{end})}, y_1\boldsymbol{\mu}_1\rangle) - \sigma(\langle\mathbf{w}_{-y_1,r}^{(T_2,t_{end})}, y_1\boldsymbol{\mu}_1\rangle)\big] - (C_8/\sqrt{2\pi})\sum_{r=1}^{m}\sigma_{p_1}^q\cdot\|\widetilde{\mathbf{w}}_{-y_1,r}^{(T_2,t_{end})}\|_2^q\right)^2}{q^2\left(6\sqrt{\log(8mn/\delta)}\sigma_0\sigma_{p_1}\sqrt{d}\right)^{2q-2}\cdot\left(\sum_{r=1}^{m}\sigma_{p_1}\cdot\|\widetilde{\mathbf{w}}_{-y_1,r}^{(T_2,t_{end})}\|_2\right)^2}\right]$$

$$\le \exp\left[\frac{-c}{q^2\left(6\sqrt{\log(8mn/\delta)}\sigma_0\sigma_{p_1}\sqrt{d}\right)^{2q-2}}\cdot\left(\frac{\sum_{r=1}^{m}\big[\sigma(\langle\mathbf{w}_{y_1,r}^{(T_2,t_{end})}, y_1\boldsymbol{\mu}_1\rangle) - \sigma(\langle\mathbf{w}_{-y_1,r}^{(T_2,t_{end})}, y_1\boldsymbol{\mu}_1\rangle)\big]}{\sigma_{p_1}\sum_{r=1}^{m}\|\widetilde{\mathbf{w}}_{-y_1,r}^{(T_2,t_{end})}\|_2} - C_8/\sqrt{2\pi}\right)^2\right]$$

$$\le \exp\left[\frac{1}{q^2\left(6\sqrt{\log(8mn/\delta)}\sigma_0\sigma_{p_1}\sqrt{d}\right)^{2q-2}}\left(\frac{cC_8^2}{2\pi} - \frac{c}{2}\left(\frac{\sum_{r=1}^{m}\big[\sigma(\langle\mathbf{w}_{y_1,r}^{(T_2,t_{end})}, y_1\boldsymbol{\mu}_1\rangle) - \sigma(\langle\mathbf{w}_{-y_1,r}^{(T_2,t_{end})}, y_1\boldsymbol{\mu}_1\rangle)\big]}{\sigma_{p_1}\sum_{r=1}^{m}\|\widetilde{\mathbf{w}}_{-y_1,r}^{(T_2,t_{end})}\|_2}\right)^2\right)\right],$$

where the second inequality is by (F.5) and (F.6), the third inequality is by $\sigma_{p_1}\|\widetilde{\mathbf{w}}_{-y_1,r}^{(T_2,t_{end})}\|_2 \le \widetilde{O}(\sigma_0\sigma_{p_1}\sqrt{d})$ according to (F.5) and $\sigma_0\sigma_{p_1}\sqrt{d} \le \widetilde{O}(m^{-1}n^{-1})$ in Condition 3.1, the last inequality is due to the fact that $(s-t)^2 \ge s^2/2 - t^2, \forall s, t \ge 0$. Then by Lemma F.19, we further have that

$$\mathbb{P}_{(\mathbf{x}_1,y_1)\sim\mathcal{D}_1}\big(y_1 f(\mathbf{W}^{(T_2,t_{end})}, \mathbf{x}_1) \le 0\big)$$

$$\le \exp\left[\frac{1}{q^2\left(6\sqrt{\log(8mn/\delta)}\sigma_0\sigma_{p_1}\sqrt{d}\right)^{2q-2}}\left(\frac{cC_8^2}{2\pi} - 0.5c\left(\frac{C_3}{\sigma_{p_1}\sum_{r=1}^{m}\|\widetilde{\mathbf{w}}_{-y_1,r}^{(T_2,t_{end})}\|_2}\right)^2\right)\right]$$

$$\le \exp\left[\frac{1}{q^2\left(6\sqrt{\log(8mn/\delta)}\sigma_0\sigma_{p_1}\sqrt{d}\right)^{2q-2}}\left(\frac{cC_8^2}{2\pi} - \frac{cC_3^2}{2m^2C_7^2(\sigma_0\sigma_{p_1}\sqrt{d})^2}\right)\right]$$

$$\le \exp\left[\frac{1}{q^2\left(6\sqrt{\log(8mn/\delta)}\sigma_0\sigma_{p_1}\sqrt{d}\right)^{2q-2}}\left(-\frac{cC_3^2}{4m^2C_3^2(\sigma_0\sigma_{p_1}\sqrt{d})^2}\right)\right]$$

$$\le \exp\left[\frac{-C\cdot m^{2q-2}n^{2q}}{q^2}\right],$$

where the second inequality is by $\max_{j,r}\|\widetilde{\mathbf{w}}_{j,r}^{(T_2,t)}\|_2 \le C_7\sigma_0\sqrt{d}$, the third inequality is by $\sigma_0\sigma_{p_1}\sqrt{d} \le \widetilde{O}(m^{-1}n^{-1})$ in Condition 3.1. Then we have that

$$\mathbb{P}_{(\mathbf{x}_1,y_1)\sim\mathcal{D}_1}\big(y_1 \ne \mathrm{sign}(f(\mathbf{W}^{(T_2,t_{end})}, \mathbf{x}_1))\big) \le \exp(-C\cdot m^{2q-2}n^{2q}/q^2),$$

which completes the proof. $\qquad\square$

**Lemma F.22.** *Let $C_2$ be defined in Lemma F.20. Under the same conditions as Theorem 3.2, when $-1 < \cos\theta_{1,2} \leq -C_2$, we have that $\mathbb{P}_{(\mathbf{x}_1,y_1)\sim\mathcal{D}_1}\big(y_1 \neq \text{sign}(f(\mathbf{W}^{(T_2,t_{end})}, \mathbf{x}_1))\big) \geq 1 - \exp(-C \cdot m^{2q-2}n^{2q}/q^2)$, where $C = O(1)$.*

*Proof of Lemma F.22.* We can write out the test error as

$$\mathbb{P}_{(\mathbf{x}_1,y_1)\sim\mathcal{D}_1}\big(y_1 \neq \text{sign}(f(\mathbf{W}^{(T_2,t_{end})}, \mathbf{x}_1))\big) = \mathbb{P}_{(\mathbf{x}_1,y_1)\sim\mathcal{D}_1}\big(y_1 f(\mathbf{W}^{(T_2,t_{end})}, \mathbf{x}_1) \leq 0\big)$$
$$= 1 - \mathbb{P}_{(\mathbf{x}_1,y_1)\sim\mathcal{D}_1}\big(y_1 f(\mathbf{W}^{(T_2,t_{end})}, \mathbf{x}_1) \geq 0\big).$$

It therefore suffices to provide an upper bound for $\mathbb{P}_{(\mathbf{x}_1,y_1)\sim\mathcal{D}_1}\big(y_1 f(\mathbf{W}^{(T_2,t_{end})}, \mathbf{x}_1) \geq 0\big)$. To achieve this, we write $\mathbf{x}_1 = (y_1\boldsymbol{\mu}_1, \boldsymbol{\xi}_1)$, and get

$$y_1 f(\mathbf{W}^{(T_2,t_{end})}, \mathbf{x}_1) = F_{y_1}(\mathbf{W}_{y_1}^{(T_2,t_{end})}, \mathbf{x}_1) - F_{-y_1}(\mathbf{W}_{-y_1}^{(T_2,t_{end})}, \mathbf{x}_1)$$
$$= \frac{1}{m}\sum_{r=1}^{m}\big[\sigma(\langle\mathbf{w}_{y_1,r}^{(T_2,t_{end})}, y_1\boldsymbol{\mu}_1\rangle) + \sigma(\langle\mathbf{w}_{y_1,r}^{(T_2,t_{end})}, \boldsymbol{\xi}_1\rangle)\big]$$
$$- \frac{1}{m}\sum_{r=1}^{m}\big[\sigma(\langle\mathbf{w}_{-y_1,r}^{(T_2,t_{end})}, y_1\boldsymbol{\mu}_1\rangle) + \sigma(\langle\mathbf{w}_{-y_1,r}^{(T_2,t_{end})}, \boldsymbol{\xi}_1\rangle)\big].$$

Then

$$\mathbb{P}_{(\mathbf{x}_1,y_1)\sim\mathcal{D}_1}\big(y_1 f(\mathbf{W}^{(T_2,t_{end})}, \mathbf{x}_1) \geq 0\big)$$
$$\leq \mathbb{P}_{(\mathbf{x}_1,y_1)\sim\mathcal{D}_1}\Big(\sum_{r=1}^{m}\big[\sigma(\langle\mathbf{w}_{y_1,r}^{(T_2,t_{end})}, \boldsymbol{\xi}_1\rangle)\big] \geq \sum_{r=1}^{m}\big[\sigma(\langle\mathbf{w}_{-y_1,r}^{(T_2,t_{end})}, y_1\boldsymbol{\mu}_1\rangle) - \sigma(\langle\mathbf{w}_{y_1,r}^{(T_2,t_{end})}, y_1\boldsymbol{\mu}_1\rangle)\big]\Big).$$

When $-1 < \cos\theta_{1,2} \leq -C_2$, using a proof technique similar to that in Lemma F.21, by $\sum_{r=1}^{m}\big[\sigma(\langle\mathbf{w}_{-y_1,r}^{(T_2,t_{end})}, y_1\boldsymbol{\mu}_1\rangle) - \sigma(\langle\mathbf{w}_{y_1,r}^{(T_2,t_{end})}, y_1\boldsymbol{\mu}_1\rangle)\big] \geq C_3$ in Lemma F.20, we have that

$$\mathbb{P}_{(\mathbf{x}_1,y_1)\sim\mathcal{D}_1}\big(y_1 f(\mathbf{W}^{(T_2,t_{end})}, \mathbf{x}_1) \geq 0\big) \leq \exp(-C \cdot m^{2q-2}n^{2q}/q^2).$$

Then we further have that

$$\mathbb{P}_{(\mathbf{x}_1,y_1)\sim\mathcal{D}_1}\big(y_1 \neq \text{sign}(f(\mathbf{W}^{(T_2,t_{end})}, \mathbf{x}_1))\big) \geq 1 - \exp(-C \cdot m^{2q-2}n^{2q}/q^2),$$

which completes the proof. $\square$

## G. Insights into Task T1 Learning

In this section, our intention is to present the specific details associated with the learning of task $T_1$. Specifically, the focus is on the maximum value of the inner product between the convolutional filters within $F_{+1}$ and $\boldsymbol{\mu}_1$, which is denoted as $\max_{r\in[1,m]}\langle\mathbf{w}_{+1,r}^{(T_1,t)}, \boldsymbol{\mu}_1\rangle$. If we suppose that $r^*$ is the value for which the inner product $\langle\mathbf{w}_{+1,r}^{(T_1,t)}, \boldsymbol{\mu}_1\rangle$ attains its maximum, then it follows that $\langle\mathbf{w}_{+1,r^*}^{(T_1,t)}, \boldsymbol{\mu}_1\rangle = \max_{r\in[1,m]}\langle\mathbf{w}_{+1,r}^{(T_1,t)}, \boldsymbol{\mu}_1\rangle$. Moreover, $r^{**}$ is the value that results in the second largest inner product $\langle\mathbf{w}_{+1,r}^{(T_1,t)}, \boldsymbol{\mu}_1\rangle$. In such a situation, we can state that $\langle\mathbf{w}_{+1,r^{**}}^{(T_1,t)}, \boldsymbol{\mu}_1\rangle = \max_{r\neq r^*}\langle\mathbf{w}_{+1,r}^{(T_1,t)}, \boldsymbol{\mu}_1\rangle$. Similarly, we can also obtain the values of $\langle\mathbf{w}_{-1,r^*}^{(T_1,t)}, -\boldsymbol{\mu}_1\rangle$ and $\langle\mathbf{w}_{-1,r^{**}}^{(T_1,t)}, -\boldsymbol{\mu}_1\rangle$.

**Lemma G.1.** *Under the previous conditions, we can get that when t tends to infinity*

$$\langle\mathbf{w}_{j,r^*}^{(T_1,t)}, j\boldsymbol{\mu}_1\rangle \to +\infty$$

*for $j \in \{\pm1\}$.*

*proof of Lemma G.1.* Based on the analysis of training task $T_1$ before, we know the update rule for $\mathbf{w}_{j,r}^{(T_1,t)}$ as follows:

$$\mathbf{w}_{j,r}^{(T_1,t+1)} = \mathbf{w}_{j,r}^{(T_1,t)} - \eta_1 \cdot \nabla_{\mathbf{w}_{j,r}}L_{S_1}(\mathbf{W}^{(T_1,t)})$$

$$= \mathbf{w}_{j,r}^{(T_1,t)} + \frac{\eta_1}{n_1 m} \sum_{i=1}^{n_1} (-\ell_{1,i}'^{(T_1,t)}) \cdot \sigma'(\langle \mathbf{w}_{j,r}^{(T_1,t)}, \boldsymbol{\xi}_{1,i} \rangle) \cdot j y_{1,i} \boldsymbol{\xi}_{1,i}$$

$$+ \frac{\eta_1}{n_1 m} \sum_{i=1}^{n_1} (-\ell_{1,i}'^{(T_1,t)}) \cdot \sigma'(\langle \mathbf{w}_{j,r}^{(T_1,t)}, y_{1,i} \boldsymbol{\mu}_1 \rangle) \cdot \boldsymbol{\mu}_1$$

Perform the inner product operation on the vector $\boldsymbol{\mu}_1$ on both sides of the equation, we can get

$$\langle \mathbf{w}_{j,r}^{(T_1,t+1)}, j\boldsymbol{\mu}_1 \rangle = \langle \mathbf{w}_{j,r}^{(T_1,t)}, j\boldsymbol{\mu}_1 \rangle + \frac{\eta_1 \|\boldsymbol{\mu}_1\|_2^2}{n_1 m} \sum_{i=1}^{n_1} (-\ell_{1,i}'^{(T_1,t)}) \cdot \sigma'(\langle \mathbf{w}_{j,r}^{(T_1,t)}, y_{1,i} \boldsymbol{\mu}_1 \rangle).$$

Then we have

$$\langle \mathbf{w}_{j,r^*}^{(T_1,t+1)}, j\boldsymbol{\mu}_1 \rangle = \langle \mathbf{w}_{j,r^*}^{(T_1,t)}, j\boldsymbol{\mu}_1 \rangle + \frac{\eta_1 \|\boldsymbol{\mu}_1\|_2^2}{n_1 m} \sum_{y_{1,i}=j} (-\ell_{1,i}'^{(T_1,t)}) \cdot \sigma'(\langle \mathbf{w}_{j,r^*}^{(T_1,t)}, j\boldsymbol{\mu}_1 \rangle), \tag{G.1}$$

$$\langle \mathbf{w}_{j,r^{**}}^{(T_1,t+1)}, j\boldsymbol{\mu}_1 \rangle = \langle \mathbf{w}_{j,r^{**}}^{(T_1,t)}, j\boldsymbol{\mu}_1 \rangle + \frac{\eta_1 \|\boldsymbol{\mu}_1\|_2^2}{n_1 m} \sum_{y_{1,i}=j} (-\ell_{1,i}'^{(T_1,t)}) \cdot \sigma'(\langle \mathbf{w}_{j,r^{**}}^{(T_1,t)}, j\boldsymbol{\mu}_1 \rangle). \tag{G.2}$$

Dividing the two equations (G.1) and (G.2), we can get

$$\frac{\langle \mathbf{w}_{j,r^*}^{(T_1,t+1)}, j\boldsymbol{\mu}_1 \rangle}{\langle \mathbf{w}_{j,r^{**}}^{(T_1,t+1)}, j\boldsymbol{\mu}_1 \rangle} = \frac{\langle \mathbf{w}_{j,r^*}^{(T_1,t)}, j\boldsymbol{\mu}_1 \rangle + \frac{\eta_1 \|\boldsymbol{\mu}_1\|_2^2}{n_1 m} \sum_{y_{1,i}=j} (-\ell_{1,i}'^{(T_1,t)}) \cdot \sigma'(\langle \mathbf{w}_{j,r^*}^{(T_1,t)}, j\boldsymbol{\mu}_1 \rangle)}{\langle \mathbf{w}_{j,r^{**}}^{(T_1,t)}, j\boldsymbol{\mu}_1 \rangle + \frac{\eta_1 \|\boldsymbol{\mu}_1\|_2^2}{n_1 m} \sum_{y_{1,i}=j} (-\ell_{1,i}'^{(T_1,t)}) \cdot \sigma'(\langle \mathbf{w}_{j,r^{**}}^{(T_1,t)}, j\boldsymbol{\mu}_1 \rangle)} \tag{G.3}$$

we can judge when $t$ tends to infinity, $\langle \mathbf{w}_{j,r^*}^{(T_1,t)}, j\boldsymbol{\mu}_1 \rangle$ tends to infinity. According to (G.1), we know that $\langle \mathbf{w}_{j,r^*}^{(T_1,t)}, \boldsymbol{\mu}_1 \rangle$ is an increasing sequence, so its limit is either a constant or positive infinity. Assume it converges to a positive constant $M$, take the limit of both sides of the equation (G.1) simultaneously, we can get that

$$M = M + \lim_{t \to +\infty} \frac{\eta_1 \|\boldsymbol{\mu}_1\|_2^2}{n_1 m} \sum_{y_{1,i}=j} (-\ell_{1,i}'^{(T_1,t)}) \cdot \sigma'(\langle \mathbf{w}_{j,r^*}^{(T_1,t)}, j\boldsymbol{\mu}_1 \rangle)$$

By the fact that $-\ell_{1,i}'^{(T_1,t)} > 0$, we can get that

$$M = M + \frac{\eta_1 \|\boldsymbol{\mu}_1\|_2^2}{n_1 m} \sum_{y_{1,i}=j} (-\ell_{1,i}'^{(T_1,t)}) \sigma'(M),$$

then we will come to a wrong conclusion: $0 = \frac{\eta_1 \|\boldsymbol{\mu}_1\|_2^2}{n_1 m} \sum_{y_{1,i}=j} (-\ell_{1,i}'^{(T_1,t)}) \sigma'(M)$, so $\langle \mathbf{w}_{j,r^*}^{(T_1,t)}, j\boldsymbol{\mu}_1 \rangle$ tends to infinity. $\square$

For our further work, we need to review the following mathematical lemma:

**Lemma G.2.** *Let $a_n > 0$, then the infinite product $\prod_{n=1}^{+\infty}(1 + a_n)$ converges if and only if the series $\sum_{n=1}^{+\infty} a_n$ converges.*

The proof of this lemma can be found in many mathematics books. With this tool, we will prove the following lemma.

**Lemma G.3.** *We can get that when $t$ tends to infinity*

$$\langle \mathbf{w}_{j,r^{**}}^{(T_1,t)}, j\boldsymbol{\mu}_1 \rangle = o(\langle \mathbf{w}_{j,r^*}^{(T_1,t)}, j\boldsymbol{\mu}_1 \rangle)$$

*for $j \in \{\pm 1\}$.*

*proof of Lemma G.3.* If, as $t$ approaches infinity, the inner product $\langle \mathbf{w}_{j,r^{**}}^{(T_1,t)}, j\boldsymbol{\mu}_1 \rangle$ converges to a positive constant, then the proof of this lemma becomes straightforward. Subsequently, we will investigate the case where $\langle \mathbf{w}_{j,r^{**}}^{(T_1,t)}, j\boldsymbol{\mu}_1 \rangle$ approaches infinity. To simplify the expression, we let $a_t = \langle \mathbf{w}_{j,r^*}^{(T_1,t)}, j\boldsymbol{\mu}_1 \rangle$ and $b_t = \langle \mathbf{w}_{j,r^{**}}^{(T_1,t)}, j\boldsymbol{\mu}_1 \rangle$. According to equation (G.3), we can deduce that:

$$\frac{a_{t+1}}{b_{t+1}} = \frac{a_t}{b_t} \cdot \frac{1 + \frac{q\eta_1 \|\boldsymbol{\mu}_1\|_2^2}{n_1 m} \sum_{y_{1,i}=j} (-\ell_{1,i}'^{(T_1,t)}) a_t^{q-2}}{1 + \frac{q\eta_1 \|\boldsymbol{\mu}_1\|_2^2}{n_1 m} \sum_{y_{1,i}=j} (-\ell_{1,i}'^{(T_1,t)}) b_t^{q-2}}.$$

Let $c_t = \frac{a_t}{b_t}$, then we have

$$c_{t+1} = c_t \cdot \frac{1 + \frac{q\eta_1 \|\boldsymbol{\mu}_1\|_2^2}{n_1 m} \sum_{y_{1,i}=j} (-\ell_{1,i}'^{(T_1,t)}) a_t^{q-2}}{1 + \frac{q\eta_1 \|\boldsymbol{\mu}_1\|_2^2}{n_1 m} \sum_{y_{1,i}=j} (-\ell_{1,i}'^{(T_1,t)}) b_t^{q-2}}$$

$$\geq c_t,$$

where the inequality is by $a_t \geq b_t$, so it is obvious that $c_t$ is an increasing sequence and $c_t \geq 1$ for all $t \geq 0$. Combining equation (G.1) and the definition of $c_t$, we can obtain

$$c_{t+1} b_{t+1} = c_t b_t + \frac{q\eta_1 \|\boldsymbol{\mu}_1\|_2^2}{n_1 m} \sum_{y_{1,i}=j} (-\ell_{1,i}'^{(T_1,t)}) c_t^{q-1} b_t^{q-1}.$$

Dividing this equation by (G.2) gives

$$\frac{c_{t+1}}{c_t} = \frac{b_t}{b_{t+1}} + \frac{q\eta_1 \|\boldsymbol{\mu}_1\|_2^2}{n_1 m} \sum_{y_{1,i}=j} (-\ell_{1,i}'^{(T_1,t)}) \frac{c_t^{q-2} b_t^{q-1}}{b_{t+1}}$$

$$= \frac{b_t + \frac{q\eta_1 \|\boldsymbol{\mu}_1\|_2^2}{n_1 m} \sum_{y_{1,i}=j} (-\ell_{1,i}'^{(T_1,t)}) c_t^{q-2} b_t^{q-1}}{b_t + \frac{q\eta_1 \|\boldsymbol{\mu}_1\|_2^2}{n_1 m} \sum_{y_{1,i}=j} (-\ell_{1,i}'^{(T_1,t)}) b_t^{q-1}}$$

$$= \frac{1 + \frac{q\eta_1 \|\boldsymbol{\mu}_1\|_2^2}{n_1 m} \sum_{y_{1,i}=j} (-\ell_{1,i}'^{(T_1,t)}) c_t^{q-2} b_t^{q-2}}{1 + \frac{q\eta_1 \|\boldsymbol{\mu}_1\|_2^2}{n_1 m} \sum_{y_{1,i}=j} (-\ell_{1,i}'^{(T_1,t)}) b_t^{q-2}}.$$

Next, we analyze the convergence and divergence of the infinite series:

$$\sum_{t=0}^{+\infty} \frac{\frac{q\eta_1 \|\boldsymbol{\mu}_1\|_2^2}{n_1 m} \sum_{y_{1,i}=j} (-\ell_{1,i}'^{(T_1,t)}) (c_t^{q-2} - 1) b_t^{q-2}}{1 + \frac{q\eta_1 \|\boldsymbol{\mu}_1\|_2^2}{n_1 m} \sum_{y_{1,i}=j} (-\ell_{1,i}'^{(T_1,t)}) b_t^{q-2}}$$

$$\geq \frac{c_0^{q-2} - 1}{M} \sum_{t=0}^{+\infty} \frac{q\eta_1 \|\boldsymbol{\mu}_1\|_2^2}{n_1 m} \sum_{y_{1,i}=j} (-\ell_{1,i}'^{(T_1,t)}) b_t^{q-2}$$

$$= \frac{c_0^{q-2} - 1}{M} \sum_{t=0}^{+\infty} \left( \frac{b_{t+1}}{b_t} - 1 \right),$$

where the inequality is by $c_t$ is an increasing sequence, and $\sum_{y_{1,i}=j} (-\ell_{1,i}'^{(T_1,t)}) b_t^{q-2} \leq M-1$ because $\ell'$ has an exponentially decaying tail. Since the infinite product $\prod_{t=0}^{+\infty} \frac{b_{t+1}}{b_t}$ diverges:

$$\prod_{t=0}^{+\infty} \frac{b_{t+1}}{b_t} = \frac{b_{+\infty}}{b_0} = +\infty$$

according to lemma G.2, the infinite series $\sum_{t=0}^{+\infty} \left( \frac{b_{t+1}}{b_t} - 1 \right)$ diverges. Thus, we can conclude that the infinite series $\sum_{t=0}^{+\infty} \frac{\frac{q\eta_1 \|\boldsymbol{\mu}_1\|_2^2}{n_1 m} \sum_{y_{1,i}=j} (-\ell_{1,i}'^{(T_1,t)}) (c_t^{q-2}-1) b_t^{q-2}}{1 + \frac{q\eta_1 \|\boldsymbol{\mu}_1\|_2^2}{n_1 m} \sum_{y_{1,i}=j} (-\ell_{1,i}'^{(T_1,t)}) b_t^{q-2}}$ diverges. Notice that $\frac{1 + \frac{q\eta_1 \|\boldsymbol{\mu}_1\|_2^2}{n_1 m} \sum_{y_{1,i}=j} (-\ell_{1,i}'^{(T_1,t)}) c_t^{q-2} b_t^{q-2}}{1 + \frac{q\eta_1 \|\boldsymbol{\mu}_1\|_2^2}{n_1 m} \sum_{y_{1,i}=j} (-\ell_{1,i}'^{(T_1,t)}) b_t^{q-2}} =$ $\frac{\frac{q\eta_1 \|\boldsymbol{\mu}_1\|_2^2}{n_1 m} \sum_{y_{1,i}=j} (-\ell_{1,i}'^{(T_1,t)}) (c_t^{q-2}-1) b_t^{q-2}}{1 + \frac{q\eta_1 \|\boldsymbol{\mu}_1\|_2^2}{n_1 m} \sum_{y_{1,i}=j} (-\ell_{1,i}'^{(T_1,t)}) b_t^{q-2}} + 1$, so the infinite product $\prod_{t=0}^{+\infty} \frac{c_{t+1}}{c_t} = \prod_{t=0}^{+\infty} \frac{1 + \frac{q\eta_1 \|\boldsymbol{\mu}_1\|_2^2}{n_1 m} \sum_{y_{1,i}=j} (-\ell_{1,i}'^{(T_1,t)}) c_t^{q-2} b_t^{q-2}}{1 + \frac{q\eta_1 \|\boldsymbol{\mu}_1\|_2^2}{n_1 m} \sum_{y_{1,i}=j} (-\ell_{1,i}'^{(T_1,t)}) b_t^{q-2}}$

diverges. Therefore,

$$c_{+\infty} = c_0 \prod_{t=0}^{+\infty} \frac{c_{t+1}}{c_t} = +\infty$$

so $\lim_{t\to+\infty} \frac{b_t}{a_t} = \lim_{t\to+\infty} \frac{1}{c_t} = 0$, that is to say $\langle \mathbf{w}_{j,r^{**}}^{(T_1,t)}, j\boldsymbol{\mu}_1 \rangle = o(\langle \mathbf{w}_{j,r^*}^{(T_1,t)}, j\boldsymbol{\mu}_1 \rangle)$ □

In this way we get the relationship between the maximum inner product $\langle \mathbf{w}_{j,r^*}^{(T_1,t)}, j\boldsymbol{\mu}_1 \rangle$ and other inner products, that is, $\langle \mathbf{w}_{j,r}^{(T_1,t)}, j\boldsymbol{\mu}_1 \rangle = o(\langle \mathbf{w}_{j,r^*}^{(T_1,t)}, j\boldsymbol{\mu}_1 \rangle), r \neq r^*$. Before the next proof begins, we need to first prove the following Lemma:

**Lemma G.4.** *For $j \in \{\pm 1\}$, it holds with probability at least $1 - \delta$ that*

$$\left| n_{1,j} - \frac{n_1}{2} \right| < \sqrt{\frac{n_1}{2} \log \frac{2}{\delta}},$$

*where $n_{1,j} := |\{(\mathbf{x}_{1,i}, y_{1,i}) | y_{1,i} = j, i \in [n_1]\}|$.*

*Proof of Lemma G.4.* Note that $n_{1,j} = \sum_{i=1}^{n_1} \mathbb{I}[y_{1,i} = j]$ where $y_{1,i}$ takes label $+1$ or $-1$ with equal probability $\frac{1}{2}$, according to Hoeffding's inequality, we have

$$\mathbb{P}\left( \left| \sum_{i=1}^{n_1} \mathbb{I}[y_{1,i} = j] - \mathbb{E}\left[ \sum_{i=1}^{n_1} \mathbb{I}[y_{1,i} = j] \right] \right| \geq t \right) \leq 2 \exp\left( -\frac{2t^2}{n_1} \right), j \in \{\pm 1\},$$

and it follows that

$$\mathbb{P}\left( \left| n_{1,j} - \frac{n_1}{2} \right| \geq t \right) \leq 2 \exp\left( -\frac{2t^2}{n_1} \right), j \in \{\pm 1\},$$

leading to

$$\left| n_{1,j} - \frac{n_1}{2} \right| \leq \sqrt{\frac{n_1}{2} \log \frac{2}{\delta}}$$

with probability at least $1 - \delta$. □

Then we have

$$\left| \frac{n_{1,j}}{n_1/2} - 1 \right| \leq \sqrt{\frac{2}{n_1} \log \frac{2}{\delta}} = \Theta\left( \frac{1}{\sqrt{n_1}} \right),$$

leading to

$$\begin{aligned}
n_{1,j} &= \frac{n_1}{2}\left( 1 + \frac{n_{1,j}}{n_1/2} - 1 \right) \\
&= \frac{n_1}{2}\left( 1 \pm \Theta\left( \frac{1}{\sqrt{n_1}} \right) \right) \\
&= \frac{n_1}{2}[1 + o(1)],
\end{aligned}$$

where the last equation follows by $n_1 = \Omega(\mathrm{polylog}(d))$. Similarly, under Condition 3.1, we have

$$n_{2,j} = \frac{n_2}{2}[1 + o(1)] \tag{G.4}$$

for $j \in \{\pm 1\}$. Next, we will prove that the maximum value of the inner product of the positive and negative convolutional filters is approximately equal, that is,

**Lemma G.5.** *When $t$ tends to infinity, $\lim_{t\to+\infty} \frac{\langle \mathbf{w}_{1,r^*}^{(T_1,t)}, \boldsymbol{\mu}_1 \rangle}{\langle \mathbf{w}_{-1,r^*}^{(T_1,t)}, -\boldsymbol{\mu}_1 \rangle} = 1.$*

*proof of lemma G.5.* Using the lemma G.3, We can simplify the equation (G.1):

$$
\begin{aligned}
\langle \mathbf{w}_{1,r^*}^{(T_1,t+1)}, \boldsymbol{\mu}_1 \rangle &= \langle \mathbf{w}_{1,r^*}^{(T_1,t)}, \boldsymbol{\mu}_1 \rangle + \frac{\eta_1 \|\boldsymbol{\mu}_1\|_2^2}{n_1 m} \sum_{y_{1,i}=1} \frac{\sigma'(\langle \mathbf{w}_{1,r^*}^{(T_1,t)}, \boldsymbol{\mu}_1 \rangle)}{1 + \exp(\frac{1}{m}\sigma(\langle \mathbf{w}_{1,r^*}^{(T_1,t)}, \boldsymbol{\mu}_1 \rangle)[1+o(1)]} \\
&= \langle \mathbf{w}_{1,r^*}^{(T_1,t)}, \boldsymbol{\mu}_1 \rangle + \frac{\eta_1 \|\boldsymbol{\mu}_1\|_2^2}{n_1 m} \frac{n_1}{2}[1+o(1)] \frac{\sigma'(\langle \mathbf{w}_{1,r^*}^{(T_1,t)}, \boldsymbol{\mu}_1 \rangle)}{1 + \exp(\frac{1}{m}\sigma(\langle \mathbf{w}_{1,r^*}^{(T_1,t)}, \boldsymbol{\mu}_1 \rangle)[1+o(1)]} \\
&= \langle \mathbf{w}_{1,r^*}^{(T_1,t)}, \boldsymbol{\mu}_1 \rangle + \frac{\eta_1 \|\boldsymbol{\mu}_1\|_2^2}{2m} \frac{\sigma'(\langle \mathbf{w}_{1,r^*}^{(T_1,t)}, \boldsymbol{\mu}_1 \rangle)}{1 + \exp(\frac{1}{m}\sigma(\langle \mathbf{w}_{1,r^*}^{(T_1,t)}, \boldsymbol{\mu}_1 \rangle)} + o(1),
\end{aligned}
\tag{G.5}
$$

we can also simplify the equation (G.2)

$$
\begin{aligned}
\langle \mathbf{w}_{-1,r^*}^{(T_1,t+1)}, -\boldsymbol{\mu}_1 \rangle &= \langle \mathbf{w}_{-1,r^*}^{(T_1,t)}, -\boldsymbol{\mu}_1 \rangle + \frac{\eta_1 \|\boldsymbol{\mu}_1\|_2^2}{n_1 m} \sum_{y_{1,i}=-1} \frac{\sigma'(\langle \mathbf{w}_{-1,r^*}^{(T_1,t)}, -\boldsymbol{\mu}_1 \rangle)}{1 + \exp[(\frac{1}{m}\sigma(\langle \mathbf{w}_{-1,r^*}^{(T_1,t)}, -\boldsymbol{\mu}_1 \rangle)[1+o(1)]]} \\
&= \langle \mathbf{w}_{-1,r^*}^{(T_1,t)}, -\boldsymbol{\mu}_1 \rangle + \frac{\eta_1 \|\boldsymbol{\mu}_1\|_2^2}{n_1 m} \frac{n_1}{2}[1+o(1)] \frac{\sigma'(\langle \mathbf{w}_{-1,r^*}^{(T_1,t)}, -\boldsymbol{\mu}_1 \rangle)}{1 + \exp[(\frac{1}{m}\sigma(\langle \mathbf{w}_{-1,r^*}^{(T_1,t)}, -\boldsymbol{\mu}_1 \rangle)[1+o(1)]]} \\
&= \langle \mathbf{w}_{-1,r^*}^{(T_1,t)}, -\boldsymbol{\mu}_1 \rangle + \frac{\eta_1 \|\boldsymbol{\mu}_1\|_2^2}{2m} \frac{\sigma'(\langle \mathbf{w}_{-1,r^*}^{(T_1,t)}, -\boldsymbol{\mu}_1 \rangle)}{1 + \exp(\frac{1}{m}\sigma(\langle \mathbf{w}_{-1,r^*}^{(T_1,t)}, -\boldsymbol{\mu}_1 \rangle)} + o(1).
\end{aligned}
\tag{G.6}
$$

We have the function $f(x) = \frac{x^{q-1}}{1+\exp(\frac{1}{m}x^q)}$, further we get a derivative of this function $f'(x) = \frac{x^{2q-2}}{(1+\exp(\frac{1}{m}x^q))^2}(\frac{q-1}{x^q} + (\frac{q-1}{x^q} - 1)\exp(\frac{1}{m}x^q))$. When $x \geq (2\max\{m,q\})^{\frac{1}{q}}$, we have that

$$
\begin{aligned}
&\frac{q-1}{x^q} + (\frac{q-1}{x^q} - 1)\exp(\frac{1}{m}x^q) \\
\leq\ &\frac{q-1}{x^q} + (\frac{q-1}{x^q} - 1)(\frac{1}{m}x^q) \\
=\ &\frac{q-1}{x^q} + \frac{q-1}{m} - \frac{1}{m}x^q \\
<\ &\frac{q}{x^q} + \frac{q}{m} - \frac{1}{m}x^q \\
\leq\ &\frac{q}{2m} + \frac{q}{m} - \frac{2q}{m} \\
<\ &0,
\end{aligned}
$$

where the first inequality is due to the fact that $e^x > x$ for $x > 0$, and the last inequality follows by the condition that $x \geq (2\max\{m,q\})^{\frac{1}{q}}$. So when $x \geq (2\max\{m,q\})^{\frac{1}{q}}$, $f'(x) < 0$, $f(x)$ decreases monotonically. By Lemma G.1, we know that at some point in the iteration $max_r \langle \mathbf{w}_{j,r}^{(T_1,t)}, j\boldsymbol{\mu}_1 \rangle$ will reach $(2\max\{m,q\})^{\frac{1}{q}}$. Suppose that at some point, $\langle \mathbf{w}_{1,r^*}^{(T_1,t)} \rangle = \langle \mathbf{w}_{-1,r^*}^{(T_1,t)} \rangle + C$, where $C = o(1)$ is a positive constant, then we subtract the two equations (G.5) and (G.6), we have that

$$
\begin{aligned}
\langle \mathbf{w}_{1,r^*}^{(T_1,t+1)}, \boldsymbol{\mu}_1 \rangle - \langle \mathbf{w}_{-1,r^*}^{(T_1,t+1)}, -\boldsymbol{\mu}_1 \rangle &= \langle \mathbf{w}_{1,r^*}^{(T_1,t)}, \boldsymbol{\mu}_1 \rangle - \langle \mathbf{w}_{-1,r^*}^{(T_1,t)}, -\boldsymbol{\mu}_1 \rangle \\
&+ \frac{\eta_1 \|\boldsymbol{\mu}_1\|_2^2}{2m} \frac{\sigma'(\langle \mathbf{w}_{1,r^*}^{(T_1,t)}, \boldsymbol{\mu}_1 \rangle)}{1 + \exp(\frac{1}{m}\sigma(\langle \mathbf{w}_{1,r^*}^{(T_1,t)}, \boldsymbol{\mu}_1 \rangle)} - \frac{\eta_1 \|\boldsymbol{\mu}_1\|_2^2}{2m} \frac{\sigma'(\langle \mathbf{w}_{-1,r^*}^{(T_1,t)}, -\boldsymbol{\mu}_1 \rangle)}{1 + \exp(\frac{1}{m}\sigma(\langle \mathbf{w}_{-1,r^*}^{(T_1,t)}, -\boldsymbol{\mu}_1 \rangle)} + o(1) \\
&< \langle \mathbf{w}_{1,r^*}^{(T_1,t)}, \boldsymbol{\mu}_1 \rangle - \langle \mathbf{w}_{-1,r^*}^{(T_1,t)}, -\boldsymbol{\mu}_1 \rangle \\
&= C,
\end{aligned}
$$

where the equation is due to the fact that $f(x) = \frac{\eta_1 \|\boldsymbol{\mu}_1\|_2^2}{2m} \frac{\sigma'(x)}{1+\exp(\frac{1}{m}\sigma(x))}$ is a monotonically decreasing function when

$x \geq (2\max\{m,q\})^{\frac{1}{q}}$. So when t tends to infinity, $|\langle \mathbf{w}_{1,r^*}^{(T_1,t)}, \boldsymbol{\mu}_1 \rangle - \langle \mathbf{w}_{-1,r^*}^{(T_1,t)}, -\boldsymbol{\mu}_1 \rangle| \leq C$, we have that

$$\lim_{t \to +\infty} \frac{\langle \mathbf{w}_{1,r^*}^{(T_1,t)}, \boldsymbol{\mu}_1 \rangle}{\langle \mathbf{w}_{-1,r^*}^{(T_1,t)}, -\boldsymbol{\mu}_1 \rangle} = \lim_{t \to +\infty} \frac{\langle \mathbf{w}_{-1,r^*}^{(T_1,t)}, -\boldsymbol{\mu}_1 \rangle}{\langle \mathbf{w}_{-1,r^*}^{(T_1,t)}, -\boldsymbol{\mu}_1 \rangle} + \lim_{t \to +\infty} \frac{\langle \mathbf{w}_{1,r^*}^{(T_1,t)}, \boldsymbol{\mu}_1 \rangle - \langle \mathbf{w}_{-1,r^*}^{(T_1,t)}, -\boldsymbol{\mu}_1 \rangle}{\langle \mathbf{w}_{-1,r^*}^{(T_1,t)}, -\boldsymbol{\mu}_1 \rangle} = 1,$$

which completes the proof. $\qquad\square$

## H. Proof of Replay-based Methods

The replay method here refers to combining the partial data of task $T_{k-1}$ and all data of task $T_k$ for model training. The initial weight of the model is the weight value retained at the end of the previous task: $\mathbf{w}_{j,r}^{(0)} = \mathbf{w}_{j,r}^{(T_{k-1}, t_{end})}$. Let us consider the case of two tasks, task $T_1$ and task $T_2$. The signal vectors for the two tasks are $\boldsymbol{\mu}_1$ and $\boldsymbol{\mu}_2$, and the noise vector are $\boldsymbol{\xi}_{1,i}$ and $\boldsymbol{\xi}_{2,i}$. The empirical replay method is mainly to solve the forgetting situation where the angle between two tasks is obtuse, so we might as well make the angle between the two signal vectors obtuse $\theta$. To simplify the mathematical form, we let $\lambda$ represent the $-\cos\theta_{1,2}$, expressed in terms of the formula $\lambda = -\cos\theta_{1,2}$. So the inner product of $\boldsymbol{\mu}_1$ and $\boldsymbol{\mu}_2$ is

$$\langle \boldsymbol{\mu}_1, \boldsymbol{\mu}_2 \rangle = -\lambda \|\boldsymbol{\mu}_1\|_2 \|\boldsymbol{\mu}_2\|_2, \ \ \lambda \in (0,1).$$

Then $\mathbf{w}_{j,r}^{(0)} = \mathbf{w}_{j,r}^{(T_1, t_{end})}$, at this time, the loss function is defined as

$$L_{S_1 \cup S_2}(\mathbf{W}) = \frac{1}{n_1^* + n_2}(\sum_{i=1}^{n_1^*} \ell[y_{1,i} \cdot f(\mathbf{W}, \mathbf{x}_{1,i}) + \sum_{i=1}^{n_2} \ell[y_{2,i} \cdot f(\mathbf{W}, \mathbf{x}_{2,i})])$$

Moreover, we define a number $k$, which is expressed by the formula $k = \min\left\{\frac{1-C_2}{1+C_2}, \frac{(1-C_2)(3+C_2)}{8(C_2+1)}, \frac{(3+C_2)(1-C_2)}{4\left(8\left(\frac{\|\boldsymbol{\mu}_2\|_2}{\|\boldsymbol{\mu}_1\|_2}\right)^q + \left(\frac{1+C_2}{2}\right)^q\right)^{1/q}}\right\} \frac{\|\boldsymbol{\mu}_2\|_2}{\|\boldsymbol{\mu}_1\|_2}$, where the $C_1$ and $C_2$ is the same as that in Lemma F.19. The number $k$ is used to describe the growth of $\mathbf{w}_{j,r}^{(T_2,t)}$ on the vector $\boldsymbol{\mu}_2$. Then we can verify that when $0 < \lambda \leq \frac{1+C_2}{2}$, the two below inequality holds:

$$(1-\lambda^2)^q - (k\frac{\|\boldsymbol{\mu}_1\|_2}{\|\boldsymbol{\mu}_2\|_2}\lambda)^q \geq (1 - (\frac{1+C_2}{2})^2)^q - (k\frac{\|\boldsymbol{\mu}_1\|_2}{\|\boldsymbol{\mu}_2\|_2}\frac{1+C_2}{2})^q = 2\widetilde{C}$$

$$\geq (1 - (\frac{1+C_2}{2})^2 - k\frac{\|\boldsymbol{\mu}_1\|_2}{\|\boldsymbol{\mu}_2\|_2}\frac{1+C_2}{2})q(k\frac{\|\boldsymbol{\mu}_1\|_2}{\|\boldsymbol{\mu}_2\|_2}\frac{1+C_2}{2})^{q-1}$$

$$= O(1), \tag{H.1}$$

where the first inequality is by the decreasing nature of the function $f(\lambda) = (1-\lambda^2)^q - (k\frac{\|\boldsymbol{\mu}_1\|_2}{\|\boldsymbol{\mu}_2\|_2}\lambda)^q$, the second inequality is due to the Lagrange Median Theorem, and the last inequality follows by the definition of $k$, and we denote the positive constant by $2\widetilde{C}$, and we can obtain

$$\widetilde{C} \geq 4k^q, \tag{H.2}$$

where the inequality is by the definition of $k$.

**Lemma H.1.** *The gradient of loss function $L_{S_1 \cup S_2}(\mathbf{W}^{(T_2,t)})$ with respect to weight parameters $\mathbf{w}_{j,r}^{(T_2,t)}$ is*

$$\nabla_{\mathbf{w}_{j,r}^{(T_2,t)}} L_{S_1 \cup S_2}(\mathbf{W}^{(T_2,t)}) = \frac{1}{(n_1^* + n_2)m} \sum_{i=1}^{n_1^*} (-\ell_{1,i}'^{(T_2,t)}) \cdot \sigma'(\langle \mathbf{w}_{j,r}^{(T_2,t)}, \boldsymbol{\xi}_{1,i} \rangle) \cdot j y_{1,i} \boldsymbol{\xi}_{1,i}$$

$$+ \frac{1}{(n_1^* + n_2)m} \sum_{i=1}^{n_2} (-\ell_{2,i}'^{(T_2,t)}) \cdot \sigma'(\langle \mathbf{w}_{j,r}^{(T_2,t)}, \boldsymbol{\xi}_{2,i} \rangle) \cdot j y_{2,i} \boldsymbol{\xi}_{2,i}$$

$$+ \frac{1}{(n_1^* + n_2)m} \sum_{i=1}^{n_1^*} (-\ell_{1,i}'^{(T_2,t)}) \cdot \sigma'(\langle \mathbf{w}_{j,r}^{(T_2,t)}, y_{1,i} \boldsymbol{\mu}_1 \rangle) \cdot j \boldsymbol{\mu}_1$$

$$+ \frac{1}{(n_1^* + n_2)m} \sum_{i=1}^{n_2} (-\ell_{2,i}'^{(T_2,t)}) \cdot \sigma'(\langle \mathbf{w}_{j,r}^{(T_2,t)}, y_{2,i}\boldsymbol{\mu}_2 \rangle) \cdot j\boldsymbol{\mu}_2,$$

*for $j = \pm 1, 1 \le r \le m$, where $\ell'(y_{1,i}f(\mathbf{W}^{(t)}, x_{1,i})]) = -1/(1 + \exp[y_{1,i}f(\mathbf{W}^{(t)}, \mathbf{x}_{1,i})])$ is denoted by $\ell_{1,i}'^{(t)}$, and $\ell'(y_{2,i}f(\mathbf{W}^{(t)}, x_{2,i})]) = -1/(1 + \exp[y_{2,i}f(\mathbf{W}^{(t)}, \mathbf{x}_{2,i})])$ is denoted by $\ell_{2,i}'^{(t)}$.*

*proof of Lemma H.1.* When $j = \pm 1, 1 \le r \le m$,

$$\nabla_{\mathbf{w}_{j,r}^{(T_2,t)}} \ell[y_{1,i} \cdot f(\mathbf{W}^{(T_2,t)}, \mathbf{x}_{1,i}) = \ell'(y_{1,i}f(\mathbf{W}^{(T_2,t)}, \mathbf{x}_{1,i})]) \cdot y_{1,i} \cdot \nabla_{\mathbf{w}_{j,r}^{(T_2,t)}} f(\mathbf{W}^{(T_2,t)}, \mathbf{x}_{1,i})$$
$$= \ell_{1,i}'^{(T_2,t)} \cdot y_{1,i} \cdot (j \cdot \sigma'(\langle \mathbf{w}_{j,r}^{(T_2,t)}, y_{1,i}\boldsymbol{\mu}_1 \rangle)y_{1,i}\boldsymbol{\mu}_1 + j \cdot \sigma'(\langle \mathbf{w}_{j,r}^{(T_2,t)}, \boldsymbol{\xi}_{1,i} \rangle)\boldsymbol{\xi}_{1,i}),$$

$$\nabla_{\mathbf{w}_{j,r}^{(T_2,t)}} \ell[y_{2,i} \cdot f(\mathbf{W}^{(T_2,t)}, \mathbf{x}_{2,i}) = \ell'(y_{2,i}f(\mathbf{W}^{(T_2,t)}, \mathbf{x}_{2,i})]) \cdot y_{2,i} \cdot \nabla_{\mathbf{w}_{j,r}^{(T_2,t)}} f(\mathbf{W}^{(T_2,t)}, \mathbf{x}_{2,i})$$
$$= \ell_{2,i}'^{(T_2,t)} \cdot y_{2,i} \cdot (j \cdot \sigma'(\langle \mathbf{w}_{j,r}^{(T_2,t)}, y_{2,i}\boldsymbol{\mu}_2 \rangle)y_{2,i}\boldsymbol{\mu}_2 + j \cdot \sigma'(\langle \mathbf{w}_{j,r}^{(T_2,t)}, \boldsymbol{\xi}_{2,i} \rangle)\boldsymbol{\xi}_{2,i}).$$

Note that $\nabla_{\mathbf{w}_{j,r}} L_{S_1 \cup S_2}(\mathbf{W}) = \frac{1}{n_1^* + n_2}(\sum_{i=1}^{n_1^*} \nabla_{\mathbf{w}_{j,r}} \ell[y_{1,i} \cdot f(\mathbf{W}, \mathbf{x}_{1,i})] + \sum_{i=1}^{n_2} \nabla_{\mathbf{w}_{j,r}} \ell[y_{2,i} \cdot f(\mathbf{W}, \mathbf{x}_{2,i})])$, and bringing the results of our calculations into this equation, we can get that

$$\nabla_{\mathbf{w}_{j,r}^{(T_2,t)}} L_{S_1 \cup S_2}(\mathbf{W}^{(T_2,t)}) = \frac{1}{(n_1^* + n_2)m} \sum_{i=1}^{n_1^*} (-\ell_{1,i}'^{(T_2,t)}) \cdot \sigma'(\langle \mathbf{w}_{j,r}^{(T_2,t)}, \boldsymbol{\xi}_{1,i} \rangle) \cdot jy_{1,i}\boldsymbol{\xi}_{1,i}$$
$$+ \frac{1}{(n_1^* + n_2)m} \sum_{i=1}^{n_2} (-\ell_{2,i}'^{(T_2,t)}) \cdot \sigma'(\langle \mathbf{w}_{j,r}^{(T_2,t)}, \boldsymbol{\xi}_{2,i} \rangle) \cdot jy_{2,i}\boldsymbol{\xi}_{2,i}$$
$$+ \frac{1}{(n_1^* + n_2)m} \sum_{i=1}^{n_1^*} (-\ell_{1,i}'^{(T_2,t)}) \cdot \sigma'(\langle \mathbf{w}_{j,r}^{(T_2,t)}, y_{1,i}\boldsymbol{\mu}_1 \rangle) \cdot j\boldsymbol{\mu}_1$$
$$+ \frac{1}{(n_1^* + n_2)m} \sum_{i=1}^{n_2} (-\ell_{2,i}'^{(T_2,t)}) \cdot \sigma'(\langle \mathbf{w}_{j,r}^{(T_2,t)}, y_{2,i}\boldsymbol{\mu}_2 \rangle) \cdot j\boldsymbol{\mu}_2, \tag{H.3}$$

which completes the proof. $\qquad\square$

When the model is trained by gradient descent, the update rule can be formulated by

$$\mathbf{w}_{j,r}^{(T_2,t+1)} = \mathbf{w}_{j,r}^{(T_2,t)} - \eta_2 \cdot \nabla_{\mathbf{w}_{j,r}} L_{S_1 \cup S_2}(\mathbf{W}^{(T_2,t)}), \tag{H.4}$$

for $j = \pm 1$ and $r \in [m]$. In the empirical replay method, we place equal emphasis on the laws of change in the inner products. Specifically, $\langle \mathbf{w}_{j,r}^{(T_2,t)}, j\boldsymbol{\mu}_1 \rangle$ and $\langle \mathbf{w}_{j,r}^{(T_2,t)}, j\boldsymbol{\mu}_1 \rangle$ represent feature learning, while $\langle \mathbf{w}_{j,r}^{(T_2,t)}, \boldsymbol{\xi}_{1,i} \rangle$ and $\langle \mathbf{w}_{j,r}^{(T_2,t)}, \boldsymbol{\xi}_{2,i} \rangle$ pertain to noise memorization. And then we have following lemma for the inner product update rule.

**Lemma H.2.** *The performance of gradient descent with respect to feature learning and noise memorization can be formulated by*

$$\langle \mathbf{w}_{j,r}^{(T_2,t)}, j\boldsymbol{\mu}_1 \rangle = \langle \mathbf{w}_{j,r}^{(T_2,0)}, j\boldsymbol{\mu}_1 \rangle + \frac{\eta_2 \|\boldsymbol{\mu}_1\|_2^2}{(n_1^* + n_2)m} \sum_{s=0}^{t-1} \sum_{i=1}^{n_1^*} (-\ell_{1,i}'^{(T_2,s)}) \cdot \sigma'(\langle \mathbf{w}_{j,r}^{(T_2,s)}, y_{1,i}\boldsymbol{\mu}_1 \rangle)$$
$$- \frac{\lambda\eta_2 \|\boldsymbol{\mu}_1\|_2 \|\boldsymbol{\mu}_2\|_2}{(n_1^* + n_2)m} \sum_{s=0}^{t-1} \sum_{i=1}^{n_2} (-\ell_{2,i}'^{(T_2,s)}) \cdot \sigma'(\langle \mathbf{w}_{j,r}^{(T_2,s)}, y_{2,i}\boldsymbol{\mu}_2 \rangle), \tag{H.5}$$

$$\langle \mathbf{w}_{j,r}^{(T_2,t)}, j\boldsymbol{\mu}_2 \rangle = \langle \mathbf{w}_{j,r}^{(T_2,0)}, j\boldsymbol{\mu}_2 \rangle - \frac{\lambda\eta_2 \|\boldsymbol{\mu}_1\|_2 \|\boldsymbol{\mu}_2\|_2}{(n_1^* + n_2)m} \sum_{s=0}^{t-1} \sum_{i=1}^{n_1^*} (-\ell_{1,i}'^{(T_2,s)}) \cdot \sigma'(\langle \mathbf{w}_{j,r}^{(T_2,s)}, y_{1,i}\boldsymbol{\mu}_1 \rangle)$$

$$+ \frac{\eta_2 \|\boldsymbol{\mu}_2\|_2^2}{(n_1^* + n_2)m} \sum_{s=0}^{t-1} \sum_{i=1}^{n_2} (-\ell_{2,i}'^{(T_2,s)}) \cdot \sigma'(\langle \mathbf{w}_{j,r}^{(T_2,s)}, y_{2,i}\boldsymbol{\mu}_2 \rangle), \tag{H.6}$$

$$\langle \mathbf{w}_{j,r}^{(T_2,t)}, \boldsymbol{\xi}_{1,i} \rangle = \langle \mathbf{w}_{j,r}^{(T_2,0)}, \boldsymbol{\xi}_{1,i} \rangle + \frac{\eta_2 \|\boldsymbol{\xi}_{1,i}\|_2^2}{(n_1^* + n_2)m} \sum_{s=0}^{t-1} \sum_{i=1}^{n_1^*} (-\ell_{1,i}'^{(T_2,s)}) \cdot \sigma'(\langle \mathbf{w}_{j,r}^{(T_2,s)}, \boldsymbol{\xi}_{1,i} \rangle) \cdot j y_{1,i}$$

$$+ \frac{\eta_2}{(n_1^* + n_2)m} \sum_{s=0}^{t-1} \sum_{i=1}^{n_2} (-\ell_{2,i}'^{(T_2,s)}) \cdot \sigma'(\langle \mathbf{w}_{j,r}^{(T_2,s)}, \boldsymbol{\xi}_{2,i} \rangle) \cdot j y_{2,i} \langle \boldsymbol{\xi}_{1,i}, \boldsymbol{\xi}_{2,i} \rangle, \tag{H.7}$$

$$\langle \mathbf{w}_{j,r}^{(T_2,t)}, \boldsymbol{\xi}_{2,i} \rangle = \langle \mathbf{w}_{j,r}^{(T_2,0)}, \boldsymbol{\xi}_{2,i} \rangle + \frac{\eta_2}{(n_1^* + n_2)m} \sum_{s=0}^{t-1} \sum_{i=1}^{n_1^*} (-\ell_{1,i}'^{(T_2,s)}) \cdot \sigma'(\langle \mathbf{w}_{j,r}^{(T_2,s)}, \boldsymbol{\xi}_{1,i} \rangle) \cdot j y_{1,i} \langle \boldsymbol{\xi}_{1,i}, \boldsymbol{\xi}_{2,i} \rangle$$

$$+ \frac{\eta_2 \|\boldsymbol{\xi}_{2,i}\|_2^2}{(n_1^* + n_2)m} \sum_{s=0}^{t-1} \sum_{i=1}^{n_2} (-\ell_{2,i}'^{(T_2,s)}) \cdot \sigma'(\langle \mathbf{w}_{j,r}^{(T_2,s)}, \boldsymbol{\xi}_{2,i} \rangle) \cdot j y_{2,i}. \tag{H.8}$$

*proof of Lemma H.2.* From equation (H.4) and lemma H.1, we can get that

$$\mathbf{w}_{j,r}^{(T_2,t+1)} = \mathbf{w}_{j,r}^{(T_2,t)} - \eta_2 \cdot \nabla_{\mathbf{w}_{j,r}^{(T_2,t)}} L_{S_1 \cup S_2}(\mathbf{W}^{(T_2,t)})$$

$$= \mathbf{w}_{j,r}^{(T_2,t)} + \frac{\eta_2}{(n_1^* + n_2)m} \sum_{i=1}^{n_1^*} (-\ell_{1,i}'^{(T_2,t)}) \cdot \sigma'(\langle \mathbf{w}_{j,r}^{(T_2,t)}, \boldsymbol{\xi}_{1,i} \rangle) \cdot j y_{1,i} \boldsymbol{\xi}_{1,i}$$

$$+ \frac{\eta_2}{(n_1^* + n_2)m} \sum_{i=1}^{n_2} (-\ell_{2,i}'^{(T_2,t)}) \cdot \sigma'(\langle \mathbf{w}_{j,r}^{(T_2,t)}, \boldsymbol{\xi}_{2,i} \rangle) \cdot j y_{2,i} \boldsymbol{\xi}_{2,i}$$

$$+ \frac{\eta_2}{(n_1^* + n_2)m} \sum_{i=1}^{n_1^*} (-\ell_{1,i}'^{(T_2,t)}) \cdot \sigma'(\langle \mathbf{w}_{j,r}^{(T_2,t)}, y_{1,i}\boldsymbol{\mu}_1 \rangle) \cdot j\boldsymbol{\mu}_1$$

$$+ \frac{\eta_2}{(n_1^* + n_2)m} \sum_{i=1}^{n_2} (-\ell_{2,i}'^{(T_2,t)}) \cdot \sigma'(\langle \mathbf{w}_{j,r}^{(T_2,t)}, y_{2,i}\boldsymbol{\mu}_2 \rangle) \cdot j\boldsymbol{\mu}_2. \tag{H.9}$$

Adding up the left side of the equation (H.9) gives us the following equation:

$$\mathbf{w}_{j,r}^{(T_2,t)} = \mathbf{w}_{j,r}^{(T_2,0)} + \frac{\eta_2}{(n_1^* + n_2)m} \sum_{s=0}^{t-1} \sum_{i=1}^{n_1^*} (-\ell_{1,i}'^{(T_2,s)}) \cdot \sigma'(\langle \mathbf{w}_{j,r}^{(T_2,s)}, \boldsymbol{\xi}_{1,i} \rangle) \cdot j y_{1,i} \boldsymbol{\xi}_{1,i}$$

$$+ \frac{\eta_2}{(n_1^* + n_2)m} \sum_{s=0}^{t-1} \sum_{i=1}^{n_2} (-\ell_{2,i}'^{(T_2,s)}) \cdot \sigma'(\langle \mathbf{w}_{j,r}^{(T_2,s)}, \boldsymbol{\xi}_{2,i} \rangle) \cdot j y_{2,i} \boldsymbol{\xi}_{2,i}$$

$$+ \frac{\eta_2}{(n_1^* + n_2)m} \sum_{s=0}^{t-1} \sum_{i=1}^{n_1^*} (-\ell_{1,i}'^{(T_2,s)}) \cdot \sigma'(\langle \mathbf{w}_{j,r}^{(T_2,s)}, y_{1,i}\boldsymbol{\mu}_1 \rangle) \cdot j\boldsymbol{\mu}_1$$

$$+ \frac{\eta_2}{(n_1^* + n_2)m} \sum_{s=0}^{t-1} \sum_{i=1}^{n_2} (-\ell_{2,i}'^{(T_2,s)}) \cdot \sigma'(\langle \mathbf{w}_{j,r}^{(T_2,s)}, y_{2,i}\boldsymbol{\mu}_2 \rangle) \cdot j\boldsymbol{\mu}_2 \tag{H.10}$$

By doing a vector inner product on each side of the equation, we can get the Lemma H.2. This completes the proof. □

We define the coefficients $\gamma(\boldsymbol{\mu}_1)_{j,r}^{(T_2,t)}, \gamma(\boldsymbol{\mu}_2)_{j,r}^{(T_2,t)}, \rho(\boldsymbol{\mu}_1)_{j,r,i}^{(T_2,t)}, \rho(\boldsymbol{\mu}_2)_{j,r,i}^{(T_2,t)}$ as follows, which satisfy the following iterative equations:

$$\gamma(\boldsymbol{\mu}_1)_{j,r}^{(T_2,0)}, \gamma(\boldsymbol{\mu}_2)_{j,r}^{(T_2,0)}, \rho(\boldsymbol{\mu}_1)_{j,r,i}^{(T_2,0)}, \rho(\boldsymbol{\mu}_2)_{j,r,i}^{(T_2,0)} = 0$$

$$\gamma(\boldsymbol{\mu}_1)_{j,r}^{(T_2,t+1)} = \gamma(\boldsymbol{\mu}_1)_{j,r}^{(T_2,t)} + \frac{\eta_2}{(n_1^* + n_2)m} \sum_{i=1}^{n_1^*} (-\ell_{1,i}'^{(T_2,t)}) \cdot \sigma'(\langle \mathbf{w}_{j,r}^{(T_2,t)}, y_{1,i}\boldsymbol{\mu}_1 \rangle) \cdot \|\boldsymbol{\mu}_1\|_2^2,$$

$$\gamma(\boldsymbol{\mu}_2)_{j,r}^{(T_2,t+1)} = \gamma(\boldsymbol{\mu}_2)_{j,r}^{(T_2,t)} + \frac{\eta_2}{(n_1^* + n_2)m} \sum_{i=1}^{n_2} (-\ell_{2,i}'^{(T_2,t)}) \cdot \sigma'(\langle \mathbf{w}_{j,r}^{(T_2,t)}, y_{2,i}\boldsymbol{\mu}_2 \rangle) \cdot \|\boldsymbol{\mu}_2\|_2^2,$$

$$\rho(\boldsymbol{\mu}_1)_{j,r,i}^{(T_2,t+1)} = \rho(\boldsymbol{\mu}_1)_{j,r,i}^{(T_2,t)} + \frac{\eta_2}{(n_1^* + n_2)m} (-\ell_{1,i}'^{(T_2,t)}) \sigma'(\langle \mathbf{w}_{j,r}^{(T_2,t)}, \boldsymbol{\xi}_{1,i} \rangle) \|\boldsymbol{\xi}_{1,i}\|_2^2 j y_{1,i},$$

$$\rho(\boldsymbol{\mu}_2)_{j,r,i}^{(T_2,t+1)} = \rho(\boldsymbol{\mu}_2)_{j,r,i}^{(T_2,t)} + \frac{\eta_2}{(n_1^* + n_2)m} (-\ell_{2,i}'^{(T_2,t)}) \sigma'(\langle \mathbf{w}_{j,r}^{(T_2,t)}, \boldsymbol{\xi}_{2,i} \rangle) \|\boldsymbol{\xi}_{2,i}\|_2^2 j y_{2,i}.$$

**Lemma H.3.** *By the definition of coefficients* $\gamma(\boldsymbol{\mu}_1)_{j,r}^{(T_2,t)}, \gamma(\boldsymbol{\mu}_2)_{j,r}^{(T_2,t)}, \rho(\boldsymbol{\mu}_1)_{j,r,i}^{(T_2,t)}, \rho(\boldsymbol{\mu}_2)_{j,r,i}^{(T_2,t)}$ *as above, We can simplify the equation* (H.10)

$$\mathbf{w}_{j,r}^{(T_2,t)} = \mathbf{w}_{j,r}^{(T_2,0)} + j \cdot \gamma(\boldsymbol{\mu}_1)_{j,r}^{(T_2,t)} \cdot \|\boldsymbol{\mu}_1\|_2^{-2} \boldsymbol{\mu}_1 + j \cdot \gamma(\boldsymbol{\mu}_2)_{j,r}^{(T_2,t)} \cdot \|\boldsymbol{\mu}_2\|_2^{-2} \boldsymbol{\mu}_2$$

$$+ \sum_{i=1}^{n_1^*} \rho(\boldsymbol{\mu}_1)_{j,r,i}^{(T_2,t)} \|\boldsymbol{\xi}_{1,i}\|_2^{-2} \boldsymbol{\xi}_{1,i} + \sum_{i=1}^{n_2} \rho(\boldsymbol{\mu}_2)_{j,r,i}^{(T_2,t)} \|\boldsymbol{\xi}_{2,i}\|_2^{-2} \boldsymbol{\xi}_{2,i}. \tag{H.11}$$

*proof of Lemma H.3.* According to the equation, we can easily calculate that

$$\gamma(\boldsymbol{\mu}_1)_{j,r}^{(T_2,t)} = \gamma(\boldsymbol{\mu}_1)_{j,r}^{(T_2,0)} + \frac{\eta_2}{(n_1^* + n_2)m} \sum_{s=0}^{t-1} \sum_{i=1}^{n_1^*} (-\ell_{1,i}'^{(T_2,s)}) \cdot \sigma'(\langle \mathbf{w}_{j,r}^{(T_2,s)}, y_{1,i}\boldsymbol{\mu}_1 \rangle) \cdot \|\boldsymbol{\mu}_1\|_2^2,$$

$$\gamma(\boldsymbol{\mu}_2)_{j,r}^{(T_2,t)} = \gamma(\boldsymbol{\mu}_2)_{j,r}^{(T_2,0)} + \frac{\eta_2}{(n_1^* + n_2)m} \sum_{s=0}^{t-1} \sum_{i=1}^{n_2} (-\ell_{2,i}'^{(T_2,s)}) \cdot \sigma'(\langle \mathbf{w}_{j,r}^{(T_2,s)}, y_{2,i}\boldsymbol{\mu}_2 \rangle) \cdot \|\boldsymbol{\mu}_2\|_2^2,$$

$$\rho(\boldsymbol{\mu}_1)_{j,r,i}^{(T_2,t)} = \rho(\boldsymbol{\mu}_1)_{j,r,i}^{(T_2,0)} + \frac{\eta_2}{(n_1^* + n_2)m} \sum_{s=0}^{t-1} (-\ell_{1,i}'^{(T_2,s)}) \sigma'(\langle \mathbf{w}_{j,r}^{(T_2,s)}, \boldsymbol{\xi}_{1,i} \rangle) \|\boldsymbol{\xi}_{1,i}\|_2^2 j y_{1,i},$$

$$\rho(\boldsymbol{\mu}_2)_{j,r,i}^{(T_2,t)} = \rho(\boldsymbol{\mu}_2)_{j,r,i}^{(T_2,0)} + \frac{\eta_2}{(n_1^* + n_2)m} \sum_{s=0}^{t-1} (-\ell_{2,i}'^{(T_2,s)}) \sigma'(\langle \mathbf{w}_{j,r}^{(T_2,s)}, \boldsymbol{\xi}_{2,i} \rangle) \|\boldsymbol{\xi}_{2,i}\|_2^2 j y_{2,i}.$$

$\square$

Bringing these equations into equation (H.10) completes the proof. Through the action of these four coefficients, we have succeeded in simplifying the equation (H.10). Further, the inner product update rule can also be simplified, they can be rewritten as

$$\langle \mathbf{w}_{j,r}^{(T_2,t)}, j\boldsymbol{\mu}_1 \rangle = \langle \mathbf{w}_{j,r}^{(T_2,0)}, j\boldsymbol{\mu}_1 \rangle + \gamma(\boldsymbol{\mu}_1)_{j,r}^{(T_2,t)} - \lambda \frac{\|\boldsymbol{\mu}_1\|_2}{\|\boldsymbol{\mu}_2\|_2} \gamma(\boldsymbol{\mu}_2)_{j,r}^{(T_2,t)} \tag{H.12}$$

$$\langle \mathbf{w}_{j,r}^{(T_2,t)}, j\boldsymbol{\mu}_2 \rangle = \langle \mathbf{w}_{j,r}^{(T_2,0)}, j\boldsymbol{\mu}_2 \rangle + \gamma(\boldsymbol{\mu}_2)_{j,r}^{(T_2,t)} - \lambda \frac{\|\boldsymbol{\mu}_2\|_2}{\|\boldsymbol{\mu}_1\|_2} \gamma(\boldsymbol{\mu}_1)_{j,r}^{(T_2,t)} \tag{H.13}$$

$$\langle \mathbf{w}_{j,r}^{(T_2,t)}, \boldsymbol{\xi}_{1,i} \rangle = \langle \mathbf{w}_{j,r}^{(T_2,0)}, \boldsymbol{\xi}_{1,i} \rangle + \rho(\boldsymbol{\mu}_1)_{j,r,i}^{(T_2,t)} + \sum_{i' \neq i} \rho(\boldsymbol{\mu}_1)_{j,r,i'}^{(T_2,t)} \|\boldsymbol{\xi}_{1,i'}\|_2^{-2} \langle \boldsymbol{\xi}_{1,i'}, \boldsymbol{\xi}_{1,i} \rangle + \sum_{i=1}^{n_2} \rho(\boldsymbol{\mu}_2)_{j,r,i}^{(T_2,t)} \|\boldsymbol{\xi}_{2,i}\|_2^{-2} \langle \boldsymbol{\xi}_{2,i}, \boldsymbol{\xi}_{1,i} \rangle$$

$$\langle \mathbf{w}_{j,r}^{(T_2,t)}, \boldsymbol{\xi}_{2,i} \rangle = \langle \mathbf{w}_{j,r}^{(T_2,0)}, \boldsymbol{\xi}_{2,i} \rangle + \rho(\boldsymbol{\mu}_2)_{j,r,i}^{(T_2,t)} + \sum_{i=1}^{n_1^*} \rho(\boldsymbol{\mu}_1)_{j,r,i}^{(T_2,t)} \|\boldsymbol{\xi}_{1,i}\|_2^{-2} \langle \boldsymbol{\xi}_{1,i}, \boldsymbol{\xi}_{2,i} \rangle + \sum_{i' \neq i} \rho(\boldsymbol{\mu}_2)_{j,r,i'}^{(T_2,t)} \|\boldsymbol{\xi}_{2,i'}\|_2^{-2} \langle \boldsymbol{\xi}_{2,i'}, \boldsymbol{\xi}_{2,i} \rangle$$

When performing training for task $T_1$, let's assume that we plan to stop training when the training loss reaches $\epsilon$. Then at the end of the task $T_1$ training, we have that

$$L_{S_1}(W^{(T_1,t_{end})}) = \frac{1}{n_1} \sum_{i=1}^{n_1} \ell(y_{1,i} f(\mathbf{W}^{(T_1,t_{end})}, x_{1,i}))$$

$$= \frac{1}{n_1} \sum_{i=1}^{n_1} \log(1 + e^{-\frac{1}{m}\sigma(\langle \mathbf{w}_{y_{1,i},r^*}^{(T_1,t_{end})}, y_{1,i}\boldsymbol{\mu}_1 \rangle)[1+o(1)]})$$

$$\in [\frac{\epsilon}{C_0}, \epsilon], \tag{H.14}$$

where $C_0$ is a positive constant and $C_0 > 1$. Suppose the maximum value $\langle \mathbf{w}_{j,r^*}^{(T_1,t_{end})}, j\boldsymbol{\mu}_1 \rangle$ has reached $\Lambda_{max}$, then we can get

$$\frac{1}{n_1} \sum_{i=1}^{n_1} \log(1 + e^{-\frac{1}{m}\sigma(\Lambda_{max})[1+o(1)]}) \in [\frac{\epsilon}{C_0}, \epsilon]$$

$$\Rightarrow \Lambda_{max} = \left[ m\log(\frac{1}{e^\epsilon - 1}) \right]^{\frac{1}{q}} [1 + o(1)] = \left[ m\log(\frac{1}{\epsilon}) \right]^{\frac{1}{q}} [1 + o(1)], \tag{H.15}$$

where the last equation is due to the fact that $e^\epsilon - 1 = \epsilon$ when $\epsilon$ converges to 0.

Next we will give a lemma, which will be used in the proof that follows.

**Lemma H.4.** *Under Condition 3.1, if* $\gamma(\boldsymbol{\mu}_1)_{j,r}^{(T_2,t)} = \lambda \frac{\|\boldsymbol{\mu}_1\|_2}{\|\boldsymbol{\mu}_2\|_2} \gamma(\boldsymbol{\mu}_2)_{j,r}^{(T_2,t)}$, *when t tends to infinity, for* $r \in I_{j,2}$, *we have*

- $\max_r \gamma(\boldsymbol{\mu}_2)_{j,r}^{(T_2,t)} \to +\infty$ *for* $j \in \{\pm 1\}$.

- $\gamma(\boldsymbol{\mu}_2)_{j,r}^{(T_2,t)} = o(\gamma(\boldsymbol{\mu}_2)_{j,r^*}^{(T_2,t)})$, *where* $r^* = \arg\max \gamma(\boldsymbol{\mu}_2)_{j,r}^{(T_2,t)}, r \neq r^*$ *and* $j \in \{\pm 1\}$.

- $\gamma(\boldsymbol{\mu}_2)_{1,r}^{(T_2,t)} = \gamma(\boldsymbol{\mu}_2)_{-1,r}^{(T_2,t)}[1 + o(1)]$.

*Proof of Lemma H.4.* Recall the update rule of $\gamma(\boldsymbol{\mu}_2)_{j,r}^{(T_2,t)}$, we can get

$$\gamma(\boldsymbol{\mu}_2)_{j,r}^{(T_2,t+1)} = \gamma(\boldsymbol{\mu}_2)_{j,r}^{(T_2,t)} + \frac{\eta_2}{(n_1^* + n_2)m} \sum_{y_{2,i}=j} (-\ell_{2,i}'^{(T_2,t)}) \cdot \sigma'(\langle \mathbf{w}_{j,r}^{(T_2,0)}, j\boldsymbol{\mu}_2 \rangle + (1-\lambda^2)\gamma(\boldsymbol{\mu}_2)_{j,r}^{(T_2,t)}) \cdot \|\boldsymbol{\mu}_2\|_2^2.$$

Similar to the analysis of the $\max\langle \mathbf{w}_{j,r}^{(T_2,t)}, j\boldsymbol{\mu}_1 \rangle$ in the previous section G, using the same proof method, we can prove that when $t$ tends to infinity

- $\max_r \gamma(\boldsymbol{\mu}_2)_{j,r}^{(T_2,t)} \to +\infty$ for $j \in \{\pm 1\}$.

- $\gamma(\boldsymbol{\mu}_2)_{j,r}^{(T_2,t)} = o(\gamma(\boldsymbol{\mu}_2)_{j,r^*}^{(T_2,t)})$, where $r^* = \arg\max \gamma(\boldsymbol{\mu}_2)_{j,r}^{(T_2,t)}, r \neq r^*$ and $j \in \{\pm 1\}$.

- $\gamma(\boldsymbol{\mu}_2)_{1,r}^{(T_2,t)} = \gamma(\boldsymbol{\mu}_2)_{-1,r}^{(T_2,t)}[1 + o(1)]$.

Here the proof completes. $\square$

**Proposition H.5.** *Under Condition 3.1, for* $0 \leq t \leq T_2^*$, *we have that*

$$0 \leq \gamma(\boldsymbol{\mu}_1)_{j,r}^{(T_2,t)}, \gamma(\boldsymbol{\mu}_2)_{j,r}^{(T_2,t)} \leq \alpha_2 \tag{H.16}$$

*for all* $j \in \{\pm 1\}$ *and* $r \in [m]$, *and*

$$0 \leq |\rho(\boldsymbol{\mu}_1)_{j,r,i}^{(T_2,t)}|, |\rho(\boldsymbol{\mu}_2)_{j,r,i}^{(T_2,t)}| \leq \alpha_2 \tag{H.17}$$

*for all* $j \in \{\pm 1\}$, $r \in [m]$, $i \in [n_1^*]$ *and* $i' \in [n_2]$.

We will use induction to prove Proposition H.5. We will elaborate on the following theorems, which will be used to prove Proposition H.5.

**Lemma H.6.** *For any $t \geq 0$, we have that*

$$\langle \mathbf{w}_{j,r}^{(T_2,t)}, \boldsymbol{\xi}_{1,i} \rangle \leq \langle \mathbf{w}_{j,r}^{(T_2,0)}, \boldsymbol{\xi}_{1,i} \rangle + 4n_1^* \sqrt{\frac{\log(4n_1^2/\delta)}{d}} \alpha_2 + 4n_2 \frac{\sigma_{p_1}}{\sigma_{p_2}} \sqrt{\frac{\log(4n_2^2/\delta)}{d}} \alpha_2,$$

$$\langle \mathbf{w}_{j,r}^{(T_2,t)}, \boldsymbol{\xi}_{2,i} \rangle \leq \langle \mathbf{w}_{j,r}^{(T_2,0)}, \boldsymbol{\xi}_{2,i} \rangle + 4n_1^* \frac{\sigma_{p_2}}{\sigma_{p_1}} \sqrt{\frac{\log(4n_1^2/\delta)}{d}} \alpha_2 + 4n_2 \sqrt{\frac{\log(4n_2^2/\delta)}{d}} \alpha_2,$$

*for any $r \in [m]$ and $y_{1,i}, y_{2,i} = -j$.*

*Proof of Lemma H.6.* Recall the update rule for $\langle \mathbf{w}_{j,r}^{(T_2,t)}, \boldsymbol{\xi}_{1,i} \rangle$:

$$\langle \mathbf{w}_{j,r}^{(T_2,t)}, \boldsymbol{\xi}_{1,i} \rangle = \langle \mathbf{w}_{j,r}^{(T_2,0)}, \boldsymbol{\xi}_{1,i} \rangle + \rho(\boldsymbol{\mu}_1)_{j,r,i}^{(T_2,t)} + \sum_{i' \neq i} \rho(\boldsymbol{\mu}_1)_{j,r,i'}^{(T_2,t)} \|\boldsymbol{\xi}_{1,i'}\|_2^{-2} \langle \boldsymbol{\xi}_{1,i'}, \boldsymbol{\xi}_{1,i} \rangle + \sum_{i=1}^{n_2} \rho(\boldsymbol{\mu}_2)_{j,r,i}^{(T_2,t)} \|\boldsymbol{\xi}_{2,i}\|_2^{-2} \langle \boldsymbol{\xi}_{2,i}, \boldsymbol{\xi}_{1,i} \rangle,$$

so we have

$$\begin{aligned}
\langle \mathbf{w}_{j,r}^{(T_2,t)}, \boldsymbol{\xi}_{1,i} \rangle &\leq \langle \mathbf{w}_{j,r}^{(T_2,0)}, \boldsymbol{\xi}_{1,i} \rangle + \sum_{i' \neq i} \rho(\boldsymbol{\mu}_1)_{j,r,i'}^{(T_2,t)} \|\boldsymbol{\xi}_{1,i'}\|_2^{-2} \langle \boldsymbol{\xi}_{1,i'}, \boldsymbol{\xi}_{1,i} \rangle + \sum_{i=1}^{n_2} \rho(\boldsymbol{\mu}_2)_{j,r,i}^{(T_2,t)} \|\boldsymbol{\xi}_{2,i}\|_2^{-2} \langle \boldsymbol{\xi}_{2,i}, \boldsymbol{\xi}_{1,i} \rangle \\
&\leq \langle \mathbf{w}_{j,r}^{(T_2,0)}, \boldsymbol{\xi}_{1,i} \rangle + 4\sqrt{\frac{\log(4n_1^2/\delta)}{d}} \sum_{i' \neq i} \rho(\boldsymbol{\mu}_1)_{j,r,i'}^{(T_2,t)} + 4\frac{\sigma_{p_1}}{\sigma_{p_2}} \sqrt{\frac{\log(4n_2^2/\delta)}{d}} \sum_{i=1}^{n_2} \rho(\boldsymbol{\mu}_2)_{j,r,i}^{(T_2,t)} \\
&\leq \langle \mathbf{w}_{j,r}^{(T_2,0)}, \boldsymbol{\xi}_{1,i} \rangle + 4n_1^* \sqrt{\frac{\log(4n_1^2/\delta)}{d}} \alpha_2 + 4n_2 \frac{\sigma_{p_1}}{\sigma_{p_2}} \sqrt{\frac{\log(4n_2^2/\delta)}{d}} \alpha_2,
\end{aligned}$$

where the first inequality is by that $\rho(\boldsymbol{\mu}_1)_{j,r,i}^{(T_2,t)} \leq 0$ when $y_{i,i} = -j$, the second inequality follows by Lemma C.1 and the last inequality is due to our induction hypothesis. Similarly, we can prove that

$$\langle \mathbf{w}_{j,r}^{(T_2,t)}, \boldsymbol{\xi}_{2,i} \rangle \leq \langle \mathbf{w}_{j,r}^{(T_2,0)}, \boldsymbol{\xi}_{2,i} \rangle + 4n_1^* \frac{\sigma_{p_2}}{\sigma_{p_1}} \sqrt{\frac{\log(4n_1^2/\delta)}{d}} \alpha_2 + 4n_2 \sqrt{\frac{\log(4n_2^2/\delta)}{d}} \alpha_2,$$

which completes the proof. $\qquad\square$

**Lemma H.7.** *Under Condition 3.1, for $0 \leq t \leq T_2^*$, we have*

$$F_j(\mathbf{W}_j^{(T_2,t)}, \mathbf{x}_{2,i}) \leq 2^{q+1} (\lambda \frac{\|\boldsymbol{\mu}_2\|_2}{\|\boldsymbol{\mu}_1\|_2} \alpha)^q,$$

*for $j = -y_{2,i}$, where the constant $\alpha$ is defined in Cao et al. (2022)*

*Proof of Lemma H.7.* Firstly, we know that $\gamma(\boldsymbol{\mu}_1)_{j,r}^{(T_1,t_{end})} \leq \alpha$, which is proved in Cao et al. (2022) So when $y_{2,i} = -j$, we calculate $\langle \mathbf{w}_{j,r}^{(T_2,0)}, -j\boldsymbol{\mu}_2 \rangle$ as follows:

$$\langle \mathbf{w}_{j,r}^{(T_2,0)}, -j\boldsymbol{\mu}_2 \rangle = \langle \mathbf{w}_{j,r}^{(T_1,0)}, -j\boldsymbol{\mu}_2 \rangle + \lambda \frac{\|\boldsymbol{\mu}_2\|_2}{\|\boldsymbol{\mu}_1\|_2} \gamma(\boldsymbol{\mu}_1)_{j,r}^{(T_1,t_{end})} \leq 2\lambda \frac{\|\boldsymbol{\mu}_2\|_2}{\|\boldsymbol{\mu}_1\|_2} \alpha, \qquad (\text{H.18})$$

where the inequality is due to Lemma C.2 that $\langle \mathbf{w}_{j,r}^{(T_1,0)}, -j\boldsymbol{\mu}_2 \rangle \leq \sqrt{2\log(8m/\delta)} \cdot \sigma_0 \|\boldsymbol{\mu}_2\|_2 \leq \lambda \frac{\|\boldsymbol{\mu}_2\|_2}{\|\boldsymbol{\mu}_1\|_2} \alpha$. We also have

$$\begin{aligned}
\langle \mathbf{w}_{j,r}^{(T_2,0)}, \boldsymbol{\xi}_{2,i} \rangle &= \langle \mathbf{w}_{j,r}^{(T_1,0)}, \boldsymbol{\xi}_{2,i} \rangle + \sum_{i=1}^{n_1} \rho_{j,r,i}^{(T_1,t)} \|\boldsymbol{\xi}_{1,i}\|_2^{-2} \cdot \langle \boldsymbol{\xi}_{1,i}, \boldsymbol{\xi}_{2,i} \rangle \\
&\leq 2\sqrt{\log(8mn_2/\delta)} \cdot \sigma_0 \sigma_{p_1} \sqrt{d} + 4n_1 \frac{\sigma_{p_2}}{\sigma_{p_1}} \sqrt{\frac{\log(4n_1^2/\delta)}{d}} \alpha
\end{aligned}$$

$$\leq 2\lambda \frac{\|\boldsymbol{\mu}_2\|_2}{\|\boldsymbol{\mu}_1\|_2}\alpha, \tag{H.19}$$

where the second inequality is by Lemma C.1 and Lemma C.2, the third inequality is due to Condition 3.1 that $\sigma_0 \leq \widetilde{O}(m^{-[2/(q-2)]\vee 1} \cdot \max\{n_1, n_2\}^{-[1/(q-2)]\vee 1}) \cdot \min\{(\sigma_{p_1}\sqrt{d})^{-1}, \|\boldsymbol{\mu}_1\|_2^{-1}, (\sigma_{p_2}\sqrt{d})^{-1}, \|\boldsymbol{\mu}_2\|_2^{-1}\}$ and $d = \widetilde{\Omega}(m^{2\vee[4/(q-2)]} \cdot \max\{n_1, n_2\}^{4\vee[(2q-2)/(q-2)]})$. Then we can get

$$F_j(\mathbf{W}_j^{(T_2,t)}, \mathbf{x}_{2,i}) = \frac{1}{m}\sum_{r=1}^{m}[\sigma(\langle \mathbf{w}_{j,r}^{(T_2,t)}, -j\boldsymbol{\mu}_2\rangle) + \sigma(\langle \mathbf{w}_{j,r}^{(T_2,t)}, \boldsymbol{\xi}_{2,i}\rangle)]$$

$$\leq 2^{q+1}\max_{j,r,i}\left\{|\langle \mathbf{w}_{j,r}^{(T_2,0)}, -j\boldsymbol{\mu}_2\rangle|, |\langle \mathbf{w}_{j,r}^{(T_2,0)}, \boldsymbol{\xi}_{2,i}\rangle|, 4n_1^*\frac{\sigma_{p_2}}{\sigma_{p_1}}\sqrt{\frac{\log(4n_1^2/\delta)}{d}}\alpha_2 + 4n_2\sqrt{\frac{\log(4n_2^2/\delta)}{d}}\alpha_2\right\}^q$$

$$\leq 2^{2q+1}(\lambda \frac{\|\boldsymbol{\mu}_2\|_2}{\|\boldsymbol{\mu}_1\|_2}\alpha)^q,$$

where the first inequality is due to (H.18) and (H.19), and the last inequality is by Condition 3.1, which completes the proof. $\qquad\square$

Then we will prove Proposition H.5.

*Proof of Proposition H.5.* When $y_{2,i} = j$, denote $t_{j,r,i}$ the last time that $\rho(\boldsymbol{\mu}_2)_{j,r,i}^{(T_2,t)} \leq 0.5\alpha_2$, and recall the update rule for $\rho(\boldsymbol{\mu}_2)_{j,r,i}^{(T_2,t)}$, we have that

$$\rho(\boldsymbol{\mu}_2)_{j,r,i}^{(T_2,\widetilde{T})} \leq \rho(\boldsymbol{\mu}_2)_{j,r,i}^{(T_2,t_{j,r,i})} + \underbrace{\frac{\eta_2}{(n_1^*+n_2)m}\cdot(-\ell_{2,i'}^{(T_2,t_{j,r,i})})\cdot\sigma'(\langle \mathbf{w}_{j,r}^{(T_2,t_{j,r,i})}, \boldsymbol{\xi}_{2,i}\rangle)\|\boldsymbol{\xi}_{2,i}\|_2^2}_{I_1}$$

$$+ \underbrace{\sum_{t_{j,r,i}<t<T}\frac{\eta_2}{(n_1^*+n_2)m}\cdot(-\ell_{2,i'}^{(T_2,t)})\cdot\sigma'(\langle \mathbf{w}_{j,r}^{(T_2,t)}, \boldsymbol{\xi}_{2,i}\rangle)\|\boldsymbol{\xi}_{2i}\|_2^2}_{I_2}, \tag{H.20}$$

for any $\widetilde{T} \leq T_2^*$. Then we will give $I_1$ an upper bound:

$$I_1 \leq \frac{\eta_2\|\boldsymbol{\xi}_{2,i}\|_2^2}{(n_1^*+n_2)m}\sigma'(|\langle \mathbf{w}_{j,r}^{(T_2,0)}, \boldsymbol{\xi}_{2,i}\rangle| + \rho(\boldsymbol{\mu}_2)_{j,r,i}^{(T_2,t_{j,r,i})} + 4n_1^*\frac{\sigma_{p_2}}{\sigma_{p_1}}\sqrt{\frac{\log(4n_1^2/\delta)}{d}}\alpha_2 + 4n_2\sqrt{\frac{\log(4n_2^2/\delta)}{d}}\alpha_2)$$

$$\leq \frac{2^q q\eta_2\sigma_{p_2}^2 d}{(n_1^*+n_2)m}\alpha_2^{q-1}$$

$$\leq 0.25\alpha_2,$$

where the second inequality is due to that $|\langle \mathbf{w}_{j,r}^{(T_2,0)}, \boldsymbol{\xi}_{2,i}\rangle| \leq 0.1\alpha_2$, $\rho(\boldsymbol{\mu}_2)_{j,r,i}^{(T_2,t_{j,r,i})} \leq 0.5\alpha_2$ and $4n_1^*\frac{\sigma_{p_2}}{\sigma_{p_1}}\sqrt{\frac{\log(4n_1^2/\delta)}{d}}\alpha_2 + 4n_2\sqrt{\frac{\log(4n_2^2/\delta)}{d}}\alpha_2 \leq 0.1\alpha_2$, and the last inequality follows by Condition 3.1 that $\eta_2 \leq \frac{(n_1+n_2)m}{8q\sigma_{p_2}^2 d\alpha_2^{q-2}}$. When the time $t$ reaches $t_{j,r,i}$, we can lower bound $\langle \mathbf{w}_{j,r}^{(T_2,t)}, \boldsymbol{\xi}_{2,i}\rangle$ as follows,

$$\langle \mathbf{w}_{j,r}^{(T_2,t)}, \boldsymbol{\xi}_{2,i}\rangle \geq \rho(\boldsymbol{\mu}_2)_{j,r,i}^{(T_2,t_{j,r,i})} - |\langle \mathbf{w}_{j,r}^{(T_2,0)}, \boldsymbol{\xi}_{2,i}\rangle| - 4n_1^*\frac{\sigma_{p_2}}{\sigma_{p_1}}\sqrt{\frac{\log(4n_1^2/\delta)}{d}}\alpha_2 - 4n_2\sqrt{\frac{\log(4n_2^2/\delta)}{d}}\alpha_2$$

$$\geq 0.5\alpha_2 - 0.1\alpha_2 - 0.1\alpha_2$$

$$\geq 0.25\alpha_2,$$

where the second inequality is due to $\rho(\boldsymbol{\mu}_2)_{j,r,i}^{(T_2,t_{j,r,i})} \geq 0.5\alpha_2$, $|\langle \mathbf{w}_{j,r}^{(T_2,0)}, \boldsymbol{\xi}_{2,i}\rangle| \leq 0.1\alpha$ and $4n_1^*\frac{\sigma_{p_2}}{\sigma_{p_1}}\sqrt{\frac{\log(4n_1^2/\delta)}{d}}\alpha_2 + 4n_2\sqrt{\frac{\log(4n_2^2/\delta)}{d}}\alpha_2 \leq 0.1\alpha_2$. Then we will upper bound $\langle \mathbf{w}_{j,r}^{(T_2,t)}, \boldsymbol{\xi}_{2,i}\rangle$ in the same way,

$$\langle \mathbf{w}_{j,r}^{(T_2,t)}, \boldsymbol{\xi}_{2,i}\rangle \leq \rho(\boldsymbol{\mu}_2)_{j,r,i}^{(T_2,t)} + \langle \mathbf{w}_{j,r}^{(T_2,0)}, \boldsymbol{\xi}_{2,i}\rangle + 4n_1^*\frac{\sigma_{p_2}}{\sigma_{p_1}}\sqrt{\frac{\log(4n_1^2/\delta)}{d}}\alpha_2 + 4n_2\sqrt{\frac{\log(4n_2^2/\delta)}{d}}\alpha_2$$

$$\leq \alpha_2 + 0.1\alpha_2 + 0.1\alpha_2$$
$$\leq 2\alpha_2,$$

where the second inequality is by the induction hypothesis $\rho(\boldsymbol{\mu}_2)_{j,r,i}^{(T_2,t)} \leq \alpha_2$ and $\langle \mathbf{w}_{j,r}^{(T_2,0)}, \boldsymbol{\xi}_{2,i}\rangle, 4n_1^* \frac{\sigma_{p_2}}{\sigma_{p_1}} \sqrt{\frac{\log(4n_1^2/\delta)}{d}}\alpha_2 + 4n_2\sqrt{\frac{\log(4n_2^2/\delta)}{d}}\alpha_2 \leq 0.1\alpha_2$. Then we can plug the lower bound and upper bound of $\langle \mathbf{w}_{j,r}^{(T_2,t)}, \boldsymbol{\xi}_{2,i}\rangle$ into $I_2$:

$$
\begin{aligned}
I_2 &\leq \sum_{t_{j,r,i} < t < T} \frac{\eta_2}{(n_1^* + n_2)m} \exp\{-\sigma(\langle \mathbf{w}_{j,r}^{(T_2,t)}, \boldsymbol{\xi}_{2,i}\rangle) + 2^{2q+1}(\lambda\frac{\|\boldsymbol{\mu}_2\|_2}{\|\boldsymbol{\mu}_1\|_2}\alpha)^q\} \cdot \sigma'(2\alpha_2)\|\boldsymbol{\xi}_{2i}\|_2^2 \\
&\leq T_2^* \frac{2^q q \eta_2 \sigma_{p_2}^2 d}{(n_1^* + n_2)m} \exp\{-\alpha_2^q/4^q + 2^{2q+1}(\lambda\frac{\|\boldsymbol{\mu}_2\|_2}{\|\boldsymbol{\mu}_1\|_2}\alpha)^q\}\alpha_2^{q-1} \\
&\leq 0.25 T_2^* \exp\{-\alpha_2^q/2^{2q+1}\}\alpha_2 \\
&\leq 0.25 T_2^* \exp\{-(\log T_2^*)^q\}\alpha_2 \\
&\leq 0.25\alpha_2,
\end{aligned}
$$

where the second inequality is by Lemma C.1, the third inequality follows by the definition of $\alpha_2 = \max\{32(\lambda\frac{\|\boldsymbol{\mu}_2\|_2}{\|\boldsymbol{\mu}_1\|_2} + 1)\alpha, 6\log(T_2^*)\}$ and Condition 3.1 on $\eta_2$, the fourth inequality is due to $\alpha_2 \geq 6\log(T_2^*)$ and the last inequality follows by the fact that $(\log T_2^*)^q > \log T_2^*$. When $y_{2,i} = -j$, we can use the similar method in Cao et al. (2022) to prove that $\rho(\boldsymbol{\mu}_2)_{j,r,i}^{(T_2,t)} \geq \alpha_2$, so as to $\gamma(\boldsymbol{\mu}_1)_{j,r}^{(T_2,t)}, \gamma(\boldsymbol{\mu}_2)_{j,r}^{(T_2,t)}$ and $|\rho(\boldsymbol{\mu}_1)_{j,r,i}^{(T_2,t)}|$ for all $j \in \{\pm 1\}$. Here we complete the proof of Proposition H.5. $\qquad\square$

### H.1. Noise Analysis

In this subsection, we will give an upper bound for the noise inner product $\langle \mathbf{w}_{j,r}^{(T_2,t)}, \boldsymbol{\xi}_{1,i}\rangle$ and $\langle \mathbf{w}_{j,r}^{(T_2,t)}, \boldsymbol{\xi}_{2,i}\rangle$. At the end of the task $T_1$ training, we get an upper bound for the $\max_{j,r,i} |\rho_{j,r,i}^{(T_2,t)}|$:

$$\max_{j,r,i} |\rho(\boldsymbol{\mu}_k)_{j,r,i_k}^{(T_2,t)}| \leq \sigma_0 \sigma_{p_k}\sqrt{d},$$

for all $j \in \{\pm 1\}, r \in [m], i_k \in [n_k]$ and $k \in \{1, 2\}$. We have a similar conclusion when training the experience replay method.

**Lemma H.8.** *There exists a time* $T_1^+ = \frac{2^{2-q}(n_1^*+n_2)m\sigma_0^{2-q}\max\{\sigma_{p_1}^{-q}, \sigma_{p_2}^{-q}\}d^{-q/2}}{3^q \eta_2 q[\sqrt{\log(8m(n_1+n_2)/\delta)}]^{q-1}}$, *such that*

$$|\langle \mathbf{w}_{j,r}^{(T_2,t)}, \boldsymbol{\xi}_{1,i}\rangle|, |\langle \mathbf{w}_{j,r}^{(T_2,t)}, \boldsymbol{\xi}_{2,i}\rangle| \leq 6\sqrt{\log(8m(n_1+n_2)/\delta)} \cdot \sigma_0 \sigma_p\sqrt{d}$$

*for any* $j \in \pm 1, r \in [m], i \in [n_1]$ *and* $t \leq T_1^*$

*proof of Lemma H.8.* This lemma requires mathematical induction to prove. When the task $T_1$ training ends, we have

$$
\begin{aligned}
|\langle \mathbf{w}_{j,r}^{(T_2,0)}, \boldsymbol{\xi}_{1,i}\rangle| &\leq |\langle \mathbf{w}_{j,r}^{(T_1,0)}, \boldsymbol{\xi}_{1,i}\rangle| + \max_{j,r,i} |\rho_{j,r,i}^{(T_1,t_{end})}| + \sum_{i' \neq i} |\rho_{j,r,i}^{(T_1,t_{end})}| \cdot \|\boldsymbol{\xi}_{1,i'}\|_2^2 \cdot \langle \boldsymbol{\xi}_{1,i'}, \boldsymbol{\xi}_{1,i}\rangle| \\
&\leq |\langle \mathbf{w}_{j,r}^{(T_1,0)}, \boldsymbol{\xi}_{1,i}\rangle| + \max_{j,r,i} |\rho_{j,r,i}^{(T_1,t_{end})}| + 4n_1\sqrt{\frac{\log(4n_1^2/\delta)}{d}}\max_{j,r,i}|\rho_{j,r,i}^{(T_1,t_{end})}| \\
&\leq 2\sqrt{\log(8mn_1/\delta)} \cdot \sigma_0\sigma_{p_1}\sqrt{d} + \sigma_0\sigma_{p_1}\sqrt{d} + 4n_1\sqrt{\frac{\log(4n_1^2/\delta)}{d}}\sigma_0\sigma_{p_1}\sqrt{d} \\
&\leq 4\sqrt{\log(8mn_1/\delta)} \cdot \sigma_0\sigma_{p_1}\sqrt{d}, \qquad\qquad (H.21)
\end{aligned}
$$

where the first inequality is by the absolute value inequality, the second inequality is due to Lemma C.1 and Lemma C.2, the third inequality follows by $\max_{j,r,i} |\rho_{j,r,i}^{(T_1,t_{end})}| \leq \sigma_0\sigma_{p_1}\sqrt{d}$ proved in Cao et al. (2022), and the last inequality is by the Condition 3.1 that $d \geq 1024\log(4\max\{n_1^2, n_2^2\}/\delta)\alpha_2^2\max\{n_1^2, n_2^2\}\max\{\frac{\sigma_{p_1}^2}{\sigma_{p_2}^2}, \frac{\sigma_{p_2}^2}{\sigma_{p_1}^2}\}$.

Recall the rules for updating the inner product of the noise vector for the empirical replay method

$$\langle \mathbf{w}_{j,r}^{(T_2,t)}, \boldsymbol{\xi}_{1,i} \rangle = \langle \mathbf{w}_{j,r}^{(T_2,0)}, \boldsymbol{\xi}_{1,i} \rangle + \sum_{i'=1}^{n_1^*} \rho(\boldsymbol{\mu}_1)_{j,r,i'}^{(T_2,t)} \|\boldsymbol{\xi}_{1,i'}\|_2^{-2} \langle \boldsymbol{\xi}_{1,i'}, \boldsymbol{\xi}_{1,i} \rangle + \sum_{i=1}^{n_2} \rho(\boldsymbol{\mu}_2)_{j,r,i}^{(T_2,t)} \|\boldsymbol{\xi}_{2,i}\|_2^{-2} \langle \boldsymbol{\xi}_{2,i}, \boldsymbol{\xi}_{1,i} \rangle.$$

From the absolute inequality we can get

$$
|\langle \mathbf{w}_{j,r}^{(T_2,t)}, \boldsymbol{\xi}_{1,i} \rangle| \leq |\langle \mathbf{w}_{j,r}^{(T_2,0)}, \boldsymbol{\xi}_{1,i} \rangle| + |\rho(\boldsymbol{\mu}_1)_{j,r,i}^{(T_2,t)}|
$$
$$
+ \sum_{i' \neq i} |\rho(\boldsymbol{\mu}_1)_{j,r,i'}^{(T_2,t)}| \|\boldsymbol{\xi}_{1,i'}\|_2^{-2} |\langle \boldsymbol{\xi}_{1,i'}, \boldsymbol{\xi}_{1,i} \rangle| + \sum_{i=1}^{n_2} |\rho(\boldsymbol{\mu}_2)_{j,r,i}^{(T_2,t)}| \|\boldsymbol{\xi}_{2,i}\|_2^{-2} |\langle \boldsymbol{\xi}_{2,i}, \boldsymbol{\xi}_{1,i} \rangle|. \tag{H.22}
$$

Then we can give $|\rho(\boldsymbol{\mu}_1)_{j,r,i}^{(T_2,t)}|$ an upper bound. Firstly we have that

$$
|\rho(\boldsymbol{\mu}_1)_{j,r,i}^{(T_2,t+1)}| \leq |\rho(\boldsymbol{\mu}_1)_{j,r,i}^{(T_2,t)}| + \frac{\eta_2}{(n_1^* + n_2)m}(-\ell_{1,i}'^{(T_2,t)})\sigma'(\langle \mathbf{w}_{j,r}^{(T_2,t)}, \boldsymbol{\xi}_{1,i} \rangle)\|\boldsymbol{\xi}_{1,i}\|_2^2
$$
$$
\leq |\rho(\boldsymbol{\mu}_1)_{j,r,i}^{(T_2,t)}| + \frac{\eta_2 q}{(n_1^* + n_2)m}(\langle \mathbf{w}_{j,r}^{(T_2,t)}, \boldsymbol{\xi}_{1,i} \rangle)^{q-1}\|\boldsymbol{\xi}_{1,i}\|_2^2
$$
$$
\leq |\rho(\boldsymbol{\mu}_1)_{j,r,i}^{(T_2,t)}| + \frac{3\eta_2 q\sigma_{p_1}^2 d}{2(n_1^* + n_2)m}(|\langle \mathbf{w}_{j,r}^{(T_2,0)}, \boldsymbol{\xi}_{1,i'} \rangle| + |\rho(\boldsymbol{\mu}_1)_{j,r,i}^{(T_2,t)}|
$$
$$
+ \sum_{i \neq i'} |\rho(\boldsymbol{\mu}_1)_{j,r,i}^{(T_2,t)}| \|\boldsymbol{\xi}_{1,i}\|_2^{-2} |\langle \boldsymbol{\xi}_{1,i}, \boldsymbol{\xi}_{1,i'} \rangle| + \sum_{i=1}^{n_2} |\rho(\boldsymbol{\mu}_2)_{j,r,i}^{(T_2,t)}| \|\boldsymbol{\xi}_{2,i}\|_2^{-2} |\langle \boldsymbol{\xi}_{2,i}, \boldsymbol{\xi}_{1,i'} \rangle|)^{q-1}
$$
$$
\leq |\rho(\boldsymbol{\mu}_1)_{j,r,i}^{(T_2,t)}| + \frac{3\eta_2 q\sigma_{p_1}^2 d}{2(n_1^* + n_2)m}[4\sqrt{\log(8m\max\{n_1,n_2\}/\delta)} \cdot \sigma_0\sigma_{p_1}\sqrt{d}
$$
$$
+ \sigma_0\sigma_{p_1}\sqrt{d} + 4n_1^*\sqrt{\frac{\log(4n_1^2/\delta)}{d}}\sigma_0\sigma_{p_1}\sqrt{d} + 4n_2\frac{\sigma_{p_1}}{\sigma_{p_2}}\sqrt{\frac{\log(4n_2^2/\delta)}{d}}\sigma_0\sigma_{p_2}\sqrt{d}]^{q-1}
$$
$$
\leq |\rho(\boldsymbol{\mu}_1)_{j,r,i}^{(T_2,t)}| + \frac{3\eta_2 q\sigma_{p_1}^2 d}{2(n_1^* + n_2)m}[5\sqrt{\log(8m(n_1+n_2)/\delta)} \cdot \sigma_0\sigma_{p_1}\sqrt{d}]^{q-1},
$$

where the third inequality is by (H.22) and Lemma C.1, and the fourth inequality follows by the induction hypothesis that $\rho(\boldsymbol{\mu}_1)_{j,r,i}^{(T_2,t)} \leq \sigma_0\sigma_{p_1}\sqrt{d}$ and Lemma C.1 and Lemma C.2, and the last inequality is due to Condition 3.1 on $d$, specifically $d \geq 1024\log(4\max\{n_1^2,n_2^2\}/\delta)\alpha_2^2\max\{n_1^2,n_2^2\}\max\{\frac{\sigma_{p_1}^2}{\sigma_{p_2}^2}, \frac{\sigma_{p_2}^2}{\sigma_{p_1}^2}\}$.

When $t \leq T_1^+$, we have

$$
|\rho(\boldsymbol{\mu}_1)_{j,r,i}^{(T_2,t)}| \leq |\rho(\boldsymbol{\mu}_1)_{j,r,i}^{(T_2,t-1)}| + \frac{3\eta_2 q\sigma_{p_1}^2 d}{2(n_1^* + n_2)m}[5\sqrt{\log(8m(n_1+n_2)/\delta)} \cdot \sigma_0\sigma_{p_1}\sqrt{d}]^{q-1}
$$
$$
\leq |\rho(\boldsymbol{\mu}_1)_{j,r,i}^{(T_2,0)}| + \sum_{s=0}^{t-1}\frac{3\eta_2 q\sigma_{p_1}^2 d}{2(n_1^* + n_2)m}[5\sqrt{\log(8m(n_1+n_2)/\delta)} \cdot \sigma_0\sigma_{p_1}\sqrt{d}]^{q-1}
$$
$$
\leq 0 + t \cdot \frac{3\eta_2 q\sigma_{p_1}^2 d}{2(n_1^* + n_2)m}[5\sqrt{\log(8m(n_1+n_2)/\delta)} \cdot \sigma_0\sigma_{p_1}\sqrt{d}]^{q-1}
$$
$$
\leq T_1^+ \cdot \frac{3\eta_2 q\sigma_{p_1}^2 d}{2(n_1^* + n_2)m}[5\sqrt{\log(8m(n_1+n_2)/\delta)} \cdot \sigma_0\sigma_{p_1}\sqrt{d}]^{q-1}
$$
$$
\leq \sigma_0\sigma_{p_1}\sqrt{d},
$$

where the last inequality is by that $t \leq T_1^+$, and the last inequality is due to the definition of $T_1^+$.

Based on the above analysis, we can give $\langle \mathbf{w}_{j,r}^{(t)}, \boldsymbol{\xi}_{1,i'} \rangle$ an upper bound

$$|\langle \mathbf{w}_{j,r}^{(T_2,t)}, \boldsymbol{\xi}_{1,i} \rangle| \leq 5\sqrt{\log(8m(n_1+n_2)/\delta)} \cdot \sigma_0\sigma_{p_1}\sqrt{d}. \tag{H.23}$$

In the same way, we can also give $\langle \mathbf{w}_{j,r}^{(T_2,t)}, \boldsymbol{\xi}_{2,i} \rangle$ an upper bound. We have

$$
\begin{aligned}
|\langle \mathbf{w}_{j,r}^{(T_2,0)}, \boldsymbol{\xi}_{2,i} \rangle| &\leq |\langle \mathbf{w}_{j,r}^{(T_1,0)}, \boldsymbol{\xi}_{2,i} \rangle| + \sum_{i=1}^{n_1} |\rho_{j,r,i}^{(T_1,t_{end})}| \cdot \|\boldsymbol{\xi}_{1,i}\|_2^2 \cdot \langle \boldsymbol{\xi}_{1,i}, \boldsymbol{\xi}_{2,i} \rangle| \\
&\leq |\langle \mathbf{w}_{j,r}^{(T_1,0)}, \boldsymbol{\xi}_{2,i} \rangle| + 4n_1 \frac{\sigma_{p_2}}{\sigma_{p_1}} \sqrt{\frac{\log(4n_1^2/\delta)}{d}} \max_{j,r,i} |\rho_{j,r,i}^{(T_1,t_{end})}| \\
&\leq 2\sqrt{\log(8mn_2/\delta)} \cdot \sigma_0 \sigma_{p_2} \sqrt{d} + 4n_1 \frac{\sigma_{p_2}}{\sigma_{p_1}} \sqrt{\frac{\log(4n_1^2/\delta)}{d}} \sigma_0 \sigma_{p_1} \sqrt{d} \\
&\leq 4\sqrt{\log(8m \max\{n_1, n_2\}/\delta)} \cdot \sigma_0 \sigma_{p_2} \sqrt{d},
\end{aligned}
$$

where the first inequality is by the absolute value inequality, the second inequality is due to Lemma C.1 and Lemma C.2, the third inequality follows by $\max_{j,r,i} |\rho_{j,r,i}^{(T_1,t_{end})}| \leq \sigma_0 \sigma_{p_1} \sqrt{d}$ proved in Cao et al. (2022), and the last inequality is by the Condition 3.1 that $d \geq 1024 \log(4 \max\{n_1^2, n_2^2\}/\delta) \alpha_2^2 \max\{n_1^2, n_2^2\} \max\{\frac{\sigma_{p_1}^2}{\sigma_{p_2}^2}, \frac{\sigma_{p_2}^2}{\sigma_{p_1}^2}\}$. Further we have

$$
\begin{aligned}
|\langle \mathbf{w}_{j,r}^{(T_2,t)}, \boldsymbol{\xi}_{2,i} \rangle| &\leq |\langle \mathbf{w}_{j,r}^{(T_2,0)}, \boldsymbol{\xi}_{2,i} \rangle| + |\rho(\boldsymbol{\mu}_2)_{j,r,i}^{(T_2,t)}| \\
&+ \sum_{i'=1}^{n_1^*} |\rho(\boldsymbol{\mu}_1)_{j,r,i'}^{(T_2,t)}| \|\boldsymbol{\xi}_{1,i'}\|_2^{-2} |\langle \boldsymbol{\xi}_{1,i'}, \boldsymbol{\xi}_{2,i} \rangle| + \sum_{i' \neq i} |\rho(\boldsymbol{\mu}_2)_{j,r,i'}^{(T_2,t)}| \|\boldsymbol{\xi}_{2,i'}\|_2^{-2} |\langle \boldsymbol{\xi}_{2,i'}, \boldsymbol{\xi}_{2,i} \rangle|, \quad \text{(H.24)}
\end{aligned}
$$

which can be proved by the update rule of $\langle \mathbf{w}_{j,r}^{(T_2,t)}, \boldsymbol{\xi}_{2,i} \rangle$. Then we can get

$$
\begin{aligned}
|\rho(\boldsymbol{\mu}_2)_{j,r,i}^{(T_2,t+1)}| &\leq |\rho(\boldsymbol{\mu}_2)_{j,r,i}^{(T_2,t)}| + \frac{\eta_2}{(n_1^* + n_2)m}(-\ell_{2,i}'^{(T_2,t)}) \sigma'(\langle \mathbf{w}_{j,r}^{(T_2,t)}, \boldsymbol{\xi}_{2,i} \rangle) \|\boldsymbol{\xi}_{2,i}\|_2^2 \\
&\leq |\rho(\boldsymbol{\mu}_2)_{j,r,i}^{(T_2,t)}| + \frac{\eta_2 q}{(n_1^* + n_2)m}(\langle \mathbf{w}_{j,r}^{(T_2,t)}, \boldsymbol{\xi}_{2,i} \rangle) \|\boldsymbol{\xi}_{2,i}\|_2^2 \\
&\leq |\rho(\boldsymbol{\mu}_2)_{j,r,i}^{(T_2,t)}| + \frac{3\eta_2 q \sigma_{p_2}^2 d}{2(n_1^* + n_2)m}(|\langle \mathbf{w}_{j,r}^{(T_2,0)}, \boldsymbol{\xi}_{2,i} \rangle| + |\rho(\boldsymbol{\mu}_2)_{j,r,i}^{(T_2,t)}| \\
&+ \sum_{i'=1}^{n_1^*} |\rho(\boldsymbol{\mu}_1)_{j,r,i'}^{(T_2,t)}| \|\boldsymbol{\xi}_{1,i'}\|_2^{-2} |\langle \boldsymbol{\xi}_{1,i'}, \boldsymbol{\xi}_{2,i} \rangle| + \sum_{i' \neq i} |\rho(\boldsymbol{\mu}_2)_{j,r,i'}^{(T_2,t)}| \|\boldsymbol{\xi}_{2,i'}\|_2^{-2} |\langle \boldsymbol{\xi}_{2,i'}, \boldsymbol{\xi}_{2,i} \rangle|)^{q-1} \\
&\leq |\rho(\boldsymbol{\mu}_2)_{j,r,i}^{(T_2,t)}| + \frac{3\eta_2 q \sigma_{p_2}^2 d}{2(n_1^* + n_2)m}[4\sqrt{\log(8m \max\{n_1, n_2\}/\delta)} \cdot \sigma_0 \sigma_{p_2} \sqrt{d} \\
&+ \sigma_0 \sigma_{p_2} \sqrt{d} + 4n_1^* \frac{\sigma_{p_2}}{\sigma_{p_1}} \sqrt{\frac{\log(4n_1^2/\delta)}{d}} \sigma_0 \sigma_{p_1} \sqrt{d} + 4n_2 \sqrt{\frac{\log(4n_2^2/\delta)}{d}} \sigma_0 \sigma_{p_2} \sqrt{d}]^{q-1} \\
&\leq |\rho(\boldsymbol{\mu}_1)_{j,r,i}^{(T_2,t)}| + \frac{3\eta_2 q \sigma_p^2 d}{2(n_1^* + n_2)m}[5\sqrt{\log(8m(n_1 + n_2)/\delta)} \cdot \sigma_0 \sigma_{p_2} \sqrt{d}]^{q-1},
\end{aligned}
$$

where the third inequality is by (H.24) and Lemma C.1, and the fourth inequality follows by the induction hypothesis that $\rho(\boldsymbol{\mu}_1)_{j,r,i}^{(T_2,t)} \leq \sigma_0 \sigma_{p_1} \sqrt{d}$ and Lemma C.1 and Lemma C.2, and the last inequality is due to Condition 3.1 on $d$. Lastly, we have

$$
\begin{aligned}
|\rho(\boldsymbol{\mu}_2)_{j,r,i}^{(T_2,t)}| &\leq |\rho(\boldsymbol{\mu}_2)_{j,r,i}^{(T_2,t-1)}| + \frac{3\eta_2 q \sigma_{p_2}^2 d}{2(n_1^* + n_2)m}[5\sqrt{\log(8m(n_1 + n_2)/\delta)} \cdot \sigma_0 \sigma_{p_2} \sqrt{d}]^{q-1} \\
&\leq |\rho(\boldsymbol{\mu}_1)_{j,r,i}^{(T_2,0)}| + \sum_{s=0}^{t-1} \frac{3\eta_2 q \sigma_{p_2}^2 d}{2(n_1^* + n_2)m}[5\sqrt{\log(8m(n_1 + n_2)/\delta)} \cdot \sigma_0 \sigma_{p_2} \sqrt{d}]^{q-1} \\
&\leq 0 + t \cdot \frac{3\eta_2 q \sigma_{p_2}^2 d}{2(n_1^* + n_2)m}[5\sqrt{\log(8m(n_1 + n_2)/\delta)} \cdot \sigma_0 \sigma_{p_2} \sqrt{d}]^{q-1} \\
&\leq T_1^* \cdot \frac{3\eta_2 q \sigma_{p_2}^2 d}{2(n_1 + n_2)m}[5\sqrt{\log(8m(n_1 + n_2)/\delta)} \cdot \sigma_0 \sigma_{p_2} \sqrt{d}]^{q-1}
\end{aligned}
$$

$$\leq \sigma_0 \sigma_{p_2} \sqrt{d},$$

where the last inequality is by that $t \leq T_1^+$, and the last inequality is due to the definition of $T_1^+$. Then we can get that

$$|\langle \mathbf{w}_{j,r}^{(T_2,t)}, \boldsymbol{\xi}_{2,i} \rangle| \leq 5\sqrt{\log\left(8m(n_1+n_2)/\delta\right)} \cdot \sigma_0 \sigma_{p_2} \sqrt{d}. \tag{H.25}$$

Here completes the proof. □

We have obtained the upper bounds of $\langle \mathbf{w}_{j,r}^{(T_2,t)}, \boldsymbol{\xi}_{1,i} \rangle$ and $\langle \mathbf{w}_{j,r}^{(T_2,t)}, \boldsymbol{\xi}_{2,i} \rangle$, which we can mark as $C_{\boldsymbol{\xi}}$. The formula to express this meaning is

$$C_{\boldsymbol{\xi}} = 6\sqrt{\log\left(8m(n_1+n_2)/\delta\right)} \cdot \sigma_0 \max\{\sigma_{p_1}, \sigma_{p_2}\} \sqrt{d}.$$

Recall the definition in the subsection F.1, we categorize the neurons $\mathbf{w}_{j,r}^{(T_1,0)}$ into three sets: $I_{j,1} = \{r \in [m] : \langle \mathbf{w}_{j,r}^{(T_1,0)}, j\boldsymbol{\mu}_1 \rangle > 0\}$, $I_{j,1}^c = \{r \in [m] : \langle \mathbf{w}_{j,r}^{(T_1,0)}, j\boldsymbol{\mu}_1 \rangle \leq 0\}$, $I_{j,2} = \{r \in [m] : \langle \mathbf{w}_{j,r}^{(T_1,0)}, j\boldsymbol{\mu}_1 \rangle \leq 0\} \cap \{r \in [m] : \langle \mathbf{w}_{j,r}^{(T_1,0)}, j\boldsymbol{\mu}_1^\perp \rangle > 0\}$ and $I_{j,3} = \{r \in [m] : \langle \mathbf{w}_{j,r}^{(T_1,0)}, j\boldsymbol{\mu}_1 \rangle \leq 0\} \cap \{r \in [m] : \langle \mathbf{w}_{j,r}^{(T_1,0)}, j\boldsymbol{\mu}_1^\perp \rangle \leq 0\}$ for $j \in \{\pm 1\}$. Before doing the next analysis of the changes in the following three neurons, we first want to show that at within a period of time from the start of training for task $T_2$, the value of $-\ell_{1,i}^{\prime (T_2,t)}$ is very small. We have that

$$
\begin{aligned}
-\ell_{1,i}^{\prime (T_2,t)} &= 1/(1 + e^{\frac{1}{m}\sum_r (\sigma(\langle \mathbf{w}_{j,r}^{(T_2,t)}, j\boldsymbol{\mu}_1 \rangle) + \sigma(\langle \mathbf{w}_{j,r}^{(T_2,t)}, \boldsymbol{\xi}_{1,i} \rangle)) - \frac{1}{m}\sum_r (\sigma(\langle \mathbf{w}_{-j,r}^{(T_2,t)}, j\boldsymbol{\mu}_1 \rangle) + \sigma(\langle \mathbf{w}_{-j,r}^{(T_2,t)}, \boldsymbol{\xi}_{1,i} \rangle))}) \\
&\leq 1/(1 + e^{\frac{1}{m}\sum_r \sigma(\langle \mathbf{w}_{j,r}^{(T_2,t)}, j\boldsymbol{\mu}_1 \rangle) - \frac{1}{m}\sum_r \sigma(\langle \mathbf{w}_{-j,r}^{(T_2,t)}, j\boldsymbol{\mu}_1 \rangle) - 2C_{\boldsymbol{\xi}}^q}) \\
&\leq 1/(1 + e^{\frac{1}{m}[(1-\lambda^2)^q - (k\frac{\|\boldsymbol{\mu}_1\|_2}{\|\boldsymbol{\mu}_2\|_2}\lambda)^q]\Lambda_{max}^q - 2C_{\boldsymbol{\xi}}^q}) \\
&\leq 1/(1 + e^{\frac{1}{2}[(1-\lambda^2)^q - (k\frac{\|\boldsymbol{\mu}_1\|_2}{\|\boldsymbol{\mu}_2\|_2}\lambda)^q]\log(1/\epsilon)}) \\
&\leq \epsilon^{\widetilde{C}},
\end{aligned} \tag{H.26}
$$

for $j = y_{1,i}$, the equation is the specific formula of $-\ell_{1,i}^{\prime (T_2,t)}$, the first inequality is by the upper bound of $\langle \mathbf{w}_{j,r}^{(T_2,t)}, \boldsymbol{\xi}_{1,i} \rangle$ and $\langle \mathbf{w}_{j,r}^{(T_2,t)}, \boldsymbol{\xi}_{2,i} \rangle$ proved before, the third inequality follows by the definition of $\Lambda_{max}$, and the last inequality is by H.1. The proof of upper bounds for derivatives is based primarily on induction and will be addressed in subsequent proofs for correctness and specific details.

## H.2. The First Type of Neuron

The first class of neurons is those belonging to the set $I_{j,1}$, which is expressed by the formula as follows:

$$\langle \mathbf{w}_{j,r}^{(T_1,0)}, j\boldsymbol{\mu}_1 \rangle > 0.$$

By the previous analysis, we can get that

$$\langle \mathbf{w}_{j,r}^{(T_2,0)}, j\boldsymbol{\mu}_1 \rangle > 0, \langle \mathbf{w}_{j,r}^{(T_2,0)}, j\boldsymbol{\mu}_2 \rangle < 0.$$

Then, we will first study the maximum value of the inner product $\max_r \langle \mathbf{w}_{j,r}^{(T_1,t_{end})}, j\boldsymbol{\mu}_1 \rangle$ of this set, which is denoted as $\langle \mathbf{w}_{j,r^*}^{(T_1,t_{end})}, j\boldsymbol{\mu}_1 \rangle$. Then we can get the value of the inner product $\langle \mathbf{w}_{j,r^*}^{(T_1,t_{end})}, j\boldsymbol{\mu}_2 \rangle$.

**Lemma H.9.**

$$\langle \mathbf{w}_{j,r}^{(T_1,t_{end})}, j\boldsymbol{\mu}_2 \rangle = -\lambda \Lambda_{max}[1 + o(1)],$$

*for $j \in \{\pm 1\}$.*

*proof of Lemma H.9.* We know that for the maximum inner product,

$$\langle \mathbf{w}_{j,r^*}^{(T_1,t_{end})}, j\boldsymbol{\mu}_1 \rangle = \max_r \langle \mathbf{w}_{j,r}^{(T_1,t_{end})}, j\boldsymbol{\mu}_1 \rangle = \Lambda_{max} > 0, \tag{H.27}$$

And as for task $T_1$, recall that $\langle \mathbf{w}_{j,r}^{(T_1,t)}, j\boldsymbol{\mu}_1 \rangle = \langle \mathbf{w}_{j,r}^{(T_1,0)}, j\boldsymbol{\mu}_1 \rangle + \gamma(\boldsymbol{\mu}_1)_{j,r}^{(T_1,t)}$, so we have

$$
\begin{aligned}
\gamma(\boldsymbol{\mu}_1)_{j,r}^{(T_1,t_{end})} &= \max_r \langle \mathbf{w}_{j,r}^{(T_1,t_{end})}, j\boldsymbol{\mu}_1 \rangle - \max_r \langle \mathbf{w}_{j,r}^{(T_1,0)}, j\boldsymbol{\mu}_1 \rangle \\
&= \max_r \langle \mathbf{w}_{j,r}^{(T_1,t_{end})}, j\boldsymbol{\mu}_1 \rangle [1 - \frac{\max_r \langle \mathbf{w}_{j,r}^{(T_1,0)}, j\boldsymbol{\mu}_1 \rangle}{\max_r \langle \mathbf{w}_{j,r}^{(T_1,t_{end})}, j\boldsymbol{\mu}_1 \rangle}] \\
&= \Lambda_{max}[1 + o(1)],
\end{aligned}
$$

where the last equation is by that $\max_r \langle \mathbf{w}_{j,r}^{(T_1,0)}, j\boldsymbol{\mu}_1 \rangle$ is a definite constant.

Then, we can do a calculation of the value of the inner product $\langle \mathbf{w}_{j,r^*}^{(T_1,t_{end})}, j\boldsymbol{\mu}_2 \rangle$:

$$
\begin{aligned}
\langle \mathbf{w}_{j,r^*}^{(T_1,t_{end})}, j\boldsymbol{\mu}_2 \rangle &= \max_r \langle \mathbf{w}_{j,r}^{(T_1,t_{end})}, j\boldsymbol{\mu}_2 \rangle \\
&= \max_r \langle \mathbf{w}_{j,r}^{(T_1,0)}, j\boldsymbol{\mu}_2 \rangle + \gamma(\boldsymbol{\mu}_1)_{j,r}^{(T_1,t_{end})} \|\boldsymbol{\mu}\|_2^{-2} \langle \boldsymbol{\mu}_1, \boldsymbol{\mu}_2 \rangle \\
&= \max_r \langle \mathbf{w}_{j,r}^{(T_1,0)}, j\boldsymbol{\mu}_2 \rangle - \lambda \frac{\|\boldsymbol{\mu}_2\|_2}{\|\boldsymbol{\mu}_1\|_2} \gamma(\boldsymbol{\mu}_1)_{j,r}^{(T_1,t_{end})} \\
&= -\lambda \frac{\|\boldsymbol{\mu}_2\|_2}{\|\boldsymbol{\mu}_1\|_2} \gamma(\boldsymbol{\mu}_1)_{j,r}^{(T_1,t_{end})} \cdot [1 + \frac{\max_r \langle \mathbf{w}_{j,r}^{(T_1,0)}, j\boldsymbol{\mu}_2 \rangle}{-\lambda \frac{\|\boldsymbol{\mu}_2\|_2}{\|\boldsymbol{\mu}_1\|_2} \gamma(\boldsymbol{\mu}_1)_{j,r}^{(T_1,t_{end})}}] \\
&= -\lambda \frac{\|\boldsymbol{\mu}_2\|_2}{\|\boldsymbol{\mu}_1\|_2} \Lambda_{max}[1 + o(1)],
\end{aligned}
$$

this completes the proof. $\qquad\square$

**Lemma H.10.** *There exists time*

$$
T_2^+ = \frac{m^{\frac{1}{2q}} (n_1^* + n_2) \epsilon^{-\tilde{C}/2}}{3q\eta_2 n_1^* \|\boldsymbol{\mu}_1\|_2^2} \tag{H.28}
$$

*such that*

- $\langle \mathbf{w}_{j,r^*}^{(T_2,t)}, j\boldsymbol{\mu}_1 \rangle = (1 - \lambda^2)\Lambda_{max}[1 + o(1)]$,

- $\langle \mathbf{w}_{j,r^*}^{(T_2,t)}, j\boldsymbol{\mu}_2 \rangle = o(1)$,

*for $j \in \{\pm 1\}$ and $0 \le t \le T_2^+$.*

*proof of Lemma H.10.* In this case, the update method of the $\gamma(\boldsymbol{\mu}_1)_{j,r}^{(T_2,t)}, \gamma(\boldsymbol{\mu}_2)_{j,r}^{(T_2,t)}$ is as follows

$$
\begin{aligned}
\gamma(\boldsymbol{\mu}_1)_{j,r}^{(T_2,t+1)} &= \gamma(\boldsymbol{\mu}_1)_{j,r}^{(T_2,t)} \\
&\quad + \frac{\eta_2}{(n_1^* + n_2)m} \sum_{y_{1,i}=j} (-\ell_{1,i}'^{(T_2,t)}) \cdot \sigma'(\Lambda_{max} + \gamma(\boldsymbol{\mu}_1)_{j,r}^{(T_2,t)} - \lambda \frac{\|\boldsymbol{\mu}_1\|_2}{\|\boldsymbol{\mu}_2\|_2} \gamma(\boldsymbol{\mu}_2)_{j,r}^{(T_2,t)}) \cdot \|\boldsymbol{\mu}_1\|_2^2, \\
\gamma(\boldsymbol{\mu}_2)_{j,r}^{(T_2,t+1)} &= \gamma(\boldsymbol{\mu}_2)_{j,r}^{(T_2,t)} \\
&\quad + \frac{\eta_2}{(n_1^* + n_2)m} \sum_{y_{2,i}=-j} (-\ell_{2,i}'^{(T_2,t)}) \cdot \sigma'(\lambda \frac{\|\boldsymbol{\mu}_2\|_2}{\|\boldsymbol{\mu}_1\|_2} \Lambda_{max}[1 + o(1)] - \gamma(\boldsymbol{\mu}_2)_{j,r}^{(T_2,t)} + \lambda \frac{\|\boldsymbol{\mu}_2\|_2}{\|\boldsymbol{\mu}_1\|_2} \gamma(\boldsymbol{\mu}_1)_{j,r}^{(T_2,t)}) \cdot \|\boldsymbol{\mu}_2\|_2^2.
\end{aligned}
$$

Firstly, we will use induction to show that when $t \le T_2^+$, $\gamma(\boldsymbol{\mu}_1)_{j,r}^{(T_2,t)}$ has a very little change

$$
\gamma(\boldsymbol{\mu}_1)_{j,r}^{(T_2,\tilde{T})} \le m^{-\frac{1}{2q}}.
$$

By the definition, we know that $\gamma(\boldsymbol{\mu}_1)_{j,r}^{(T_2,0)} = 0$. According to the inductive hypothesis (H.26), we can get an upper bound for $-\ell_{1,i}^{'(T_2,t)}$. Thus, plugging the upper bound of $-\ell_{1,i}^{'(T_2,t)}$ into $\gamma(\boldsymbol{\mu}_1)_{j,r}^{(T_2,t+1)}$ update rule gives

$$
\begin{aligned}
\gamma(\boldsymbol{\mu}_1)_{j,r}^{(T_2,t+1)} &= \gamma(\boldsymbol{\mu}_1)_{j,r}^{(T_2,t)} + \frac{\eta_2\|\boldsymbol{\mu}_1\|_2^2}{(n_1^*+n_2)m} \sum_{y_i=j}^{n_1^*} (-\ell_{1,i}^{'(T_2,t)})\sigma'(\langle \mathbf{w}_{j,r}^{(T_2,0)}, j\boldsymbol{\mu}_1\rangle + \gamma(\boldsymbol{\mu}_1)_{j,r}^{(T_2,t)} - \lambda\frac{\|\boldsymbol{\mu}_1\|_2}{\|\boldsymbol{\mu}_2\|_2}\gamma(\boldsymbol{\mu}_2)_{j,r}^{(T_2,t)}) \\
&\le \gamma(\boldsymbol{\mu}_1)_{j,r}^{(T_2,t)} + \frac{3\eta_2\|\boldsymbol{\mu}_1\|_2^2 q n_1^*}{4(n_1^*+n_2)m}\epsilon^{\widetilde{C}}(\Lambda_{max} + \gamma(\boldsymbol{\mu}_1)_{j,r}^{(T_2,t)})^{q-1} \\
&\le \gamma(\boldsymbol{\mu}_1)_{j,r}^{(T_2,t)} + \frac{3\eta_2\|\boldsymbol{\mu}_1\|_2^2 q n_1^*}{2(n_1^*+n_2)m}\epsilon^{\widetilde{C}}\Lambda_{max}^{q-1} \\
&\le \gamma(\boldsymbol{\mu}_1)_{j,r}^{(T_2,t)} + \frac{3\eta_2\|\boldsymbol{\mu}_1\|_2^2 q n_1^*}{(n_1^*+n_2)m^{1/q}}\epsilon^{\widetilde{C}}\log(1/\epsilon)^{1-1/q},
\end{aligned}
\tag{H.29}
$$

where the first inequality is by the upper bound of $-\ell_{1,i}^{'(T_2,t)}$, and the last inequality is by the induction. For any $0 \le \widetilde{T} \le T_2^+$, taking a telescoping sum over $t = 0, 1, \ldots, \widetilde{T}-1$ then gives

$$
\begin{aligned}
\gamma(\boldsymbol{\mu}_1)_{j,r}^{(T_2,\widetilde{T})} &\le \gamma(\boldsymbol{\mu}_1)_{j,r}^{(T_2,0)} + \sum_{t=0}^{\widetilde{T}-1} \frac{3\eta_2\|\boldsymbol{\mu}_1\|_2^2 q n_1^*}{(n_1^*+n_2)m^{1/q}}\epsilon^{\widetilde{C}}\log(1/\epsilon)^{1-1/q} \\
&\le \gamma(\boldsymbol{\mu}_1)_{j,r}^{(0)} + \widetilde{T}\frac{3\eta_2\|\boldsymbol{\mu}_1\|_2^2 q n_1^*}{(n_1^*+n_2)m^{1/q}}\epsilon^{\widetilde{C}}\log(1/\epsilon)^{1-1/q} \\
&\le T_2^+ \frac{3\eta_2\|\boldsymbol{\mu}_1\|_2^2 q n_1^*}{(n_1^*+n_2)m^{1/q}}\epsilon^{\widetilde{C}/2} \\
&\le m^{-\frac{1}{2q}},
\end{aligned}
$$

where the third inequality follows by the $\widetilde{T} \le T_2^+$ in our induction hypothesis and $\epsilon^{\widetilde{C}/2}\log(1/\epsilon)^{1-1/q} \le 1$ for $\frac{1}{\epsilon} \ge \exp\{\widetilde{C}e^{\widetilde{C}}\}$, and the last inequality is due to the definition of $T_2^+$.

Secondly, we are going to analyze update changes of $\gamma(\boldsymbol{\mu}_2)_{j,r}^{(T_2,t)}$.

**Lemma H.11.** *Under Condition 3.1, we suppose $\gamma(\boldsymbol{\mu}_2)_{j,r^*}^{(T_2,t)} \le \frac{m^{1/q}}{1-\lambda^2}$. Then there exists a time*

$$
T_3^+ = \frac{4(n_1^*+n_2)m^q \log(\lambda\frac{\|\boldsymbol{\mu}_2\|_2}{\|\boldsymbol{\mu}_1\|_2}m\Lambda_{max})}{q n_2 \eta_2 C_2'\|\boldsymbol{\mu}_2\|_2^2}
$$

*such that $\langle \mathbf{w}_{j,r}^{(T_2,t_{end})}, j\boldsymbol{\mu}_2\rangle = o(1)$ for all $r \in I_{j,1}$.*

*Proof of Lemma H.11.* When $t \le T^*$, we have

$$
\begin{aligned}
-\ell_{2,i}^{'(T_2,t)} &= 1/(1 + e^{\frac{1}{m}\sum_r(\sigma(\langle \mathbf{w}_{j,r}^{(T_2,t)}, y_{2,i}\boldsymbol{\mu}_2\rangle) + \sigma(\langle \mathbf{w}_{j,r}^{(T_2,t)}, \boldsymbol{\xi}_{2,i}\rangle)) - \frac{1}{m}\sum_r(\sigma(\langle \mathbf{w}_{-j,r}^{(T_2,t)}, -y_{2,i}\boldsymbol{\mu}_1\rangle) + \sigma(\langle \mathbf{w}_{-j,r}^{(T_2,t)}, \boldsymbol{\xi}_{2,i}\rangle))}) \\
&\ge 1/(1 + e^{\frac{1}{m}\sigma(\langle \mathbf{w}_{j,r^*}^{(T_2,t)}, j\boldsymbol{\mu}_2\rangle)})[1 + o(1)] \\
&\ge 1/(1 + e^{\frac{1}{m}\sigma((1-\lambda^2)\gamma(\boldsymbol{\mu}_2)_{j,r^*}^{(T_2,t)})})[1 + o(1)] \\
&\ge \frac{1}{2(1+e)} = C_2',
\end{aligned}
$$

where the second inequality is by Lemma H.4, and we denote $\frac{1}{2(1+e)}$ by $C_2'$. Within the $T_2^+$, $\gamma(\boldsymbol{\mu}_1)_{j,r}^{(T_2,t)}$ is no more than $m^{-\frac{1}{2q}}$, so we have

$$
\gamma(\boldsymbol{\mu}_2)_{j,r}^{(T_2,t+1)} = \gamma(\boldsymbol{\mu}_2)_{j,r}^{(T_2,t)} + \frac{\eta_2\|\boldsymbol{\mu}_2\|_2^2}{(n_1^*+n_2)m} \sum_{y_{2,i}=-j} (-\ell_{2,i}^{'(T_2,t)}) \cdot \sigma'(\lambda\frac{\|\boldsymbol{\mu}_2\|_2}{\|\boldsymbol{\mu}_1\|_2}(\Lambda_{max} + \gamma(\boldsymbol{\mu}_1)_{j,r}^{(T_2,t)}) - \gamma(\boldsymbol{\mu}_2)_{j,r}^{(T_2,t)})
$$

$$= \gamma(\boldsymbol{\mu}_2)_{j,r}^{(T_2,t)} + \frac{\eta_2}{(n_1^* + n_2)m} \sum_{y_{2,i}=-j} (-\ell_{2,i}'^{(T_2,t)}) \cdot \sigma'(\lambda \frac{\|\boldsymbol{\mu}_2\|_2}{\|\boldsymbol{\mu}_1\|_2} \Lambda_{max}[1 + o(1)] - \gamma(\boldsymbol{\mu}_2)_{j,r}^{(T_2,t)}) \cdot \|\boldsymbol{\mu}_2\|_2^2.$$

From this equation, we can see that $\gamma(\boldsymbol{\mu}_2)_{j,r}^{(T_2,t)}$ is an increasing sequence. In order to simplify the form, we order $\lambda \frac{\|\boldsymbol{\mu}_2\|_2}{\|\boldsymbol{\mu}_1\|_2} \Lambda_{max}[1 + o(1)] - \gamma(\boldsymbol{\mu}_2)_{j,r}^{(T_2,t)} = a_t$, so that

$$a_{t+1} = a_t - \frac{\eta_2}{(n_1^* + n_2)m} \sum_{y_{2,i}=-j} (-\ell_{2,i}'^{(T_2,t)}) \cdot \sigma'(a_t) \cdot \|\boldsymbol{\mu}_2\|_2^2.$$

Giving the specific conditions for Theorem 3.3 $n_1^* \geq n_2 \cdot \{2(-\cos\theta_{1,2})^q \|\boldsymbol{\mu}_2\|_2^q \|\boldsymbol{\mu}_1\|_2^{2-q} \eta_2 q m^{-2/q} [\log(\frac{1}{\epsilon})]^{1/q} - 1\}$, the inequality $\frac{(n_1^* + n_2)m}{\eta_2 q \|\boldsymbol{\mu}_2\|_2^2 n_2} \geq 2(\lambda \frac{\|\boldsymbol{\mu}_2\|_2}{\|\boldsymbol{\mu}_1\|_2} \Lambda_{max})^{q-2}$ holds, we have that

$$a_{t+1} = a_t - \frac{\eta_2}{(n_1^* + n_2)m} \sum_{y_{2,i}=-j} (-\ell_{2,i}'^{(t)}) \cdot \sigma'(a_t) \cdot \|\boldsymbol{\mu}_2\|_2^2$$

$$\geq a_t - \frac{\eta_2}{(n_1^* + n_2)m} \sum_{y_{2,i}=-j} \cdot \sigma'(a_t) \cdot \|\boldsymbol{\mu}_2\|_2^2$$

$$\geq a_t - \frac{\eta q n_2 \|\boldsymbol{\mu}_2\|_2^2}{(n_1^* + n_2)m} (a_t)^{q-1}$$

$$\geq 0,$$

where the last inequality is due to that $a_t \leq a_0 = \lambda \frac{\|\boldsymbol{\mu}_2\|_2}{\|\boldsymbol{\mu}_1\|_2} \Lambda_{max}[1 + o(1)]$. If $0 \leq a_t \leq \lambda \frac{\|\boldsymbol{\mu}_2\|_2}{\|\boldsymbol{\mu}_1\|_2} \Lambda_{max}$, we can have that $0 \leq a_{t+1} \leq \lambda \frac{\|\boldsymbol{\mu}_2\|_2}{\|\boldsymbol{\mu}_1\|_2} \Lambda_{max}$ for that $a_t$ is a decreasing series. Since $a_0 = \lambda \frac{\|\boldsymbol{\mu}_2\|_2}{\|\boldsymbol{\mu}_1\|_2} \Lambda_{max}[1 + o(1)]$, according to the iterative method, we can know that $a_t \geq 0$ for $t \geq 0$. Denoting by $\widetilde{T}$ the last time in $[0, T^*]$ that satisfying $a_t \geq \frac{1}{m}$, then for $0 \leq t \leq \widetilde{T}$, we have

$$a_{t+1} = a_t - \frac{\eta_2}{(n_1^* + n_2)m} \sum_{y_{2,i}=-j} (-\ell_{2,i}'^{(t)}) \cdot \sigma'(a_t) \cdot \|\boldsymbol{\mu}_2\|_2^2$$

$$\leq a_t - \frac{q \eta_2 n_2 C_2' \|\boldsymbol{\mu}_2\|_2^2}{4(n_1^* + n_2)m} (a_t)^{q-1}$$

$$\leq a_t (1 - \frac{q \eta_2 n_2 C_2' \|\boldsymbol{\mu}_2\|_2^2}{4(n_1^* + n_2)m^q}),$$

where the first inequality is by $(-\ell_{2,i}'^{(t)}) \geq C_2'$, and the last inequality is due to the assumption that $a_t \geq \frac{1}{m}$. Taking a telescoping multiplication over $t = 0, 1, ..., t-1$ for all $t \leq \widetilde{T}$, we obtain

$$a_t \leq a_0 (1 - \frac{q \eta_2 n_2 C_2' \|\boldsymbol{\mu}_2\|_2^2}{4(n_1^* + n_2)m^q})^t$$

$$\leq \lambda \frac{\|\boldsymbol{\mu}_2\|_2}{\|\boldsymbol{\mu}_1\|_2} \Lambda_{max} (1 - \frac{q \eta_2 n_2 C_2' \|\boldsymbol{\mu}_2\|_2^2}{4(n_1^* + n_2)m^q})^t,$$

where the last inequality is by $a_0 \leq \lambda \frac{\|\boldsymbol{\mu}_2\|_2}{\|\boldsymbol{\mu}_1\|_2} \Lambda_{max}$. So when $t = \left\lfloor \frac{\log(\lambda \frac{\|\boldsymbol{\mu}_2\|_2}{\|\boldsymbol{\mu}_1\|_2} m \Lambda_{max})}{\log(1/(1 - \frac{q \eta_2 n_2 C_2' \|\boldsymbol{\mu}_2\|_2^2}{4(n_1^* + n_2)m^q}))} \right\rfloor$, we have that $a_t \leq \frac{1}{m}$. We can verify that

$$T_3^+ = \frac{4(n_1^* + n_2)m^q \log(\lambda \frac{\|\boldsymbol{\mu}_2\|_2}{\|\boldsymbol{\mu}_1\|_2} m \Lambda_{max})}{q n_2 \eta_2 C_2' \|\boldsymbol{\mu}_2\|_2^2} \geq \left\lfloor \frac{\log(\lambda \frac{\|\boldsymbol{\mu}_2\|_2}{\|\boldsymbol{\mu}_1\|_2} m \Lambda_{max})}{\log(1/(1 - \frac{q \eta_2 n_2 C_2' \|\boldsymbol{\mu}_2\|_2^2}{4(n_1^* + n_2)m^q}))} \right\rfloor,$$

where the inequality is by $\log(\frac{1}{1-x}) \geq x$ for $x > 0$. By the definition of $\widetilde{T}$, we have $\widetilde{T} \leq T_3^+$. Now let us sort out the time relationship. By $\epsilon \leq \left\{ \frac{[\log(8m(n_1+n_2)/\delta)]^{(q-1)/2} \min\{\sigma_{p_1}^q, \sigma_{p_2}^q\} \|\boldsymbol{\mu}_2\|_2^q}{2^{4-2q} m^{1-\frac{1}{2q}} \|\boldsymbol{\mu}_1\|_2^2 \sigma_{p_2}^q \text{SNR}_2^q \sigma_0^{2-q}} \right\}^{\frac{2}{C}}$, we have $T_2^+ \geq T_1^+$. By $\sigma_0 \leq$

$O\left(\frac{n_2 m^{1-q} \max\{\sigma_{p_1}^{-q}, \sigma_{p_2}^{-q}\} d^{-q/2} \|\boldsymbol{\mu}_2\|_2^2}{2^{3q} [\sqrt{\log(8m(n_1+n_2)/\delta)}]^{q-1} \log(\lambda \frac{\|\boldsymbol{\mu}_2\|_2}{\|\boldsymbol{\mu}_1\|_2} m\Lambda_{max})}\right)^{\frac{1}{q-2}}$, we have $T_1^+ \geq T_3^+$. Here we would like to make it clear that these

two inequalities imply when $\lambda \frac{\|\boldsymbol{\mu}_2\|_2}{\|\boldsymbol{\mu}_1\|_2} \Lambda_{max}[1+o(1)] - \gamma(\boldsymbol{\mu}_2)_{j,r}^{(T_2,t)}$ down to $\frac{1}{m}$, $\max_{j,r,i} |\rho(\boldsymbol{\mu}_k)_{j,r,i_k}^{(T_2,t)}| \leq \sigma_0 \sigma_{p_k} \sqrt{d}$ and $\gamma(\boldsymbol{\mu}_1)_{j,r}^{(T_2,t)} \leq m^{-\frac{1}{2q}}$ still hold. Here we complete the proof. $\qquad\square$

Then we can calculate $\langle \mathbf{w}_{j,r^*}^{(T_2,t)}, j\boldsymbol{\mu}_1 \rangle$ and $\langle \mathbf{w}_{j,r^*}^{(T_2,t)}, j\boldsymbol{\mu}_2 \rangle$:

$$
\begin{aligned}
\langle \mathbf{w}_{j,r^*}^{(T_2,t)}, j\boldsymbol{\mu}_1 \rangle &= \langle \mathbf{w}_{j,r^*}^{(T_1,0)}, j\boldsymbol{\mu}_1 \rangle + \gamma(\boldsymbol{\mu}_1)_{j,r^*}^{(T_2,t)} - \lambda \frac{\|\boldsymbol{\mu}_1\|_2}{\|\boldsymbol{\mu}_2\|_2} \gamma(\boldsymbol{\mu}_2)_{j,r^*}^{(T_2,t_{end})} \\
&= \langle \mathbf{w}_{j,r^*}^{(T_1,0)}, j\boldsymbol{\mu}_1 \rangle + \Lambda_{max}[1+o(1)] - \lambda^2 \Lambda_{max}[1+o(1)] \\
&= (1-\lambda^2)\Lambda_{max}[1+o(1)], \\
\langle \mathbf{w}_{j,r^*}^{(T_2,t)}, j\boldsymbol{\mu}_2 \rangle &= \langle \mathbf{w}_{j,r^*}^{(T_1,0)}, j\boldsymbol{\mu}_2 \rangle + \gamma(\boldsymbol{\mu}_2)_{j,r^*}^{(T_2,t)} - \lambda \frac{\|\boldsymbol{\mu}_2\|_2}{\|\boldsymbol{\mu}_1\|_2} \gamma(\boldsymbol{\mu}_1)_{j,r^*}^{(T_2,t_{end})} \\
&\leq \frac{1}{m} + O(m^{-\frac{1}{2q}}) \\
&= o(1).
\end{aligned}
$$

Here the proof of Lemma H.10 is completed. $\qquad\square$

When $\gamma(\boldsymbol{\mu}_1)_{j,r}^{(T_2,t)} \leq m^{-\frac{1}{2q}}$ no longer holds, we still have

$$
\begin{aligned}
\langle \mathbf{w}_{j,r^*}^{(T_2,t)}, j\boldsymbol{\mu}_1 \rangle &= \langle \mathbf{w}_{j,r^*}^{(T_1,0)}, j\boldsymbol{\mu}_1 \rangle + \gamma(\boldsymbol{\mu}_1)_{j,r^*}^{(T_2,t)} - \lambda \frac{\|\boldsymbol{\mu}_1\|_2}{\|\boldsymbol{\mu}_2\|_2} \gamma(\boldsymbol{\mu}_2)_{j,r^*}^{(T_2,t_{end})} \\
&\geq (1-\lambda^2)\Lambda_{max}(1+\gamma(\boldsymbol{\mu}_1)_{j,r^*}^{(T_2,t)})[1+o(1)] \\
&\geq (1-\lambda^2)\Lambda_{max}[1+o(1)].
\end{aligned}
$$

From here, we can see that $\langle \mathbf{w}_{j,r^*}^{(T_2,t)}, j\boldsymbol{\mu}_1 \rangle$ will enter the re-ascending phase again. And we still have $\langle \mathbf{w}_{j,r^*}^{(T_2,t_{end})}, j\boldsymbol{\mu}_2 \rangle = o(1)$. As for the other neurons inside the set $I_{j,1}$, using the similar method in our previous section G, we can get $\langle \mathbf{w}_{j,r}^{(T_2,t_{end})}, j\boldsymbol{\mu}_1 \rangle = o(\langle \mathbf{w}_{j,r^*}^{(T_2,t_{end})}, j\boldsymbol{\mu}_1 \rangle)$ and $\langle \mathbf{w}_{j,r}^{(T_2,t_{end})}, j\boldsymbol{\mu}_2 \rangle = o(1)$ for $r \neq r^*$.

### H.3. The Second Type of Neuron

The second class of neurons is those belonging to the set $I_{j,2}$, which is expressed by the formula as follows:

$$
\langle \mathbf{w}_{j,r}^{(T_1,0)}, j\boldsymbol{\mu}_1 \rangle < 0, \langle \mathbf{w}_{j,r}^{(T_1,0)}, j\boldsymbol{\mu}_1^\perp \rangle > 0.
$$

By the previous analysis, we can get that

$$
\langle \mathbf{w}_{j,r}^{(T_2,0)}, j\boldsymbol{\mu}_1 \rangle = -C_1 < 0, \langle \mathbf{w}_{j,r}^{(T_2,0)}, j\boldsymbol{\mu}_1^\perp \rangle = C_2 > 0.
$$

In this case, the update method of the $\gamma(\boldsymbol{\mu}_1)_{j,r}^{(T_2,t)}, \gamma(\boldsymbol{\mu}_2)_{j,r}^{(T_2,t)}$ is as follows

$$
\gamma(\boldsymbol{\mu}_1)_{j,r}^{(T_2,t+1)} = \gamma(\boldsymbol{\mu}_1)_{j,r}^{(T_2,t)} + \frac{\eta_2}{(n_1^* + n_2)m} \sum_{y_{1,i}=-j} (-\ell_{1,i}'^{(T_2,t)}) \cdot \sigma'(C_1 - \gamma(\boldsymbol{\mu}_1)_{j,r}^{(T_2,t)} + \lambda \frac{\|\boldsymbol{\mu}_1\|_2}{\|\boldsymbol{\mu}_2\|_2} \gamma(\boldsymbol{\mu}_2)_{j,r}^{(T_2,t)}) \cdot \|\boldsymbol{\mu}_1\|_2^2,
$$

$$
\gamma(\boldsymbol{\mu}_2)_{j,r}^{(T_2,t+1)} = \gamma(\boldsymbol{\mu}_2)_{j,r}^{(T_2,t)} + \frac{\eta_2}{(n_1^* + n_2)m} \sum_{y_{2,i}=j} (-\ell_{2,i}'^{(T_2,t)}) \cdot \sigma'(C_2 + \gamma(\boldsymbol{\mu}_2)_{j,r}^{(T_2,t)} - \lambda \frac{\|\boldsymbol{\mu}_2\|_2}{\|\boldsymbol{\mu}_1\|_2} \gamma(\boldsymbol{\mu}_1)_{j,r}^{(T_2,t)}) \cdot \|\boldsymbol{\mu}_2\|_2^2.
$$

The following lemma shows that with a certain time, $\gamma(\boldsymbol{\mu}_1)_{j,r}^{(T_2,t)}$ changes little for $r \in I_{j,2}$. First of all, similar to the proof of Lemma H.11, we have $\gamma(\boldsymbol{\mu}_1)_{j,r}^{(T_2,t)} = \lambda \frac{\|\boldsymbol{\mu}_1\|_2}{\|\boldsymbol{\mu}_2\|_2} \gamma(\boldsymbol{\mu}_2)_{j,r}^{(T_2,t)} + \langle \mathbf{w}_{j,r}^{(T_2,0)}, j\boldsymbol{\mu}_1 \rangle$, plugging this into $\gamma(\boldsymbol{\mu}_2)_{j,r}^{(T_2,t)}$ update rule,

we can obtain

$$\gamma(\boldsymbol{\mu}_2)_{j,r}^{(T_2,t+1)} = \gamma(\boldsymbol{\mu}_2)_{j,r}^{(T_2,t)} - \frac{\eta_2\|\boldsymbol{\mu}_2\|_2^2}{(n_1^* + n_2)m} \sum_{y_{2,i}=j} \ell_{2,i}'^{(T_2,t)} \cdot \sigma'(\langle \mathbf{w}_{j,r}^{(T_1,0)}, j(\boldsymbol{\mu}_2 + \lambda\frac{\|\boldsymbol{\mu}_2\|_2}{\|\boldsymbol{\mu}_1\|_2}\boldsymbol{\mu}_1)\rangle) + (1-\lambda^2)\gamma(\boldsymbol{\mu}_2)_{j,r}^{(T_2,t)})$$

$$= \gamma(\boldsymbol{\mu}_2)_{j,r}^{(T_2,t)} + \frac{\eta_2}{(n_1^* + n_2)m} \sum_{y_{2,i}=j} (-\ell_{2,i}'^{(T_2,t)}) \cdot \sigma'(C_2 + (1-\lambda^2)\gamma(\boldsymbol{\mu}_2)_{j,r}^{(T_2,t)}) \cdot \|\boldsymbol{\mu}_2\|_2^2.$$

Then we can estimate $\langle \mathbf{w}_{j,r}^{(T_2,t)}, j\boldsymbol{\mu}_2 \rangle$ by $\gamma(\boldsymbol{\mu}_2)_{j,r}^{(T_2,t)}$, we have

$$\langle \mathbf{w}_{j,r^*}^{(T_2,t)}, j\boldsymbol{\mu}_2 \rangle = \langle \mathbf{w}_{j,r^*}^{(T_2,0)}, j\boldsymbol{\mu}_2 \rangle + \gamma(\boldsymbol{\mu}_2)_{j,r^*}^{(T_2,t)} - \lambda\frac{\|\boldsymbol{\mu}_2\|_2}{\|\boldsymbol{\mu}_1\|_2}\gamma(\boldsymbol{\mu}_1)_{j,r^*}^{(T_2,t)}$$

$$= (1-\lambda^2)\gamma(\boldsymbol{\mu}_2)_{j,r^*}^{(T_2,t)}[1 + o(1)],$$

where the equation is due to Lemma H.4 that $\gamma(\boldsymbol{\mu}_2)_{j,r^*}^{(T_2,t)} \to +\infty$ and $\langle \mathbf{w}_{j,r^*}^{(T_2,0)}, j\boldsymbol{\mu}_2 \rangle, \gamma(\boldsymbol{\mu}_1)_{j,r^*}^{(T_2,t)} \leq O(1)$. Then by Lemma H.4 we can get

$$-\ell_{2,i}'^{(T_2,t)} = 1/(1 + e^{\frac{1}{m}\sum_r(\sigma(\langle \mathbf{w}_{j,r}^{(T_2,t)}, y_{2,i}\boldsymbol{\mu}_2\rangle) + \sigma(\langle \mathbf{w}_{j,r}^{(T_2,t)}, \boldsymbol{\xi}_{2,i}\rangle)) - \frac{1}{m}\sum_r(\sigma(\langle \mathbf{w}_{-j,r}^{(T_2,t)}, -y_{2,i}\boldsymbol{\mu}_1\rangle) + \sigma(\langle \mathbf{w}_{-j,r}^{(T_2,t)}, \boldsymbol{\xi}_{2,i}\rangle))})$$

$$= 1/(1 + e^{\frac{1}{m}\sigma(\langle \mathbf{w}_{j,r^*}^{(T_2,t)}, j\boldsymbol{\mu}_2\rangle)})[1 + o(1)]$$

$$= 1/(1 + e^{\frac{1}{m}\sigma((1-\lambda^2)\gamma(\boldsymbol{\mu}_2)_{j,r^*}^{(T_2,t)})})[1 + o(1)].$$

**Lemma H.12.** *With probability at least $1 - \delta$,*

$$\sqrt{1-\lambda^2}\sigma_0\|\boldsymbol{\mu}_2\|_2^2 < \max_{r\in[m]} \langle \mathbf{w}_{j,r}^{(T_1,0)}, j(\boldsymbol{\mu}_2 + \lambda\frac{\|\boldsymbol{\mu}_2\|_2}{\|\boldsymbol{\mu}_1\|_2}\boldsymbol{\mu}_1)\rangle$$

*for all $j \in \{\pm1\}$.*

*Proof of Lemma H.12.* Similar to the previous proof of Lemma C.2, we have for each $r \in [m]$, $\langle \mathbf{w}_{j,r}^{(T_1,0)}, j(\boldsymbol{\mu}_2+\lambda\frac{\|\boldsymbol{\mu}_2\|_2}{\|\boldsymbol{\mu}_1\|_2}\boldsymbol{\mu}_1)\rangle$ is a Gaussian random variable with mean zero and variance $(1-\lambda^2)\sigma_0^2\|\boldsymbol{\mu}_2\|_2^2$. And, $\mathbb{P}(\sqrt{1-\lambda^2}\sigma_0\|\boldsymbol{\mu}_2\|_2 > \langle \mathbf{w}_{j,r}^{(T_1,0)}, j(\boldsymbol{\mu}_2+\lambda\frac{\|\boldsymbol{\mu}_2\|_2}{\|\boldsymbol{\mu}_1\|_2}\boldsymbol{\mu}_1)\rangle)$ is an absolute constant, and therefore by the condition on $m$, we have

$$\mathbb{P}(\sqrt{1-\lambda^2}\sigma_0\|\boldsymbol{\mu}_2\|_2 < \max_{r\in[m]} \langle \mathbf{w}_{j,r}^{(T_1,0)}, j(\boldsymbol{\mu}_2 + \lambda\frac{\|\boldsymbol{\mu}_2\|_2}{\|\boldsymbol{\mu}_1\|_2}\boldsymbol{\mu}_1)\rangle)$$

$$= 1 - \mathbb{P}(\sqrt{1-\lambda^2}\sigma_0\|\boldsymbol{\mu}_2\|_2 > \max_{r\in[m]} \langle \mathbf{w}_{j,r}^{(T_1,0)}, j(\boldsymbol{\mu}_2 + \lambda\frac{\|\boldsymbol{\mu}_2\|_2}{\|\boldsymbol{\mu}_1\|_2}\boldsymbol{\mu}_1)\rangle)$$

$$= 1 - \mathbb{P}(\sqrt{1-\lambda^2}\sigma_0\|\boldsymbol{\mu}_2\|_2 > \langle \mathbf{w}_{j,r}^{(T_1,0)}, j(\boldsymbol{\mu}_2 + \lambda\frac{\|\boldsymbol{\mu}_2\|_2}{\|\boldsymbol{\mu}_1\|_2}\boldsymbol{\mu}_1)\rangle)^m$$

$$\geq 1 - \delta,$$

here we completes the proof. $\square$

**Lemma H.13.** *Under Condition 3.1, if we choose*

$$n_2\mathrm{SNR}_2^q \geq C\frac{2^{3q+2}\min\{\sigma_{p_1}^q, \sigma_{p_2}^q\}}{(1-\lambda^2)^{q/2}\sigma_{p_2}^q} \log\left(\frac{2m^{\frac{1}{q}}}{\sqrt{1-\lambda^2}\sigma_0\|\boldsymbol{\mu}_2\|_2}\right)[\log(8m(n_1+n_2)/\delta)]^{(q-1)/2}, \tag{H.30}$$

*where $C = O(1)$ is a positive constant, there exists time*

$$T_4^+ = \frac{C(n_1^* + n_2)2^4 m \log\left(\frac{2m^{\frac{1}{q}}}{\sqrt{1-\lambda^2}\sigma_0\|\boldsymbol{\mu}_2\|_2}\right)}{\eta_2 q n_2 (1-\lambda^2)^{q/2}\sigma_0^{q-2}\|\boldsymbol{\mu}_2\|_2^q}$$

*such that*

- $\max_r \gamma(\boldsymbol{\mu}_2)_{j,r}^{(T_2,t)} \geq \frac{m^{\frac{1}{q}}}{1-\lambda^2}$ for $j \in \{\pm 1\}$.

- $\max_{j,r,i} |\rho(\boldsymbol{\mu}_k)_{j,r,i_k}^{(T_2,t)}| \leq \sigma_0 \sigma_{p_k} \sqrt{d}$ for all $j \in \{\pm 1\}$, $r \in [m]$, $i_k \in [n_k]$ and $0 \leq t \leq T_4^+$.

*Proof of Lemma H.13.* Denote by $\widetilde{T_5}^+$ the last time satisfying that $\gamma(\boldsymbol{\mu}_2)_{j,r^*}^{(T_2,t)} \leq \frac{m^{\frac{1}{q}}}{(1-\lambda^2)}$. Thus, we can have an upper bound for $-\ell_{2,i}^{'(T_2,t)}$:

$$-\ell_{2,i}^{'(T_2,t)} = 1/(1 + e^{\frac{1}{m}\sigma((1-\lambda^2)\gamma(\boldsymbol{\mu}_2)_{j,r^*}^{(T_2,t)})})[1 + o(1)] \geq \frac{1}{2(1 + e^{\frac{1}{m}\sigma((1-\lambda^2)\gamma(\boldsymbol{\mu}_2)_{j,r^*}^{(T_2,t)})})} \geq \frac{1}{2(1+e)} = C_2',$$

where the second inequality is by our hypothesis, and we denote $\frac{1}{2(1+e)}$ as $C_2'$. Thus we get $C_2'$ as the upper bound for $-\ell_{2,i}^{'(T_2,t)}$. Then we compute the growth of $\gamma(\boldsymbol{\mu}_2)_{j,r}^{(T_2,t)}$ as below:

$$\gamma(\boldsymbol{\mu}_2)_{j,r}^{(T_2,t+1)} = \gamma(\boldsymbol{\mu}_2)_{j,r}^{(T_2,t)} + \frac{\eta_2 \|\boldsymbol{\mu}_2\|_2^2}{(n_1^* + n_2)m} \sum_{y_i=j} (-\ell_{2,i}^{'(T_2,t)})\sigma'(C_2 + \gamma(\boldsymbol{\mu}_2)_{j,r}^{(T_2,t)} - \lambda\frac{\|\boldsymbol{\mu}_2\|_2}{\|\boldsymbol{\mu}_1\|_2}\gamma(\boldsymbol{\mu}_1)_{j,r}^{(T_2,t)})$$

$$= \gamma(\boldsymbol{\mu}_2)_{j,r}^{(T_2,t)} + \frac{\eta_2 \|\boldsymbol{\mu}_2\|_2^2}{(n_1^* + n_2)m} \sum_{y_i=j} (-\ell_{2,i}^{'(T_2,t)})\sigma'(C_2 + (1-\lambda^2)\gamma(\boldsymbol{\mu}_2)_{j,r}^{(T_2,t)}(1 + o(1)))$$

$$\geq \gamma(\boldsymbol{\mu}_2)_{j,r}^{(T_2,t)} + \frac{\eta_2 C_2' q\|\boldsymbol{\mu}_2\|_2^2 n_2}{8(n_1^* + n_2)m}(C_2 + (1-\lambda^2)\gamma(\boldsymbol{\mu}_2)_{j,r}^{(T_2,t)})^{q-1},$$

where the inequality is by (G.4) and $-\ell_{2,i}^{'(T_2,t)} \geq C_2'$. Let $A_j^{(t)} = \max_r\{C_2 + (1-\lambda^2)\gamma(\boldsymbol{\mu}_2)_{j,r}^{(T_2,t)}\}$, then we have that

$$A_j^{(t+1)} \geq A_j^{(t)} + \frac{\eta_2(1-\lambda^2)C_2' q\|\boldsymbol{\mu}_2\|_2^2 n_2}{8(n_1^* + n_2)m}(A_j^{(t)})^{q-1}$$

$$\geq A_j^{(t)}\left[1 + \frac{\eta_2(1-\lambda^2)C_2' q\|\boldsymbol{\mu}_2\|_2^2 n_2}{8(n_1^* + n_2)m}(A_j^{(t)})^{q-2}\right]$$

$$\geq A_j^{(t)}\left[1 + \frac{\eta_2(1-\lambda^2)C_2' q\|\boldsymbol{\mu}_2\|_2^2 n_2}{8(n_1^* + n_2)m}(A_j^{(0)})^{q-2}\right]$$

$$\geq A_j^{(t)}\left[1 + \frac{\eta_2(1-\lambda^2)C_2' q\|\boldsymbol{\mu}_2\|_2^2 n_2}{8(n_1^* + n_2)m}(\sqrt{1-\lambda^2}\sigma_0\|\boldsymbol{\mu}_2\|_2)^{q-2}\right],$$

where the third inequality is due to that $A_j^{(t)}$ is an increasing sequence, and the last inequality follows by Lemma H.12 and $A_j^{(0)} = \langle \mathbf{w}_{j,r}^{(T_1,0)}, j(\boldsymbol{\mu}_2 + \lambda\frac{\|\boldsymbol{\mu}_2\|_2}{\|\boldsymbol{\mu}_1\|_2}\boldsymbol{\mu}_1)\rangle$. Then we have

$$A_j^{(t)} \geq A_j^{(t-1)}\left[1 + \frac{\eta_2(1-\lambda^2)C_2' q\|\boldsymbol{\mu}_2\|_2^2 n_2}{8(n_1^* + n_2)m}(\sqrt{1-\lambda^2}\sigma_0\|\boldsymbol{\mu}_2\|_2)^{q-2}\right]$$

$$\geq A_j^{(0)}\left[1 + \frac{\eta_2(1-\lambda^2)C_2' q\|\boldsymbol{\mu}_2\|_2^2 n_2}{8(n_1^* + n_2)m}(\sqrt{1-\lambda^2}\sigma_0\|\boldsymbol{\mu}_2\|_2)^{q-2}\right]^t$$

$$\geq A_j^{(0)}\exp\left\{\frac{\eta(1-\lambda^2)C_2' q\|\boldsymbol{\mu}_2\|_2^2 n_2}{16(n_1^* + n_2)m}(\sqrt{1-\lambda^2}\sigma_0\|\boldsymbol{\mu}_2\|_2)^{q-2}t\right\}$$

$$\geq \exp\left\{\frac{\eta_2(1-\lambda^2)C_2' q\|\boldsymbol{\mu}_2\|_2^2 n_2}{16(n_1^* + n_2)m}(\sqrt{1-\lambda^2}\sigma_0\|\boldsymbol{\mu}_2\|_2)^{q-2}t\right\}\sqrt{1-\lambda^2}\sigma_0\|\boldsymbol{\mu}_2\|_2,$$

where the second inequality is due to the cumulative product of the series $A_j^{(t)}$, the third inequality is by $1 + x \geq e^{\frac{x}{2}}$ for $x \leq 2$ and our condition of $\eta_2$ and $\sigma_0$ listed in Condition 3.1, and the last inequality follows by Lemma H.12. Therefore $A_j^{(t)}$ will reach $2m^{\frac{1}{q}}$ within $T_4^+ = \frac{16(n_1^* + n_2)m \log\left(\frac{2m^{\frac{1}{q}}}{\sqrt{1-\lambda^2}\sigma_0\|\boldsymbol{\mu}_2\|_2}\right)}{\eta_2 C_2' q n_2(1-\lambda^2)^{q/2}\sigma_0^{q-2}\|\boldsymbol{\mu}_2\|_2^q}$ iterations. Therefore, we can get $\gamma(\boldsymbol{\mu}_2)_{j,r^*}^{(T_2,t)} = \frac{A_j - C}{1-\lambda^2} \geq$

$\frac{2m^{\frac{1}{q}}-C}{1-\lambda^2} \geq \frac{m^{\frac{1}{q}}}{1-\lambda^2}$. By (H.30), we can obtain that $T_4^+ \leq T_1^+$. By $\sigma_0 \leq O\left(\frac{m^{1-q}\|\boldsymbol{\mu}_2\|_2^{2-q}\log\left(\frac{2m^{\frac{1}{q}}}{\sqrt{1-\lambda^2}\sigma_0\|\boldsymbol{\mu}_2\|_2}\right)}{(1-\lambda^2)^{q/2}\log(\lambda\frac{\|\boldsymbol{\mu}_2\|_2}{\|\boldsymbol{\mu}_1\|_2}m\Lambda_{max})}\right)^{\frac{1}{q-2}}$, we have

$T_4^+ \geq T_3^+$, so Lemma H.11 hypothesis holds and $\max_{j,r,i}|\rho(\boldsymbol{\mu}_k)_{j,r,i_k}^{(T_2,t)}| \leq \sigma_0\sigma_{p_k}\sqrt{d}$ obviously holds. Here the proof completes. $\square$

**Lemma H.14.** *Under Condition 3.1, let* $T_5^+ = T_4^+ + \left\lceil \frac{6(n_1^*+n_2)(a^*-a_{T_2^+})^2}{\eta_2\|\boldsymbol{\mu}_2\|_2^2 n_2\epsilon^{k^q}} \right\rceil = T_4^+ + \widetilde{\Theta}(m^{\frac{2}{q}}\eta_2^{-1}\epsilon^{-k^q})$. *Then we have*

$\max_{j,r,i}|\rho(\boldsymbol{\mu}_k)_{j,r,i_k}^{(t)}| \leq 2\sigma_0\sigma_{p_k}\sqrt{d}$ *for all* $T_4^+ \leq t \leq T_5^+$. *And we can also find a time* $t^*$ *that* $\gamma(\boldsymbol{\mu}_2)_{j,r^*}^{(T_2,t^*)} \geq k\Lambda_{max}$, *where* $T_4^+ \leq t^* \leq T_5^+$.

*Proof of Lemma H.14.* When $\gamma(\boldsymbol{\mu}_2)_{j,r^*}^{(T_2,t)}$ reaches $\frac{m^{\frac{1}{q}}}{1-\lambda^2}$, we have

$$\gamma(\boldsymbol{\mu}_2)_{j,r^*}^{(T_2,t+1)} = \gamma(\boldsymbol{\mu}_2)_{j,r^*}^{(T_2,t)} + \frac{\eta_2\|\boldsymbol{\mu}_2\|_2^2}{(n_1^*+n_2)m}\sum_{y_i=j}(-\ell_{2,i}^{'(T_2,t)})\sigma'(C_2 + \gamma(\boldsymbol{\mu}_2)_{j,r^*}^{(T_2,t)} - \lambda\frac{\|\boldsymbol{\mu}_2\|_2}{\|\boldsymbol{\mu}_1\|_2}\gamma(\boldsymbol{\mu}_1)_{j,r}^{(T_2,t)})$$

$$= \gamma(\boldsymbol{\mu}_2)_{j,r^*}^{(T_2,t)} + \frac{q\eta_2\|\boldsymbol{\mu}_2\|_2^2 n_2}{2(n_1^*+n_2)m}\frac{((1-\lambda^2)\gamma(\boldsymbol{\mu}_2)_{j,r^*}^{(T_2,t)})^{q-1}}{1+e^{\frac{1}{m}((1-\lambda^2)\gamma(\boldsymbol{\mu}_2)_{j,r^*}^{(T_2,t)})^q}}[1+o(1)],$$

where the last equation is by (G.4). Let $a_t = (1-\lambda^2)\gamma(\boldsymbol{\mu}_2)_{j,r^*}^{(T_2,t)}$, and function $\widetilde{\ell}(x) = \log(1+\exp(-\frac{1}{m}x^q))$, then we can verify that

$$a_{t+1} = a_t - \frac{\eta_2\|\boldsymbol{\mu}_2\|_2^2 n_2}{2(n_1^*+n_2)}\widetilde{\ell}'(a_t).$$

Firstly, we will give an upper bound for $(\widetilde{\ell}'(x))^2$:

$$(\widetilde{\ell}'(x))^2 = \left(\frac{\frac{q}{m}x^{q-1}}{1+\exp(\frac{1}{m}x^q)}\right)^2 \leq \frac{\frac{q^2}{m^2}x^{2q-2}}{1+\exp(\frac{1}{m}x^q)}\widetilde{\ell}(x) \leq O(\widetilde{\ell}(x)), \tag{H.31}$$

where the second inequality is due to $\frac{1}{1+x} \leq \log(1+\frac{1}{x})$ for $x > 0$, and the last inequality is by $\frac{x^{2q-2}}{1+\exp(\frac{1}{m}x^q)} < +\infty$ for

$x > 0$. Denote $\max\left\{\left[m\log\frac{3}{\epsilon^{k^q}}\right]^{1/q}, 2k\Lambda_{max}\right\}$ by $a^*$, then we have

$$(a^*-a_t)^2 - (a^*-a_{t+1})^2 = (a_{t+1}-a_t)(2a^*-a_t-a_{t+1})$$

$$= -\frac{\eta_2\|\boldsymbol{\mu}_2\|_2^2 n_2}{2(n_1^*+n_2)}\widetilde{\ell}'(a_t)\left(2a^*-2a_t+\frac{\eta_2\|\boldsymbol{\mu}_2\|_2^2 n_2}{2(n_1^*+n_2)}\widetilde{\ell}'(a_t)\right)$$

$$= \frac{\eta_2\|\boldsymbol{\mu}_2\|_2^2 n_2}{(n_1^*+n_2)}\ell'(a_t)(a_t-a^*)[1+o(1)] - \left(\frac{\eta_2\|\boldsymbol{\mu}_2\|_2^2 n_2}{2(n_1^*+n_2)}\widetilde{\ell}'(a_t)[1+o(1)]\right)^2$$

$$\geq \frac{3\eta_2\|\boldsymbol{\mu}_2\|_2^2 n_2}{4(n_1^*+n_2)}\widetilde{\ell}(a_t) - \frac{3\eta_2\|\boldsymbol{\mu}_2\|_2^2 n_2}{4(n_1^*+n_2)}\widetilde{\ell}(a^*) - \left(\frac{\eta_2\|\boldsymbol{\mu}_2\|_2^2 n_2}{(n_1^*+n_2)}\right)^2 O(\widetilde{\ell}(a_t))$$

$$\geq \frac{\eta_2\|\boldsymbol{\mu}_2\|_2^2 n_2}{2(n_1^*+n_2)}\widetilde{\ell}(a_t) - \frac{3\eta_2\|\boldsymbol{\mu}_2\|_2^2 n_2}{4(n_1^*+n_2)}\widetilde{\ell}(a^*),$$

where the first inequality follows by the convexity of $\widetilde{\ell}(x)$ and (H.31), and the last inequality is by Condition 3.1 that $\eta_2 \leq \widetilde{O}(\min\{\|\boldsymbol{\mu}_2\|_2^{-2}, \sigma_{p_2}^{-2}d^{-1}\})$. Taking a summation over $t = T_4^+, T_4^++1, ..., T_5^+$, we have

$$\sum_{t=T_4^+}^{T_5^+}\widetilde{\ell}(a_t) \leq 2(T_5^+-T_4^++1)\widetilde{\ell}(a^*) + \frac{2(n_1^*+n_2)}{\eta_2\|\boldsymbol{\mu}_2\|_2^2 n_2}(a^*-a_{T_4^+})^2$$

$$\leq 2(T_5^+-T_4^++1)\widetilde{\ell}(a^*) + \frac{2(n_1^*+n_2)}{\eta_2\|\boldsymbol{\mu}_2\|_2^2 n_2}(a^*-a_{T52+})^2$$

$$\leq 2(T_5^+ - T_4^+ + 1)\widetilde{\ell}(a^*) + \frac{2(n_1^* + n_2)}{\eta_2\|\boldsymbol{\mu}_2\|_2^2 n_2}\widetilde{\Theta}(m^{\frac{2}{q}}),$$

where the last inequality is by $(a^* - a_{T_4^+})^2 = \widetilde{\Theta}(m^{\frac{2}{q}})$. Let $T_5^+ = T_4^+ + \left\lfloor \frac{6(n_1^* + n_2)(a^* - a_{T_2^+})^2}{\eta_2\|\boldsymbol{\mu}_2\|_2^2 n_2 \epsilon^{kq}} \right\rfloor = T_4^+ + \widetilde{\Theta}\left(\frac{(n_1^* + n_2)m^{\frac{2}{q}}}{\eta_2\|\boldsymbol{\mu}_2\|_2^2 n_2 \epsilon^{kq}}\right),$

then dividing $T_5^+ - T_4^+ + 1$ on both side of the last inequality gives us

$$\frac{1}{T_5^+ - T_4^+ + 1}\sum_{t=T_4^+}^{T_6^*}\widetilde{\ell}(a_t) \leq 2\widetilde{\ell}(a^*) + \frac{2(n_1^* + n_2)}{\eta_2\|\boldsymbol{\mu}_2\|_2^2 n_2(T_5^+ - T_4^+ + 1)}\widetilde{\Theta}(m^{\frac{2}{q}})$$

$$\leq \frac{2}{3}\epsilon^{kq} + \frac{1}{3}\epsilon^{kq}$$

$$\leq \epsilon^{kq},$$

where the second inequality is by $\log(1 + x) \leq x$ for $x > 0$ and the definition of $a^*$. Since the mean is less than $\epsilon^{kq}$, there must exist a $t^*$ $(T_4^+ \leq t^* \leq T_5^+)$ such that

$$\widetilde{\ell}(a_{t^*}) \leq \epsilon^{kq},$$

and we can obtain that $\gamma(\boldsymbol{\mu}_2)_{j,r^*}^{(T_2,t^*)} \geq \frac{k\Lambda_{max}}{1-\lambda^2}$. By the definition of $T_5^+$, we can obtain

$$\sum_{t=T_4^+}^{T_5^+}\widetilde{\ell}(a_t) \leq 2(T_5^+ - T_4^+ + 1)\widetilde{\ell}(a^*) + \frac{2(n_1^* + n_2)}{\eta_2\|\boldsymbol{\mu}_2\|_2^2 n_2}\widetilde{O}(m^{\frac{2}{q}}) = \widetilde{\Theta}\left(\frac{(n_1^* + n_2)m^{\frac{2}{q}}}{\eta_2\|\boldsymbol{\mu}_2\|_2^2 n_2}\right). \tag{H.32}$$

Letting $\Psi(\boldsymbol{\mu}_1)^{(t)} = \max_{j,r,i}|\rho(\boldsymbol{\mu}_1)_{j,r,i}^{(t)}|$, obviously we have $\Psi(\boldsymbol{\mu}_1)^{(t)} \leq \sigma_0\sigma_{p_1}\sqrt{d} \leq 2\sigma_0\sigma_{p_1}\sqrt{d}$. Now suppose that there exists $\widetilde{T} \in [T_4^+, T_5^+]$ such that $\Psi(\boldsymbol{\mu}_1)^{(t)} \leq 2\sigma_0\sigma_{p_1}\sqrt{d}$ for all $t \in [T_4^+, \widetilde{T} - 1]$. Then for $t \in [T_4^+, \widetilde{T} - 1]$, we have

$$\Psi(\boldsymbol{\mu}_1)^{(t+1)} \leq \Psi(\boldsymbol{\mu}_1)^{(t)} + \max\left\{\frac{\eta_2}{(n_1^* + n_2)m}(-\ell_{2,i}'^{(T_2,t)})\sigma'(\langle\mathbf{w}_{j,r}^{(T_2,t)}, \boldsymbol{\xi}_{2,i}\rangle)\|\boldsymbol{\xi}_{2,i}\|_2^2\right\}$$

$$\leq \Psi(\boldsymbol{\mu}_1)^{(t)} + \frac{3\eta_2 q\sigma_{p_1}^2 d}{2(n_1^* + n_2)m} \cdot \max_{j,r,i}\left\{(-\ell_{2,i}'^{(T_2,t)})\left(|\langle\mathbf{w}_{j,r}^{(T_2,0)}, \boldsymbol{\xi}_{1,i'}\rangle| + |\rho(\boldsymbol{\mu}_1)_{j,r,i}^{(T_2,t)}|\right.\right.$$

$$\left.\left.+ \sum_{i\neq i'}|\rho(\boldsymbol{\mu}_1)_{j,r,i}^{(T_2,t)}|\|\boldsymbol{\xi}_{1,i}\|_2^{-2}|\langle\boldsymbol{\xi}_{1,i}, \boldsymbol{\xi}_{1,i'}\rangle| + \sum_{i=1}^{n_2}|\rho(\boldsymbol{\mu}_2)_{j,r,i}^{(T_2,t)}|\|\boldsymbol{\xi}_{2,i}\|_2^{-2}|\langle\boldsymbol{\xi}_{2,i}, \boldsymbol{\xi}_{1,i'}\rangle|\right)^{q-1}\right\}$$

$$\leq \Psi(\boldsymbol{\mu}_1)^{(t)} + \frac{3\eta_2 q\sigma_{p_1}^2 d\epsilon^{\widetilde{C}}}{2(n_1^* + n_2)m}[4\sqrt{\log(8m\max\{n_1, n_2\}/\delta)} \cdot \sigma_0\sigma_{p_1}\sqrt{d}$$

$$+ \Psi(\boldsymbol{\mu}_1)^{(t)} + 4n_1^*\sqrt{\frac{\log(4n_1^2/\delta)}{d}}\Psi(\boldsymbol{\mu}_1)^{(t)} + 4n_2\frac{\sigma_{p_1}}{\sigma_{p_2}}\sqrt{\frac{\log(4n_2^2/\delta)}{d}}\Psi(\boldsymbol{\mu}_1)^{(t)}]^{q-1},$$

where the the second inequality is by Lemma C.1, and the last inequality follows by (H.26) and Lemma C.1. Taking a telescoping sum over $t = T_4^+, T_4^+ + 1, ..., \widetilde{T} - 1$, we have

$$\Psi(\boldsymbol{\mu}_1)^{(\widetilde{T})} \leq \Psi(\boldsymbol{\mu}_1)^{(T_4^+)} + \frac{\eta_2 q\epsilon^{\widetilde{C}}}{(n_1^* + n_2)m}\sum_{T_4^+}^{\widetilde{T}-1}\widetilde{O}(\sigma_{p_1}^2 d)(\sigma_0\sigma_{p_1}\sqrt{d})^{q-1}$$

$$\leq \Psi(\boldsymbol{\mu}_1)^{(T_4^+)} + \frac{\eta_2 q\epsilon^{\widetilde{C}}}{(n_1^* + n_2)m}(T_5^+ - T_4^+)\widetilde{O}(\sigma_{p_1}^2 d)(\sigma_0\sigma_{p_1}\sqrt{d})^{q-1}$$

$$\leq \sigma_0\sigma_{p_1}\sqrt{d} + \widetilde{O}\left\{\epsilon^{\widetilde{C}-kq}\frac{\sigma_{p_1}^2}{\sigma_{p_2}^2}m^{2/q-1}n_2^{-1}\mathrm{SNR}_2^{-2}(\sigma_0\sigma_{p_1}\sqrt{d})^{q-2}\right\}\sigma_0\sigma_{p_1}\sqrt{d}$$

$$\leq \sigma_0\sigma_{p_1}\sqrt{d} + \widetilde{O}\left\{\epsilon^{\widetilde{C}-kq}\frac{\sigma_{p_1}^2}{\sigma_{p_2}^2}m^{2/q-1}n_2^{2/q-1}(\sigma_0\sigma_{p_1}\sqrt{d})^{q-2}\right\}\sigma_0\sigma_{p_1}\sqrt{d}$$

$$\leq 2\sigma_0\sigma_{p_1}\sqrt{d},$$

where the first inequality follows by the assumption that $d \geq 1024 \log(4\max\{n_1^2, n_2^2\}/\delta)\alpha_2^2 \max\{n_1^2, n_2^2\} \max\{\frac{\sigma_{p_1}^2}{\sigma_{p_2}^2}, \frac{\sigma_{p_2}^2}{\sigma_{p_1}^2}\}$, the second inequality is by $\widetilde{T} \leq T_5^+$, the third inequality is due to Lemma H.8, the fourth inequality is due to $n_2\text{SNR}_2^q = \widetilde{\Omega}(1)$ and the last inequality is by Condition 3.1 on $\sigma_0$. Similarly we will use the same way to obtain that $\Psi(\boldsymbol{\mu}_2)^{(t)} \leq 2\sigma_0\sigma_{p_2}\sqrt{d}$ for all $T_4^+ \leq t \leq T_5^+$. Obviously we have $\Psi(\boldsymbol{\mu}_2)^{(T_4^+)} \leq \sigma_0\sigma_{p_2}\sqrt{d} \leq 2\sigma_0\sigma_{p_2}\sqrt{d}$. Now suppose that there exists $\widetilde{T} \in [T_4^+, T_5^+]$ such that $\Psi(\boldsymbol{\mu}_1)^{(t)} \leq 2\sigma_0\sigma_{p_1}\sqrt{d}$ for all $t \in [T_4^+, \widetilde{T} - 1]$. Then for $t \in [T_4^+, \widetilde{T} - 1]$, we have

$$\begin{aligned}
\Psi(\boldsymbol{\mu}_2)^{(t+1)} &\leq \Psi(\boldsymbol{\mu}_2)^{(t)} + \max_{j,r,i}\left\{\frac{\eta_2}{(n_1^* + n_2)m}(-\ell_{2,i}'^{(T_2,t)})\sigma'(\langle\mathbf{w}_{j,r}^{(T_2,t)}, \boldsymbol{\xi}_{2,i}\rangle)\|\boldsymbol{\xi}_{2,i}\|_2^2\right\} \\
&\leq \Psi(\boldsymbol{\mu}_2)^{(t)} + \max_{j,r,i}\left\{\frac{3\eta_2 q\sigma_{p_2}^2 d}{2(n_1^* + n_2)m}(-\ell_{2,i}'^{(T_2,t)})\left(|\langle\mathbf{w}_{j,r}^{(T_2,0)}, \boldsymbol{\xi}_{2,i}\rangle| + |\rho(\boldsymbol{\mu}_2)_{j,r,i}^{(T_2,t)}|\right.\right. \\
&\quad \left.\left. + \sum_{i'=1}^{n_1^*}|\rho(\boldsymbol{\mu}_1)_{j,r,i'}^{(T_2,t)}|\|\boldsymbol{\xi}_{1,i'}\|_2^{-2}|\langle\boldsymbol{\xi}_{1,i'}, \boldsymbol{\xi}_{2,i}\rangle| + \sum_{i'\neq i}|\rho(\boldsymbol{\mu}_2)_{j,r,i'}^{(T_2,t)}|\|\boldsymbol{\xi}_{2,i'}\|_2^{-2}|\langle\boldsymbol{\xi}_{2,i'}, \boldsymbol{\xi}_{2,i}\rangle|\right)^{q-1}\right\} \\
&\leq \Psi(\boldsymbol{\mu}_2)^{(t)} + \frac{3\eta_2 q\sigma_{p_2}^2 d}{2(n_1^* + n_2)m}\max_i\{-\ell_{2,i}'^{(T_2,t)}\}\left(4\sqrt{\log(8m\max\{n_1, n_2\}/\delta)} \cdot \sigma_0\sigma_{p_2}\sqrt{d}\right. \\
&\quad \left. + \sigma_0\sigma_{p_2}\sqrt{d} + 4n_1^*\frac{\sigma_{p_2}}{\sigma_{p_1}}\sqrt{\frac{\log(4n_1^2/\delta)}{d}}\Psi(\boldsymbol{\mu}_2)^{(t)} + 4n_2\sqrt{\frac{\log(4n_2^2/\delta)}{d}}\Psi(\boldsymbol{\mu}_2)^{(t)}\right)^{q-1},
\end{aligned}$$

where the the second inequality and the last inequality follows by (H.26) and Lemma C.1. Taking a telescoping sum over $t = T_4^+, T_4^+ + 1, ..., \widetilde{T} - 1$, we have

$$\begin{aligned}
\Psi(\boldsymbol{\mu}_2)^{(\widetilde{T})} &\leq \Psi(\boldsymbol{\mu}_2)^{(T_4^+)} + \frac{3\eta_2 q\sigma_{p_2}^2 d}{2(n_1^* + n_2)m}\widetilde{O}((\sigma_0\sigma_{p_2}\sqrt{d})^{q-1})\sum_{T_4^+}^{\widetilde{T}-1}\max_i\{-\ell_{2,i}'^{(T_2,t)}\} \\
&\leq \Psi(\boldsymbol{\mu}_2)^{(T_4^+)} + \frac{3\eta_2 q\sigma_{p_2}^2 d}{2(n_1^* + n_2)m}\widetilde{O}((\sigma_0\sigma_{p_2}\sqrt{d})^{q-1})\widetilde{O}\left(\frac{(n_1^* + n_2)m^{\frac{2}{q}}}{\eta_2\|\boldsymbol{\mu}_2\|_2^2 n_2}\right) \\
&\leq \sigma_0\sigma_{p_2}\sqrt{d} + \widetilde{O}\left\{m^{2/q-1}n_2^{-1}\text{SNR}_2^{-2}(\sigma_0\sigma_{p_2}\sqrt{d})^{q-2}\right\}\sigma_0\sigma_{p_2}\sqrt{d} \\
&\leq \sigma_0\sigma_{p_2}\sqrt{d} + \widetilde{O}\left\{m^{2/q-1}n_2^{2/q-1}(\sigma_0\sigma_{p_2}\sqrt{d})^{q-2}\right\}\sigma_0\sigma_{p_2}\sqrt{d} \\
&\leq 2\sigma_0\sigma_{p_2}\sqrt{d},
\end{aligned}$$

where the first inequality is by the assumption that $d \geq 1024 \log(4\max\{n_1^2, n_2^2\}/\delta)\alpha_2^2 \max\{n_1^2, n_2^2\} \max\{\frac{\sigma_{p_1}^2}{\sigma_{p_2}^2}, \frac{\sigma_{p_2}^2}{\sigma_{p_1}^2}\}$, the second inequality is due to (H.32), the third inequality is due to Lemma H.8, the fourth inequality is due to $n_2\text{SNR}_2^q = \widetilde{\Omega}(1)$ and the last inequality is by Condition 3.1 on $\sigma_0$. Therefore, $\Psi(\boldsymbol{\mu}_k)^{(t)} \leq \sigma_0\sigma_{p_k}\sqrt{d}$ for $k \in \{1, 2\}$ and $T_4^+ \leq t \leq T_5^+$, which completes the proof. $\qquad\square$

Moreover, we can bound $\langle\mathbf{w}_{j,r}^{(T_2,t)}, \boldsymbol{\xi}_{1,i}\rangle$ and $\langle\mathbf{w}_{j,r}^{(T_2,t)}, \boldsymbol{\xi}_{2,i}\rangle$ by $C_{\boldsymbol{\xi}}$. When $T_4^+ \leq t \leq T_5^+$, we have

$$\begin{aligned}
|\langle\mathbf{w}_{j,r}^{(T_2,t)}, \boldsymbol{\xi}_{1,i}\rangle| &\leq |\langle\mathbf{w}_{j,r}^{(T_2,0)}, \boldsymbol{\xi}_{1,i}\rangle| + |\rho(\boldsymbol{\mu}_1)_{j,r,i}^{(T_2,t)}| \\
&\quad + \sum_{i'\neq i}|\rho(\boldsymbol{\mu}_1)_{j,r,i'}^{(T_2,t)}|\|\boldsymbol{\xi}_{1,i'}\|_2^{-2}|\langle\boldsymbol{\xi}_{1,i'}, \boldsymbol{\xi}_{1,i}\rangle| + \sum_{i=1}^{n_2}|\rho(\boldsymbol{\mu}_2)_{j,r,i}^{(T_2,t)}|\|\boldsymbol{\xi}_{2,i}\|_2^{-2}|\langle\boldsymbol{\xi}_{2,i}, \boldsymbol{\xi}_{1,i}\rangle| \\
&\leq 4\sqrt{\log(8m\max\{n_1, n_2\}/\delta)} \cdot \sigma_0\sigma_{p_1}\sqrt{d} \\
&\quad + 2\sigma_0\sigma_{p_1}\sqrt{d} + 4n_1^*\sqrt{\frac{\log(4n_1^2/\delta)}{d}}2\sigma_0\sigma_{p_1}\sqrt{d} + 4n_2\frac{\sigma_{p_1}}{\sigma_{p_2}}\sqrt{\frac{\log(4n_2^2/\delta)}{d}}2\sigma_0\sigma_{p_2}\sqrt{d}
\end{aligned}$$

$$\leq 6\sqrt{\log\left(8m\max\{n_1, n_2\}/\delta\right)} \cdot \sigma_0\sigma_{p_1}\sqrt{d},$$

where the second inequality is by Lemma C.1, and the last inequality follows by $d \geq 1024\log(4\max\{n_1^2, n_2^2\}/\delta)\alpha_2^2\max\{n_1^2, n_2^2\}\max\{\frac{\sigma_{p_1}^2}{\sigma_{p_2}^2}, \frac{\sigma_{p_2}^2}{\sigma_{p_1}^2}\}$. Thus, when $T_4^+ \leq t \leq T_5^+$, we can bound the $\langle \mathbf{w}_{j,r}^{(T_2,t)}, \boldsymbol{\xi}_{1,i}\rangle$ by $C_{\boldsymbol{\xi}}$. Using the same method, we can also bound the $\langle \mathbf{w}_{j,r}^{(T_2,t)}, \boldsymbol{\xi}_{2,i}\rangle$ by $C_{\boldsymbol{\xi}}$.

## H.4. The Third Type of Neuron

The first class of neurons is those belonging to the set $I_{j,3}$, which is expressed by the formula as follows:

$$\langle \mathbf{w}_{j,r}^{(T_1,0)}, j\boldsymbol{\mu}_1\rangle < 0, \langle \mathbf{w}_{j,r}^{(T_1,0)}, j\boldsymbol{\mu}_1^\perp\rangle < 0$$

By the previous analysis, we can get that

$$\langle \mathbf{w}_{j,r}^{(T_2,0)}, j\boldsymbol{\mu}_1\rangle = -C_1 = -O(1) < 0, \langle \mathbf{w}_{j,r}^{(T_2,0)}, j\boldsymbol{\mu}_2\rangle = -C_2 = -O(1) < 0.$$

**Lemma H.15.** *For any $t \geq 0$, we have the following bounds*

$$0 \leq \gamma(\boldsymbol{\mu}_1)_{j,r}^{(T_2,t)} \leq \frac{\|\boldsymbol{\mu}_2\|_2 C_1 + \lambda\|\boldsymbol{\mu}_1\|_2 C_2}{(1-\lambda^2)\|\boldsymbol{\mu}_2\|_2},$$

$$0 \leq \gamma(\boldsymbol{\mu}_2)_{j,r}^{(T_2,t)} \leq \frac{\|\boldsymbol{\mu}_1\|_2 C_2 + \lambda\|\boldsymbol{\mu}_2\|_2 C_1}{(1-\lambda^2)\|\boldsymbol{\mu}_1\|_2}.$$

*We will prove this lemma by mathematical induction.*

*proof of Lemma H.15.* In this case, the update method of the $\gamma(\boldsymbol{\mu}_1)_{j,r}^{(T_2,t)}, \gamma(\boldsymbol{\mu}_2)_{j,r}^{(T_2,t)}$ is as follows

$$\gamma(\boldsymbol{\mu}_1)_{j,r}^{(T_2,t+1)} = \gamma(\boldsymbol{\mu}_1)_{j,r}^{(T_2,t)} + \frac{\eta_2}{(n_1^* + n_2)m}\sum_{y_{1,i}=-j}(-\ell_{1,i}'^{(T_2,t)})\cdot\sigma'(C_1 - \gamma(\boldsymbol{\mu}_1)_{j,r}^{(T_2,t)} + \lambda\frac{\|\boldsymbol{\mu}_1\|_2}{\|\boldsymbol{\mu}_2\|_2}\gamma(\boldsymbol{\mu}_2)_{j,r}^{(T_2,t)})\cdot\|\boldsymbol{\mu}_1\|_2^2,$$

$$\gamma(\boldsymbol{\mu}_2)_{j,r}^{(T_2,t+1)} = \gamma(\boldsymbol{\mu}_2)_{j,r}^{(T_2,t)} + \frac{\eta_2}{(n_1^* + n_2)m}\sum_{y_{2,i}=-j}(-\ell_{2,i}'^{(T_2,t)})\cdot\sigma'(C_2 - \gamma(\boldsymbol{\mu}_2)_{j,r}^{(T_2,t)} + \lambda\frac{\|\boldsymbol{\mu}_2\|_2}{\|\boldsymbol{\mu}_1\|_2}\gamma(\boldsymbol{\mu}_1)_{j,r}^{(T_2,t)})\cdot\|\boldsymbol{\mu}_2\|_2^2$$

When $t = 0$, the inequality clearly holds. Suppose that the inequality holds true when $0 \leq t \leq T - 1$, then we can get that

$$\gamma(\boldsymbol{\mu}_1)_{j,r}^{(T_2,t+1)} \leq \gamma(\boldsymbol{\mu}_1)_{j,r}^{(T_2,t)} + \frac{\eta_2\|\boldsymbol{\mu}_1\|_2^2}{(n_1^* + n_2)m}\sum_{y_{1,i}=-j}(-\ell_{1,i}'^{(T_2,t)})\cdot\sigma'(C_1 + \lambda\frac{\|\boldsymbol{\mu}_2\|_2 C_1 + \lambda\|\boldsymbol{\mu}_1\|_2 C_2}{(1-\lambda^2)\|\boldsymbol{\mu}_1\|_2} - \gamma(\boldsymbol{\mu}_1)_{j,r}^{(T_2,t)})$$

$$\leq \gamma(\boldsymbol{\mu}_1)_{j,r}^{(T_2,t)} + \frac{\eta_2\|\boldsymbol{\mu}_1\|_2^2}{(n_1^* + n_2)m}\sum_{y_{1,i}=-j}(-\ell_{1,i}'^{(T_2,t)})\cdot\sigma'(\frac{\|\boldsymbol{\mu}_2\|_2 C_1 + \lambda\|\boldsymbol{\mu}_1\|_2 C_2}{(1-\lambda^2)\|\boldsymbol{\mu}_2\|_2} - \gamma(\boldsymbol{\mu}_1)_{j,r}^{(T_2,t)})$$

$$\leq \gamma(\boldsymbol{\mu}_1)_{j,r}^{(T_2,t)} + \frac{\eta_2 q n_1^*\|\boldsymbol{\mu}_1\|_2^2}{(n_1^* + n_2)m}(\frac{\|\boldsymbol{\mu}_2\|_2 C_1 + \lambda\|\boldsymbol{\mu}_1\|_2 C_2}{(1-\lambda^2)\|\boldsymbol{\mu}_2\|_2} - \gamma(\boldsymbol{\mu}_1)_{j,r}^{(T_2,t)})^{q-1}, \tag{H.33}$$

where the first inequality follows by the induction hypothesis H.10, the second inequality is by the fact that $\sigma'(x)$ is a monotonically increasing function, and the last inequality is by the fact that $\sum_{y_{1,i}=-j} \leq n_1^*$ and $-\ell_{1,i}'^{(T_2,t)} \leq 1$. After the same derivation, we can get that

$$\gamma(\boldsymbol{\mu}_2)_{j,r}^{(T_2,t+1)} \leq \gamma(\boldsymbol{\mu}_2)_{j,r}^{(T_2,t)} + \frac{\eta_2 q n_2\|\boldsymbol{\mu}_2\|_2^2}{(n_1^* + n_2)m}(\frac{\|\boldsymbol{\mu}_1\|_2 C_2 + \lambda\|\boldsymbol{\mu}_2\|_2 C_1}{(1-\lambda^2)\|\boldsymbol{\mu}_1\|_2} - \gamma(\boldsymbol{\mu}_2)_{j,r}^{(T_2,t)})^{q-1}.$$

In order to prove the lemma H.10, we need more mathematical transformations. Let's make $a_t = \frac{\|\boldsymbol{\mu}_2\|_2 C_1 + \lambda\|\boldsymbol{\mu}_1\|_2 C_2}{(1-\lambda^2)\|\boldsymbol{\mu}_2\|_2} - \gamma(\boldsymbol{\mu}_1)_{j,r}^{(t)}$, then according to the inequality (H.33), we can have

$$a_{t+1} \geq a_t - \frac{\eta_2 q n_1^*\|\boldsymbol{\mu}_1\|_2^2}{(n_1^* + n_2)m}(a_t)^{q-1} \tag{H.34}$$

Then, we will use the inductive method to prove $a_t \geq 0$. This is known from the $\gamma(\boldsymbol{\mu}_1)_{j,r}^{(T_2,t)}$'s iterative equation, $\gamma(\boldsymbol{\mu}_1)_{j,r}^{(T_2,t)}$ is an increasing sequence, so $a_t \geq a_{t+1}$. When $0 \leq a_t \leq [\frac{(n_1^*+n_2)m}{\eta_2 qn_1^* \|\boldsymbol{\mu}_1\|_2^2}]^{q-1}$, it can be seen from the inequality H.34, we have that $a_{t+1} \geq 0$. And because $0 \leq a_0 = \frac{\|\boldsymbol{\mu}_1\|_2 C_2 + \lambda \|\boldsymbol{\mu}_2\|_2 C_1}{(1-\lambda^2)\|\boldsymbol{\mu}_1\|_2} \leq [\frac{(n_1^*+n_2)m}{\eta_2 qn_1^* \|\boldsymbol{\mu}_1\|_2^2}]^{q-1}$, which can be proved by Condition 3.1 on $\eta_2$. Then by induction, we can know that $a_{t+1} \geq 0$, for any $t \geq 0$. So it is obvious that $0 \leq \gamma(\boldsymbol{\mu}_1)_{j,r}^{(T_2,t)} \leq \frac{\|\boldsymbol{\mu}_2\|_2 C_1 + \lambda \|\boldsymbol{\mu}_1\|_2 C_2}{(1-\lambda^2)\|\boldsymbol{\mu}_2\|_2}$. In the same way, we can get $0 \leq \gamma(\boldsymbol{\mu}_2)_{j,r}^{(T_2,t)} \leq \frac{\|\boldsymbol{\mu}_1\|_2 C_2 + \lambda \|\boldsymbol{\mu}_2\|_2 C_1}{(1-\lambda^2)\|\boldsymbol{\mu}_1\|_2}$. $\qquad\square$

Taking it a step further, we can bound the inner products $\langle \mathbf{w}_{j,r}^{(T_2,t)}, j\boldsymbol{\mu}_1 \rangle$ and $\langle \mathbf{w}_{j,r}^{(T_2,t)}, j\boldsymbol{\mu}_2 \rangle$ as follows.

**Lemma H.16.** *for any $t \geq 0$,*

$$|\langle \mathbf{w}_{j,r}^{(T_2,t)}, j\boldsymbol{\mu}_1 \rangle| \leq O(1),$$
$$|\langle \mathbf{w}_{j,r}^{(T_2,t)}, j\boldsymbol{\mu}_2 \rangle| \leq O(1).$$

*proof of Lemma H.16.* Combining Lemma H.16 and the two equations (H.12) and (H.13), we are able to derive that

$$
\begin{aligned}
\langle \mathbf{w}_{j,r}^{(T_2,t)}, j\boldsymbol{\mu}_1 \rangle &= \langle \mathbf{w}_{j,r}^{(T_2,0)}, j\boldsymbol{\mu}_1 \rangle + \gamma(\boldsymbol{\mu}_1)_{j,r}^{(T_2,t)} - \lambda \frac{\|\boldsymbol{\mu}_1\|_2}{\|\boldsymbol{\mu}_2\|_2} \gamma(\boldsymbol{\mu}_2)_{j,r}^{(T_2,t)} \\
&\leq -C_1 + \gamma(\boldsymbol{\mu}_1)_{j,r}^{(T_2,t)} \\
&\leq -C_1 + \frac{\|\boldsymbol{\mu}_2\|_2 C_1 + \lambda \|\boldsymbol{\mu}_1\|_2 C_2}{(1-\lambda^2)\|\boldsymbol{\mu}_2\|_2} \\
&= O(1),
\end{aligned}
\tag{H.35}
$$

where the first inequality is by the fact that $\gamma(\boldsymbol{\mu}_2)_{j,r}^{(T_2,t)} \geq 0$ and the second is by known result that $\gamma(\boldsymbol{\mu}_1)_{j,r}^{(T_2,t)} \leq \frac{\|\boldsymbol{\mu}_2\|_2 C_1 + \lambda \|\boldsymbol{\mu}_1\|_2 C_2}{(1-\lambda^2)\|\boldsymbol{\mu}_2\|_2}$. Further, we can get that

$$
\begin{aligned}
\langle \mathbf{w}_{j,r}^{(T_2,t)}, j\boldsymbol{\mu}_1 \rangle &= \langle \mathbf{w}_{j,r}^{(T_2,0)}, j\boldsymbol{\mu}_1 \rangle + \gamma(\boldsymbol{\mu}_1)_{j,r}^{(T_2,t)} - \lambda \frac{\|\boldsymbol{\mu}_1\|_2}{\|\boldsymbol{\mu}_2\|_2} \gamma(\boldsymbol{\mu}_2)_{j,r}^{(T_2,t)} \\
&\geq -C_1 - \lambda \frac{\|\boldsymbol{\mu}_1\|_2}{\|\boldsymbol{\mu}_2\|_2} \gamma(\boldsymbol{\mu}_2)_{j,r}^{(T_2,t)} \\
&\geq -C_1 - \lambda \frac{\|\boldsymbol{\mu}_1\|_2}{\|\boldsymbol{\mu}_2\|_2} \frac{\|\boldsymbol{\mu}_1\|_2 C_2 + \lambda \|\boldsymbol{\mu}_2\|_2 C_1}{(1-\lambda^2)\|\boldsymbol{\mu}_1\|_2} \\
&= -O(1),
\end{aligned}
\tag{H.36}
$$

where the first inequality is by the fact that $\gamma(\boldsymbol{\mu}_1)_{j,r}^{(T_2,t)} \geq 0$ and the second is by known result that $\gamma(\boldsymbol{\mu}_2)_{j,r}^{(T_2,t)} \leq \frac{\|\boldsymbol{\mu}_1\|_2 C_2 + \lambda \|\boldsymbol{\mu}_2\|_2 C_1}{(1-\lambda^2)\|\boldsymbol{\mu}_1\|_2}$. The proof for $\langle \mathbf{w}_{j,r}^{(T_2,t)}, j\boldsymbol{\mu}_2 \rangle$ is similar, we can prove that $-O(1) \leq \langle \mathbf{w}_{j,r}^{(T_2,t)}, j\boldsymbol{\mu}_2 \rangle$ and $\langle \mathbf{w}_{j,r}^{(T_2,t)}, j\boldsymbol{\mu}_2 \rangle \leq O(1)$, which completes the proof. $\qquad\square$

We denote the time to stop training for task $T_2$ by $t_{end}$, which represents the first time $\gamma(\boldsymbol{\mu}_2)_{j,r^*}^{(T_2,t)}$ reaches $k\Lambda_{max}$, and we know $T_4^+ \leq t_{end} \leq T_5^+$.

For $I_{j,1}$, we can calculate the following summation of the neuron output:

$$\sum_{r \in I_{j,1}} \sigma(\langle \mathbf{w}_{j,r}^{(T_2,t_{end})}, j\boldsymbol{\mu}_1 \rangle) = \sigma(\langle \mathbf{w}_{j,r^*}^{(T_2,t_{end})}, j\boldsymbol{\mu}_1 \rangle) + \sum_{r \neq r^*} \sigma(\langle \mathbf{w}_{j,r}^{(T_2,t_{end})}, j\boldsymbol{\mu}_1 \rangle) = [(1-\lambda^2)\Lambda_{max}]^q [1 + o(1)],$$

and

$$\sum_{r \in I_{j,1}} \sigma(\langle \mathbf{w}_{j,r}^{(T_2,t_{end})}, j\boldsymbol{\mu}_2 \rangle) = \sum_{r \in I_{j,1}} \sigma(o(1)) = o(m).$$

We also have

$$\sum_{r \in I_{j,1}} \sigma(\langle \mathbf{w}_{j,r}^{(T_2,t_{end})}, -j\boldsymbol{\mu}_1 \rangle) = o(m),$$

and

$$\sum_{r \in I_{j,1}} \sigma(\langle \mathbf{w}_{j,r}^{(T_2,t_{end})}, -j\boldsymbol{\mu}_2 \rangle) = \sum_{r \in I_{j,1}} \sigma(o(1)) = o(m).$$

Now we obtain the neurons in the set $I_{j,1}$ output values in task $T_1$ and $T_2$.

For $I_{j,2}$, we can calculate the following summation of the neuron output:

$$\sum_{r \in I_{j,2}} \sigma(\langle \mathbf{w}_{j,r}^{(T_2,t_{end})}, j\boldsymbol{\mu}_1 \rangle), \sum_{r \in I_{j,2}} \sigma(\langle \mathbf{w}_{j,r}^{(T_2,t_{end})}, -j\boldsymbol{\mu}_1 \rangle), \sum_{r \in I_{j,2}} \sigma(\langle \mathbf{w}_{j,r}^{(T_2,t_{end})}, -j\boldsymbol{\mu}_2 \rangle) = o(m),$$

and

$$\sum_{r \in I_{j,2}} \sigma(\langle \mathbf{w}_{j,r}^{(T_2,t_{end})}, j\boldsymbol{\mu}_2 \rangle) = \sigma(\langle \mathbf{w}_{j,r^*}^{(T_2,t_{end})}, j\boldsymbol{\mu}_2 \rangle) + \sum_{r \neq r^*} \sigma(\langle \mathbf{w}_{j,r}^{(T_2,t_{end})}, j\boldsymbol{\mu}_2 \rangle) = [k\Lambda_{max}]^q [1 + o(1)].$$

Now we obtain the neurons in the set $I_{j,2}$ output values in task $T_1$ and $T_2$.

For $I_{j,3}$, we can calculate the following summation of the neuron output:

$$\sum_{r \in I_{j,3}} \sigma(\langle \mathbf{w}_{j,r}^{(T_2,t_{end})}, j\boldsymbol{\mu}_1 \rangle) = \sum_{r \in I_{j,3}} \sigma(O(1)) = O(m),$$

and

$$\sum_{r \in I_{j,3}} \sigma(\langle \mathbf{w}_{j,r}^{(T_2,t_{end})}, j\boldsymbol{\mu}_2 \rangle) = \sum_{r \in I_{j,3}} \sigma(O(1)) = O(m).$$

We also have

$$\sum_{r \in I_{j,3}} \sigma(\langle \mathbf{w}_{j,r}^{(T_2,t_{end})}, -j\boldsymbol{\mu}_1 \rangle) = \sum_{r \in I_{j,3}} \sigma(O(1)) = O(m),$$

and

$$\sum_{r \in I_{j,3}} \sigma(\langle \mathbf{w}_{j,r}^{(T_2,t_{end})}, -j\boldsymbol{\mu}_2 \rangle) = \sum_{r \in I_{j,3}} \sigma(O(1)) = O(m).$$

Now we obtain the neurons in the set $I_{j,3}$ output values in task $T_1$ and $T_2$.

### H.5. Loss Analysis

In the previous section, we divided the neurons into three sets: $I_{j,1}, I_{j,2}$ and $I_{j,3}$. For the set $I_{j,1}$, we conclude that

$$\sum_{r \in I_{j,1}} \sigma(\langle \mathbf{w}_{j,r}^{(T_2,t_{end})}, j\boldsymbol{\mu}_1 \rangle) = [(1 - \lambda^2)\Lambda_{max}]^q [1 + o(1)], \sum_{r \in I_{j,1}} \sigma(\langle \mathbf{w}_{j,r}^{(T_2,t_{end})}, j\boldsymbol{\mu}_2 \rangle) = o(m),$$

$$\sum_{r \in I_{j,1}} \sigma(\langle \mathbf{w}_{j,r}^{(T_2,t_{end})}, -j\boldsymbol{\mu}_1 \rangle) = o(m), \sum_{r \in I_{j,1}} \sigma(\langle \mathbf{w}_{j,r}^{(T_2,t_{end})}, -j\boldsymbol{\mu}_2 \rangle) = o(m),$$

for $j \in \pm 1$.

For the set $I_{j,2}$, we conclude that

$$\sum_{r \in I_{j,2}} \sigma(\langle \mathbf{w}_{j,r}^{(T_2,t_{end})}, j\boldsymbol{\mu}_1 \rangle) = o(m), \ \sum_{r \in I_{j,2}} \sigma(\langle \mathbf{w}_{j,r}^{(T_2,t_{end})}, j\boldsymbol{\mu}_2 \rangle) = [k\Lambda_{max}]^q[1 + o(1)],$$

$$\sum_{r \in I_{j,2}} \sigma(\langle \mathbf{w}_{j,r}^{(T_2,t_{end})}, -j\boldsymbol{\mu}_1 \rangle) = o(m), \ \sum_{r \in I_{j,2}} \sigma(\langle \mathbf{w}_{j,r}^{(T_2,t_{end})}, -j\boldsymbol{\mu}_2 \rangle) = o(m),$$

for $j \in \pm 1$.

For the set $I_{j,3}$, we conclude that

$$\sum_{r \in I_{j,3}} \sigma(\langle \mathbf{w}_{j,r}^{(T_2,t_{end})}, j\boldsymbol{\mu}_1 \rangle) = O(m), \ \sum_{r \in I_{j,3}} \sigma(\langle \mathbf{w}_{j,r}^{(T_2,t_{end})}, j\boldsymbol{\mu}_2 \rangle) = O(m),$$

$$\sum_{r \in I_{j,3}} \sigma(\langle \mathbf{w}_{j,r}^{(T_2,t_{end})}, -j\boldsymbol{\mu}_1 \rangle) = O(m), \ \sum_{r \in I_{j,3}} \sigma(\langle \mathbf{w}_{j,r}^{(T_2,t_{end})}, -j\boldsymbol{\mu}_2 \rangle) = O(m),$$

for $j \in \pm 1$.

Then we can calculate the $\sum_{r=1}^{m} \sigma(\langle \mathbf{w}_{j,r}^{(T_2,t_{end})}, j \cdot \boldsymbol{\mu}_1 \rangle), \sum_{r=1}^{m} \sigma(\langle \mathbf{w}_{-j,r}^{(T_2,t_{end})}, j \cdot \boldsymbol{\mu}_1 \rangle), \sum_{r=1}^{m} \sigma(\langle \mathbf{w}_{j,r}^{(T_2,t_{end})}, j \cdot \boldsymbol{\mu}_2 \rangle)$ and $\sum_{r=1}^{m} \sigma(\langle \mathbf{w}_{-j,r}^{(T_2,t_{end})}, j \cdot \boldsymbol{\mu}_2 \rangle)$:

$$\sum_{r=1}^{m} \sigma(\langle \mathbf{w}_{j,r}^{(T_2,t_{end})}, j \cdot \boldsymbol{\mu}_1 \rangle)$$

$$= \sum_{r \in I_{j,1}} \sigma(\langle \mathbf{w}_{j,r}^{(T_2,t_{end})}, j \cdot \boldsymbol{\mu}_1 \rangle) + \sum_{r \in I_{j,2}} \sigma(\langle \mathbf{w}_{j,r}^{(T_2,t_{end})}, j \cdot \boldsymbol{\mu}_1 \rangle) + \sum_{r \in I_{j,3}} \sigma(\langle \mathbf{w}_{j,r}^{(T_2,t_{end})}, j \cdot \boldsymbol{\mu}_1 \rangle)$$

$$= [(1 - \lambda^2)\Lambda_{max}]^q[1 + o(1)] + O(m) + O(m)$$

$$= [(1 - \lambda^2)\Lambda_{max}]^q[1 + o(1)],$$

$$\sum_{r=1}^{m} \sigma(\langle \mathbf{w}_{-j,r}^{(T_2,t_{end})}, j \cdot \boldsymbol{\mu}_1 \rangle)$$

$$= \sum_{r \in I_{-j,1}} \sigma(\langle \mathbf{w}_{-j,r}^{(T_2,t_{end})}, j \cdot \boldsymbol{\mu}_1 \rangle) + \sum_{r \in I_{-j,2}} \sigma(\langle \mathbf{w}_{-j,r}^{(T_2,t_{end})}, j \cdot \boldsymbol{\mu}_1 \rangle) + \sum_{r \in I_{-j,3}} \sigma(\langle \mathbf{w}_{-j,r}^{(T_2,t_{end})}, j \cdot \boldsymbol{\mu}_1 \rangle)$$

$$= o(m) + o(m) + O(m)$$

$$= O(m),$$

$$\sum_{r=1}^{m} \sigma(\langle \mathbf{w}_{j,r}^{(T_2,t_{end})}, j \cdot \boldsymbol{\mu}_2 \rangle)$$

$$= \sum_{r \in I_{j,1}} \sigma(\langle \mathbf{w}_{j,r}^{(T_2,t_{end})}, j \cdot \boldsymbol{\mu}_2 \rangle) + \sum_{r \in I_{j,2}} \sigma(\langle \mathbf{w}_{j,r}^{(T_2,t_{end})}, j \cdot \boldsymbol{\mu}_2 \rangle) + \sum_{r \in I_{j,3}} \sigma(\langle \mathbf{w}_{j,r}^{(T_2,t_{end})}, j \cdot \boldsymbol{\mu}_2 \rangle)$$

$$= o(m) + [k\Lambda_{max}]^q[1 + o(1)] + O(m)$$

$$= [k\Lambda_{max}]^q[1 + o(1)],$$

$$\sum_{r=1}^{m} \sigma(\langle \mathbf{w}_{-j,r}^{(T_2,t_{end})}, j \cdot \boldsymbol{\mu}_2 \rangle)$$

$$= \sum_{r \in I_{-j,1}} \sigma(\langle \mathbf{w}_{-j,r}^{(T_2,t_{end})}, j \cdot \boldsymbol{\mu}_2 \rangle) + \sum_{r \in I_{-j,2}} \sigma(\langle \mathbf{w}_{-j,r}^{(T_2,t_{end})}, j \cdot \boldsymbol{\mu}_2 \rangle) + \sum_{r \in I_{-j,3}} \sigma(\langle \mathbf{w}_{-j,r}^{(T_2,t_{end})}, j \cdot \boldsymbol{\mu}_2 \rangle)$$

$$= o(m) + o(m) + O(m)$$
$$= O(m),$$

As for task $T_1$, note that $L_{S_1}(\mathbf{W}^{(T_2,t_{end})}) = \frac{1}{n_1}\sum_{i=1}^{n_1}\ell[y_{1,i} \cdot f(\mathbf{W}^{(T_2,t_{end})}, \mathbf{x}_{1,i})]$, when $y_{1,i} = j$, we have

$$
\begin{aligned}
&y_{1,i} \cdot f(\mathbf{W}^{(T_2,t_{end})}, \mathbf{x}_{1,i}) \\
&= F_j(\mathbf{W}_j^{(T_2,t_{end})}, \mathbf{x}_{1,i}) - F_{-j}(\mathbf{W}_{-j}^{(T_2,t_{end})}, \mathbf{x}_{1,i}) \\
&= \frac{1}{m}\sum_{r=1}^{m}\left[\sigma(\langle \mathbf{w}_{j,r}^{(T_2,t_{end})}, j \cdot \boldsymbol{\mu}_1 \rangle) + \sigma(\langle \mathbf{w}_{j,r}^{(T_2,t_{end})}, \boldsymbol{\xi}_{1,i} \rangle)\right] - \frac{1}{m}\sum_{r=1}^{m}\left[\sigma(\langle \mathbf{w}_{-j,r}^{(T_2,t_{end})}, j \cdot \boldsymbol{\mu}_1 \rangle) + \sigma(\langle \mathbf{w}_{-j,r}^{(T_2,t_{end})}, \boldsymbol{\xi}_{1,i} \rangle)\right] \\
&\geq \frac{1}{m}[(1-\lambda^2)\Lambda_{max}]^q[1+o(1)] - 2C_{\boldsymbol{\xi}}^q,
\end{aligned}
$$

where the last inequality is by $\langle \mathbf{w}_{-j,r}^{(T_2,t_{end})}, \boldsymbol{\xi}_{1,i} \rangle \leq C_{\boldsymbol{\xi}}$ for $j \in \{\pm 1\}$. As for task $T_2$, note that $L_{S_2}(\mathbf{W}^{(T_2,t_{end})}) = \frac{1}{n_2}\sum_{i=1}^{n_2}\ell[y_{2,i} \cdot f(\mathbf{W}^{(T_2,t_{end})}, \mathbf{x}_{2,i})]$, when $y_{2,i} = j$, we can also get

$$
\begin{aligned}
&y_{2,i} \cdot f(\mathbf{W}^{(T_2,t_{end})}, \mathbf{x}_{2,i}) \\
&= F_j(\mathbf{W}_j^{(T_2,t_{end})}, \mathbf{x}_{2,i}) - F_{-j}(\mathbf{W}_{-j}^{(T_2,t_{end})}, \mathbf{x}_{2,i}) \\
&= \frac{1}{m}\sum_{r=1}^{m}\left[\sigma(\langle \mathbf{w}_{j,r}^{(T_2,t_{end})}, j \cdot \boldsymbol{\mu}_2 \rangle) + \sigma(\langle \mathbf{w}_{j,r}, \boldsymbol{\xi}_{2,i} \rangle)\right] - \frac{1}{m}\sum_{r=1}^{m}\left[\sigma(\langle \mathbf{w}_{-j,r}^{(T_2,t_{end})}, j \cdot \boldsymbol{\mu}_2 \rangle) + \sigma(\langle \mathbf{w}_{-j,r}, \boldsymbol{\xi}_{2,i} \rangle)\right] \\
&\geq \frac{1}{m}[k\Lambda_{max}]^q[1+o(1)] - 2C_{\boldsymbol{\xi}}^q,
\end{aligned}
$$

leading to

$$
\begin{aligned}
L_{S_2}(\mathbf{W}^{(T_2,t_{end})}) &= \frac{1}{n_2}\sum_{i=1}^{n_2}\ell[y_{2,i} \cdot f(\mathbf{W}^{(T_2,t_{end})}, \mathbf{x}_{2,i})] \\
&\leq \ell(\frac{1}{m}[k\Lambda_{max}]^q[1+o(1)] - 2C_{\boldsymbol{\xi}}^q) \\
&\leq \ell(\frac{1}{2m}[k\Lambda_{max}]^q) = \log(1 + \epsilon^{k^q/2}) \\
&\leq \epsilon^{k^q/2},
\end{aligned}
\tag{H.37}
$$

where the third inequality is due to $\epsilon \leq e^{-8C_{\boldsymbol{\xi}}^q}$, and the last inequality follows by $\log(1+x) \leq x$ where $x \geq 0$. Applying a proof technique similar to subsection D.3 in Cao et al. (2022), we can obtain $L_{D_2}(\mathbf{W}^{(T_2,t_{end})}) \leq 6\epsilon^{k^q/2} + \exp(-n_2^2)$.

Similar to Lemma F.19, we have that

**Lemma H.17.** *Under the same conditions as Theorem 3.3, when $-\frac{1+C_2}{2} \leq \cos\theta_{1,2} \leq 0$, we have that*

$$
\sum_{r=1}^{m}\left[\sigma(\langle \mathbf{w}_{y_1,r}^{(T_2,t_{end})}, y_1\boldsymbol{\mu}_1 \rangle) - \sigma(\langle \mathbf{w}_{-y_1,r}^{(T_2,t_{end})}, y_1\boldsymbol{\mu}_1 \rangle)\right] \geq C_3,
$$

*where $C_1$ and $C_3$ are the same constants as that in the Lemma F.19.*

*Proof of Lemma H.17.* we have that

$$
\sum_{r=1}^{m}\left[\sigma(\langle \mathbf{w}_{y_1,r}^{(T_2,t_{end})}, y_1\boldsymbol{\mu}_1 \rangle) - \sigma(\langle \mathbf{w}_{-y_1,r}^{(T_2,t_{end})}, y_1\boldsymbol{\mu}_1 \rangle)\right] = \frac{1}{m}[(1-\lambda^2)\Lambda_{max}]^q[1+o(1)].
$$

So when $-\frac{1+C_2}{2} \leq \cos\theta_{1,2} \leq 0$, that is $0 \leq \lambda \leq \frac{1+C_2}{2}$, we have

$$
\frac{1}{m}[(1-\lambda^2)\Lambda_{max}]^q[1+o(1)]
$$

$$\geq \frac{1}{m}\Lambda_{max}^q \Big[(1-\lambda^2)^q - (k\frac{\|\boldsymbol{\mu}_1\|_2}{\|\boldsymbol{\mu}_2\|_2}\lambda)^q\Big][1+o(1)]$$

$$\geq \frac{1}{2m}m\log(\frac{1}{\epsilon})\Big[(1-(\frac{1+C_2}{2})^2)^q - (k\frac{\|\boldsymbol{\mu}_1\|_2}{\|\boldsymbol{\mu}_2\|_2}\frac{1+C_2}{2})^q\Big]$$

$$= \widetilde{C}\log(\frac{1}{\epsilon})$$

$$> C_3,$$

where $C_4$ is a positive constant, the first inequality is by that the function $f(\lambda) = (1-\lambda^2)^q - (k\frac{\|\boldsymbol{\mu}_1\|_2}{\|\boldsymbol{\mu}_2\|_2}\lambda)^q$ is a decreasing function, and the last inequality is by $\epsilon \leq e^{-C_3/\widetilde{C}}$. Here the proof completes. $\qquad \square$

Under the same conditions as in Lemma F.21, it can be concluded that when $-\frac{1+C_2}{2} \leq \cos\theta_{1,2} < 0$, the probability $\mathbb{P}_{(\mathbf{x}_1,y_1)\sim\mathcal{D}_1}\big(y_1 \neq \text{sign}(f(\mathbf{W}^{(T_2,t_{end})}, \mathbf{x}_1))\big)$ is less than or equal to $\exp(-C \cdot m^{2q-2}n^{2q}/q^2)$, where $C = O(1)$ is a positive constant. Moreover, since $\frac{1+C_2}{2} > C_2$, it can be inferred that the replay method is capable of expanding the angular range corresponding to benign forgetting.

