# OpenReview forum: "Understanding the Forgetting of (Replay-based) Continual Learning via Feature Learning: Angle Matters"
_ICML.cc/2025/Conference — ICML 2025 poster_

### Official Review · Reviewer_z4gT · 2025-03-11

**Overall Recommendation:** 4

**Summary:**

The paper develops a unified theoretical framework for understanding catastrophic forgetting in continual learning through the lens of feature learning. The authors focus on a two-layer convolutional neural network with a polynomial ReLU activation function that is trained sequentially on binary classification tasks. Their key findings are:

- The extent of forgetting on previously learned tasks is critically influenced by the cosine similarity (i.e., the angle) between the task signal vectors. Specifically, when the angle is acute or only slightly obtuse, the network experiences benign forgetting with only minor performance degradation on old tasks. In contrast, larger obtuse angles lead to harmful forgetting with significant performance loss.
- Replay-based continual learning methods are shown to mitigate forgetting by effectively expanding the angular range that corresponds to benign forgetting. This insight leads to the proposal of a “mid-angle sampling” strategy, where examples are selected based on having a moderate cosine similarity to the class prototype. This strategy aims to balance stability and plasticity, further enhancing the effectiveness of replay methods.
- Theoretical results are rigorously supported by a detailed analysis of neuron behavior during training (via a signal-noise decomposition) and by characterizing how the network’s weight updates interact with task signal angles.
- Experimental validations on both synthetic datasets and real-world benchmarks (such as MNIST and CIFAR100) confirm the theoretical predictions, illustrating the relationship between task angles, forgetting, and the beneficial impact of replay and mid-angle sampling.

In summary, the paper contributes a novel theoretical perspective that links the geometric relationship between tasks to the phenomenon of forgetting in continual learning, and it introduces practical replay strategies inspired by this analysis.

## update after rebuttal
The authors address most of my concerns. So I keep my positive score.

**Claims And Evidence:**

The paper’s claims are largely supported by a combination of rigorous theoretical analysis and comprehensive experimental validations. In particular:

- The core claim—that the cosine similarity between task signal vectors critically influences the degree of forgetting—is backed by detailed theoretical derivations (e.g., Theorem 3.2 and Theorem 3.3) and is further substantiated through experimental evidence on both synthetic and real-world datasets.
- The analysis of neuron behavior via signal-noise decomposition provides a convincing mechanism for how different angles lead to either benign or harmful forgetting.
- The effectiveness of replay-based methods and the proposed mid-angle sampling strategy is supported by experiments that demonstrate improved performance under various task settings.

One potential concern is that the theoretical results rely on strong assumptions (such as over-parameterized two-layer CNNs and binary classification settings), which may limit the direct generalizability to more complex or different learning scenarios.

**Essential References Not Discussed:**

While the paper cites a broad range of works on catastrophic forgetting, continual learning, and feature learning theory, there are a few related lines of research that could further contextualize its key contributions:

- There is a growing body of work examining the geometric properties of learned representations and their impact on transfer or interference between tasks. For instance, studies on Neural Collapse (e.g., Papyan et al., 2020) reveal that deep networks tend to organize their features in a highly symmetric and clustered manner, which could be directly related to how task signal vectors interact. These works might provide additional insights into the role of angular relationships in feature representations.
- In the realm of deep metric learning, methods such as CosFace or ArcFace explicitly incorporate angular margins to improve discrimination between classes. Although these works focus on face recognition, the idea that angular separation can enhance class separability is relevant to understanding why certain angles between task signals might lead to benign versus harmful forgetting.

**Experimental Designs Or Analyses:**

The experimental designs appear sound and well-motivated for validating the theoretical claims. Here are some key points:

- Synthetic experiments were designed following a controlled data distribution (as described in Definition 1.1) with parameters (e.g., training sample size, dimension, noise variance) that allow a clear examination of the relationship between task angles and forgetting. This controlled setup helps in isolating the effects predicted by the theory.

- Real-world experiments on benchmark datasets such as MNIST and CIFAR100 are standard in continual learning research. They not only test the basic hypothesis regarding the cosine similarity between task signals but also validate the effectiveness of replay-based methods and the proposed mid-angle sampling strategy.

One potential limitation is that the experiments focus on binary classification and over-parameterized two-layer networks. While this aligns with the theoretical framework, it may limit the direct applicability of the findings to more complex settings (e.g., multi-class scenarios or deeper architectures).

**Methods And Evaluation Criteria:**

The methods and evaluation criteria appear well-tailored to the problem. The paper employs a rigorous theoretical framework—developed for a two-layer CNN with polynomial ReLU activation—and supports its findings with experiments on both synthetic data and established benchmarks like MNIST and CIFAR100. These datasets are standard in continual learning research, providing a reasonable basis for evaluating both forgetting and the effectiveness of replay-based methods. Additionally, the evaluation metrics (training loss, test loss, and test error on old tasks) align closely with the objectives of mitigating catastrophic forgetting. While the setting is somewhat restricted (e.g., binary classification and overparameterized networks), within this scope, the methods and criteria make sense and are appropriate for the application at hand.

**Other Comments Or Suggestions:**

Overall, the paper is solid with clear theoretical contributions and extensive experimental validations. Here are a few additional suggestions and minor comments:
- Consider adding a brief discussion that highlights potential avenues for extending the analysis beyond binary classification and two-layer networks, which could help contextualize the work for a broader audience.
- A few sections of the proofs are mathematically dense; including a high-level summary or intuition behind the most critical steps could further aid readers who are less familiar with the technical details.

**Other Strengths And Weaknesses:**

Strengths:
- Novel Theoretical Perspective: The paper presents a unique theoretical framework that links the geometric relationship (cosine similarity) between task signal vectors to catastrophic forgetting. This approach provides a fresh angle on understanding when forgetting is benign versus harmful.
 - Integration of Theory and Practice: By combining rigorous proofs with experimental validation on both synthetic and benchmark datasets (MNIST, CIFAR100), the work offers a comprehensive study that spans theory and practical implementation.

Weaknesses:
- Restrictive Assumptions: The theoretical analysis relies on assumptions such as over-parameterized two-layer CNNs, binary classification tasks, and specific conditions on the signal-to-noise ratio. These may limit the direct applicability of the results to more complex architectures or real-world scenarios that involve multi-class problems.
- Complexity of Theoretical Arguments: While the proofs are thorough, the level of mathematical complexity and the reliance on heavy technical machinery may make it challenging for practitioners who are less familiar with the theoretical underpinnings of deep learning.

**Questions For Authors:**

- The current analysis focuses on binary classification with two-layer CNNs. Could you elaborate on how the framework might extend to multi-class classification or deeper architectures? A clear discussion on this could enhance the practical impact and generalizability of your work.

- The mid-angle sampling strategy appears promising. Can you provide additional insights on how sensitive the method is to the choice of cosine similarity thresholds? Is there any theoretical or empirical guidance on selecting these thresholds optimally?

- Your proofs rely on a signal-noise decomposition of network weights. Could you offer more intuition or illustrative examples (e.g., visualizations) that clarify how this decomposition relates to neuron behavior and forgetting in continual learning?

**Relation To Broader Scientific Literature:**

The paper’s contributions are deeply connected with several strands of prior work in continual learning and theoretical deep learning. In particular:

- It extends earlier theoretical studies on catastrophic forgetting, which often rely on linear models or lazy training regimes (e.g., Evron et al., 2022; Doan et al., 2021), by analyzing a two-layer CNN with a polynomial ReLU activation in a feature learning setting. This move addresses limitations in capturing the dynamics of practical neural networks.
- The work builds on feature learning theory advances (e.g., Allen-Zhu & Li, 2020; Cao et al., 2022; Huang et al., 2023), adapting these ideas to the continual learning scenario. This integration allows for a unified framework that links the geometry (via cosine similarity between task signals) to the degree of forgetting.

**Theoretical Claims:**

I reviewed the proof sketches provided for the main theoretical results—specifically Theorem 3.2 (for standard continual learning) and Theorem 3.3 (for replay-based continual learning)—along with the supporting lemmas (e.g., Lemma 4.1 through Lemma 4.8). Within the framework of their stated assumptions (such as the over-parameterization of a two-layer polynomial ReLU CNN, binary classification settings, and certain conditions on the signal-to-noise ratio and network initialization), the proofs are logically consistent and appear to be correctly derived.

---

> ### Author Rebuttal · Authors · 2025-04-01
>
> Thanks for your constructive feedback! We address your questions and concerns as follows.
> > **Q1. The current analysis focuses on binary classification with two-layer CNNs. Could you elaborate on how the framework might extend to multi-class classification or deeper architectures? A clear discussion on this could enhance the practical impact and generalizability of your work.**
>
> While our current analysis focuses on two binary tasks for clarity and tractability, the angle-based framework naturally extends to multi-class and multi-task settings. Specifically, complex configurations can be decomposed into pairwise class-level interactions across tasks, with angular relationships capturing the core learning dynamics. This forms the basis for our analysis. We plan to extend our theory by studying how the accumulation of such pairwise interactions drives forgetting.
>
> Additionally, Jiang et al. recently employed a feature learning theory based on signal-noise decomposition to study benign overfitting in Vision Transformers [1]. Our exploration of continual learning in CNNs through this lens may serve as a foundation for extending such analysis to more complex models like Transformers.
>
> > **Q2. The mid-angle sampling strategy appears promising. Can you provide additional insights on how sensitive the method is to the choice of cosine similarity thresholds? Is there any theoretical or empirical guidance on selecting these thresholds optimally?**
>
> In fact, our mid-angle sampling strategy does not require threshold selection. The experiment follows the classical iCaRL framework for CL based on CNNs [2], which consists of a feature extractor and a classification layer. Since iCaRL allows only a fixed number of replay samples to be stored, our Mid-angle sampling strategy selects the most intermediate examples by first sorting all examples within each class based on their cosine similarity to the class prototype, and then selecting those closest to the median. We conduct experiments on the CIFAR100-5 (5 tasks with 20 classes each) and CIFAR100-10 benchmarks. As shown in Table 1, our mid-angle sampling outperforms herding—a nontrivial result given that herding is a commonly used sampling method in CL with replay.
>
> |Sampling|Random|Small-angle|Mid-angle|Big-angle|Herding|
> |-|-|-|-|-|-|
> |ave-accuracy ↑ (CIFAR100-10)|47.17 ± 0.45|45.63 ± 0.12|**48.02 ± 0.27**|45.34 ± 0.76|47.40 ± 0.17|
> |ave-forgetting ↓ (CIFAR100-10)|15.72 ± 0.31|19.47 ± 0.39|**14.84 ± 0.26**|18.04 ± 0.49|15.51 ± 0.08|
> |ave-accuracy ↑ (CIFAR100-5)|56.08 ± 0.12|54.36 ± 0.35|**56.51 ± 0.06**|54.77 ± 0.29| 56.12 ± 0.20|
> |ave-forgetting ↓ (CIFAR100-5)|11.15 ± 0.46 |14.10 ± 0.26| **10.15 ± 0.28**|12.50 ± 0.72|10.64 ± 0.13|
>
> *Table 1: Experimental Results with std and Average Forgetting on CIFAR100.*
>
> > **Q3. Your proofs rely on a signal-noise decomposition of network weights. Could you offer more intuition or illustrative examples (e.g., visualizations) that clarify how this decomposition relates to neuron behavior and forgetting in continual learning?**
>
> We first provide an intuitive explanation of the relationship between signal-noise decomposition and neuron behavior. In our setting, the signal vector $\mu$ is orthogonal to the noise $\xi$, forming a basis for a plane. After training, the CNN weights evolve as $\mathbf{w}_{j,r} = j\gamma\mu + \rho\xi$, where $\gamma \gg \rho$, indicating that the weights grow predominantly along the signal direction. This suggests that the network has effectively learned the signal. In multi-task scenarios, the weights adjust according to the signal vectors of tasks.
>
> Regarding forgetting in CL, we show that harmful forgetting mainly arises in the obtuse-angle case. As stated in Lemma 4.3 (page 5), if $\sum\_{r = 1}^{m}[\sigma(\langle\mathbf{w}\_{y\_{1},r}^{(T\_{2},t\_{end})}, y\_{1}\mu_{1}\rangle)-\sum\_{r = 1}^{m}\sigma(\langle\mathbf{w}\_{-y\_{1},r}^{(T\_{2},t\_{end})}, y\_{1}\mu\_{1}\rangle)] \geq C\_{3}$, then the CNN will achieve benign forgetting. We further derive that $\sum\_{r = 1}^{m}\sigma(\langle\mathbf{w}\_{y\_{1},r}^{(T\_{2},t_{end})}, y\_{1}\mu\_{1}\rangle)$ and $\sum\_{r = 1}^{m}\sigma(\langle\mathbf{w}\_{-y\_{1},r}^{(T\_{2},t\_{end})}, y\_{1}\mu\_{1}\rangle)$ can be characterized by $\Theta(m\overline{\gamma(\mu\_{1})\_{y\_{1}}}(1 - \cos^{2}\theta\_{1,2})^{q})$ and $\Theta(m\overline{\gamma(\mu\_{2})\_{-y\_{1}}}\frac{||\mu\_{1}||\_{2}^{q}(-\cos\theta\_{1,2})^{q}}{||\mu
> \_{2}||\_{2}^{q}})$ respectively. Then we obtain the angle range corresponding to benign forgetting; the analysis for harmful forgetting follows similarly.
>
> **References**
>
> [1] Jiang J, et al. Unveil benign overfitting for transformer in vision: Training dynamics, convergence, and generalization. In NeurIPS 2024
>
> [2] Rebuffi S A, et al. icarl: Incremental classifier and representation learning. In CVPR 2017

---

> > ### Comment · Reviewer_z4gT · 2025-04-05
> >
> > Thank you for the detailed responses, which address my questions about multi-class extensions, mid-angle sampling sensitivity, and signal-noise decomposition intuition. I keep my positive score.

---

### Official Review · Reviewer_N92G · 2025-03-11

**Overall Recommendation:** 4

**Summary:**

The authors propose a theoretical analysis of catastrophic forgetting in the two class setup for two layer convolutional neural networks, with polynomial RELU activations. They prove that for rehearsal free CL, forgetting is significant when the angle is between the new task and previous task is small enough. They also prove that if this angle is large enough, the forgetting can be upper bounded.
For CL methods with rehearsal, the authors prove that the benign forgetting range is larger therefore incurring less forgetting for more dissimilar tasks. \
Based on the findings above, the authors present a rehearsal method with mid angle rehearsal, to mitigate forgetting more effectively, compared to random rehearsal. This method is compared experimentally against other baselines on CIFAR-100 CL benchmarks.

**Claims And Evidence:**

The main claims stated in the contributions sections are supported with proofs.

In terms of experimental evidence :

Convincing evidence
- the claim about the forgetting regions with and without replay (Fig 1) is verified experimentally in Figure 2. However, while the experiments show that the replay setup has a larger range of non forgetting, it looks like the region that was identified as a grey area in the analysis is partially a significant forgetting region.

Missing evidence :
- It would be very informative to validate experimentally the tightness of the forgetting bounds in Theorems 3.2 and 3.3
- To validate the proposed rehearsal scheme, the std is missing from Table 1. The std is necessary to conclude on the significance of the mean improvement. More so given that the metrics are very close across the baseline. Also, I think that it's important to report the Average Forgetting as well, because conclusions cannot be made only based on the Average Accuracy.

**Essential References Not Discussed:**

I am not aware of any missing references.

**Experimental Designs Or Analyses:**

I checked all the experiments and shared some related comments in the "Methods And Evaluation Criteria" section.

Some additional comments :
- I may have misunderstood the rightmost plot in Figure 2, but isn't there a color error, shouldn't the blue curve decay to zero and the green increase monotonically ?
- Could you clarify how the experiment in Figure 3 translates to or validates the analysis ?
- In Table 1, do you consider the full CIFAR dataset or only two tasks ? Could you clarify it ?
- Optional : Could you run the synthetic experiments in the multitask setup, to see the impact of the angle on the final accuracy ? Do you think it would be sensible ?

**Methods And Evaluation Criteria:**

The proposed evaluation criteria is sensible overall.

The missing critical elements are the following :
- Reporting the std in Table 1
- Reporting the Average Forgetting in Table 1

Nice to have evaluations :
- Experimentally validating the tightness of the forgetting bounds in Theorems 3.2 and 3.3
- Experimentally validating the over-parameterisation lower bound for the Theorems

**Other Comments Or Suggestions:**

- Definition 1.1 : I would suggest clarifying that the intuition behind the covariance matrix is the orthogonality wrt the U
- Definition 1.1 : I suggest clarifying the intuition behind mu, it only became clear to me in the experiments section

**Other Strengths And Weaknesses:**

- Strengths :
   - Theoretical analysis of forgetting for a practical architecture (CNN), and interesting derivation of bounds on forgetting regimes depending on the angle between the tasks
   - Experiments in the same setup as the theory to validate some analytical observations
   - Deriving a practical application from the analysis - though the significance of the improvement is still unclear for now
   - Overall clear presentation of the intuition behind the theorems even though the notation is heavy
- Weaknesses :
   - Very restrictive assumptions (overparameterisation and architecture), it's unclear to which extend they could apply to more complex and widely used architectures.
   - Missing std in Table 1, therefore no conclusion is possible yet about this experiment
   - Unclear tightness of the bounds

**Questions For Authors:**

In addition to the questions in the other sections, I wanted to ask the following questions :

- Definition 1.1 : why is x_k subdivided into two vectors ? Is it the definition of a CNN in this analysis ?
- Definition 1.1 : what are the assumptions about the distribution D_k ?
- L88 : Could you explain the choice of the loss function and to which extend it is restrictive compared to the cross entropy loss ?
- Theorem 3.1 : is the upper bound on cos theta a typo, shouldn't it be 1 ? if not why ?

General questions :
- A large enough over-parametrisation is one of the assumptions of the theorems, is the lower bound not too large enough to fall in the lazy regime ?
- Why the choice of polynomial RELU activations, to which extent is it restrictive or does it translate to commonly used activations ?
- Does the analysis apply to non convolutional models ?
- What happens if the convolutions are not split between the signal and noise ?
- Optional :  In the multitask setup, under the same assumptions, is the final accuracy impacted by the angle ?

**Relation To Broader Scientific Literature:**

This work relates to the theoretical Continual Learning literature. Several works quantify the impact of task similarity on CF under different task, model and data assumptions : [2], [3], [4], [5], [6].

The analysis is also based on Feature Learning Theory. I am not familiar with this research area, however the analysis is significantly inspired by [1].


- [1] Cao, Yuan et al. “Benign Overfitting in Two-layer Convolutional Neural Networks.” ArXiv abs/2202.06526 (2022): n. Pag.
- [2] Bennani, Mehdi et al. “Generalisation Guarantees for Continual Learning with Orthogonal Gradient Descent.” ArXiv abs/2006.11942 (2020): n. Pag.
- [3] Doan, Thang Van et al. “A Theoretical Analysis of Catastrophic Forgetting through the NTK Overlap Matrix.” International Conference on Artificial Intelligence and Statistics (2020).
- [4] Lee, Sebastian et al. “Continual Learning in the Teacher-Student Setup: Impact of Task Similarity.” International Conference on Machine Learning (2021).
- [5] Evron, Itay et al. “How catastrophic can catastrophic forgetting be in linear regression?” ArXiv abs/2205.09588 (2022): n. Pag.
- [6] Evron, Itay et al. “The Joint Effect of Task Similarity and Overparameterization on Catastrophic Forgetting - An Analytical Model.” ArXiv abs/2401.12617 (2024): n. pag.
- [7] Hiratani, N. (2024). Disentangling and Mitigating the Impact of Task Similarity for Continual Learning. ArXiv, abs/2405.20236.

**Theoretical Claims:**

- Definition 1.1 : Could you clarify the design choice of splitting x into two vectors ?
- Sec 2. Could you clarify why the proposed neural network definition is a convolutional neural network ? Also isn’t it too simplistic to define one convolution wrt the signal and the second convolution wrt the noise ? Could you discuss the assumption and its possible limitations ?
- Could you discuss the loss assumption, and to which extent it is limiting to generalise the takeaways to more commonly used losses such as the cross entropy loss ?
- Is the analysis extensible to the multitask learning setup, where the data mixture has large obstute angles ?

I haven’t checked the proof in the Appendix.

---

> ### Author Rebuttal · Authors · 2025-04-01
>
> Thanks for your constructive feedback! We address your questions and concerns as follows.
> > **Is the grey area a significant forgetting region ?**
>
> The grey area is a region for uncertainty, either harmful or benign forgetting, ensuring that our claims remain rigorous. The yellow area is for harmful forgetting.
>
> > **Reporting the std and Average Forgetting in Table 1.**
>
> Due to space limitations, the answer is provided in our response to Reviewer #4 (z4gT), Comment Q2.
>
> > **Experimentally validating Theorem 3.2 and 3.3.**
>
> As Reviewer #2 (qzcp) noted, "the authors present a coherent chain of reasoning plus experiments that affirm their principal angle-based explanations for forgetting". Figure 2 shows that benign forgetting leads to near-zero error, while harmful forgetting approaches one, validating our Theorem 3.2 and 3.3. Figure 3 confirms both on MNIST. The second plot in Figure 2 shows that our over-parameterized model avoids the lazy training regime, with significant weight increase in the signal direction.
>
> > **I may have misunderstood the rightmost plot in Figure 2.**
>
> Blue indicates the maximum value, and green indicates the second-largest value, rather than a single continuous line. The intersection of the blue and green lines shows where the maximum value shifts between the two variables.
>
> > **Why is x_k subdivided into two vectors ?**
>
> Due to space limitations, the answer is provided in our response to Reviewer #2 (qzcp), Comment Q2.
>
> > **What are the assumptions about the distribution D_k ?**
>
> We assume $\xi_k \sim N\left(0, \sigma_{p_k}^2 \cdot \left(I - U (U^T U)^{-1} U^T\right)\right)$. The label $y_k$ is a Rademacher random variable. One of $x^{(1)}_k$ and $x^{(2)}_k$ is $y_k\cdot\mu_k$, the other $\xi_k$, with $(x_k, y_k)\sim D_k$.
>
> > **Explain the choice of the logistic loss. Is it restrictive ?**
>
> Logistic loss is a special case of cross-entropy loss for binary classification. We use $L_{CE}=-\frac{1}{n}\sum_{i = 1}^{n}[y_i\log(\hat{y}_i)+(1- y_i)\log(1-\hat{y}_i)]$, where $\hat{y}_i$ is softmax-normalized. After softmax calculations in $L\_{CE}$, we get the logistic loss used in our analysis.
>
> > **Theorem 3.1 : is the upper bound on cos theta a typo, shouldn't it be 1 ?**
>
> Theorem 3.1 does not appear in the paper. If the intended reference is Theorem 3.2 or 3.3, the range should be $1 \geq \cos\theta_{1,2} \geq 0$ for Theorem 3.2 and $-\frac{1+C_2}{2} \leq \cos\theta_{1,2} \leq 1$ for 3.3.
>
> > **Does the network fall in the lazy regime ?**
>
> Our approach avoids the lazy training regime by using smaller initialization, in contrast to the NTK setting. This allows the weights to move significantly and learn the signal ($\gamma_{j,r}^{(t)}$), rather than staying near initialization. Large initialization typically induces NTK-like behavior with minimal updates, while small initialization enables meaningful parameter growth along the signal direction, thus avoiding lazy regime.
>
> > **Why is the $Relu^q$ activation chosen ? Is it restrictive or does it translate to commonly used activations ?**
>
> Polynomial RELU can speed up both signal learning and noise memorization, to further boost the gap between them. Our theoretical framework can be extensible to RELU activation by techniques similar to those by Kou et al. (2023) [1].
>
> > **Does the analysis apply to non convolutional models ?**
>
> The answer is provided in our response to Reviewer #2 (qzcp), Comment Q2.
>
> > **Why the network is a CNN ? What happens if the convolutions are not split between the signal and noise ?**
>
> Our network structure, a common choice for theoretical analysis (including our references [1,2]), retains the core features of a CNN. We use a dual-channel input model $\mathbf{x}=[\mathbf{x}^{(1)},\mathbf{x}^{(2)}]$ with shared weights across channels. The output is:
> $ f=\frac{1}{m}\sum_{r = 1}^m\left[\sigma(\mathbf{w}\_{+1,r}^\top\mathbf{x}^{(1)})+\sigma(\mathbf{w}\_{+1,r}^\top\mathbf{x}^{(2)})\right]-\frac{1}{m}\sum_{r = 1}^m\left[\sigma(\mathbf{w}\_{-1,r}^\top\mathbf{x}^{(1)})+\sigma(\mathbf{w}\_{-1,r}^\top\mathbf{x}^{(2)})\right] $.
> Here, $m$ is the number of filters, $\sigma(z)=(\max\{0,z\})^q$ ($q > 2$), and $\mathbf{w}\_{j,r}$ are filter weights. The structure approximates $\sigma(\mathbf{w}\_{j,r}^\top(\mathbf{x}^{(1)}+\mathbf{x}^{(2)}))$, which is commomly used in feature learning theory [1,2]. Furthermore, We can extend the two-patch data to multi-patch data model similar to Allen-Zhu et al. (2020) [3].
>
> > **In the multitask setup, is the final accuracy impacted by the angle ?**
>
> The answer is provided in our response to Reviewer #2 (qzcp), Comment Q1.
>
> **References**
>
> [1] Kou, Y., et al. Benign overfitting in two-layer ReLU convolutional neural networks. In ICML 2023
>
> [2] Cao, Y., et al. Benign overfitting in two-layer convolutional neural networks. In NeurIPS 2022
>
> [3] Allen-Zhu, Z., et al. Towards understanding ensemble, knowledge distillation and self-distillation in deep learning. arXiv, 2020

---

### Official Review · Reviewer_qzcp · 2025-03-12

**Overall Recommendation:** 4

**Summary:**

The paper develops a theoretical framework for understanding continual learning (CL) and catastrophic forgetting using a two-layer polynomial ReLU CNN. It focuses on how the angle between two tasks’ “signal vectors” (representing core features for each task) influences forgetting: if the angle is acute or only mildly obtuse, forgetting from the first task remains “benign,” but if the angle is large (i.e., vectors are nearly opposite), the model experiences “harmful” forgetting. The authors prove these claims by characterizing network training through a signal-versus-noise decomposition and analyzing neuron behavior under gradient descent. They also show that replay-based methods expand the range of angles for which forgetting stays benign and introduce a “mid-angle sampling” strategy that selects replay samples with moderate angles to their class prototypes, demonstrating an improvement over standard sampling techniques in empirical tests on synthetic data, MNIST, and CIFAR100. ## update after rebuttal

**Claims And Evidence:**

The paper’s main claims revolve around (1) the theoretical relationship between the angle of two tasks’ signal vectors and the severity of forgetting, (2) the ability of replay methods to expand the range of angles over which forgetting remains benign, and (3) the benefit of “mid-angle sampling.” Below is how the paper substantiates these claims:

1.Angle and Forgetting: The authors provide a formal derivation under a polynomial ReLU two-layer CNN. They track the gradient-based evolution of weights with a “signal-noise decomposition,” proving that the inner-product behaviors align with angle-based predictions. They construct a controlled “signal + noise” dataset with varying angles. The observed forgetting closely matches the theoretically predicted angular thresholds.

Comments: The proofs are detailed and logically consistent with prior feature-learning analyses. The empirical curves on synthetic data indeed show changes in forgetting severity at about the angles predicted.

2.Replay Expands the Benign-Forgotting Range: The authors augment their theoretical framework to account for stored samples of the previous task and show that, under certain buffer size conditions, harmful forgetting is avoided even when the angle is moderately large. They run experiments (both synthetic and on MNIST, CIFAR100) contrasting the “no replay” condition against “with replay,” then measure performance on the old task.

Comments: The paper does not fully generalize this to many tasks, but the data for the two-task scenario supports the claim well.
3.Mid-Angle Sampling: On MNIST and CIFAR100, they compare mid-angle sampling with random sampling and herding. Results show small but consistent accuracy improvements on older tasks.

Comments: margin of improvement is not enormous. The mechanism for why it works is rooted in their angle-based theoretical analysis, which is coherent within their two-task scope.

Overall, the authors present a coherent chain of reasoning plus experiments that affirm their principal angle-based explanations for forgetting, along with replay’s benefits and the utility of mid-angle sampling. However, there are several weaknesses:
●Simplicity of the Data Model: The paper’s theoretical and synthetic experiments heavily rely on a fairly stylized “signal + noise” data model. Real-world data can be more varied, so it may be difficult to guarantee the same clean angle-based properties in practice.
●Limited Improvements in Experiments: While the mid-angle sampling approach does show some gains over standard replay sampling, the performance boost is not markedly large. The experiments, though suggestive, do not represent a major breakthrough in empirical performance.

**Essential References Not Discussed:**

It appears that the paper adequately addresses the relevant prior work for its main theoretical results and replay-based methods.

**Experimental Designs Or Analyses:**

Strengths
●Clear Connection to Theory: The synthetic setup precisely matches the assumptions in the paper, making it easy to see the influence of angles on forgetting.
●Use of Standard Benchmarks: Validating on MNIST and CIFAR100 shows that the angle-based findings and replay strategy improvements hold in relatively common experimental contexts, beyond purely synthetic data.
●Systematic Comparisons: They compare no replay vs. replay, as well as different sampling approaches (including mid-angle sampling), which cleanly highlights each method’s impact on forgetting.
Weaknesses
●Limited Scope: The experiments focus on binary classification in a two-task scenario. It’s unclear how the angle-based conclusions might extend to more tasks or multi-class settings.
●Incremental Gains: While mid-angle sampling does yield improvements, the empirical boost over standard sampling strategies (like random or herding) is not very large.
●Data Model Simplifications: The synthetic data strictly follows a “signal + noise” model, which may not capture all complexities of real-world datasets.

**Methods And Evaluation Criteria:**

In the context of a theoretical study on continual learning, the paper’s methods and chosen evaluation criteria are generally reasonable for the goals it aims to achieve, but there are also some limitations:

Signal+Noise Synthetic Model: The authors use a carefully controlled synthetic dataset to verify their angle-based theoretical predictions. Because the paper is largely focused on proving formal guarantees, having a simple generative setup where signal vectors, noise, and angles can be precisely controlled is sensible. It helps isolate and confirm the paper’s core theory on how angles influence forgetting.

Two-Layer Polynomial ReLU CNN: This is a restricted but analytically tractable network architecture. For a primarily theoretical analysis, using a simplified architecture that captures core nonlinear effects (rather than a purely linear or kernelized model) is a rational choice. It allows them to go beyond lazy training assumptions and linear analyses.

Nevertheless, the choices of methods and datasets reflect a clear effort to validate both theoretical and practical aspects of the approach. By anchoring the proofs in a specifically designed synthetic model, the authors can rigorously pinpoint the conditions under which angle-based insights hold. At the same time, employing MNIST and CIFAR100—despite being relatively standard benchmarks—demonstrates that the proposed ideas and replay strategy are not confined to purely toy examples.

**Other Comments Or Suggestions:**

Additional Suggestions
●Typos and Minor Clarifications: A quick proofreading pass could help catch minor linguistic issues, particularly in the theorem statements and figure captions. Ensuring complete alignment of notation between the main text and supplementary would also enhance clarity.
●Further Exploration: For readers seeking deeper insight into how angles evolve across multiple tasks, the paper could briefly outline potential extensions beyond the two-task, two-layer setup—even if only at a conceptual level.

**Other Strengths And Weaknesses:**

Other Strengths
1.Originality of Angle-Based Analysis: Although researchers have long recognized that task similarity can affect forgetting, framing this in terms of a precise angle between “signal vectors” provides a fresh, more mathematically rigorous viewpoint.
2.Balanced Theoretical and Empirical Components: By combining rigorous proofs with both synthetic and real-data experiments, the paper goes beyond many purely theoretical treatments and offers a more complete picture.
Other Weaknesses： Focus on Two-Task Setting: While the theoretical insights may be extended, the paper’s primary focus is a two-task scenario, which leaves open how well these angle-based insights hold for longer task sequences.

**Questions For Authors:**

Question: Your theory focuses primarily on two binary tasks. Could you outline how you would expect the angle-based framework and replay analysis to extend if there were multiple sequential tasks or multi-class tasks for each session?
Question: Do you see a straightforward way to relax the “signal + noise” assumption to capture more varied real-world data distributions? Or do you view the current model as primarily a stepping-stone for further exploration?
Question: How significant is the computational cost of measuring and comparing angles (or proxies for them) in mid-angle sampling, particularly for large networks or large-scale datasets?

**Relation To Broader Scientific Literature:**

The paper’s central focus—analyzing catastrophic forgetting through the lens of feature learning and the geometry between tasks—connects directly to several threads in the existing continual learning literature.

**Theoretical Claims:**

The paper’s most notable highlight is how it thoroughly compares two methods—standard continual learning versus a replay-enhanced version—and demonstrates, both theoretically and empirically, how replay expands the range of angles for benign forgetting. Additionally, the technical derivation that underpins these findings is quite detailed, showcasing a clear, step-by-step structure. The logical flow—spanning from the setup of the signal-noise decomposition, to the rigorous lemmas about inner products and gradient dynamics, and ultimately to the theorems on angle-based forgetting—reflects a methodical and well-organized presentation. Together, these aspects make the core results not only transparent but also easy to follow.

---

> ### Author Rebuttal · Authors · 2025-04-01
>
> Thanks for your constructive feedback! We address your questions and concerns as follows.
> > **Q1. Your theory focuses primarily on two binary tasks. Could you outline how you would expect the angle-based framework and replay analysis to extend if there were multiple sequential tasks or multi-class tasks for each session?**
>
> We validate the relationship between forgetting and angle in multi-task settings using the synthetic dataset. We conduct two sets of experiments, each consisting of three tasks. In Experiment E1, Task 3 forms angles $\theta_{1,3} = 150°$ and $\theta_{2,3} = 100°$ with Tasks 1 and 2, respectively. In Experiment E2, these angles are $\theta_{1,3} = 170°$ and $\theta_{2,3} = 80°$. Let $Acc_{1}$ and $Acc_{2}$ denote the accuracy on Tasks 1 and 2, respectively, after learning all three tasks. As shown in Table 1, the conclusion consistently holds.
>
> ||$(\theta\_{1,3},Acc\_{1})$|$(\theta\_{2,3},Acc\_{2})$|
> |-|-|-|
> |$E\_1$|$(150°,2.5\\%)$|$(100°,99.7\\%)$|
> | $E\_2$|$(170°,0.3\\%)$|$(80°,100\\%)$|
>
> *Table 1: Experimental Results in multi-task settings on the synthetic dataset.*
>
> While our current analysis focuses on two binary tasks for clarity and tractability, the angle-based framework naturally extends to multi-class and multi-task settings. Specifically, complex configurations can be decomposed into pairwise class-level interactions across tasks, with angular relationships capturing the core learning dynamics. This forms the basis for our analysis. We plan to extend our theory by studying how the accumulation of such pairwise interactions drives forgetting and how subproblems influence one another in more general settings.
>
> > **Q2. Do you see a straightforward way to relax the “signal + noise” assumption to capture more varied real-world data distributions? Or do you view the current model as primarily a stepping-stone for further exploration?**
>
> The signal-noise data model takes inspiration from image data, where the inputs are composed of various patches, and only certain patches are relevant to the class label of the image. This model has been widely adopted in recent theoretical studies, including our references [1,2]. Furthermore, the two-patch setting can be extended to a multi-patch model by techniques similar to those in Allen-Zhu et al. (2020) [3].
>
> Jiang et al. recently employed a feature learning theory based on signal-noise decomposition to study benign overfitting in Vision Transformers [4]. Our exploration of continual learning in CNNs through this lens may serve as a foundation for extending such analysis to more complex models like Transformers.
>
> > **Q3. How significant is the computational cost of measuring and comparing angles (or proxies for them) in mid-angle sampling, particularly for large networks or large-scale datasets?**
>
> We primarily implement the mid-angle sampling strategy by computing cosine similarity, whose computational cost should be comparable to the original herding strategy adopted in the iCaRL framework (which relies on Euclidean distance) [5].
>
> > **Typos and Minor Clarifications**
>
> We appreciate the thoroughness of the review and the opportunity to improve the accuracy and professionalism of our paper. We commit to fixing all minor writing issues and ensuring consistent notation to further improve clarity.
>
> **References**
>
> [1] Cao, Y., et al. Benign overfitting in two-layer convolutional neural networks. In NeurIPS 2022
>
> [2] Kou, Y., et al. Benign overfitting in two-layer ReLU convolutional neural networks. In ICML 2023
>
> [3] Allen-Zhu, Z., et al. Towards understanding ensemble, knowledge distillation and self-distillation in deep learning. arXiv, 2020
>
> [4] Jiang J, et al. Unveil benign overfitting for transformer in vision: Training dynamics, convergence, and generalization. In NeurIPS 2024
>
> [5] Rebuffi S A, et al. icarl: Incremental classifier and representation learning. In CVPR 2017

---

### Official Review · Reviewer_2zRv · 2025-03-15

**Overall Recommendation:** 2

**Summary:**

The paper provides a mathematical framework of forgetting in continual learning, for the specific case of a two-layer convolutional neural network with polynomial ReLU activation. The authors show that replay has the effect of increasing the range of settings under which forgetting is limited. Based on their analysis, they also propose a scheme for sampling mid-angle examples for the buffer, which has a slightly positive effect in an experiment on Cifar 10 and Cifar 100.

**Claims And Evidence:**

The main claims of the paper seem accurate.

My main concern is related to the relevance of the studied setting, which is quite different from a practical neural network setup. This applies not so much to the fact that it's only two layers, as most theoretical works focus on simplified architectures, even just one layer or linear models. More importantly, they seem to work with a shared head, whereas separate heads for each tasks is mostly used in practice, and they assume the noise is orthogonal to the signal vectors from all tasks (definition 1.1). I realize the latter setting is adopted from earlier work (Cao et al., 2022), but still I find it a strong assumption that is not well motivated. Finally, the signal and noise vector are processed separately by the network, which again is not how it works in practice.

**Essential References Not Discussed:**

NA

**Ethical Review Concerns:**

/

**Experimental Designs Or Analyses:**

Experiments are limited, as this is mostly a theoretical paper. Experiment 1 illustrates the theoretical setup.  Experiment 2, classifying MNIST digits vs. their inverse, is quite extreme, yet by design still very close to the theory. A multi-head setup would have made it more realistic. Experiment 3 lacks details (in main paper): what network is used ? what range is actually sampled for "moderate cosine similarity"  ? The differences between the different sampling methods are very small, making one wonder if they are significant at all.

**Methods And Evaluation Criteria:**

1. The definition of Forgetting (end of section 1.1) is weird. Instead of measuring the true error on the first task, it should measure the increase in test error on the first task, i.e. L_D1^{0-1}(W^(T2)) - L_D_1^{0-1}(W^(T1)). In practice, in the simple setting and with all the assumptions made, the test error after training task 1 (second term) is close to zero, so it doesn't impact too much, but still...

2. Initially, I was charmed by the idea of a theoretical paper that, at the same time, included a practical algorithm evaluated on more common CL setups using CIFAR and some larger network (not specified though). However, he proposed method applying mid-angle sampling is only very weakly related to the theory given earlier in the paper. I would appreciate if the authors could elaborate how the proposed strategy follows directly from the earlier theorems.

**Other Comments Or Suggestions:**

## update after rebuttal ##
I stick to my original score, as I'm still not convinced what the added value of this paper is.

First, I don't think the theoretical analysis really brings us more insights. That angles matter is, in fact, rather intuitive. It's equivalent to saying something like 'distance to the decision boundary matters', but then on a unit sphere. I'm not impressed.

Second, I emphasize the impact of the 'single head' setting. It's not just a "slight difference between theoretical settings and practical setup", as stated by the authors. It's a completely different setup that influences the analysis drastically, and it wasn't even discussed in the paper as being a deviation from the practical setting. At the very least authors should be transparant about all the simplifications they make, rather than hiding them and hope readers overlook. Without such transparancy, papers like this bring a false impression of theoretical foundation. Using a single head only makes sense to me for domain incremental settings, where the angles are typically small anyway. No one with some common sense in continual learning would try a domain incremental setting with extreme domain changes. If that were the case, probably a task incremental setting would be selected instead (i.e., first identifying the task).

**Other Strengths And Weaknesses:**

The structure of the paper should be revised.
Section 1.2 ('Main Contributions') is impossible to follow, as many of the symbols used are only introduced later in Section 2.

**Questions For Authors:**

1. How does the proposed 'mid-angle' sampling strategy relate to the theoretical theorems given earlier ?
2. Please discuss the choice for a single head setup.
3. The choice for noise that is orthogonal to all signal vectors (from all tasks) seems a very restrictive one. Please discuss.

**Relation To Broader Scientific Literature:**

Overall, the relation to the broader scientific literature is well described.

I just found it a bit condescending to refer to methods such as EWC as "empirical methods".

**Theoretical Claims:**

I did not check all proofs in the appendix (50 pages!). As far as I checked, the math is correct under the defined setting / assumptions.

---

> ### Author Rebuttal · Authors · 2025-04-01
>
> Thanks for your constructive feedback! We address your questions as follows.
> > **Q1. How the mid-angle sampling strategy relates to the theoretical theorems**
>
> We sincerely appreciate the opportunity to clarify the connection between our theoretical findings and mid-angle sampling. Our theoretical results show that smaller angles between task signal vectors lead to benign forgetting, while larger angles cause harmful forgetting. In practice, we treat each class prototype—the mean feature of its examples—as the signal vector.
>
> We focus on the case where the angle between task prototypes is obtuse, as harmful forgetting arises only in this setting. If a sample’s feature forms a larger angle with its own prototype, it tends to form a smaller angle with the second task’s prototype, likely falling within the benign forgetting range in standard CL (i.e., it can be remembered without replay). Conversely, if the angle with its own prototype is smaller, the angle with the second task’s prototype may be larger—possibly beyond the benign range under CL with replay (i.e., it will be forgotten despite replay).
>
> In contrast, samples with mid-range angles are more likely to fall outside the benign range of standard CL but within that of replay-based CL—making them the most effective candidates for replay. Thus, mid-angle sampling offers a more efficient and targeted replay strategy.
>
> > **Q2. Discuss the choice for a single head setup.**
>
> We adopt the single-head setting primarily to facilitate theoretical analysis. However, our theoretical framework can extend to the multi-head setting. In fact, our final forgetting results are derived by analyzing the behavior of individual neurons, through which we characterize the angle-dependent antagonism between tasks. While the multi-head setup may increase the number of neurons involved in learning the feature of the second task, the core antagonism between tasks (driven by the angle) still persists.
>
> Moreover, We emphasize that slight differences between theoretical settings and practical setups are common to ensure analytical robustness. As Reviewer #2 (qzcp) noted, "For a primarily theoretical analysis, using a simplified architecture that captures core nonlinear effects (rather than a purely linear or kernelized model) is a rational choice." In this regard, our framework goes beyond linear and NTK-based models, making it more aligned with practical scenarios.
>
> > **Q3. The noise that is orthogonal to all signal vectors seems restrictive.**
>
> We adopt the orthogonality assumption primarily to simplify the proof and reduce the length of the manuscript. In fact, by techniques similar to those used by Kou et al. (2023) [1], we can extend our theoreical results to non-orthogonality case.
>
> > **The signal and noise vector are processed separately by the network.**
>
> This setting, commonly used in analyses of feature learning theory [1,2], mainly serves to simplify the proof. In practice, we can extend to multi-patch model using techniques similar to those of Allen-Zhu et al. (2020) [3].
>
> > **The definition of Forgetting is weird.**
>
> As you noted, the approximation using $L_{D_1}^{0-1} (W^{(T_2)})$ has limited impact since $L_{D_1}^{0-1} (W^{(T_1)})$ approaches zero. We commit to clarifying this to avoid any misunderstanding.
>
> > **Experiment 3 lacks details with mid-angle sampling a slightly positive effect**
>
> The experiment follows the classical iCaRL framework for CL based on CNNs [4], which consists of a feature extractor and a classification layer. Since iCaRL allows only a fixed number of replay samples to be stored, our Mid-angle sampling strategy selects the most intermediate examples by first sorting all examples within each class based on their cosine similarity to the class prototype, and then selecting those closest to the median. As shown in Table 1 in our response to Reviewer #4 (z4gT), Comment Q2, our mid-angle sampling outperforms herding—a nontrivial result given that herding is a widely used sampling method in CL.
>
> > **Condescending to refer to methods such as EWC as "empirical methods".**
>
> Here, we follow the description from Ding et al. (2024) [5]. We will remove the misleading term "empirical methods" and revise the description for clarity.
>
> > **The structure of the paper should be revised.**
>
> Thanks for your suggestion regarding the structure of our manuscript. We will move Section 2 before 1.2 to improve readability.
>
> **References**
>
> [1] Kou, Y., et al. Benign overfitting in two-layer ReLU convolutional neural networks. In ICML 2023
>
> [2] Cao, Y., et al. Benign overfitting in two-layer convolutional neural networks. In NeurIPS 2022
>
> [3] Allen-Zhu, Z., et al. Towards understanding ensemble, knowledge distillation and self-distillation in deep learning. arXiv, 2020
>
> [4] Rebuffi S A, et al. icarl: Incremental classifier and representation learning. In CVPR 2017
>
> [5] Ding, M., et al. Understanding forgetting in continual learning with linear regression. arXiv, 2024

---

### Decision · Program_Chairs · 2025-05-01

**Decision:**

Accept (poster)

**Comment:**

The paper develops a unified theoretical framework for continual learning (CL) based on feature learning theory, focusing on a two-layer CNN with polynomial ReLU activation. It shows that the angle between task signal vectors significantly affects forgetting. The study also explains how replay methods reduce forgetting by expanding the range of benign angles. A new mid-angle sampling strategy is proposed to further improve replay performance. Experimental results on both synthetic and real-world data validate the theoretical findings.

While reviewers acknowledge the value of the proposed framework,  they share some common concerns, including the strong assumptions and lack of generalization to multiple sequential tasks. The authors' rebuttal has addressed these important concerns to certain extent  and most reviewers support the acceptance of the paper in the end.

Overall, the theoretical results are not entirely surprising as pointed some reviewers. Meanwhile, the use of a single-head setup also deviates from practical continual learning settings, particularly in class-incremental scenarios. The paper does not clearly state its simplifying assumptions, which may undermine its theoretical contribution and create a less strict sense of generality.

The authors are suggested to carefully incorporate these comments and further clarify the scope and significance of their contributions to strengthen the presentation.